# TabEBM: A Tabular Data Augmentation Method with Distinct Class-Specific Energy-Based Models

**Andrei Margeloiu[1]\*, Xiangjian Jiang[1]\*, Nikola Simidjievski[2,1], Mateja Jamnik[1]**

[1]Department of Computer Science and Technology, University of Cambridge, UK
[2]PBCI, Department of Oncology, University of Cambridge, UK
{am2770, xj265, ns779, mj201}@cam.ac.uk

## Abstract

Data collection is often difficult in critical fields such as medicine, physics, and chemistry, yielding typically only small tabular datasets. However, classification methods tend to struggle with these small datasets, leading to poor predictive performance. Increasing the training set with additional synthetic data, similar to data augmentation in images, is commonly believed to improve downstream tabular classification performance. However, current tabular generative methods that learn either the joint distribution $p(\mathbf{x}, y)$ or the class-conditional distribution $p(\mathbf{x} \mid y)$ often overfit on small datasets, resulting in poor-quality synthetic data, usually worsening classification performance compared to using real data alone. To solve these challenges, we introduce TabEBM, a novel class-conditional generative method using Energy-Based Models (EBMs). Unlike existing tabular methods that use a shared model to approximate all class-conditional densities, our key innovation is to create distinct EBM generative models for each class, each modelling its class-specific data distribution individually. This approach creates robust energy landscapes, even in ambiguous class distributions. Our experiments show that TabEBM generates synthetic data with higher quality and better statistical fidelity than existing methods. When used for data augmentation, our synthetic data consistently leads to improved classification performance across diverse datasets of various sizes, especially small ones. Code is available at https://github.com/andreimargeloiu/TabEBM.

## 1 Introduction

Many scientific domains within medicine, physics, and chemistry often rely on intricate and challenging data acquisition procedures [5, 50, 4, 32, 76, 12] that typically render small-size tabular datasets [5, 46]. Using these to train machine learning models that can aid in tasks such as disease diagnosis [52, 37], material property prediction [35], and chemical compound classification [11], can lead to poor performance [74, 52, 37]. In the case of learning tasks which leverage image and text data, a standard remedy to address performance issues due to data scarcity is employing data augmentation techniques [72, 73, 60, 71] that generate additional synthetic samples from existing data.

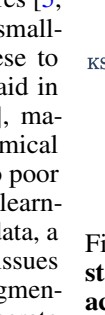

Figure 1: **Evaluation of TabEBM and other state-of-the-art tabular generative methods across six key metrics** (larger area indicates better performance). The results demonstrate that TabEBM excels in data augmentation (utility), with a larger area than all other methods.

---

\*Equal contribution.

38th Conference on Neural Information Processing Systems (NeurIPS 2024).

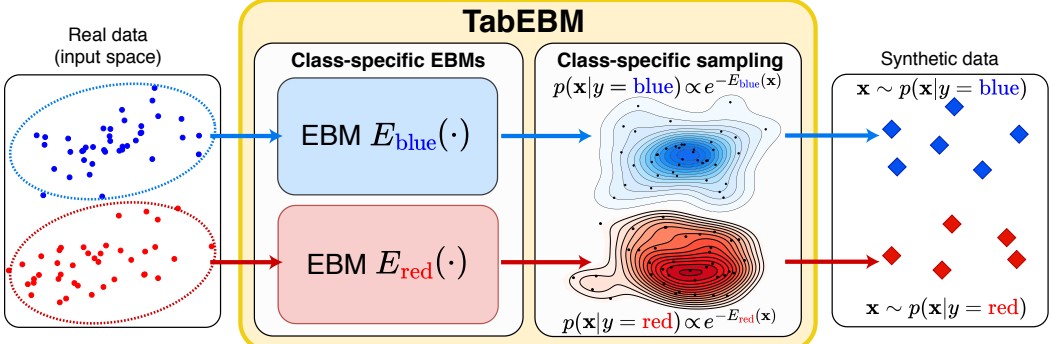

Figure 2: **An overview of TabEBM**. We learn *distinct* class-specific Energy-Based Models (EBMs) $E_{\text{blue}}(\mathbf{x})$ and $E_{\text{red}}(\mathbf{x})$ exclusively on the points of their respective class. Each EBM approximates a class-conditional distribution $p(\mathbf{x}|y)$. TabEBM allows synthetic data generation by sampling from the estimated distributions for each class $p(\mathbf{x}|y = \text{blue})$ and $p(\mathbf{x}|y = \text{red})$.

However, applying data augmentation to tabular data introduces additional challenges, as tabular datasets are often very diverse and lack explicit symmetries [8], such as rotations or translations seen in images. Consequently, existing tabular data augmentation methods often yield mixed results and can even degrade model performance [51, 71, 48], hindering their widespread adoption.

Tabular augmentation typically involves training generative models to approximate either the joint distribution $p(\mathbf{x}, y)$ [85, 24] or the class-conditional distribution $p(\mathbf{x}|y)$ [85, 42, 83, 47, 48]. A key challenge of joint distribution methods is maintaining the original training label distribution, as sampling from such generators can produce label distributions that deviate from the original and even fail to generate data for specific classes (see Appendix C for an example). These issues compromise the effectiveness of data augmentation [51] by undermining the label accuracy and distribution. On the other hand, while class-conditional models that learn $p(\mathbf{x}|y)$ preserve the stratification of the original data, they often employ a *shared* model to represent all class-conditional densities. This, however, can lead to overfitting, particularly in imbalanced datasets where the model may prioritise more frequent classes [21], ignoring unique features needed for generating label-invariant samples. Additionally, in datasets with limited data, this can lead to mode collapse [68, 70], where the model does not effectively capture the diversity of each class [70], and thus tends to perform poorly in a multi-class setting.

To address the challenges of class-conditional tabular generation, we introduce TabEBM (Figure 2), a new method for tabular data augmentation utilising Energy-Based Models (EBMs). Our method introduces two innovations: (i) *Distinct class-specific models:* TabEBM constructs a collection of individual models – one for each class – which, by design, enables learning distinct marginal distributions for the inputs associated with each class. This, in turn, enables performing data augmentation while maintaining the original label distribution. (ii) *Generative models:* we build novel class-specific generators that produce high-quality synthetic data even from extremely few samples. Specifically, we create a surrogate binary classification task for each class and fit it with a pre-trained tabular in-context classifier. We then convert the binary classifier into an EBM, a generative model, without additional training. Using class-specific EBMs makes the energy landscape more robust to class overlaps, compared to using a single shared EBM to approximate the class-conditional distribution.

Our contributions can be summarised as:

- **Technical:** We propose TabEBM, which is the first generative method to create class-specific EBMs, learning the marginal distribution for each class separately.

- **Empirical:** We present the first comprehensive analysis of tabular data augmentation across different dataset sizes and use cases beyond predictive performance. Our analysis compares TabEBM with eight leading tabular generative models across various datasets, demonstrating that TabEBM consistently improves data augmentation performance on small datasets, while our generated data demonstrates better statistical fidelity and privacy-preserving properties (Figure 1).

- **Library:** We release TabEBM as an open-source library, available at https://github.com/andreimargeloiu/TabEBM. Our library enables off-the-shelf data generation and data augmentation on any tabular dataset without requiring training. Further details are available in Appendix B.5.

## 2 TabEBM

**Notation.** We address classification problems with $C$ classes, denoted by $\mathcal{Y} = \{1, 2, \ldots, C\}$. Let $\{(\mathbf{x}^{(i)}, y_i)\}_{i=1}^N$ represent a dataset of $N$ samples, each being a $D$-dimensional vector $\mathbf{x}^{(i)} \in \mathbb{R}^D$, with a corresponding label $y_i \in \mathcal{Y}$. For each class $c \in \mathcal{Y}$, we define $\mathcal{X}_c = \{\mathbf{x}^{(i)} \mid y_i = c\}$ as the subset of samples labelled with class $c$. Let $f_\theta(\cdot)$ denote a classifier. The expression $f_\theta(\mathbf{x})[y]$ represents the (unnormalised) logit assigned to the class $y$ for the input $\mathbf{x}$.

### 2.1 Preliminaries on Energy-Based Models

An Energy-Based Model (EBM) [43] defines a probability density function $p_\theta(\mathbf{x})$ through an energy function $E(\mathbf{x})$. Specifically, the model posits that $p(\mathbf{x}) \propto e^{-E(\mathbf{x})}$, where $E(\mathbf{x})$ represents the unnormalised negative log-density of the input $\mathbf{x}$. In this framework, lower energy values correspond to higher probability densities. This relationship allows EBMs to model distributions by learning to assign lower energy to more probable configurations of $\mathbf{x}$ and higher energy to less probable ones.

An important observation is that energy-based models can utilise the same model architectures as standard classification models [29]. Typically, the logits $f_\theta(\mathbf{x})[y]$ from a classification model define a discriminative distribution through the softmax function, expressed as $p_\theta(y|\mathbf{x}) = \mathrm{softmax}(f_\theta(\mathbf{x})[y])$. Intriguingly, these same logits can be reinterpreted to define an energy-based model for the joint distribution $p(\mathbf{x}, y)$. This is achieved by setting the energy function to $E(\mathbf{x}, y) = -f_\theta(\mathbf{x})$. Furthermore, the energy function for the marginal distribution $p(\mathbf{x})$ is obtained by marginalising over $p(\mathbf{x}, y)$, resulting in $E(\mathbf{x}) = -\mathrm{LogSumExp}_{y'} f_\theta(\mathbf{x})[y']$.

Such an energy-based model, trained with EBM-specific protocols on multiple classes, is typically used as a classifier, as demonstrated on several computer vision tasks in [29]. In contrast, in this work our focus is the opposite: we propose employing trained classifiers, one for each specific class, as a generative energy-based model for the class-conditional distributions $p(\mathbf{x}|y)$. We apply our TabEBM method for generative tasks on tabular data.

### 2.2 Distinct Class-Specific Energy-Based Models

TabEBM is a class-conditional generative model $p(\mathbf{x}|y)$ implemented using a set of EBMs, $\{E_1(\mathbf{x}), E_2(\mathbf{x}), \ldots, E_C(\mathbf{x})\}$. Our approach assumes that the class-conditional density $p(\mathbf{x}|y = c)$ is best modelled using its class-specific data $\mathcal{X}_c$. Thus, for each class $c$, we construct a class-specific EBM, $E_c(\mathbf{x})$, using only the data from that class, $\mathcal{X}_c$, such that $p(\mathbf{x}|y = c) \propto \exp(-E_c(\mathbf{x}))$.

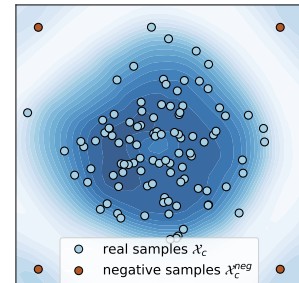

We derive each class-specific EBM $E_c(\mathbf{x})$ by training a classifier on a novel task and reinterpreting its logits. Specifically, for each class $c$, we propose a *surrogate binary classification task* to determine if a sample belongs to class $c$ by comparing $\mathcal{X}_c$ against a set of surrogate negative samples $\mathcal{X}_c^{\mathrm{neg}}$, which we show in Figure 3. Specifically, we generate the negative samples at the corners of a hypercube in $R^D$. For each dimension $d$, the coordinates of a negative sample are either $\alpha_{\mathrm{dist}}^{\mathrm{neg}} \sigma_d$ or $-\alpha_{\mathrm{dist}}^{\mathrm{neg}} \sigma_d$, where $\alpha_{\mathrm{dist}}^{\mathrm{neg}}$ is a fixed constant and $\sigma_d$ is the standard deviation of dimension $d$. For example, in $R^3$, a negative sample might have coordinates $[\alpha_{\mathrm{dist}}^{\mathrm{neg}} \sigma_1, \alpha_{\mathrm{dist}}^{\mathrm{neg}} \sigma_2, -\alpha_{\mathrm{dist}}^{\mathrm{neg}} \sigma_3]$. Placing the negative samples at the corners of a hypercube ensures they are easily distinguishable from the real data, which is crucial for an accurate energy function (see Appendix D.1.1). This placement is also robust to variations in the number and distance of the negative samples (see Appendices D.1.2 and D.1.3).

We create the combined dataset $\mathcal{D}_c$ for the surrogate binary classification task by labelling $\mathcal{X}_c$ as 1 and $\mathcal{X}_c^{\mathrm{neg}}$ as 0:

$$\mathcal{D}_c = (\mathcal{X}_c \cup \mathcal{X}_c^{\mathrm{neg}}, \{1\}^{|\mathcal{X}_c|} \cup \{0\}^{|\mathcal{X}_c^{\mathrm{neg}}|}) \quad (1)$$

Figure 3: The class-specific energy function $E_c(\mathbf{x})$ from the surrogate binary task, where the blue region represents low energy (i.e., high data density). Placing the negative samples in a hypercube distant from the data results in an accurate energy function.

We then train a binary classifier $f_\theta^c(\cdot)$ on $\mathcal{D}_c$ and use it to construct the class-specific energy $E_c(\mathbf{x})$ for class $c$. To do this, we reinterpret the logits $\{f_\theta^c(\mathbf{x})[0], f_\theta^c(\mathbf{x})[1]\}$ of the trained binary classifier as components of an approximated joint distribution for the surrogate binary task:

$$p_c(\mathbf{x}, 0) = \frac{\exp(f_\theta^c(\mathbf{x})[0])}{Z}, \quad p_c(\mathbf{x}, 1) = \frac{\exp(f_\theta^c(\mathbf{x})[1])}{Z} \quad (Z \text{ is the normalisation constant)} \quad (2)$$

Next, we derive the approximated distribution $p_c(\mathbf{x})$ by marginalisation:

$$
\begin{aligned}
p_c(\mathbf{x}) &= p_c(\mathbf{x}, 0) + p_c(\mathbf{x}, 1) \\
&= \frac{\exp(f_\theta^c(\mathbf{x})[0]) + \exp(f_\theta^c(\mathbf{x})[1])}{Z} \\
&= \frac{\exp\left(\log\left(\exp(f_\theta^c(\mathbf{x})[0]) + \exp(f_\theta^c(\mathbf{x})[1])\right)\right)}{Z} \\
\rightarrow E_c(\mathbf{x}) &= -\log\left(\exp(f_\theta^c(\mathbf{x})[0]) + \exp(f_\theta^c(\mathbf{x})[1])\right) \quad \text{(TabEBM class-specific energy)} \quad (3)
\end{aligned}
$$

For the binary classifier $f_\theta^c(\cdot)$ in the surrogate binary classification, we use TabPFN [33], a pre-trained tabular in-context model. Note that TabPFN is intended for inference only, with no updates to its parameters (see Section 4 for more details about TabPFN). In this context, "training" the TabPFN classifier is analogous to the K-Nearest Neighbour algorithm, which simply performs inference based on a training dataset provided to the model. We apply TabPFN multiple times on separate datasets $\{\mathcal{D}_1, \mathcal{D}_2, \ldots, \mathcal{D}_C\}$ to obtain multiple classifiers $\{f_\theta^1, f_\theta^2, \ldots, f_\theta^C\}$. In Section 3.4, we explore why reinterpreting TabPFN's logits, trained on our surrogate binary tasks, can be useful for estimating an energy function. We emphasise that TabEBM is a general method, capable of using any gradient-based classifier that computes logits (using Equation (3)), and is not limited to TabPFN.

**Generating data with TabEBM** involves two steps. First, we sample a class $c$ from the empirical distribution $c \sim p(y)$. Then, we sample a data point $\mathbf{x}$ from the conditional distribution $\mathbf{x} \sim p(\mathbf{x}|y = c)$ approximated by the class-specific energy-based model $E_c(\mathbf{x})$, as outlined in Algorithm 1. We employ Stochastic Gradient Langevin Dynamics (SGLD) [84] to perform this sampling. SGLD is an efficient method for high-dimensional data, combining stochastic gradient descent (SGLD) with Langevin dynamics. The update rule for SGLD at each iteration is:

$$\mathbf{x}_{t+1} = \mathbf{x}_t - \frac{\eta}{2}\nabla E(\mathbf{x}_t) + \epsilon_t, \quad \epsilon_t \sim \mathcal{N}(0, \eta\mathbf{I}) \quad (4)$$

where a Gaussian noise term $\epsilon_t$ introduces randomness into the sampling process, enhancing the exploration of the distribution. In practice, the step size and the noise standard deviations are often chosen separately, resulting in a biased sampler that allows for faster training. Appendix D.2 further shows that TabEBM is stable to hyperparameters for the sampling process.

In our method, SGLD performs iterative augmentation. We start by sampling close to real data and iteratively adjust these synthetic data points, steering them towards regions of higher probability under the learned energy model. TabEBM enables sampling from any specified class distribution, including the original class distribution, which is crucial for data augmentation.

---

**Algorithm 1** TabEBM sampling from Class-Specific EBM $E_c(\mathbf{x})$

---

**Input:** Training data $\mathcal{X}_c$ for class $c$, step size $\alpha_{\text{step}}$, noise scale $\alpha_{\text{noise}}$, initial perturbation $\sigma_{\text{start}}$, number of steps $T$
**Output:** Set of synthetic samples for class $c$

    *Initialise a surrogate binary classification task and train the model*
1: Assign new labels to the samples $\mathcal{X}_c$ from class $c$, setting them to class 1
2: Generate a set of surrogate negative samples $\mathcal{X}_c^{\text{neg}}$ and assign them class 0 labels
3: Train a binary classifier $f_\theta^c$ on the dataset $\mathcal{D}_c = (\mathcal{X}_c \cup \mathcal{X}_c^{\text{neg}}, \{1\}^{|\mathcal{X}_c|} \cup \{0\}^{|\mathcal{X}_c^{\text{neg}}|})$
    *Synthesise samples using Stochastic Gradient Langevin Dynamics (SGLD)*
4: Initialise synthetic data points $\mathbf{x}_0^{\text{synth}}$ by sampling from $\mathcal{N}(\mathcal{X}_c, \sigma_{\text{start}}^2\mathbf{I})$
5: **for** each iteration $t = 0, 1, \ldots, T-1$ **do**
6:     $E_c(\mathbf{x}_t^{\text{synth}}) = -\log\left(\exp(f_\theta^c(\mathbf{x}_t^{\text{synth}})[0]) + \exp(f_\theta^c(\mathbf{x}_t^{\text{synth}})[1])\right)$
7:     $\mathbf{x}_{t+1}^{\text{synth}} = \mathbf{x}_t^{\text{synth}} - \alpha_{\text{step}}\nabla E_c(\mathbf{x}_t^{\text{synth}}) + \mathcal{N}(0, \alpha_{\text{noise}}^2\mathbf{I})$
8: **end for**
9: **return** $\mathbf{x}_T^{\text{synth}}$ as the generated synthetic data for class $c$

---

# 3 Experiments

We evaluate TabEBM by focusing on four research questions:

- **Data Augmentation Improvement (Q1, Section 3.1):** Can TabEBM generate synthetic data that improves the accuracy of downstream predictors via data augmentation?
- **Statistical Fidelity (Q2, Section 3.2):** Can TabEBM generate synthetic data with high statistical fidelity (i.e., with similar distributions to those of real data)?
- **Privacy Preservation (Q3, Section 3.3):** Can TabEBM generate synthetic data that finds a competitive trade-off between downstream performance and privacy preservation?
- **Understanding TabEBM's energy formulation (Q4, Section 3.4):** Why is TabEBM's class-specific energy effective, and how do the proposed surrogate tasks influence this?

**Datasets.** We utilise eight open-source tabular datasets from OpenML [7] across five domains: Medicine, Chemistry, Engineering, Language and Economics. As TabPFN utilises many small-size OpenML datasets in its meta-validation [33], it can lead to data leakage when evaluating TabEBM. Therefore, to provide fair comparisons, we select six additional leakage-free datasets from UCI [22]. These diverse datasets contain 7 to 77 features and 698 to 5500 samples across 2 to 26 classes. Five datasets contain both numerical and categorical features, while the remaining are numerical only. We further enlarge the evaluation scope by varying the degrees of data availability (i.e., $N_{real}$), leading up to 33 different test cases for the eight OpenML datasets. Appendix B.1 provides detailed descriptions.

**Benchmark generators.** We compare TabEBM against eight existing tabular data generation methods of eight different categories: (i) a standard interpolation method SMOTE [13]; (ii) a Variational Autoencoders (VAE) based method TVAE [85]; (iii) a Generative Adversarial Networks (GAN) method CTGAN [85]; (iv) a normalising flow model Neural Spine Flows (NFLOW) [24]; (v) a diffusion model TabDDPM [42]; (vi) a tree-based method Adversarial Random Forests (ARF) [83]; (vii) a Graph Neural Network (GNN) based method GOGGLE [47]; and (viii) a Prior-Data Fitted Networks (PFN) based method TabPFGen [48]. Furthermore, we also include a "Baseline" model, where no data augmentation is applied (i.e., only real data is used to train downstream predictors). In Appendix B.6, we detail the settings used for TabEBM and all other generators.

**Downstream predictors.** We select six representative downstream predictors, including three standard baselines: Logistic Regression (LR) [16], KNN [27] and MLP [28]; two tree-based methods: Random Forest (RF) [10] and XGBoost [14]; and a PFN method: TabPFN [33].

**General experimental setup.** For each dataset of $N$ samples, we first split it into stratified train and test sets. We create large test sets to reduce the likelihood that the model's performance is accidentally inflated due to a small, unrepresentative set of samples [69], and thus the test size is computed via $N_{test} = \min\left(\frac{N}{2}, 500\right)$. The full train set approximates the upper bound of the quality of synthetic data, and we call this set "oracle". We subsample the full train set to simulate different levels of data availability, thus the subset size $N_{real}$ varies over $\{20, 50, 100, 200, 500\}$. We split each subset into stratified training and validation sets with a ratio of 4:1. We provide detailed descriptions of data splitting in Appendix B.2 and preprocessing in Appendix B.3. We repeat the splitting ten times, summing up to 10 runs per subset size. The reported results are averaged by default over ten runs on the test sets. When aggregating results across datasets, we use the average distance to the minimum (ADTM) metric via affine renormalisation between the top-performing and worse-performing models [30, 54]. We provide the evaluation results averaged over six downstream predictors for a general conclusion, and the fine-grained numerical results for each predictor are in Appendix D.

**Data augmentation setup.** Given $N_{real}$ real samples, we first train generators on the real training data and then generate $N_{syn}$ synthetic samples. For training the downstream predictors, we expand the real training split by adding the synthetic samples. The real validation data is used for early stopping, and the real test set is used for evaluating the predictor's performance. The optimal $N_{syn}$ remains an open problem for tabular data [51, 71, 31]. Prior works [47, 48] mainly use synthetic sets with equivalent sizes to the real sets (i.e., $N_{real} = N_{syn}$). However, we observe that $N_{real} = N_{syn}$ can lead to highly unstable results, especially on small datasets that we investigate. Recent work has used different $N_{syn}$ for various generators, such as by applying post-processing [31, 71]. In this work, we want to provide a head-to-head comparison of the effect of data augmentation across subsampled datasets of varying sizes $N_{real} \in \{20, 50, 100, 200, 500\}$. Therefore, we perform data augmentation with a large synthetic set ($N_{syn} = 500$) across all splits, and the synthetic data has the same class distribution as the real training data. We provide an illustrative figure of the data splitting setup in Appendix B.2.

Table 1: **Classification accuracy** (%) aggregated over six downstream predictors, comparing data augmentation on eight real-world tabular datasets with varied real data availability. We report the mean ± std balanced accuracy and average accuracy rank across datasets. A higher rank implies higher accuracy. Note that "N/A" denotes that a specific generator was not applicable, and the rank is computed with the mean balanced accuracy of other methods. We **bold** the highest accuracy for each dataset of different sample size. Our method, TabEBM, consistently outperforms training on real data alone, and achieves the best overall performance against Baseline and benchmark generators.

| Datasets | | $N_{real}$ | Baseline (Real data) | SMOTE | TVAE | CTGAN | NFLOW | TabDDPM | ARF | GOGGLE | TabPFGen | **TabEBM** |
|---|---|---|---|---|---|---|---|---|---|---|---|---|
| *At most 10 classes* | protein | 20 | $28.14_{\pm6.83}$ | N/A | $21.18_{\pm1.48}$ | $22.00_{\pm3.43}$ | $21.30_{\pm2.84}$ | $22.12_{\pm5.30}$ | $24.82_{\pm2.88}$ | $22.40_{\pm9.28}$ | $33.25_{\pm5.01}$ | $\mathbf{33.84_{\pm4.92}}$ |
| | | 50 | $50.72_{\pm10.53}$ | $54.52_{\pm8.59}$ | $39.54_{\pm5.19}$ | $36.32_{\pm7.17}$ | $35.37_{\pm8.00}$ | $35.11_{\pm11.78}$ | $41.99_{\pm5.24}$ | $37.53_{\pm14.72}$ | $54.45_{\pm7.96}$ | $\mathbf{55.91_{\pm6.41}}$ |
| | | 100 | $67.83_{\pm11.72}$ | $73.25_{\pm7.48}$ | $59.28_{\pm7.20}$ | $57.64_{\pm9.95}$ | $52.57_{\pm9.55}$ | $56.37_{\pm9.64}$ | $57.01_{\pm8.56}$ | $51.69_{\pm16.68}$ | $71.53_{\pm9.87}$ | $\mathbf{73.31_{\pm6.77}}$ |
| | | 200 | $81.66_{\pm10.18}$ | $85.65_{\pm6.24}$ | $76.42_{\pm7.71}$ | $74.88_{\pm8.20}$ | $72.10_{\pm10.04}$ | $75.86_{\pm9.30}$ | $74.07_{\pm8.74}$ | $73.57_{\pm6.74}$ | $84.95_{\pm7.47}$ | $\mathbf{86.14_{\pm5.50}}$ |
| | | 500 | $93.49_{\pm5.28}$ | $94.73_{\pm3.67}$ | $92.24_{\pm3.73}$ | $91.48_{\pm4.43}$ | $90.44_{\pm5.54}$ | $90.62_{\pm5.63}$ | $91.79_{\pm4.53}$ | $91.31_{\pm5.20}$ | $94.87_{\pm3.70}$ | $\mathbf{95.18_{\pm3.10}}$ |
| | fourier | 20 | $28.30_{\pm12.09}$ | N/A | $21.32_{\pm4.06}$ | $18.19_{\pm3.90}$ | $17.30_{\pm3.03}$ | $15.35_{\pm3.26}$ | $21.75_{\pm2.76}$ | $16.70_{\pm2.91}$ | $36.72_{\pm7.30}$ | $\mathbf{37.13_{\pm6.01}}$ |
| | | 50 | $53.69_{\pm8.04}$ | $55.51_{\pm7.43}$ | $37.96_{\pm4.48}$ | $35.09_{\pm7.46}$ | $31.94_{\pm8.99}$ | $35.99_{\pm13.06}$ | $40.32_{\pm6.70}$ | $33.56_{\pm14.02}$ | $55.11_{\pm10.66}$ | $\mathbf{56.57_{\pm7.12}}$ |
| | | 100 | $63.70_{\pm6.76}$ | $64.10_{\pm6.89}$ | $50.46_{\pm8.61}$ | $49.26_{\pm9.15}$ | $44.58_{\pm8.40}$ | $52.79_{\pm10.04}$ | $51.13_{\pm6.35}$ | $41.93_{\pm15.60}$ | $63.86_{\pm7.76}$ | $\mathbf{65.21_{\pm6.42}}$ |
| | | 200 | $70.99_{\pm4.88}$ | $71.43_{\pm4.47}$ | $62.17_{\pm7.29}$ | $62.92_{\pm7.87}$ | $59.15_{\pm8.33}$ | $68.05_{\pm6.91}$ | $62.53_{\pm6.97}$ | $56.44_{\pm10.13}$ | $71.81_{\pm5.35}$ | $\mathbf{72.36_{\pm3.77}}$ |
| | | 500 | $77.72_{\pm2.36}$ | $77.51_{\pm2.60}$ | $73.29_{\pm4.97}$ | $74.61_{\pm4.89}$ | $71.74_{\pm6.54}$ | $77.04_{\pm3.64}$ | $74.31_{\pm4.40}$ | $70.61_{\pm6.01}$ | $77.15_{\pm2.57}$ | $\mathbf{78.20_{\pm2.87}}$ |
| | biodeg | 20 | $66.20_{\pm4.26}$ | $68.59_{\pm1.17}$ | $66.77_{\pm2.64}$ | $58.03_{\pm2.47}$ | $59.37_{\pm1.74}$ | $52.72_{\pm2.38}$ | $61.17_{\pm2.00}$ | $61.39_{\pm6.39}$ | $68.99_{\pm2.54}$ | $\mathbf{69.79_{\pm2.15}}$ |
| | | 50 | $72.66_{\pm3.98}$ | $72.80_{\pm3.08}$ | $71.31_{\pm2.71}$ | $67.99_{\pm3.63}$ | $62.40_{\pm4.28}$ | $60.72_{\pm10.11}$ | $71.62_{\pm2.43}$ | $66.68_{\pm6.00}$ | $73.29_{\pm3.53}$ | $\mathbf{73.78_{\pm3.42}}$ |
| | | 100 | $\mathbf{76.69_{\pm2.70}}$ | $76.31_{\pm2.42}$ | $75.38_{\pm2.06}$ | $74.82_{\pm2.89}$ | $69.50_{\pm4.59}$ | $68.28_{\pm9.54}$ | $74.42_{\pm2.38}$ | $71.68_{\pm3.72}$ | $76.22_{\pm2.31}$ | $76.45_{\pm3.08}$ |
| | | 200 | $80.01_{\pm2.66}$ | $79.67_{\pm2.56}$ | $78.11_{\pm2.68}$ | $78.19_{\pm1.78}$ | $75.05_{\pm4.68}$ | $74.43_{\pm8.09}$ | $77.97_{\pm2.32}$ | $77.13_{\pm3.01}$ | $79.76_{\pm2.63}$ | $\mathbf{80.11_{\pm2.33}}$ |
| | | 500 | $82.63_{\pm2.43}$ | $\mathbf{82.85_{\pm1.93}}$ | $82.13_{\pm1.94}$ | $82.42_{\pm1.58}$ | $81.11_{\pm3.23}$ | $79.19_{\pm6.60}$ | $81.92_{\pm2.28}$ | $81.24_{\pm2.30}$ | $82.35_{\pm2.21}$ | $82.29_{\pm2.15}$ |
| | steel | 20 | $57.51_{\pm4.58}$ | $58.32_{\pm3.27}$ | $57.99_{\pm3.06}$ | $56.61_{\pm1.70}$ | $53.89_{\pm1.73}$ | $55.74_{\pm6.02}$ | $54.24_{\pm2.08}$ | $53.04_{\pm2.36}$ | $63.21_{\pm5.86}$ | $\mathbf{63.27_{\pm5.45}}$ |
| | | 50 | $75.06_{\pm10.43}$ | $65.63_{\pm4.00}$ | $64.18_{\pm3.95}$ | $63.70_{\pm6.10}$ | $58.90_{\pm6.39}$ | $65.85_{\pm14.84}$ | $61.72_{\pm3.39}$ | $56.72_{\pm3.47}$ | $78.67_{\pm11.79}$ | $\mathbf{80.50_{\pm8.67}}$ |
| | | 100 | $86.87_{\pm12.49}$ | $74.61_{\pm5.99}$ | $70.12_{\pm5.76}$ | $69.89_{\pm5.58}$ | $65.67_{\pm9.10}$ | $76.01_{\pm17.54}$ | $67.33_{\pm5.15}$ | $60.56_{\pm5.37}$ | $90.58_{\pm9.50}$ | $\mathbf{92.71_{\pm7.57}}$ |
| | | 200 | $92.90_{\pm9.14}$ | $81.97_{\pm4.12}$ | $78.73_{\pm5.06}$ | $78.36_{\pm6.98}$ | $75.90_{\pm9.57}$ | $85.45_{\pm15.03}$ | $78.65_{\pm6.70}$ | $68.20_{\pm5.30}$ | $95.56_{\pm5.85}$ | $\mathbf{96.29_{\pm4.64}}$ |
| | | 500 | $97.52_{\pm3.76}$ | $92.44_{\pm4.46}$ | $92.47_{\pm3.66}$ | $92.42_{\pm4.76}$ | $88.20_{\pm8.36}$ | $96.34_{\pm4.67}$ | $90.41_{\pm5.35}$ | $84.23_{\pm10.90}$ | $98.14_{\pm2.67}$ | $\mathbf{98.47_{\pm2.15}}$ |
| | stock | 20 | $78.75_{\pm4.39}$ | $82.18_{\pm2.15}$ | $74.11_{\pm3.71}$ | $64.25_{\pm6.29}$ | $72.64_{\pm2.01}$ | $78.61_{\pm3.57}$ | $69.54_{\pm1.65}$ | $76.35_{\pm5.08}$ | $82.42_{\pm2.17}$ | $\mathbf{83.49_{\pm1.60}}$ |
| | | 50 | $86.10_{\pm3.62}$ | $87.82_{\pm3.41}$ | $82.81_{\pm3.51}$ | $79.63_{\pm3.93}$ | $80.14_{\pm3.90}$ | $86.72_{\pm4.29}$ | $82.48_{\pm2.95}$ | $83.36_{\pm5.23}$ | $88.14_{\pm3.01}$ | $\mathbf{88.44_{\pm3.14}}$ |
| | | 100 | $89.07_{\pm3.71}$ | $89.99_{\pm3.22}$ | $87.55_{\pm4.25}$ | $86.44_{\pm4.40}$ | $84.64_{\pm4.79}$ | $89.40_{\pm4.26}$ | $87.32_{\pm4.42}$ | $87.44_{\pm5.46}$ | $90.27_{\pm3.33}$ | $\mathbf{90.36_{\pm3.51}}$ |
| | | 200 | $90.85_{\pm4.39}$ | $\mathbf{91.75_{\pm3.73}}$ | $90.12_{\pm5.44}$ | $89.44_{\pm4.94}$ | $88.47_{\pm6.06}$ | $90.76_{\pm5.27}$ | $89.59_{\pm5.37}$ | $89.62_{\pm6.29}$ | $91.56_{\pm3.91}$ | $91.71_{\pm3.77}$ |
| *More than 10 classes* | energy | 50 | $17.77_{\pm6.15}$ | N/A | $12.30_{\pm2.59}$ | $12.11_{\pm3.16}$ | $10.14_{\pm2.87}$ | $10.55_{\pm2.44}$ | $11.99_{\pm2.27}$ | $15.46_{\pm3.54}$ | N/A | $\mathbf{23.98_{\pm2.73}}$ |
| | | 100 | $25.94_{\pm4.86}$ | N/A | $17.78_{\pm4.73}$ | $18.60_{\pm6.09}$ | $18.56_{\pm6.39}$ | $18.84_{\pm6.23}$ | $19.91_{\pm5.21}$ | $17.65_{\pm5.88}$ | N/A | $\mathbf{31.24_{\pm5.53}}$ |
| | | 200 | $35.99_{\pm8.92}$ | N/A | $27.65_{\pm11.12}$ | $27.77_{\pm10.55}$ | $28.37_{\pm10.82}$ | $29.50_{\pm10.33}$ | $29.57_{\pm9.18}$ | $28.95_{\pm10.40}$ | N/A | $\mathbf{41.28_{\pm7.66}}$ |
| | collins | 100 | $11.44_{\pm2.77}$ | N/A | $8.38_{\pm1.52}$ | $8.11_{\pm1.00}$ | $7.93_{\pm1.40}$ | $12.67_{\pm2.16}$ | $7.53_{\pm1.10}$ | $9.21_{\pm2.35}$ | N/A | $\mathbf{13.07_{\pm2.51}}$ |
| | | 200 | $15.74_{\pm3.73}$ | $\mathbf{17.45_{\pm3.46}}$ | $12.08_{\pm3.03}$ | $11.37_{\pm1.20}$ | $10.74_{\pm1.72}$ | $15.39_{\pm3.37}$ | $10.71_{\pm1.37}$ | $14.30_{\pm3.42}$ | N/A | $17.03_{\pm3.20}$ |
| | texture | 50 | $72.40_{\pm13.07}$ | $76.40_{\pm10.50}$ | $55.32_{\pm6.20}$ | $54.80_{\pm12.97}$ | $55.39_{\pm10.65}$ | $62.27_{\pm8.01}$ | $55.65_{\pm10.58}$ | $62.94_{\pm12.06}$ | N/A | $\mathbf{78.90_{\pm7.96}}$ |
| | | 100 | $82.42_{\pm10.38}$ | $84.35_{\pm9.67}$ | $66.00_{\pm7.21}$ | $69.49_{\pm10.93}$ | $71.78_{\pm9.06}$ | $76.25_{\pm7.40}$ | $70.93_{\pm9.71}$ | $76.34_{\pm9.55}$ | N/A | $\mathbf{86.01_{\pm7.36}}$ |
| | | 200 | $87.54_{\pm7.62}$ | $89.29_{\pm6.20}$ | $78.37_{\pm6.03}$ | $82.44_{\pm7.15}$ | $81.94_{\pm6.30}$ | $84.67_{\pm4.79}$ | $83.29_{\pm6.32}$ | $82.53_{\pm7.99}$ | N/A | $\mathbf{89.77_{\pm5.77}}$ |
| | | 500 | $92.96_{\pm4.07}$ | $93.69_{\pm3.83}$ | $90.09_{\pm3.56}$ | $91.48_{\pm3.50}$ | $90.50_{\pm2.71}$ | $91.53_{\pm3.29}$ | $91.76_{\pm3.98}$ | $91.24_{\pm3.56}$ | N/A | $\mathbf{93.76_{\pm3.64}}$ |
| **Average rank** | | | $3.30_{\pm1.02}$ | $3.03_{\pm1.25}$ | $6.79_{\pm1.80}$ | $7.48_{\pm1.50}$ | $8.94_{\pm0.70}$ | $6.39_{\pm2.41}$ | $6.94_{\pm1.50}$ | $7.76_{\pm2.03}$ | $3.15_{\pm1.27}$ | $\mathbf{1.21_{\pm0.74}}$ |

## 3.1 Data Augmentation Improvement (Q1)

We evaluate the effect of using synthetic data for data augmentation by comparing the *balanced accuracy* of downstream predictors before and after augmentation. Typically, higher classification accuracy (i.e., $ACC_{Generator}$) and accuracy improvements (i.e., $ACC_{Generator} - ACC_{Baseline} > 0$) demonstrate the effectiveness of the synthetic data for data augmentation.

**TabEBM effectively improves downstream performance across sample sizes, especially for very low-sample-size regimes.** Table 1 and Figure 4 (Left) show that TabEBM exhibits competitive performance in data augmentation, generally achieving the highest downstream accuracy and average rank across most datasets and sample sizes. Notably, TabEBM is the only generator that consistently improves performance across sample sizes. A key observation is that most modern benchmark generators underperform even the Baseline, indicating poor approximated distributions in the low-sample-size regime. Moreover, TabEBM achieves the largest overall performance improvement on six leakage-free UCI datasets, further supporting its effectiveness (see Appendix D.5.2 for details).

Furthermore, TabEBM is the most widely applicable method among the top three competitive generators on the considered datasets: (i) SMOTE requires at least two samples per class for interpolation, and thus it is not applicable for some datasets, such as the "protein" dataset ($N_{real} = 20$); (ii) TabPFGen cannot scale up to more than ten classes, such as the "collins" dataset. In addition, TabEBM can stabilise downstream performance, especially when real data is very scarce ($N_{real} = 20$): TabEBM leads to smaller standard deviations than Baseline on seven out of eight datasets.

**TabEBM effectively improves downstream performance across any number of classes, especially for more than ten classes.** Figure 4 (Right) shows that TabEBM consistently outperforms the

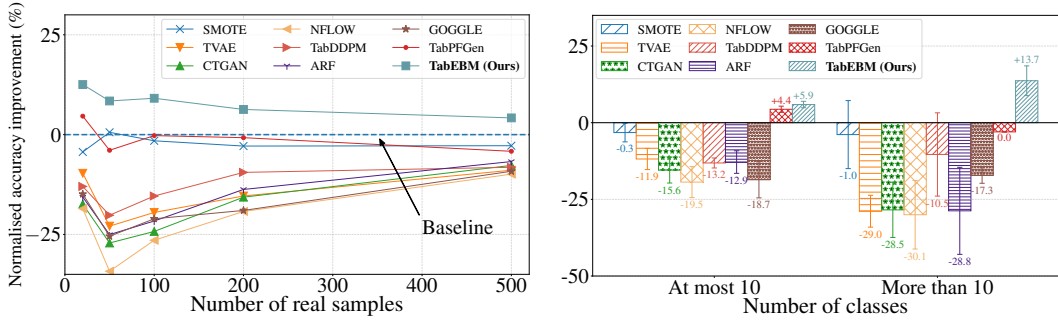

Figure 4: Mean normalised balanced accuracy improvement (%) across different sample sizes (**Left**) and across datasets with varying numbers of classes (**Right**). Because TabPFGen is not applicable for datasets with more than ten classes, we plot short bars at zeros for visual clearance. Positive values indicate that the generator improves downstream classification performance. TabEBM generally outperforms benchmark generators across varying sample sizes and number of classes.

Baseline with notable improvements, particularly in datasets with more than ten classes. In contrast, an increased number of classes tends to cause a performance degradation in the benchmark generators.

**TabEBM is robust on imbalanced datasets.** For the three binary OpenML datasets (i.e., "biodeg", "steel" and "stock"), we adjust the class distribution in the training data to vary the class imbalance, while keeping the test data fixed. Figure 5 shows that TabEBM consistently outperforms Baseline, while the other generators exhibit performance degradation as data imbalance increases.

**TabEBM is computationally efficient.** Figure 6 shows the trade-off between accuracy and the time needed for generating stratified synthetic data (for data augmentation). We measure the total duration of (i) training the model and (ii) generating 500 synthetic samples. The results show that TabEBM is practical, as it achieves higher downstream accuracy with lower time costs.

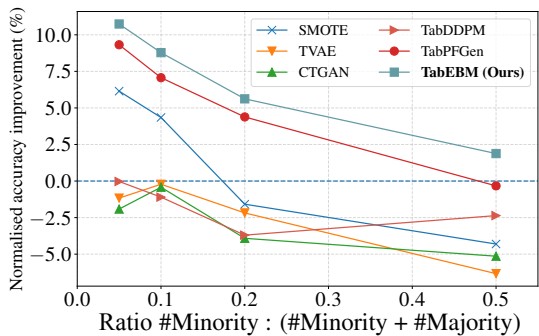

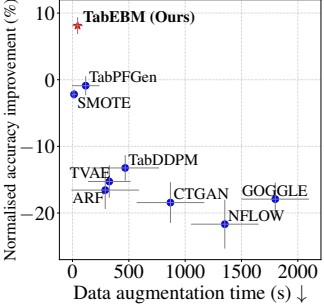

Figure 5: **Mean normalised balanced accuracy improvement (%) on imbalanced datasets.** TabEBM consistently outperforms the Baseline and other generators across different levels of data imbalance.

Figure 6: **Median data augmentation time vs. mean normalised balanced accuracy.** TabEBM achieves higher downstream accuracy while typically operating 3-30 times faster than most other methods.

## 3.2 Statistical Fidelity (Q2)

We evaluate the fidelity of synthetic data by measuring the similarity of synthetic data to real *train* data and to real *test* data (Figure 7). We evaluate this similarity via (i) *average inverse of the Kullback–Leibler Divergence* (inverse KL) [17], (ii) p-value of *Kolmogorov–Smirnov test* (KS test) [39] and (iii) p-value of *Chi-squared test* ($\chi^2$ test) [55]. For full numerical results, including $\chi^2$ test, see Appendix D.6. For all three metrics, a bigger value denotes that synthetic data is more likely to have the same distribution as real data.

In Figure 7 (a1&a2), TabEBM consistently exhibits the highest accuracy and distribution similarity between real train data and synthetic data, indicating that TabEBM learns the distributions of real train data better than benchmark generators. In Appendix D.6, we further show that TabEBM remains the most competitive method in similarity between real test data and synthetic data. This indicates that TabEBM can extrapolate beyond real train data and thus generate synthetic data from a more

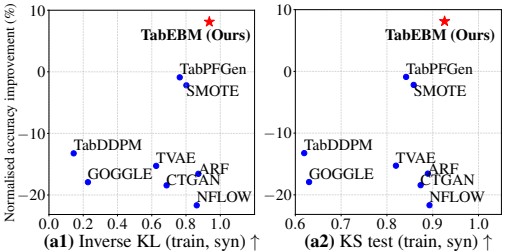 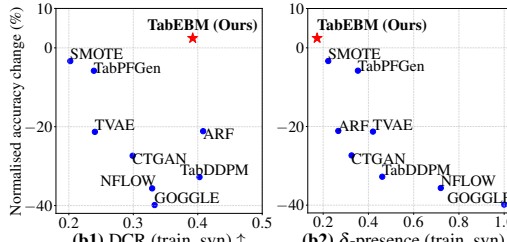

**(a1)** Inverse KL (train, syn) ↑ | **(a2)** KS test (train, syn) ↑ | **(b1)** DCR (train, syn) ↑ | **(b2)** δ-presence (train, syn) ↓

Figure 7: **(a1&a2):** Median inverse KL and KS test vs. mean normalised balanced accuracy improvement (%) between real train data and synthetic data. **(b1&b2):** Median DCR and δ-presence vs. mean normalised balanced accuracy change (%) between real train data and synthetic data. Note that "accuracy improvement" is for data augmentation, and "accuracy change" is for data sharing. Complete results with standard deviations are in Appendix D.4. TabEBM generates high-fidelity synthetic data that can also be used for privacy preservation.

general distribution that aligns with both train and test data. This extrapolation ability also explains why TabEBM can outperform Baseline via data augmentation (Section 3.1).

## 3.3 Privacy Preservation (Q3)

More broadly, data privacy is a critical concern for organisations and governments handling sensitive data [75]. Privacy-preserving synthetic data allows researchers and practitioners to bypass ethical and logistical issues while enabling model training and testing [38]. We further explore the use of TabEBM-generated data for data sharing, where only synthetic data is accessible for downstream users [75, 89, 86, 23, 42]. In this case, downstream models are trained exclusively on synthetic data.

Specifically, we evaluate synthetic data via three metrics: (i) *balanced accuracy* of downstream predictors trained with only synthetic data (i.e., train-on-synthetic, test-on-real [85, 42, 87]); (ii) median Distance to Closest Record (DCR) [88], where a greater DCR denotes synthetic data is less likely to be copied from real data; and (iii) δ-presense [62], where a smaller value denotes a lower re-identification risk for real data from synthetic data. Full numerical results are in Appendix D.7.

Figure 7 (b1&b2) shows that TabEBM consistently finds a better trade-off between accuracy and privacy preservation. Notably, the "train-on-synthetic, test-on-real" scenario poses a greater challenge for generators in achieving high accuracy because real data is inaccessible for model training and data augmentation. Despite this difficulty, TabEBM is the only generator that surpasses the overall performance of training on real data (i.e., Baseline). The relatively high DCR for TabEBM indicates that it can extrapolate beyond real train data, aligning with the finding that TabEBM's synthetic data is statistically similar to real test data (Section 3.2). These results further suggest that TabEBM learns the general distribution of real data, and can generate high-quality synthetic data suitable for various purposes, including privacy preservation.

## 3.4 Why is TabEBM effective for estimating Energy-Based Models? (Q4)

Having established that TabEBM excels in data augmentation, we explore why classifier logits can be useful when reinterpreted as a class-conditional energy function. Figure 8 shows the logit distribution of TabPFN trained on surrogate binary tasks and the corresponding energy function of TabEBM (with TabPFN as the binary classifier) as the Euclidean distance from the real data increases.

We found it essential to place the negative samples far from the real data, since TabPFN, which is pre-trained to approximate Bayesian inference [33], has its confi-

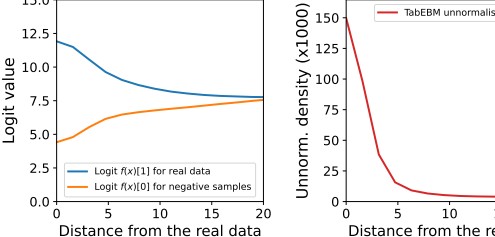

Figure 8: **(Left)** Logit distribution of TabPFN trained on our surrogate binary tasks at increasing distances from the real data (on "steel"). **(Right)** The corresponding unnormalised density approximated by TabEBM. TabEBM assigns higher density closer to the real data.

dence influenced by the distance from the training data [53]. Figure 8 (left) shows that TabPFN outputs high logit values near the real data. As the distance from the real data increases, the logit $f(\mathbf{x})[1]$ decreases smoothly until the two logits become similar, making the classifier uncertain (because the class probabilities become equal). Figure 8 (right) shows that TabEBM's inferred density drops significantly as the maximum logit decreases, because $p_c(\mathbf{x}) \propto (\exp(f(\mathbf{x})[0]) + \exp(f(\mathbf{x})[1]))$ from Equation (3). Since SGLD sampling performs gradient ascent on the density, the TabEBM-generated samples will be close to the real data. These findings are consistent across datasets (see Appendix D.3), where TabPFN's logits remain positive, with similar ranges and a relatively constant sum as distance increases, warranting further investigation. Overall, TabPFN's distance-based uncertainty is useful for inferring accurate energy functions within our TabEBM framework. Since TabEBM can be paired with any other gradient-based classifier that produces logits, we leave these extensions for future work.

## 4    Discussion & Related Work

Section 3 showed that TabEBM efficiently generates high-fidelity data that can effectively improve the downstream performance via data augmentation. In Table 2, we further provide a summary of tabular data generative models analysed from three important perspectives: (i) *Training:* the type of distribution that the generators learn (crucial for preserving the original training label distribution), and the training costs associated with learning; (ii) *Generation:* do the generators employ class-specific models (reflecting their capability to capture unique features essential for label-invariant generation), and do models support stratified generation (crucial for effective data augmentation); (iii) *Practicability:* the scalability of the generators with respect to the number of classes (a common requirement in real-world multi-class tasks), and consistent downstream performance improvement across different class sizes.

**Generative Models for Tabular Data.** The common paradigm for tabular data generation is to adapt Generative Adversarial Networks (GANs) and Variational Autoencoders (VAEs) [85, 63]. For instance, TableGAN employs a convolutional neural network to optimise the label quality [63], and TVAE is introduced in [85] as a variant of VAE for tabular data. However, these methods learn the joint distribution and thus cannot preserve the stratification of the original data (Appendix C). CTGAN [85] refines the generation to be class-conditional. The recent ARF [83] is an adversarial variant of random forest for density estimation, and GOGGLE [47] enhances VAE by learning relational structure with a Graph Neural Network (GNN). Some recent work focuses on generation with denoising diffusion models [42, 87, 40, 44]. For instance, TabDDPM [42] demonstrates that diffusion models can approximate typical distributions of tabular data. Although these class-conditional models can preserve the label distribution, they struggle to outperform Baseline and standard SMOTE in data augmentation [71, 48].

We attribute the performance degradation in current class-conditional models to their reliance on a single shared model to approximate all class-conditional densities. For instance, another promising generative approach uses pre-trained models like Prior-Data Fitted Networks (PFNs), and the recent TabPFGen [48] adapts such models into one shared class-conditional generator. However, TabPFGen's shared generator can lead to inaccurate density estimates, particularly in high-noise and class-imbalance situations (see examples in Appendix C). As noise increases, TabPFGen's inferred densities fluctuate significantly and diverge from the true data distributions. In contrast, TabEBM uses class-specific EBMs to model each class's marginal distributions, and the results in Appendix C reveal that our design choice reduces the impact of noise and data imbalance. TabEBM focuses on approximating and generating for one class at a time, remaining unaffected by noise from other classes. Overall, our results demonstrate that TabEBM consistently improves performance across different datasets and sample sizes, outperforming TabPFGen. Moreover, TabPFGen is limited in usability (e.g., it supports only up to ten classes), while TabEBM scales to any number of classes.

In a broader context, some recent work attempts to adapt Large Language Models (LLMs) for tabular data generation [25, 71, 9]. However, data contamination is an inherent issue with such LLM-based models [19, 36, 18, 49]. As the pre-training data is not typically open-source, these models can have unfair advantages in downstream tasks (i.e., the full real dataset, including the real test data, may have been used for pre-training). Therefore, in this paper, we focus on models without support from LLMs, thus avoiding potential biases from data contamination.

**Data Augmentation (DA) for Tabular Data.** DA is an omnipresent technique in computer vision and natural language processing [82, 73, 72, 60, 26, 2]. However, DA for tabular data remains

Table 2: **Comparison of the properties between TabEBM and prior tabular generative methods.** TabEBM has novel design rationales of training-free class-specific models, and TabEBM is highly practicable with wide applicability and consistent accuracy improvement.

| Methods | Category | Training | | Generation | | Practicability | | |
|---|---|---|---|---|---|---|---|---|
| | | Learned distribution | Training-free | Class-specific models | Stratified generation | Unlimited classes | ACC improve ($\leq$ 10 classes) | ACC improve ($>$ 10 classes) |
| SMOTE [13] | Interpolation | N/A | ✔ | N/A | ✔ | ✔ | ✗ | ✗ |
| TVAE [85] | VAE | $p(\mathbf{x}, y)$ | ✗ | ✗ | ✗ | ✔ | ✗ | ✗ |
| CTGAN [85] | GAN | $p(\mathbf{x} \mid y)$ | ✗ | ✗ | ✔ | ✔ | ✗ | ✗ |
| NFLOW [24] | Normal. Flows | $p(\mathbf{x}, y)$ | ✗ | ✗ | ✗ | ✔ | ✗ | ✗ |
| TabDDPM [42] | Diffusion | $p(\mathbf{x} \mid y)$ | ✗ | ✗ | ✔ | ✔ | ✗ | ✗ |
| ARF [83] | Random Forest | $p(\mathbf{x}, y)$ | ✗ | ✗ | ✗ | ✔ | ✗ | ✗ |
| GOGGLE [47] | GNN | $p(\mathbf{x} \mid y)$ | ✗ | ✗ | ✔ | ✔ | ✗ | ✗ |
| TabPFGen [48] | PFN | $p(\mathbf{x} \mid y)$ | ✔ | ✗ | ✔ | ✗ | ✔ | ✗ |
| **TabEBM (Ours)** | PFN | $p(\mathbf{x} \mid y)$ | ✔ | ✔ | ✔ | ✔ | ✔ | ✔ |

underexplored, and existing methods often perform poorly in real-world tasks, sometimes even reduce performance [51]. Recent studies show that using the same transformations across all classes leads to varied performance impacts [3, 41], indicating that data augmentation effects are class-specific and suggesting that different classes may require distinct augmentations. Given the lack of symmetries in tabular data, we believe this class-dependent effect is even more pronounced. Therefore, we propose TabEBM as a class-specific generative model to produce tailored augmentations for each class.

**Prior-fitted Networks (PFNs) for Tabular Data.** Recent work proposes to approximate the posterior predictive distribution with transformers [59, 33, 61, 79, 20]. PFNs can be adapted for various purposes by pre-training the transformer with corresponding "prior data", and then it can make in-context predictions with unseen downstream data. For instance, TabPFN is a variant that is pre-trained on a prior designed for tabular data [33]. We note that prior data is different to synthetic data in this paper. Specifically, prior data refers to manually crafted fake data (e.g., $y = 2x$) with no real-world semantics. In contrast, synthetic data from generators is expected to have the same semantics as real data. Inspired by TabPFN's success in small-size classification tasks, TabEBM converts TabPFN into multiple EBMs that learn the marginal distribution for each class. The training-free nature of TabPFN enables TabEBM to generate high-quality tabular data without introducing extra training costs. Additionally, our class-specific design lets TabEBM surpass TabPFN's limits and scale to more than ten classes.

**Limitations and Future Work.** TabEBM is a general method that relies on an underlying binary classifier, and as such, its strengths and weaknesses are directly tied to this classifier. We used TabPFN because it is a well-established open-source pre-trained model for tabular data. Therefore, TabEBM inherits some of TabPFN's limitations, particularly in scaling to a larger number of features. TabEBM can handle datasets with over 1000 samples, overcoming TabPFN's limitation, as it processes one class at a time. In Appendix D.5.3, we show that TabEBM outperforms other generators on larger datasets, though the performance gains decrease as the sample size increases. Although we implement TabEBM with TabPFN in this paper, we stress that TabEBM is compatible with any classifier that can be adapted into EBMs, as described in Section 2. As foundational models for tabular data evolve [81], new models capable of handling more features and samples are expected. Integrating them into TabEBM will enhance its ability to manage high-dimensional datasets, increasing its versatility and utility. Finally, note that, generators that are limited in modelling multivariate distributions may still perform well on univariate fidelity metrics, which is a standard approach to evaluating such models. However, evaluating their ability to learn more complex, high-order, relationships between features remains an open research question [78], which we leave for future work.

## 5   Conclusion

We introduced TabEBM, the first tabular data augmentation method that creates class-specific EBM generators, learning the marginal distribution for each class separately. We also provide the first comprehensive analysis of tabular data augmentation across various dataset sizes. Our results demonstrate that TabEBM improves downstream performance through data augmentation on real-world datasets, outperforming other benchmark generators. The statistical evaluation confirms that TabEBM generates high-fidelity synthetic data, particularly for small datasets. We release our method as an open-source library, allowing users to generate data immediately without additional training.

## Acknowledgments and Disclosure of Funding

The authors would like to thank Francisco Vargas, Randall Balestriero, and Otilia Stretcu for their insightful discussions and valuable input early in the project. NS and MJ acknowledge the support of the U.S. Army Medical Research and Development Command of the Department of Defense; through the FY22 Breast Cancer Research Program of the Congressionally Directed Medical Research Programs, Clinical Research Extension Award GRANT13769713. Opinions, interpretations, conclusions, and recommendations are those of the authors and are not necessarily endorsed by the Department of Defense.

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

# Appendix

## TabEBM: A Tabular Data Augmentation Method with Distinct Class-Specific Energy-Based Models

## Table of Contents

# A    Broader Impact Statement

This paper introduces a novel data augmentation approach, TabEBM, that aims to advance the field of machine learning by addressing challenges in the low-sample-size regime. Furthermore, TabEBM offers an elegant solution to learning the unique features in generating samples for each class, leading to high-fidelity synthetic data that can effectively improve downstream performance. These characteristics can be particularly useful in data-scarce domains like healthcare (e.g., pre-clinical drug evaluation in early-stage clinical trials [6, 58]). Moreover, we also demonstrate that TabEBM is readily applicable for privacy-preserving data sharing in high-stake tasks [89, 77].

TabEBM's impact further extends to enabling broader machine learning applications in data-scarce domains, for instance, facilitating data analysis in clinical scenarios with limited access to data collection techniques. Improving the performance of machine learning models in such applications can further foster the uptake of more sophisticated ML approaches and, ultimately, help improve the quality of healthcare [1, 15, 57]. TabEBM can further facilitate research and enhance machine learning accessibility in various communities across societal and scientific domains. To this end, our work has only been evaluated in a strictly research setting. Further applications of our work in scenarios with sensitive data bear some risks. As TabEBM is a generative model, training models with the resulting generated samples can bias the downstream model. Therefore, this risk, together with other data privacy risks during downstream deployment, must be carefully managed.

# B    Reproducibility

## B.1    Datasets

All eight datasets are publicly available on OpenML [7], and their details are listed in Table 3. To ensure consistent stratified data-splitting across all datasets, we remove classes with fewer than 10 samples. For example, the original "energy" dataset contains 14 classes with fewer than 10 samples, which could result in a validation set lacking samples from these classes, leading to unstratified data splitting.

Table 3: Details of the eight real-world tabular datasets.

| Dataset | OpenML ID | Not evaluated in TabPFN [33] | # Samples ($N$) | # Features ($D$) | # Classes | $N/D$ | # Samples per class (Min) | # Samples per class (Max) |
|---|---|---|---|---|---|---|---|---|
| At most 10 classes | | | | | | | | |
| protein | 40966 | ✗ | 1,080 | 77 | 8 | 14.03 | 105 | 150 |
| fourier | 14 | ✗ | 2,000 | 76 | 10 | 26.32 | 200 | 200 |
| biodeg | 1494 | ✗ | 1,055 | 41 | 2 | 25.73 | 356 | 699 |
| steel | 1504 | ✗ | 1,941 | 33 | 2 | 58.82 | 673 | 1,268 |
| stock | 841 | ✗ | 950 | 9 | 2 | 105.56 | 462 | 488 |
| More than 10 classes | | | | | | | | |
| energy | 1472 | ✗ | 698 | 9 | 23 | 77.56 | 10 | 74 |
| collins | 40971 | ✔ | 970 | 19 | 26 | 51.05 | 17 | 80 |
| texture | 40499 | ✔ | 5,500 | 40 | 11 | 137.5 | 500 | 500 |

## B.2    Data Splitting

Figure 9 shows the data splitting setup used across all datasets. Note that data sharing (Section 3.3) shares the same data splitting as data augmentation, except that the "Training set" and "Validation set" containing real data are no longer used for training the downstream predictors.

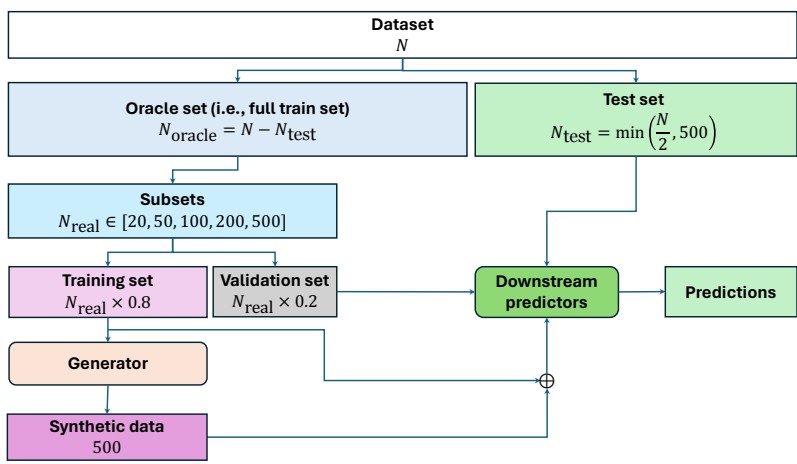

Figure 9: Data splitting strategies for data augmentation for all datasets.

## B.3 Data Preprocessing

Following the procedures presented in prior work [54, 30], we perform preprocessing in two steps. We first compute the required statistics with training data and then transform it. Firstly, we impute the missing values with the mean value for numerical features and the most mode value for categorical features. Secondly, we convert the categorical features into numerical features equal to Leave-one-out Target Statistic [66, 56]. Next, we perform Z-score normalisation for each feature. Specifically, we compute each feature's mean and standard deviation in the training data and then transform the training samples to have a mean of zero and a variance of one for each feature. Finally, we apply the same transformation to the validation and test data before conducting evaluations.

## B.4 Software and Computing Resources

**Software implementation.** *(i) For generators:* We implemented TabEBM using PyTorch 1.13 [64], an open-source deep learning library with a BSD licence. We implemented SMOTE with Imbalanced-learn [45], an open-source Python library for imbalanced datasets with an MIT licence. For other benchmark generators, we used their open-source implementations in Synthcity [67], a library for generating and evaluating synthetic tabular data with an Apache-2.0 license. *(ii) For downstream predictors:* We implemented TabPFN with its open-source implementation (https://github.com/automl/TabPFN). We implemented the other five downstream predictors (i.e., Logistic Regression, KNN, MLP, Random Forest and XGBoost) with their open-source implementation in scikit-learn [65], an open-source Python library under the 3-Clause BSD license. *(iii) For result analysis and visualisation:* All numerical plots and graphics have been generated using Matplotlib 3.7 [34], a Python-based plotting library with a BSD licence. The model architecture was generated using draw.io (https://github.com/jgraph/drawio), a free drawing software under Apache License 2.0.

We ensure the consistency and reproducibility of experimental results by implementing a uniform pipeline using PyTorch Lightning, an open-source library under an Apache-2.0 licence. We further fixed the random seeds for data loading and evaluation throughout the training and evaluation process. This ensured that TabEBM and all benchmark models were trained and evaluated on the same set of samples. The experimental environment settings, including library dependencies, are specified in the open-source library for reference and reproduction purposes.

**Computing Resources.** We trained 140,000 models for evaluations (including over 35,000 of generators and over 10,500 for downstream predictors). All our experiments are run on a single machine from an internal cluster with a GPU Nvidia Quadro RTX 8000 with 48GB memory and an Intel(R) Xeon(R) Gold 5218 CPU with 16 cores (at 2.30GHz). The operating system was Ubuntu 20.4.4 LTS.

### B.5 TabEBM open-source library

We implemented TabEBM as an extensible library, and the code is available on `https://github.com/andreimargeloiu/TabEBM`. For practitioners, it offers an easy-to-use, domain-agnostic tool that requires no training, making it particularly suitable for data augmentation, especially in small datasets. For researchers, the library includes the complete implementation of TabEBM, facilitating future extensions and investigations into class-specific energy-based models.

The library has two core functionalities:

1. **Generate synthetic data**: The library can generate data for augmentation.

   ```
   from tabebm.TabEBM import TabEBM

   tabebm = TabEBM()
   augmented_data = tabebm.generate(X_train, y_train, num_samples=100)
   % augmented_data[class_id] = numpy.ndarray of generated data
   %                            for a specific ''class_id''
   ```

2. **Compute and visualise the energy function**: The library allows computation of TabEBM's energy function and the unnormalised data density. The demo notebook, `TabEBM_approximated_density.ipynb`, shows the TabEBM-inferred densities under conditions of data noise and class imbalance (thus recreating the plots from Appendix C).

### B.6 Implementation of Generators

**TabEBM.** In all our experiments, the surrogate binary classifier in TabEBM is a pretrained in-context model, TabPFN [33], using the official model weights released by the authors (`https://github.com/automl/TabPFN/raw/main/tabpfn/models_diff/prior_diff_real_checkpoint_n_0_epoch_42.cpkt`). We use TabPFN with three ensembles. We use four surrogate negative samples, $\mathcal{X}_c^{\text{neg}}$, positioned at $\alpha_{\text{dist}}^{\text{neg}} = 5$ standard deviations from zero, in random corners of a hypercube in $\mathbb{R}^D$ (as explained in Section 2.2), distant from any real data. In Appendix D.1, we show that TabEBM is robust to the distribution of the negative samples.

We use SGLD [84] for sampling from TabEBM, where the starting points $\mathbf{x}_0^{\text{synth}}$ are initialised by adding Gaussian noise with zero mean and standard deviation $\sigma_{\text{start}} = 0.01$ to a randomly selected sample of the specific class, i.e., $\mathbf{x}_0^{\text{synth}} \sim \mathcal{N}(\mathcal{X}_c, \sigma_{\text{start}}^2 \mathbf{I})$. For SGLD, we used the following parameters: step size $\alpha_{\text{step}} = 0.1$, noise scale $\alpha_{\text{noise}} = 0.01$ and number of steps $T = 200$. We found TabEBM to be robust to the SGLD settings (see Appendix D.2).

**TabPFGen.** We re-implemented TabPFGen [48] by closely following the original paper since no official implementation is available. As recommended in [48], the starting points are initialised by adding Gaussian noise with zero mean and standard deviation of 0.01 to the training points.

**SMOTE.** We use the open-source implementation of SMOTE from Imbalanced-learn [45], and the number neighbours $k$ is set within the range of $\{1, 3, 5\}$. When applicable, we always set the maximum value for nearest neighbours (i.e., $k = 5$). However, very low-sample-size datasets may not contain sufficient samples for large $k$. For instance, the "fourier" dataset ($N_{\text{real}} = 20$) only has two samples per class. We set $k = 1$ to generate synthetic data with SMOTE in these cases.

For the other six benchmark generators, we use their open-source implementations in Synthcity [67]. Following prior studies [87, 80, 71, 48], we use the default settings for all generators.

### B.7 Implementation of Downstream Predictors

We implemented TabPFN with its official implementation [33] and the other five downstream predictors with the scikit-learn library [65]. Following prior studies [80, 71], we use the default settings for all downstream predictors.

## C  Limitations of Existing Generative Methods

We showcase three limitations of current generative models: (1) Appendix C shows that models approximating the joint distribution $p(\mathbf{x}, y)$ may fail to preserve the stratification of the real data and even fail to generate samples from specific classes. (2) Appendix C evaluates the approximated class-conditional distributions $p(\mathbf{x} \mid y)$ on data with increasing noise levels, and (3) Appendix C evaluates the approximated class-conditional distributions $p(\mathbf{x} \mid y)$ on data with increasing class imbalance.

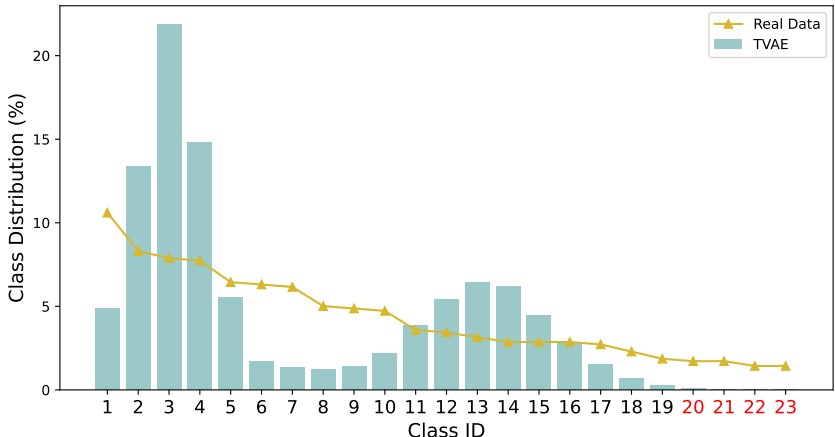

Figure 10: Comparison of class distribution between real data and synthetic data from TVAE. We first train TVAE on the "energy-efficiency" dataset and then randomly generate 10,000 samples with it. We highlight the classes where no synthetic samples are generated. TVAE fails to generate samples for 4 of 23 classes, showing the impracticability to preserve stratification by generative methods that learn joint distribution $p(\mathbf{x}, y)$.

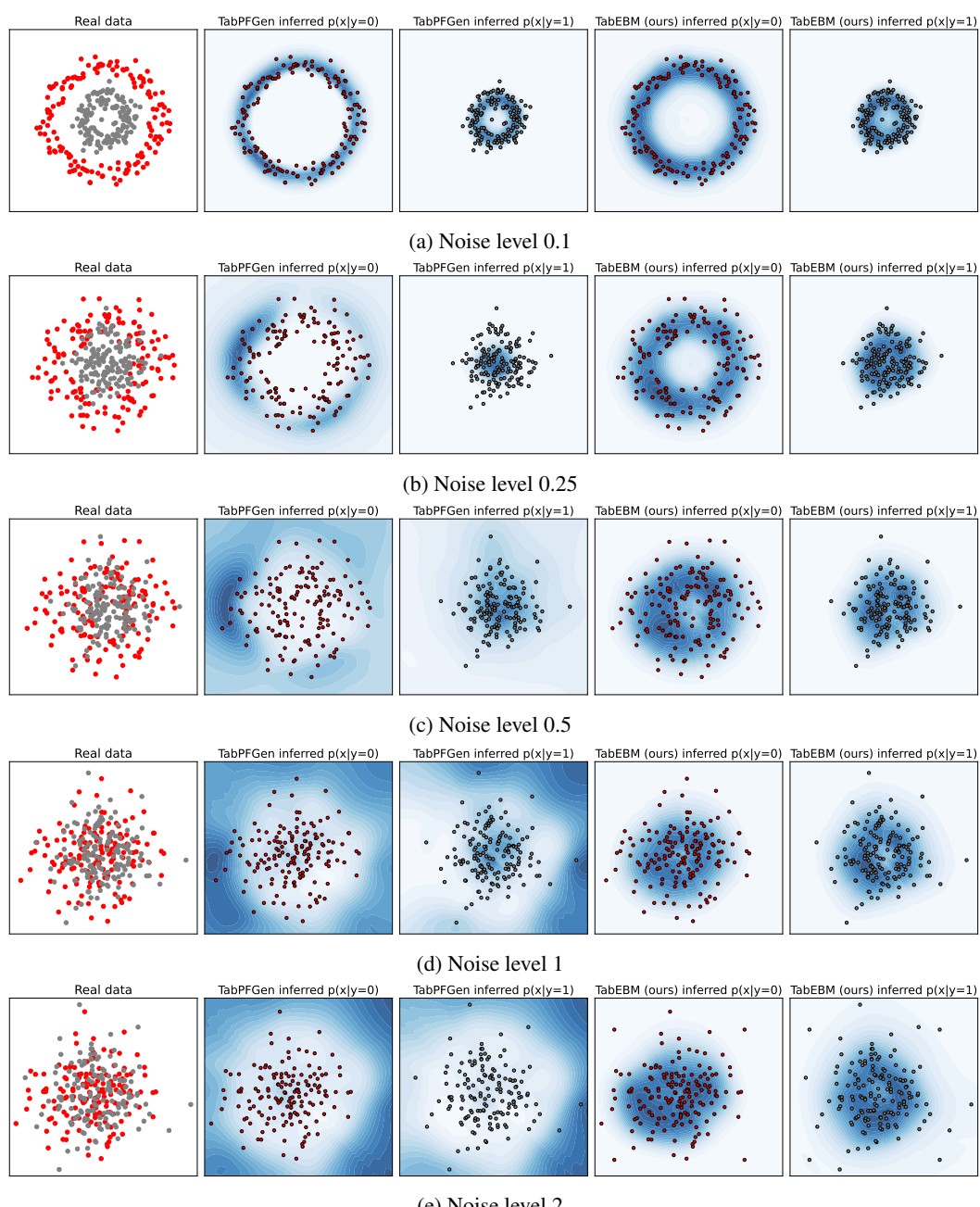

(a) Noise level 0.1

(b) Noise level 0.25

(c) Noise level 0.5

(d) Noise level 1

(e) Noise level 2

Figure 11: Evaluating the approximated class-conditional distributions on data with increasing noise levels. Darker blue indicates a higher assigned probability. TabPFGen uses a single shared energy-based model to infer the class-conditional distribution $p(\mathbf{x}|y)$. As noise increases, TabPFGen's probability assignments vary significantly and end up assigning very high probabilities that are far from the real data. For instance, the areas of assigned probability for $p(\mathbf{x}|y = 1)$ completely flip when noise increases from 0.5 to 1. In contrast, our TabEBM uses class-specific energy models, resulting in robust inferred conditionals. TabEBM performs well even under very high noise (see $p(\mathbf{x}|y = 0)$ for noise level 2), while TabPFGen struggles.

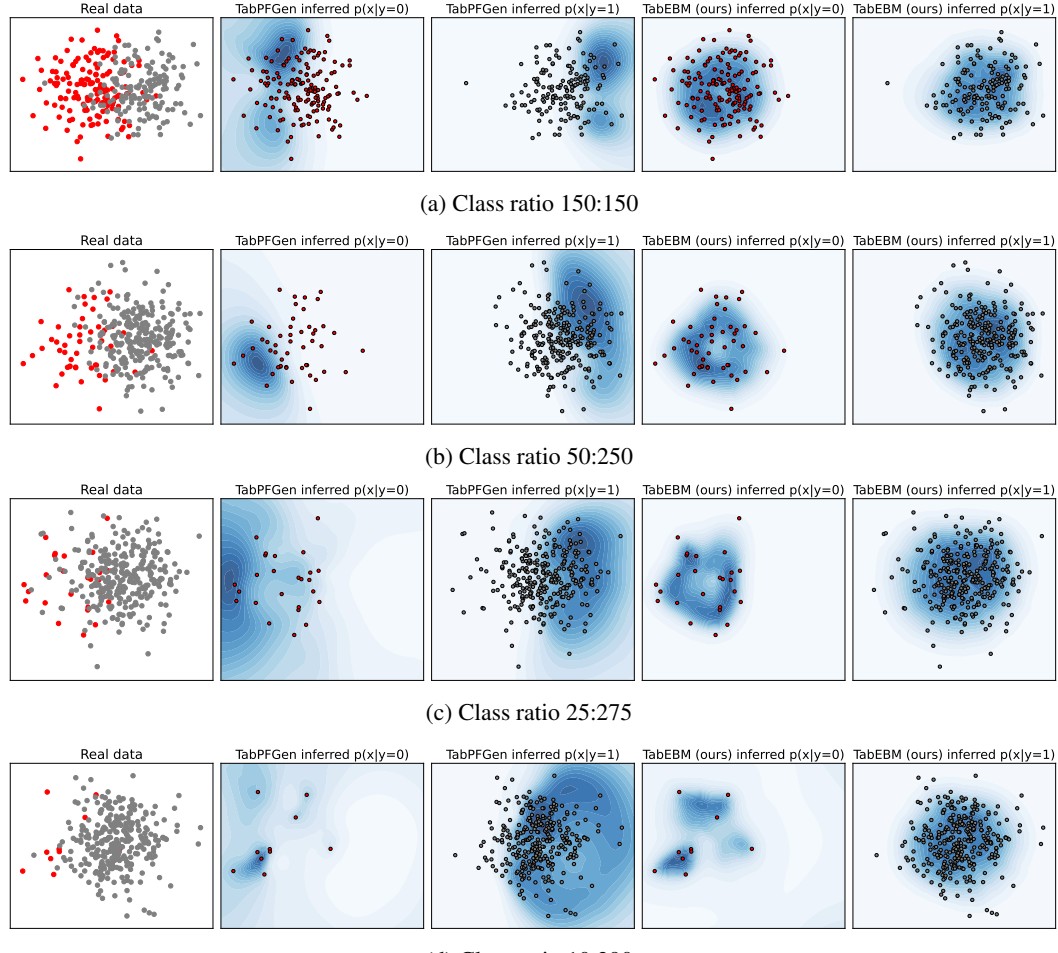

(a) Class ratio 150:150

(b) Class ratio 50:250

(c) Class ratio 25:275

(d) Class ratio 10:290

Figure 12: Evaluating the approximated class-conditional distributions on a toy dataset of 300 samples with varying class imbalances. The two clusters maintain their positions. Darker blue indicates a higher assigned probability. TabPFGen uses a single shared energy-based model to infer the class-conditional distribution $p(\mathbf{x}|y)$. As class imbalance increases, TabPFGen starts assigning high probability in areas far from the real data, for instance, in the case of $p(\mathbf{x}|y = 1)$ for class ratio 10:290. In contrast, our TabEBM fits class-specific energy models only on the class-wise data $\mathcal{X}_c = \{\mathbf{x}^{(i)} \mid y_i = c\}$. This results in very robust inferred conditional distributions even under heavy class imbalance (e.g., see that $p(\mathbf{x}|y = 1)$ remains relatively constant).

## D    Extended Experimental Results

### D.1    Ablations on the distribution of the surrogate negative samples

#### D.1.1    Ablations on placing the negative samples

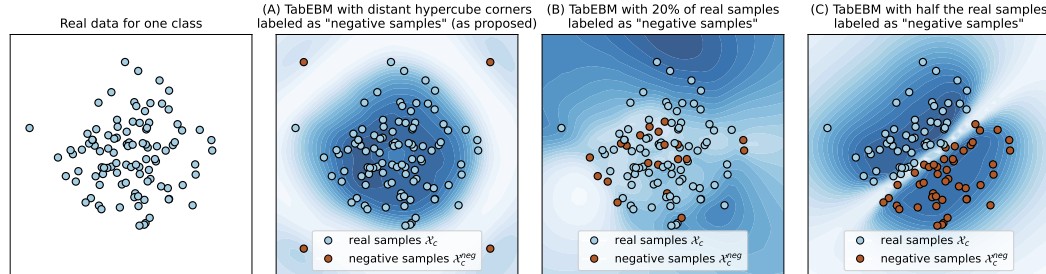

Figure 13: TabEBM energy $E_c(x)$ for different choices of negative samples. The blue region represents low energy, indicating high data density. In (A), TabEBM, with the proposed negative samples placed in a hypercube far from the data, infers an accurate energy surface, resulting in generated data close to the real points. In (B), labelling a random subset of the real data as negative samples leads to a completely inaccurate energy surface. In (C), labelling half of the real points as negative samples reduces density near the decision boundary, as TabPFN assigns low maximal logit due to the high uncertainty. In conclusion, placing negative samples far from the real data results in a robust energy surface.

Appendix D.1.1 shows TabEBM's energy $E_c(x)$ when varying the selection of the negative samples. TabEBM infers an accurate energy surface with distant negative samples, and the energy surface becomes inaccurate when negative samples resemble real samples. This occurs because TabPFN is uncertain when points of different classes are close, affecting its logits magnitude and making them unsuitable for density estimation.

### D.1.2    Varying the number of negative samples

We evaluate the impact of the ratio $|\mathcal{X}_c^{\text{neg}}| : |\mathcal{X}_c|$ between the negative samples $\mathcal{X}_c^{\text{neg}}$ and the real samples $|\mathcal{X}_c|$. We vary $|\mathcal{X}_c^{\text{neg}}|$ while keeping $|\mathcal{X}_c|$ fixed, simulating both balanced and highly imbalanced scenarios. The negative samples are placed in random corners of the hypercube (as described in Section 2), at five standard deviations in each direction (i.e., $\alpha_{\text{dist}}^{\text{neg}} = 5$). To ensure reliable outcomes, we maintained a consistent ratio across all classes, keeping the same proportion of negative samples for each class.

Table 4 shows the results across six datasets with $N_{\text{real}} = 100$ real samples, demonstrating that TabEBM is robust to imbalances in the surrogate binary tasks. The column with $|\mathcal{X}_c^{\text{neg}}| = 4$ represents the TabEBM results from the main paper, where four negative samples were placed in the corners (as described in Section 2). There are negligible differences in performance, and TabEBM consistently outperforms both the baseline and other generators (as shown in Table 1).

Table 4: Evaluating the impact of varying the ratio $|\mathcal{X}_c^{\text{neg}}| : |\mathcal{X}_c|$. We show the test classification accuracy performance (%) of TabEBM on data augmentation averaged over six datasets and ten repeats. TabEBM shows consistent performance and outperforms the baseline, regardless of the number of negative samples.

| | TabEBM | | | | | Baseline (Real data) |
|---|---|---|---|---|---|---|
| **Ratio** $|\mathcal{X}_c^{\text{neg}}| : |\mathcal{X}_c|$ | 0.1 | 0.2 | 0.5 | 1 | Fixed $|\mathcal{X}_c^{\text{neg}}| = 4$ | - |
| biodeg | $76.59_{\pm 3.95}$ | $76.54_{\pm 3.95}$ | $76.47_{\pm 4.05}$ | $76.81_{\pm 3.58}$ | $76.45_{\pm 3.08}$ | $76.69_{\pm 2.70}$ |
| steel | $92.71_{\pm 7.46}$ | $92.60_{\pm 7.45}$ | $92.79_{\pm 7.50}$ | $92.63_{\pm 7.59}$ | $92.71_{\pm 7.57}$ | $86.87_{\pm 12.4}$ |
| stock | $90.46_{\pm 3.49}$ | $90.41_{\pm 3.65}$ | $90.52_{\pm 3.52}$ | $90.31_{\pm 3.63}$ | $90.36_{\pm 3.14}$ | $89.07_{\pm 3.71}$ |
| energy | $31.20_{\pm 6.22}$ | $31.20_{\pm 6.22}$ | $30.89_{\pm 5.83}$ | $30.90_{\pm 6.09}$ | $31.24_{\pm 5.53}$ | $25.94_{\pm 4.86}$ |
| collins | $13.06_{\pm 2.88}$ | $13.02_{\pm 2.85}$ | $13.05_{\pm 2.89}$ | $12.97_{\pm 2.79}$ | $13.07_{\pm 2.51}$ | $11.44_{\pm 2.77}$ |
| texture | $85.91_{\pm 6.92}$ | $85.91_{\pm 6.92}$ | $85.94_{\pm 6.76}$ | $86.26_{\pm 6.72}$ | $86.01_{\pm 7.36}$ | $82.42_{\pm 10.38}$ |
| **Average accuracy** | 64.99 | 64.95 | 64.94 | 64.98 | 64.97 | 62.07 |

### D.1.3 Varying the distance of the negative samples

We assess the effect of varying the distance of negative samples. We use TabEBM with four negative samples positioned randomly at the corners of the hypercube, as outlined in Section 2 (this corresponds to the experimental setup from the main paper). The distance of the negative samples, denoted as $\alpha_{\text{dist}}^{\text{neg}}$, is varied. Table 5 demonstrates that TabEBM remains generally robust to changes in this distance, with only small performance variations across different datasets. Importantly, using TabEBM for data augmentation consistently improves performance by approximately 3% compared to the Baseline, regardless of the distance used.

Table 5: Evaluating the impact of varying the distance of the negative samples $\alpha_{\text{dist}}^{\text{neg}}$ across various datasets. We show the test classification accuracy performance (%) of TabEBM on data augmentation averaged over six datasets and ten repeats. TabEBM is robust, and optional tuning of the negative samples could slightly improve performance.

| | TabEBM | | | | | | Baseline (Real data) |
|---|---|---|---|---|---|---|---|
| **Per-dimension distance** $\alpha_d$ **of the negative samples** | 0.1 | 0.2 | 0.5 | 1 | 2 | 5 | - |
| biodeg | $76.72_{\pm 3.33}$ | $76.62_{\pm 3.40}$ | $77.12_{\pm 2.60}$ | $76.85_{\pm 3.14}$ | $76.50_{\pm 3.93}$ | $76.45_{\pm 3.08}$ | $76.69_{\pm 2.70}$ |
| steel | $93.97_{\pm 5.76}$ | $93.46_{\pm 6.24}$ | $93.00_{\pm 6.92}$ | $92.60_{\pm 7.31}$ | $92.68_{\pm 7.38}$ | $92.71_{\pm 7.57}$ | $86.87_{\pm 12.4}$ |
| stock | $90.42_{\pm 3.46}$ | $90.29_{\pm 3.61}$ | $90.56_{\pm 3.46}$ | $90.38_{\pm 3.64}$ | $90.43_{\pm 3.56}$ | $90.36_{\pm 3.14}$ | $89.07_{\pm 3.71}$ |
| energy | $31.73_{\pm 6.21}$ | $31.42_{\pm 6.08}$ | $31.86_{\pm 6.12}$ | $32.53_{\pm 5.96}$ | $31.65_{\pm 6.06}$ | $31.24_{\pm 5.53}$ | $25.94_{\pm 4.86}$ |
| collins | $13.03_{\pm 2.59}$ | $12.92_{\pm 2.60}$ | $12.97_{\pm 2.69}$ | $13.03_{\pm 2.84}$ | $13.08_{\pm 2.93}$ | $13.07_{\pm 2.51}$ | $11.44_{\pm 2.77}$ |
| texture | $85.62_{\pm 7.41}$ | $85.58_{\pm 7.49}$ | $85.50_{\pm 7.65}$ | $85.05_{\pm 8.21}$ | $85.20_{\pm 7.95}$ | $86.01_{\pm 7.36}$ | $82.42_{\pm 10.38}$ |
| **Average accuracy** | 65.25 | 65.05 | 65.17 | 65.07 | 64.92 | 64.97 | 62.07 |

### D.2 Ablations on the sensitivity to the hyperparameters of SGLD sampling

We vary two key hyperparameters of SGLD on the "biodeg" binary dataset with $N_{\text{real}} = 100$: the step size $\alpha_{\text{step}}$ and the noise scale $\alpha_{\text{noise}}$. Table 6 shows that TabEBM remains stable with respect to these hyperparameters. Note that smaller values of $\alpha_{\text{noise}}$ are expected to perform better because SGLD sampling adds noise at each iteration (see Line 7 in Algorithm 1), thus larger values of $\alpha_{\text{noise}}$ will hinder convergence of the SGLD sampler.

Table 6: Test classification accuracy (%) of TabEBM (averaged over six downstream predictors) with different SGLD settings. Increasing $\alpha_{\text{noise}}$ (added at each SGLD step) is expected to degrade performance, as it causes the sampling to diverge further from the real data.

| $\alpha_{\text{noise}}$ | $\alpha_{\text{step}}$ | | | |
|---|---|---|---|---|
| | 0.1 | 0.3 | 0.5 | 1.0 |
| 0.01 | 76.45 | **77.09** | 77.04 | 76.58 |
| 0.02 | 76.86 | 76.96 | 76.77 | 76.26 |
| 0.05 | 75.93 | 75.89 | 75.94 | **75.70** |

## D.3 Distribution of Logits and Unnormalized Density in TabEBM

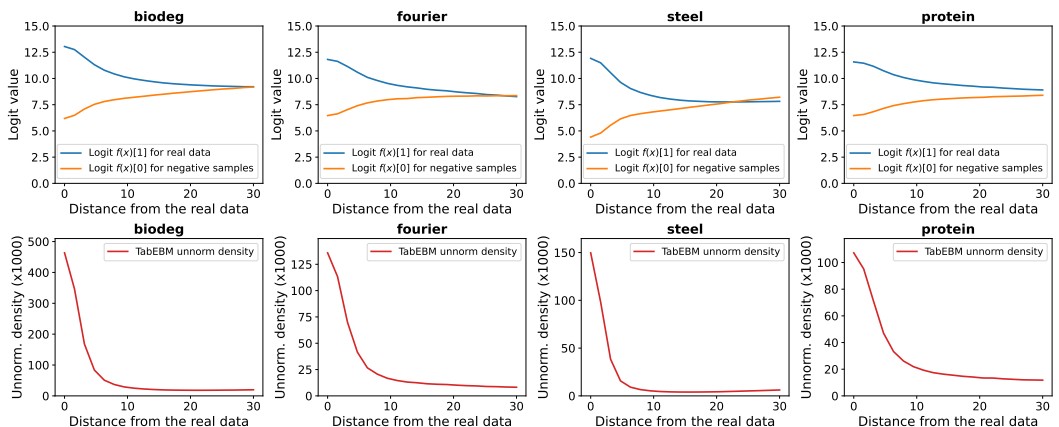

Figure 14: Additional results for Section 3.4. The logit distribution of TabPFN trained on our surrogate binary tasks across four datasets. Starting from the real samples, random points are selected at increasing distances (shown on the x-axis). The **top row** shows the logit distributions for the surrogate task. Close to the real data, TabPFN outputs a high logit value. As the distance increases, the logits converge due to increased predictive uncertainty, leading to equal class probabilities after applying softmax. Notably, across datasets, TabPFN's logits are always positive, have similar ranges, and maintain a relatively constant sum as distance increases. The **bottom row** TabEBM's unnormalized density, $p_c(x) \propto \exp(-E_c(x)) \to p_c(x) \propto (\exp(f(x)[0]) + \exp(f(x)[1]))$. The density decreases significantly far from the data, becoming negligible. Because sampling using SGLD perform gradient ascent on the density, the TabEBM-generated samples will be similar when using one or both logits.

## D.4 Complete Trade-off Figures with Error Bars

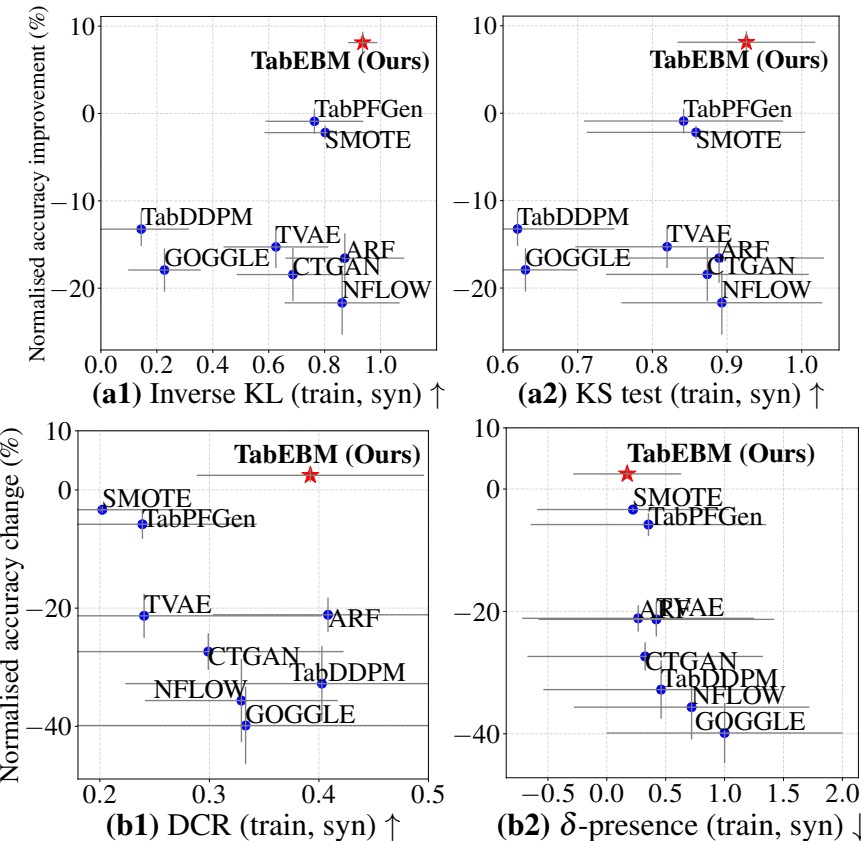

Figure 15: **(a1&a2):** Median inverse KL and KS test vs. mean normalised balanced accuracy improvement (%) between real train data and synthetic data. **(b1&b2):** Median DCR and $\delta -$ $presence$ vs. mean normalised balanced accuracy change (%) between real train data and synthetic data. Note that "accuracy improvement" is for data augmentation, and "accuracy change" is for data sharing. TabEBM generates high-fidelity synthetic data that can also be used for privacy preservation.

## D.5 Results on Data Augmentation

### D.5.1 Results on eight OpenML datasets.

Table 7: **Classification accuracy** (%) of LR, comparing data augmentation on eight real-world tabular datasets with varied real data availability. We report the mean $\pm$ std balanced accuracy and average accuracy rank across datasets. A higher rank implies higher accuracy. Note that "N/A" denotes that a specific generator was not applicable or the downstream predictor failed to converge, and the rank is computed with the mean balanced accuracy of other methods. We **bold** the highest accuracy for each dataset of different sample sizes. TabEBM achieves the best overall performance against Baseline and benchmark generators.

| | Datasets | $N_{\text{real}}$ | Baseline (Real data) | SMOTE | TVAE | CTGAN | NFLOW | TabDDPM | ARF | GOGGLE | TabPFGen | **TabEBM** |
|---|---|---|---|---|---|---|---|---|---|---|---|---|
| *At most 10 classes* | protein | 20 | N/A | $22.02_{\pm2.91}$ | $21.04_{\pm4.76}$ | $18.40_{\pm4.82}$ | $18.77_{\pm3.84}$ | $25.92_{\pm4.30}$ | $36.61_{\pm2.53}$ | $\mathbf{38.07}_{\pm1.25}$ | $38.01_{\pm2.38}$ | |
| | | 50 | $62.14_{\pm3.77}$ | $61.43_{\pm4.34}$ | $37.04_{\pm2.79}$ | $33.10_{\pm5.99}$ | $31.25_{\pm4.21}$ | $23.98_{\pm2.75}$ | $43.64_{\pm5.07}$ | $54.95_{\pm3.28}$ | $63.00_{\pm3.69}$ | $\mathbf{63.05}_{\pm3.84}$ |
| | | 100 | $79.97_{\pm3.24}$ | $79.53_{\pm3.37}$ | $61.07_{\pm5.06}$ | $55.44_{\pm1.92}$ | $46.37_{\pm4.10}$ | $45.55_{\pm4.24}$ | $56.77_{\pm3.06}$ | $67.25_{\pm4.50}$ | $\mathbf{80.54}_{\pm3.27}$ | $80.32_{\pm3.12}$ |
| | | 200 | $91.53_{\pm1.58}$ | $90.92_{\pm1.81}$ | $77.43_{\pm2.75}$ | $71.27_{\pm3.07}$ | $66.16_{\pm4.31}$ | $66.37_{\pm3.42}$ | $70.52_{\pm2.17}$ | $76.30_{\pm3.70}$ | $\mathbf{91.69}_{\pm1.66}$ | $91.34_{\pm1.77}$ |
| | | 500 | $97.86_{\pm0.83}$ | $97.69_{\pm0.80}$ | $90.77_{\pm0.93}$ | $89.05_{\pm1.52}$ | $85.09_{\pm1.99}$ | $83.58_{\pm2.22}$ | $88.55_{\pm1.54}$ | $90.64_{\pm0.81}$ | $\mathbf{97.97}_{\pm0.61}$ | $97.88_{\pm0.86}$ |
| | fourier | 20 | $42.90_{\pm5.30}$ | N/A | $22.46_{\pm5.88}$ | $16.00_{\pm4.70}$ | $15.48_{\pm3.79}$ | $13.58_{\pm4.30}$ | $22.04_{\pm4.42}$ | $15.80_{\pm4.15}$ | $\mathbf{44.67}_{\pm8.85}$ | $43.02_{\pm5.14}$ |
| | | 50 | $\mathbf{60.62}_{\pm1.64}$ | $58.40_{\pm1.95}$ | $33.42_{\pm2.98}$ | $31.18_{\pm5.47}$ | $28.70_{\pm3.74}$ | $26.18_{\pm3.80}$ | $39.04_{\pm3.12}$ | $40.00_{\pm4.97}$ | $60.07_{\pm2.14}$ | $60.36_{\pm1.55}$ |
| | | 100 | $\mathbf{67.76}_{\pm2.49}$ | $65.84_{\pm2.35}$ | $41.36_{\pm2.85}$ | $40.32_{\pm3.49}$ | $40.32_{\pm5.82}$ | $41.44_{\pm5.02}$ | $47.90_{\pm3.74}$ | $39.78_{\pm3.99}$ | $67.40_{\pm1.51}$ | $67.44_{\pm2.46}$ |
| | | 200 | $\mathbf{73.13}_{\pm2.41}$ | $71.56_{\pm2.67}$ | $54.76_{\pm3.46}$ | $55.00_{\pm3.72}$ | $52.40_{\pm3.18}$ | $58.08_{\pm3.52}$ | $58.48_{\pm2.08}$ | $50.98_{\pm2.68}$ | $70.30_{\pm2.91}$ | $72.38_{\pm3.01}$ |
| | | 500 | $77.44_{\pm1.20}$ | $76.42_{\pm1.28}$ | $68.28_{\pm2.12}$ | $70.18_{\pm1.89}$ | $68.12_{\pm1.62}$ | $72.36_{\pm1.65}$ | $71.54_{\pm1.95}$ | $69.48_{\pm1.71}$ | $76.52_{\pm1.69}$ | $\mathbf{77.50}_{\pm2.14}$ |
| | biodeg | 20 | $\mathbf{71.34}_{\pm5.63}$ | $70.10_{\pm5.49}$ | $70.16_{\pm5.75}$ | $58.17_{\pm8.00}$ | $58.05_{\pm9.91}$ | $49.99_{\pm5.88}$ | $62.61_{\pm6.45}$ | $69.47_{\pm6.00}$ | $70.76_{\pm3.95}$ | $71.24_{\pm4.85}$ |
| | | 50 | $76.35_{\pm2.88}$ | $75.69_{\pm3.03}$ | $73.63_{\pm2.64}$ | $67.44_{\pm3.83}$ | $62.87_{\pm7.30}$ | $49.44_{\pm2.63}$ | $74.44_{\pm2.77}$ | $71.75_{\pm5.27}$ | $75.68_{\pm2.31}$ | $\mathbf{76.41}_{\pm2.93}$ |
| | | 100 | $\mathbf{78.91}_{\pm1.40}$ | $78.39_{\pm1.53}$ | $77.09_{\pm2.80}$ | $74.89_{\pm2.54}$ | $68.62_{\pm5.21}$ | $55.61_{\pm3.56}$ | $75.62_{\pm2.77}$ | $72.45_{\pm3.31}$ | $77.92_{\pm2.41}$ | $78.34_{\pm2.18}$ |
| | | 200 | $\mathbf{82.00}_{\pm1.47}$ | $81.42_{\pm1.39}$ | $80.07_{\pm1.82}$ | $78.56_{\pm3.43}$ | $72.35_{\pm1.72}$ | $59.06_{\pm4.65}$ | $78.03_{\pm1.95}$ | $73.73_{\pm2.09}$ | $81.24_{\pm1.71}$ | $81.43_{\pm1.78}$ |
| | | 500 | $\mathbf{83.83}_{\pm0.57}$ | $83.74_{\pm0.90}$ | $81.69_{\pm0.82}$ | $82.12_{\pm1.17}$ | $78.06_{\pm2.13}$ | $66.86_{\pm5.43}$ | $81.47_{\pm0.93}$ | $77.98_{\pm1.27}$ | $83.43_{\pm0.82}$ | $83.10_{\pm0.98}$ |
| | steel | 20 | $63.66_{\pm8.98}$ | $57.88_{\pm5.72}$ | $60.27_{\pm7.47}$ | $57.90_{\pm4.45}$ | $53.10_{\pm7.28}$ | $54.20_{\pm6.99}$ | $55.41_{\pm4.92}$ | $53.29_{\pm4.31}$ | $66.81_{\pm9.74}$ | $\mathbf{67.03}_{\pm9.35}$ |
| | | 50 | $87.91_{\pm5.88}$ | $69.01_{\pm6.60}$ | $66.22_{\pm3.63}$ | $66.22_{\pm5.77}$ | $57.05_{\pm5.51}$ | $57.46_{\pm8.48}$ | $64.81_{\pm4.64}$ | $57.20_{\pm5.19}$ | $\mathbf{93.63}_{\pm4.78}$ | $92.20_{\pm4.81}$ |
| | | 100 | $98.85_{\pm1.20}$ | $82.67_{\pm4.30}$ | $74.33_{\pm3.85}$ | $70.49_{\pm5.35}$ | $65.09_{\pm7.30}$ | $52.77_{\pm7.06}$ | $67.85_{\pm4.94}$ | $61.62_{\pm4.05}$ | $\mathbf{99.24}_{\pm0.82}$ | $99.21_{\pm0.86}$ |
| | | 200 | $99.43_{\pm0.58}$ | $87.18_{\pm3.06}$ | $82.77_{\pm3.21}$ | $80.34_{\pm2.93}$ | $70.49_{\pm5.27}$ | $72.99_{\pm13.98}$ | $80.27_{\pm7.32}$ | $64.52_{\pm2.16}$ | $99.45_{\pm0.69}$ | $\mathbf{99.51}_{\pm0.69}$ |
| | | 500 | $99.75_{\pm0.29}$ | $96.63_{\pm2.11}$ | $94.59_{\pm2.98}$ | $96.32_{\pm1.52}$ | $84.15_{\pm2.69}$ | $98.07_{\pm1.37}$ | $95.35_{\pm2.06}$ | $70.11_{\pm2.58}$ | $99.84_{\pm0.20}$ | $99.84_{\pm0.20}$ |
| | stock | 20 | $77.99_{\pm4.40}$ | $80.45_{\pm3.98}$ | $74.21_{\pm6.36}$ | $59.20_{\pm12.69}$ | $72.50_{\pm7.92}$ | $72.09_{\pm9.75}$ | $69.04_{\pm6.25}$ | $\mathbf{80.59}_{\pm3.59}$ | $79.54_{\pm4.46}$ | $80.39_{\pm3.42}$ |
| | | 50 | $80.68_{\pm2.65}$ | $81.49_{\pm2.95}$ | $76.41_{\pm3.95}$ | $72.95_{\pm2.17}$ | $75.41_{\pm6.00}$ | $78.44_{\pm4.40}$ | $76.91_{\pm2.36}$ | $75.49_{\pm5.31}$ | $\mathbf{82.37}_{\pm3.20}$ | $82.21_{\pm2.60}$ |
| | | 100 | $82.11_{\pm1.11}$ | $\mathbf{83.86}_{\pm1.97}$ | $79.85_{\pm2.79}$ | $78.47_{\pm2.71}$ | $76.99_{\pm3.49}$ | $80.82_{\pm3.52}$ | $78.89_{\pm2.36}$ | $77.65_{\pm2.60}$ | $83.67_{\pm1.60}$ | $83.52_{\pm1.76}$ |
| | | 200 | $82.18_{\pm0.81}$ | $\mathbf{84.29}_{\pm1.19}$ | $79.24_{\pm2.82}$ | $79.86_{\pm2.42}$ | $76.49_{\pm1.37}$ | $80.21_{\pm2.13}$ | $78.87_{\pm2.46}$ | $76.91_{\pm1.04}$ | $83.75_{\pm1.53}$ | $84.17_{\pm1.42}$ |
| *More than 10 classes* | energy | 50 | $\mathbf{22.22}_{\pm2.36}$ | N/A | $10.11_{\pm2.20}$ | $9.58_{\pm3.15}$ | $7.70_{\pm1.83}$ | $8.20_{\pm2.01}$ | $10.51_{\pm1.28}$ | $17.10_{\pm5.03}$ | N/A | $21.66_{\pm1.54}$ |
| | | 100 | $24.00_{\pm2.30}$ | N/A | $13.80_{\pm2.23}$ | $13.01_{\pm1.71}$ | $12.14_{\pm1.87}$ | $10.79_{\pm3.19}$ | $15.65_{\pm2.40}$ | $14.45_{\pm2.90}$ | N/A | $\mathbf{28.10}_{\pm2.19}$ |
| | | 200 | $29.37_{\pm2.63}$ | N/A | $16.39_{\pm2.68}$ | $16.56_{\pm3.58}$ | $16.78_{\pm3.15}$ | $18.11_{\pm1.71}$ | $20.10_{\pm2.48}$ | $20.92_{\pm2.79}$ | N/A | $\mathbf{34.38}_{\pm2.60}$ |
| | collins | 100 | $\mathbf{14.28}_{\pm1.63}$ | N/A | $10.57_{\pm1.72}$ | $8.69_{\pm1.17}$ | $9.59_{\pm1.35}$ | $13.31_{\pm1.67}$ | $8.69_{\pm1.80}$ | $12.08_{\pm1.56}$ | N/A | $14.01_{\pm2.55}$ |
| | | 200 | $19.20_{\pm1.71}$ | $\mathbf{19.39}_{\pm1.88}$ | $16.03_{\pm1.74}$ | $11.64_{\pm1.76}$ | $10.97_{\pm1.46}$ | $17.06_{\pm1.51}$ | $11.31_{\pm1.58}$ | $17.80_{\pm1.21}$ | N/A | $19.33_{\pm1.55}$ |
| | texture | 50 | $86.56_{\pm2.96}$ | $86.93_{\pm2.77}$ | $55.01_{\pm5.77}$ | $42.17_{\pm6.36}$ | $44.63_{\pm5.41}$ | $60.07_{\pm10.11}$ | $44.46_{\pm6.63}$ | $77.68_{\pm4.33}$ | N/A | $\mathbf{88.54}_{\pm2.88}$ |
| | | 100 | $94.07_{\pm1.70}$ | $93.87_{\pm1.82}$ | $65.36_{\pm4.49}$ | $60.07_{\pm6.81}$ | $60.76_{\pm5.18}$ | $73.16_{\pm5.11}$ | $64.69_{\pm4.79}$ | $84.13_{\pm1.97}$ | N/A | $\mathbf{94.38}_{\pm1.24}$ |
| | | 200 | $\mathbf{96.65}_{\pm1.24}$ | $96.53_{\pm1.33}$ | $75.91_{\pm5.58}$ | $80.02_{\pm5.13}$ | $77.07_{\pm3.89}$ | $86.24_{\pm3.62}$ | $85.90_{\pm2.78}$ | $85.94_{\pm2.88}$ | N/A | $96.53_{\pm1.27}$ |
| | | 500 | $98.03_{\pm0.36}$ | $\mathbf{98.05}_{\pm0.23}$ | $91.87_{\pm0.93}$ | $92.93_{\pm1.78}$ | $90.01_{\pm1.80}$ | $93.92_{\pm0.81}$ | $94.83_{\pm0.89}$ | $91.72_{\pm1.49}$ | N/A | $97.75_{\pm0.42}$ |
| **Average rank** | | | $2.36_{\pm1.14}$ | $3.45_{\pm1.35}$ | $6.52_{\pm1.48}$ | $7.53_{\pm1.42}$ | $9.08_{\pm0.77}$ | $7.61_{\pm2.33}$ | $6.70_{\pm1.47}$ | $6.67_{\pm2.53}$ | $3.17_{\pm1.81}$ | $\mathbf{1.92}_{\pm0.75}$ |

Table 8: **Classification accuracy** (%) of KNN, comparing data augmentation on eight real-world tabular datasets with varied real data availability. We report the mean ± std balanced accuracy and average accuracy rank across datasets. A higher rank implies higher accuracy. Note that "N/A" denotes that a specific generator was not applicable or the downstream predictor failed to converge, and the rank is computed with the mean balanced accuracy of other methods. We **bold** the highest accuracy for each dataset of different sample sizes. TabEBM achieves the best overall performance against Baseline and benchmark generators.

| | Datasets | $N_{real}$ | Baseline (Real data) | SMOTE | TVAE | CTGAN | NFLOW | TabDDPM | ARF | GOGGLE | TabPFGen | **TabEBM** |
|---|---|---|---|---|---|---|---|---|---|---|---|---|
| *At most 10 classes* | protein | 20 | $21.34_{\pm2.93}$ | N/A | $21.78_{\pm2.06}$ | $21.18_{\pm4.22}$ | $21.30_{\pm1.90}$ | $22.00_{\pm2.70}$ | $22.69_{\pm3.86}$ | $16.99_{\pm3.45}$ | $\mathbf{35.78_{\pm4.46}}$ | $35.76_{\pm4.37}$ |
| | | 50 | $36.41_{\pm4.33}$ | $\mathbf{55.24_{\pm3.81}}$ | $35.85_{\pm2.50}$ | $36.13_{\pm4.24}$ | $35.40_{\pm4.27}$ | $36.77_{\pm4.06}$ | $36.84_{\pm4.05}$ | $31.02_{\pm4.11}$ | $53.38_{\pm3.53}$ | $53.49_{\pm3.30}$ |
| | | 100 | $50.17_{\pm3.11}$ | $\mathbf{70.11_{\pm2.82}}$ | $51.97_{\pm2.84}$ | $50.61_{\pm3.15}$ | $50.62_{\pm3.27}$ | $50.63_{\pm3.55}$ | $50.36_{\pm3.44}$ | $44.70_{\pm2.22}$ | $67.99_{\pm2.43}$ | $68.27_{\pm2.51}$ |
| | | 200 | $65.84_{\pm2.78}$ | $80.43_{\pm2.44}$ | $65.52_{\pm2.96}$ | $66.05_{\pm2.74}$ | $66.14_{\pm2.42}$ | $67.50_{\pm2.57}$ | $66.52_{\pm3.16}$ | $63.92_{\pm3.26}$ | $79.94_{\pm2.26}$ | $\mathbf{80.55_{\pm2.02}}$ |
| | | 500 | $85.63_{\pm1.41}$ | $90.92_{\pm1.42}$ | $87.08_{\pm1.86}$ | $85.77_{\pm1.43}$ | $85.51_{\pm1.50}$ | $86.47_{\pm1.40}$ | $85.87_{\pm1.63}$ | $85.64_{\pm1.94}$ | $91.32_{\pm1.09}$ | $\mathbf{91.67_{\pm1.11}}$ |
| | fourier | 20 | $18.06_{\pm3.30}$ | N/A | $26.56_{\pm4.92}$ | $24.88_{\pm3.66}$ | $19.80_{\pm3.77}$ | $19.30_{\pm3.53}$ | $23.42_{\pm3.45}$ | $18.78_{\pm2.17}$ | $41.08_{\pm6.56}$ | $\mathbf{42.78_{\pm5.83}}$ |
| | | 50 | $48.00_{\pm2.47}$ | $\mathbf{60.38_{\pm1.67}}$ | $39.86_{\pm3.73}$ | $46.82_{\pm3.52}$ | $43.56_{\pm3.45}$ | $49.54_{\pm2.78}$ | $42.98_{\pm2.80}$ | $28.12_{\pm2.75}$ | $59.50_{\pm1.99}$ | $58.54_{\pm1.86}$ |
| | | 100 | $58.36_{\pm3.26}$ | $\mathbf{66.96_{\pm2.47}}$ | $48.44_{\pm4.14}$ | $53.94_{\pm3.47}$ | $53.50_{\pm2.54}$ | $60.80_{\pm4.28}$ | $52.74_{\pm3.15}$ | $35.70_{\pm2.40}$ | $63.88_{\pm2.53}$ | $65.08_{\pm2.47}$ |
| | | 200 | $68.60_{\pm2.55}$ | $\mathbf{71.90_{\pm2.02}}$ | $59.66_{\pm3.31}$ | $66.54_{\pm2.75}$ | $65.16_{\pm2.71}$ | $70.22_{\pm2.26}$ | $64.52_{\pm2.44}$ | $51.24_{\pm7.29}$ | $70.32_{\pm1.94}$ | $71.08_{\pm1.87}$ |
| | | 500 | $76.90_{\pm1.30}$ | $77.64_{\pm1.07}$ | $73.20_{\pm1.68}$ | $76.22_{\pm1.62}$ | $75.72_{\pm1.40}$ | $\mathbf{78.88_{\pm1.58}}$ | $76.54_{\pm0.77}$ | $63.66_{\pm2.49}$ | $74.30_{\pm1.51}$ | $75.35_{\pm1.34}$ |
| | biodeg | 20 | $65.23_{\pm5.01}$ | $68.99_{\pm3.31}$ | $66.63_{\pm7.83}$ | $56.99_{\pm5.55}$ | $59.91_{\pm6.09}$ | $55.85_{\pm4.94}$ | $58.77_{\pm5.93}$ | $56.62_{\pm7.29}$ | $67.79_{\pm4.64}$ | $\mathbf{69.76_{\pm4.43}}$ |
| | | 50 | $71.26_{\pm3.13}$ | $73.19_{\pm2.46}$ | $70.80_{\pm2.14}$ | $70.00_{\pm5.92}$ | $65.90_{\pm3.57}$ | $65.90_{\pm3.57}$ | $73.50_{\pm4.43}$ | $65.29_{\pm4.57}$ | $72.08_{\pm3.84}$ | $\mathbf{73.58_{\pm3.57}}$ |
| | | 100 | $76.12_{\pm1.98}$ | $76.07_{\pm1.74}$ | $74.02_{\pm2.78}$ | $75.36_{\pm2.18}$ | $73.24_{\pm2.61}$ | $\mathbf{77.34_{\pm2.19}}$ | $74.28_{\pm2.02}$ | $72.26_{\pm2.46}$ | $74.56_{\pm1.58}$ | $75.60_{\pm1.55}$ |
| | | 200 | $78.86_{\pm2.19}$ | $\mathbf{79.67_{\pm1.68}}$ | $77.31_{\pm2.93}$ | $78.05_{\pm3.07}$ | $77.64_{\pm2.71}$ | $77.84_{\pm2.62}$ | $78.81_{\pm2.66}$ | $76.82_{\pm2.29}$ | $77.46_{\pm1.68}$ | $78.46_{\pm1.69}$ |
| | | 500 | $82.59_{\pm1.17}$ | $\mathbf{83.07_{\pm1.50}}$ | $82.13_{\pm1.21}$ | $82.17_{\pm1.32}$ | $82.80_{\pm1.28}$ | $81.06_{\pm1.22}$ | $82.15_{\pm1.33}$ | $82.10_{\pm0.79}$ | $79.99_{\pm1.76}$ | $81.01_{\pm1.66}$ |
| | steel | 20 | $56.40_{\pm4.48}$ | $63.95_{\pm3.14}$ | $59.45_{\pm8.27}$ | $57.04_{\pm5.05}$ | $54.59_{\pm5.81}$ | $65.46_{\pm6.10}$ | $56.97_{\pm5.43}$ | $52.90_{\pm3.76}$ | $\mathbf{70.68_{\pm3.87}}$ | $69.31_{\pm4.02}$ |
| | | 50 | $73.95_{\pm4.76}$ | $70.24_{\pm3.44}$ | $67.60_{\pm4.10}$ | $68.77_{\pm2.85}$ | $67.00_{\pm4.58}$ | $\mathbf{85.14_{\pm8.76}}$ | $64.02_{\pm3.71}$ | $57.54_{\pm2.34}$ | $82.09_{\pm3.09}$ | $80.47_{\pm3.48}$ |
| | | 100 | $84.70_{\pm5.57}$ | $77.46_{\pm3.67}$ | $71.87_{\pm2.98}$ | $72.94_{\pm4.62}$ | $77.09_{\pm2.63}$ | $\mathbf{94.05_{\pm3.84}}$ | $72.62_{\pm5.21}$ | $61.08_{\pm1.93}$ | $87.77_{\pm3.13}$ | $87.67_{\pm3.22}$ |
| | | 200 | $90.44_{\pm2.80}$ | $82.46_{\pm1.43}$ | $80.83_{\pm2.65}$ | $82.73_{\pm3.88}$ | $85.49_{\pm3.59}$ | $\mathbf{98.99_{\pm0.75}}$ | $83.38_{\pm2.67}$ | $69.12_{\pm2.56}$ | $92.01_{\pm1.73}$ | $92.06_{\pm1.48}$ |
| | | 500 | $94.99_{\pm1.09}$ | $89.97_{\pm0.88}$ | $91.34_{\pm1.69}$ | $92.42_{\pm1.36}$ | $93.37_{\pm1.13}$ | $\mathbf{99.71_{\pm0.21}}$ | $92.02_{\pm2.03}$ | $80.79_{\pm1.93}$ | $95.08_{\pm1.30}$ | $95.50_{\pm1.49}$ |
| | stock | 20 | $71.89_{\pm4.37}$ | $84.41_{\pm5.28}$ | $73.80_{\pm4.68}$ | $66.38_{\pm9.10}$ | $68.93_{\pm10.49}$ | $81.82_{\pm8.38}$ | $67.53_{\pm8.58}$ | $71.80_{\pm4.99}$ | $84.41_{\pm4.22}$ | $\mathbf{84.69_{\pm4.16}}$ |
| | | 50 | $85.03_{\pm3.39}$ | $\mathbf{89.77_{\pm1.99}}$ | $84.32_{\pm3.97}$ | $83.49_{\pm3.67}$ | $84.43_{\pm2.04}$ | $89.34_{\pm1.59}$ | $84.33_{\pm3.22}$ | $83.64_{\pm2.53}$ | $89.67_{\pm1.88}$ | $89.68_{\pm1.87}$ |
| | | 100 | $89.66_{\pm1.39}$ | $92.32_{\pm0.99}$ | $89.58_{\pm1.22}$ | $89.61_{\pm1.36}$ | $89.66_{\pm1.01}$ | $89.44_{\pm1.01}$ | $91.40_{\pm1.41}$ | $89.66_{\pm2.10}$ | $92.02_{\pm0.81}$ | $\mathbf{92.47_{\pm0.83}}$ |
| | | 200 | $91.65_{\pm1.08}$ | $93.46_{\pm0.82}$ | $92.37_{\pm1.18}$ | $91.55_{\pm1.19}$ | $91.43_{\pm1.34}$ | $92.92_{\pm1.00}$ | $91.14_{\pm1.58}$ | $91.53_{\pm1.05}$ | $93.15_{\pm0.72}$ | $\mathbf{93.62_{\pm1.14}}$ |
| *More than 10 classes* | energy | 50 | $10.85_{\pm1.76}$ | N/A | $10.64_{\pm2.36}$ | $8.22_{\pm2.03}$ | $8.83_{\pm1.53}$ | $8.92_{\pm2.51}$ | $9.14_{\pm1.95}$ | $11.86_{\pm2.33}$ | N/A | $\mathbf{25.36_{\pm2.27}}$ |
| | | 100 | $18.60_{\pm1.83}$ | N/A | $13.71_{\pm1.66}$ | $15.81_{\pm1.50}$ | $14.67_{\pm1.55}$ | $16.18_{\pm1.75}$ | $15.71_{\pm2.79}$ | $17.64_{\pm2.68}$ | N/A | $\mathbf{29.82_{\pm2.74}}$ |
| | | 200 | $26.45_{\pm1.49}$ | N/A | $20.71_{\pm1.02}$ | $21.71_{\pm3.23}$ | $23.40_{\pm2.15}$ | $23.95_{\pm2.94}$ | $23.09_{\pm2.56}$ | $27.35_{\pm2.28}$ | N/A | $\mathbf{35.93_{\pm2.85}}$ |
| | collins | 100 | $10.59_{\pm1.48}$ | N/A | $7.58_{\pm0.74}$ | $7.95_{\pm1.12}$ | $7.55_{\pm1.32}$ | $14.24_{\pm1.48}$ | $7.42_{\pm1.17}$ | $8.79_{\pm0.93}$ | N/A | $\mathbf{15.16_{\pm1.92}}$ |
| | | 200 | $15.84_{\pm1.74}$ | $\mathbf{19.81_{\pm1.73}}$ | $9.79_{\pm1.14}$ | $11.21_{\pm1.45}$ | $12.24_{\pm1.65}$ | $16.30_{\pm1.54}$ | $10.96_{\pm1.43}$ | $12.86_{\pm1.50}$ | N/A | $18.05_{\pm1.65}$ |
| | texture | 50 | $62.96_{\pm2.49}$ | $\mathbf{78.80_{\pm2.75}}$ | $55.51_{\pm3.69}$ | $61.86_{\pm4.48}$ | $62.08_{\pm3.17}$ | $61.91_{\pm2.24}$ | $62.67_{\pm2.29}$ | $56.81_{\pm2.98}$ | N/A | $75.57_{\pm2.67}$ |
| | | 100 | $77.16_{\pm1.25}$ | $\mathbf{86.15_{\pm2.62}}$ | $69.54_{\pm2.66}$ | $76.53_{\pm2.22}$ | $76.85_{\pm1.56}$ | $77.77_{\pm1.80}$ | $76.70_{\pm2.05}$ | $72.64_{\pm1.81}$ | N/A | $84.83_{\pm1.67}$ |
| | | 200 | $85.34_{\pm1.18}$ | $89.07_{\pm1.74}$ | $81.70_{\pm1.32}$ | $85.46_{\pm1.28}$ | $84.62_{\pm1.09}$ | $85.94_{\pm1.35}$ | $85.11_{\pm1.20}$ | $84.72_{\pm0.80}$ | N/A | $\mathbf{89.48_{\pm2.01}}$ |
| | | 500 | $91.40_{\pm1.60}$ | $93.14_{\pm1.28}$ | $89.88_{\pm1.44}$ | $91.40_{\pm1.55}$ | $91.34_{\pm1.60}$ | $92.31_{\pm1.60}$ | $91.46_{\pm1.51}$ | $91.91_{\pm1.63}$ | N/A | $\mathbf{93.46_{\pm0.66}}$ |
| **Average rank** | | | $5.15_{\pm2.06}$ | $2.70_{\pm1.97}$ | $7.67_{\pm2.10}$ | $7.03_{\pm1.55}$ | $7.27_{\pm1.68}$ | $4.12_{\pm2.34}$ | $6.82_{\pm1.76}$ | $8.42_{\pm2.33}$ | $3.67_{\pm1.96}$ | $\mathbf{2.15_{\pm1.75}}$ |

Table 9: **Classification accuracy** (%) of MLP, comparing data augmentation on eight real-world tabular datasets with varied real data availability. We report the mean ± std balanced accuracy and average accuracy rank across datasets. A higher rank implies higher accuracy. Note that "N/A" denotes that a specific generator was not applicable or the downstream predictor failed to converge, and the rank is computed with the mean balanced accuracy of other methods. We **bold** the highest accuracy for each dataset of different sample sizes. TabEBM achieves the best overall performance against Baseline and benchmark generators.

| | Datasets | $N_{real}$ | Baseline (Real data) | SMOTE | TVAE | CTGAN | NFLOW | TabDDPM | ARF | GOGGLE | TabPFGen | **TabEBM** |
|---|---|---|---|---|---|---|---|---|---|---|---|---|
| *At most 10 classes* | protein | 20 | 35.12$_{\pm2.59}$ | N/A | 21.89$_{\pm3.62}$ | 26.95$_{\pm4.13}$ | 24.56$_{\pm5.06}$ | 27.30$_{\pm3.23}$ | 27.96$_{\pm4.51}$ | 27.09$_{\pm3.47}$ | 36.19$_{\pm2.84}$ | **36.26**$_{\pm2.65}$ |
| | | 50 | 58.11$_{\pm4.13}$ | 57.24$_{\pm4.60}$ | 40.77$_{\pm3.59}$ | 43.04$_{\pm5.43}$ | 44.78$_{\pm3.92}$ | 49.43$_{\pm2.51}$ | 46.84$_{\pm5.08}$ | 44.78$_{\pm2.67}$ | 58.62$_{\pm4.41}$ | **58.75**$_{\pm4.48}$ |
| | | 100 | 76.82$_{\pm3.33}$ | 76.78$_{\pm3.10}$ | 62.00$_{\pm3.21}$ | 64.14$_{\pm3.19}$ | 63.24$_{\pm4.05}$ | 69.20$_{\pm2.75}$ | 65.08$_{\pm2.67}$ | 62.45$_{\pm3.22}$ | **77.84**$_{\pm3.49}$ | 77.63$_{\pm3.69}$ |
| | | 200 | 89.53$_{\pm2.34}$ | 90.28$_{\pm2.13}$ | 80.28$_{\pm3.49}$ | 81.94$_{\pm2.56}$ | 81.85$_{\pm2.46}$ | 85.48$_{\pm2.07}$ | 82.57$_{\pm2.19}$ | 78.04$_{\pm2.66}$ | **90.74**$_{\pm2.13}$ | 90.48$_{\pm2.06}$ |
| | | 500 | 98.23$_{\pm0.91}$ | 98.25$_{\pm0.78}$ | 95.08$_{\pm1.34}$ | 95.74$_{\pm0.95}$ | 96.15$_{\pm1.09}$ | 96.01$_{\pm0.86}$ | 96.50$_{\pm0.87}$ | 96.23$_{\pm1.67}$ | **98.52**$_{\pm0.80}$ | 98.50$_{\pm0.70}$ |
| | fourier | 20 | 33.66$_{\pm3.92}$ | N/A | 23.20$_{\pm5.54}$ | 17.08$_{\pm2.75}$ | 19.40$_{\pm4.03}$ | 18.32$_{\pm3.82}$ | 23.26$_{\pm4.21}$ | 19.64$_{\pm2.40}$ | **37.00**$_{\pm2.85}$ | 35.02$_{\pm3.77}$ |
| | | 50 | 53.72$_{\pm1.67}$ | 53.02$_{\pm1.96}$ | 37.16$_{\pm3.08}$ | 37.60$_{\pm4.52}$ | 35.14$_{\pm2.44}$ | 40.90$_{\pm2.80}$ | 42.82$_{\pm2.83}$ | 32.66$_{\pm5.19}$ | **55.40**$_{\pm2.23}$ | 55.34$_{\pm1.40}$ |
| | | 100 | 62.78$_{\pm1.60}$ | 61.44$_{\pm2.74}$ | 43.68$_{\pm3.15}$ | 48.80$_{\pm2.66}$ | 46.18$_{\pm3.96}$ | 56.52$_{\pm5.04}$ | 52.50$_{\pm2.66}$ | 37.74$_{\pm2.99}$ | 63.00$_{\pm1.95}$ | **63.54**$_{\pm1.83}$ |
| | | 200 | 70.18$_{\pm1.85}$ | 70.06$_{\pm2.10}$ | 58.90$_{\pm2.56}$ | 62.36$_{\pm2.86}$ | 58.40$_{\pm2.53}$ | 70.08$_{\pm1.90}$ | 62.14$_{\pm2.00}$ | 50.92$_{\pm5.13}$ | **71.49**$_{\pm1.41}$ | 71.36$_{\pm1.36}$ |
| | | 500 | 77.94$_{\pm1.65}$ | 77.18$_{\pm1.35}$ | 72.14$_{\pm1.79}$ | 74.30$_{\pm1.65}$ | 71.38$_{\pm1.54}$ | 77.78$_{\pm1.26}$ | 74.32$_{\pm1.56}$ | 67.28$_{\pm2.91}$ | 78.34$_{\pm1.72}$ | **79.30**$_{\pm0.99}$ |
| | biodeg | 20 | 71.31$_{\pm5.13}$ | 68.84$_{\pm5.95}$ | 66.64$_{\pm8.27}$ | 62.11$_{\pm4.95}$ | 62.61$_{\pm6.78}$ | 52.96$_{\pm4.22}$ | 62.06$_{\pm3.69}$ | 65.81$_{\pm7.24}$ | 72.04$_{\pm5.12}$ | **72.09**$_{\pm4.81}$ |
| | | 50 | 76.73$_{\pm3.16}$ | 74.97$_{\pm2.51}$ | 72.02$_{\pm4.74}$ | 71.83$_{\pm3.17}$ | 67.86$_{\pm6.02}$ | 69.92$_{\pm4.83}$ | 74.03$_{\pm3.05}$ | 71.01$_{\pm2.78}$ | **77.17**$_{\pm2.93}$ | 77.11$_{\pm3.20}$ |
| | | 100 | **79.13**$_{\pm1.91}$ | 78.20$_{\pm1.68}$ | 76.78$_{\pm2.79}$ | 77.85$_{\pm2.73}$ | 76.01$_{\pm2.88}$ | 76.74$_{\pm3.62}$ | 76.08$_{\pm2.39}$ | 76.24$_{\pm2.45}$ | 78.23$_{\pm2.29}$ | 79.08$_{\pm2.03}$ |
| | | 200 | **82.39**$_{\pm1.48}$ | 81.70$_{\pm1.22}$ | 80.43$_{\pm2.02}$ | 79.96$_{\pm2.35}$ | 79.92$_{\pm1.55}$ | 80.51$_{\pm1.26}$ | 79.59$_{\pm1.72}$ | 80.34$_{\pm1.93}$ | 81.74$_{\pm1.36}$ | 82.24$_{\pm1.54}$ |
| | | 500 | **84.50**$_{\pm0.61}$ | 84.50$_{\pm0.81}$ | 83.78$_{\pm1.51}$ | 83.67$_{\pm0.81}$ | 84.13$_{\pm1.20}$ | 84.09$_{\pm0.84}$ | 83.76$_{\pm1.27}$ | 82.97$_{\pm1.21}$ | 84.37$_{\pm0.48}$ | 84.14$_{\pm0.59}$ |
| | steel | 20 | 62.35$_{\pm6.30}$ | 60.34$_{\pm5.73}$ | 61.63$_{\pm8.82}$ | 59.09$_{\pm4.25}$ | 56.99$_{\pm7.13}$ | 60.67$_{\pm9.18}$ | 55.23$_{\pm3.92}$ | 55.78$_{\pm3.01}$ | **64.49**$_{\pm6.03}$ | 64.22$_{\pm5.89}$ |
| | | 50 | 79.65$_{\pm5.53}$ | 68.18$_{\pm3.16}$ | 69.01$_{\pm3.48}$ | 70.30$_{\pm4.77}$ | 66.96$_{\pm5.12}$ | **84.04**$_{\pm8.03}$ | 64.79$_{\pm4.07}$ | 58.95$_{\pm1.66}$ | 82.72$_{\pm6.02}$ | 82.15$_{\pm5.78}$ |
| | | 100 | 92.18$_{\pm2.93}$ | 78.44$_{\pm3.67}$ | 76.37$_{\pm3.06}$ | 76.92$_{\pm3.63}$ | 76.50$_{\pm3.18}$ | **95.83**$_{\pm1.90}$ | 71.85$_{\pm3.20}$ | 67.35$_{\pm2.51}$ | 95.16$_{\pm3.13}$ | 95.41$_{\pm3.23}$ |
| | | 200 | 97.31$_{\pm1.63}$ | 83.93$_{\pm1.83}$ | 82.42$_{\pm2.75}$ | 84.70$_{\pm2.54}$ | 84.06$_{\pm4.46}$ | 98.75$_{\pm0.68}$ | 79.66$_{\pm3.26}$ | 78.36$_{\pm3.86}$ | 98.83$_{\pm0.80}$ | **98.84**$_{\pm0.67}$ |
| | | 500 | 99.78$_{\pm0.30}$ | 91.37$_{\pm2.26}$ | 93.08$_{\pm1.82}$ | 94.82$_{\pm1.54}$ | 93.99$_{\pm1.83}$ | 99.47$_{\pm0.44}$ | 90.34$_{\pm2.19}$ | 96.06$_{\pm1.03}$ | **99.81**$_{\pm0.24}$ | 99.81$_{\pm0.24}$ |
| | stock | 20 | 83.56$_{\pm3.89}$ | 83.62$_{\pm4.06}$ | 77.25$_{\pm5.02}$ | 69.90$_{\pm9.27}$ | 72.85$_{\pm7.77}$ | 80.60$_{\pm5.73}$ | 69.30$_{\pm6.76}$ | 83.34$_{\pm3.38}$ | 83.81$_{\pm3.92}$ | **83.89**$_{\pm4.05}$ |
| | | 50 | 89.57$_{\pm2.01}$ | 89.71$_{\pm2.21}$ | 82.62$_{\pm3.49}$ | 79.35$_{\pm2.71}$ | 81.52$_{\pm1.99}$ | 88.48$_{\pm2.18}$ | 83.36$_{\pm2.30}$ | 88.38$_{\pm2.41}$ | 90.23$_{\pm2.02}$ | **90.38**$_{\pm2.17}$ |
| | | 100 | 90.63$_{\pm0.83}$ | 91.17$_{\pm0.83}$ | 88.37$_{\pm2.40}$ | 86.60$_{\pm3.27}$ | 83.19$_{\pm3.77}$ | 92.19$_{\pm0.78}$ | 90.65$_{\pm1.39}$ | 91.61$_{\pm0.70}$ | 91.70$_{\pm1.17}$ | **91.75**$_{\pm0.97}$ |
| | | 200 | 91.25$_{\pm0.74}$ | 92.58$_{\pm0.91}$ | 91.27$_{\pm0.95}$ | 90.34$_{\pm1.60}$ | 89.32$_{\pm2.45}$ | 92.19$_{\pm0.78}$ | 90.37$_{\pm1.32}$ | 91.89$_{\pm1.32}$ | **92.91**$_{\pm0.62}$ | 92.47$_{\pm0.63}$ |
| *More than 10 classes* | energy | 50 | **24.79**$_{\pm1.76}$ | N/A | 12.51$_{\pm2.89}$ | 12.61$_{\pm3.45}$ | 8.43$_{\pm2.11}$ | 10.91$_{\pm2.11}$ | 12.45$_{\pm1.87}$ | 20.42$_{\pm4.41}$ | N/A | 24.04$_{\pm1.39}$ |
| | | 100 | 26.86$_{\pm1.51}$ | N/A | 16.20$_{\pm2.12}$ | 15.78$_{\pm2.50}$ | 15.70$_{\pm2.44}$ | 17.75$_{\pm3.18}$ | 18.16$_{\pm2.46}$ | 20.72$_{\pm2.99}$ | N/A | **29.30**$_{\pm2.32}$ |
| | | 200 | 33.36$_{\pm2.98}$ | N/A | 22.30$_{\pm2.44}$ | 23.00$_{\pm4.24}$ | 23.53$_{\pm1.84}$ | 26.12$_{\pm1.78}$ | 26.28$_{\pm3.03}$ | 33.03$_{\pm3.38}$ | N/A | **41.27**$_{\pm2.93}$ |
| | collins | 100 | **14.16**$_{\pm1.31}$ | N/A | 9.24$_{\pm1.71}$ | 9.16$_{\pm1.57}$ | 9.04$_{\pm1.79}$ | 14.03$_{\pm1.24}$ | 8.59$_{\pm1.84}$ | 10.81$_{\pm1.68}$ | N/A | 14.07$_{\pm1.58}$ |
| | | 200 | 19.35$_{\pm1.24}$ | 19.06$_{\pm1.49}$ | 14.62$_{\pm2.00}$ | 13.17$_{\pm0.94}$ | 12.38$_{\pm1.56}$ | 18.63$_{\pm1.56}$ | 12.48$_{\pm1.91}$ | 17.65$_{\pm1.76}$ | N/A | **19.53**$_{\pm1.44}$ |
| | texture | 50 | 84.50$_{\pm2.81}$ | 84.12$_{\pm3.02}$ | 62.92$_{\pm4.09}$ | 67.69$_{\pm5.49}$ | 61.99$_{\pm3.58}$ | 69.69$_{\pm4.62}$ | 64.45$_{\pm7.84}$ | 69.68$_{\pm3.53}$ | N/A | **85.51**$_{\pm2.89}$ |
| | | 100 | 91.50$_{\pm1.34}$ | 91.57$_{\pm1.59}$ | 74.53$_{\pm3.39}$ | 79.96$_{\pm4.37}$ | 80.36$_{\pm4.58}$ | 85.42$_{\pm2.74}$ | 80.23$_{\pm2.18}$ | 85.59$_{\pm1.49}$ | N/A | **92.17**$_{\pm1.31}$ |
| | | 200 | 93.81$_{\pm1.35}$ | 94.18$_{\pm1.26}$ | 86.57$_{\pm2.33}$ | 90.68$_{\pm1.55}$ | 88.97$_{\pm2.12}$ | 90.10$_{\pm2.26}$ | 89.14$_{\pm1.85}$ | 91.66$_{\pm1.43}$ | N/A | **94.35**$_{\pm1.57}$ |
| | | 500 | 96.55$_{\pm0.63}$ | **97.21**$_{\pm0.40}$ | 94.66$_{\pm1.17}$ | 96.27$_{\pm0.74}$ | 94.34$_{\pm1.36}$ | 94.72$_{\pm0.61}$ | 95.83$_{\pm1.03}$ | 96.49$_{\pm0.48}$ | N/A | 97.13$_{\pm0.53}$ |
| **Average rank** | | | 3.00$_{\pm1.32}$ | 4.06$_{\pm1.62}$ | 7.82$_{\pm1.63}$ | 7.48$_{\pm1.50}$ | 8.45$_{\pm1.33}$ | 5.55$_{\pm2.14}$ | 7.48$_{\pm1.72}$ | 6.82$_{\pm2.57}$ | 2.67$_{\pm1.69}$ | **1.67**$_{\pm0.74}$ |

Table 10: **Classification accuracy** (%) of RF, comparing data augmentation on eight real-world tabular datasets with varied real data availability. We report the mean ± std balanced accuracy and average accuracy rank across datasets. A higher rank implies higher accuracy. Note that "N/A" denotes that a specific generator was not applicable or the downstream predictor failed to converge, and the rank is computed with the mean balanced accuracy of other methods. We **bold** the highest accuracy for each dataset of different sample sizes. TabEBM achieves the best overall performance against Baseline and benchmark generators.

| | Datasets | $N_{\text{real}}$ | Baseline (Real data) | SMOTE | TVAE | CTGAN | NFLOW | TabDDPM | ARF | GOGGLE | TabPFGen | **TabEBM** |
|---|---|---|---|---|---|---|---|---|---|---|---|---|
| *At most 10 classes* | protein | 20 | $28.52_{\pm2.19}$ | N/A | $22.74_{\pm4.23}$ | $24.94_{\pm5.15}$ | $24.61_{\pm2.46}$ | $29.62_{\pm4.06}$ | $27.69_{\pm3.73}$ | $25.76_{\pm1.62}$ | $32.04_{\pm2.40}$ | $\mathbf{34.19}_{\pm2.21}$ |
| | | 50 | $53.40_{\pm3.26}$ | $55.69_{\pm2.61}$ | $46.95_{\pm3.13}$ | $43.91_{\pm4.98}$ | $43.28_{\pm4.07}$ | $47.93_{\pm4.49}$ | $44.48_{\pm3.35}$ | $47.25_{\pm4.81}$ | $54.29_{\pm2.57}$ | $\mathbf{56.85}_{\pm2.49}$ |
| | | 100 | $68.13_{\pm3.19}$ | $\mathbf{72.89}_{\pm2.60}$ | $63.24_{\pm1.78}$ | $61.19_{\pm2.10}$ | $59.64_{\pm3.48}$ | $65.19_{\pm3.22}$ | $60.05_{\pm2.88}$ | $65.05_{\pm3.00}$ | $71.47_{\pm3.61}$ | $72.57_{\pm2.50}$ |
| | | 200 | $80.34_{\pm2.35}$ | $83.60_{\pm2.71}$ | $78.51_{\pm2.58}$ | $75.84_{\pm1.61}$ | $76.84_{\pm2.35}$ | $78.74_{\pm2.24}$ | $75.61_{\pm2.90}$ | $79.44_{\pm2.73}$ | $83.36_{\pm2.40}$ | $\mathbf{84.30}_{\pm1.97}$ |
| | | 500 | $93.01_{\pm1.12}$ | $93.82_{\pm0.67}$ | $92.86_{\pm1.66}$ | $91.37_{\pm1.25}$ | $93.00_{\pm1.08}$ | $92.93_{\pm0.97}$ | $92.38_{\pm0.90}$ | $92.95_{\pm0.92}$ | $\mathbf{94.49}_{\pm1.16}$ | $93.94_{\pm1.23}$ |
| | fourier | 20 | $35.10_{\pm4.56}$ | N/A | $19.06_{\pm3.91}$ | $17.52_{\pm2.84}$ | $20.78_{\pm2.54}$ | $16.98_{\pm2.31}$ | $23.78_{\pm3.12}$ | $19.00_{\pm2.93}$ | $34.88_{\pm5.93}$ | $\mathbf{38.60}_{\pm5.66}$ |
| | | 50 | $64.10_{\pm3.80}$ | $64.76_{\pm4.00}$ | $37.20_{\pm3.35}$ | $32.82_{\pm4.56}$ | $37.78_{\pm3.11}$ | $51.76_{\pm3.50}$ | $47.22_{\pm4.35}$ | $53.86_{\pm3.61}$ | $\mathbf{66.92}_{\pm3.05}$ | $66.26_{\pm3.16}$ |
| | | 100 | $73.86_{\pm3.06}$ | $73.78_{\pm3.22}$ | $64.40_{\pm2.51}$ | $60.82_{\pm3.71}$ | $51.64_{\pm4.16}$ | $66.14_{\pm1.91}$ | $58.62_{\pm3.73}$ | $68.16_{\pm3.12}$ | $73.13_{\pm2.70}$ | $\mathbf{74.84}_{\pm3.10}$ |
| | | 200 | $78.54_{\pm2.15}$ | $79.18_{\pm1.92}$ | $74.86_{\pm1.60}$ | $74.26_{\pm2.20}$ | $69.36_{\pm2.61}$ | $76.42_{\pm1.95}$ | $72.88_{\pm1.22}$ | $76.64_{\pm1.99}$ | $\mathbf{82.20}_{\pm0.85}$ | $79.18_{\pm2.08}$ |
| | | 500 | $81.84_{\pm1.01}$ | $82.14_{\pm1.49}$ | $81.02_{\pm1.59}$ | $81.18_{\pm1.43}$ | $80.08_{\pm1.62}$ | $81.26_{\pm1.40}$ | $80.28_{\pm1.54}$ | $80.62_{\pm1.52}$ | $81.45_{\pm1.45}$ | $\mathbf{83.40}_{\pm1.24}$ |
| | biodeg | 20 | $61.11_{\pm7.87}$ | $\mathbf{68.38}_{\pm5.90}$ | $65.44_{\pm8.89}$ | $56.29_{\pm7.96}$ | $58.19_{\pm6.60}$ | $52.90_{\pm4.74}$ | $62.33_{\pm6.14}$ | $63.52_{\pm7.29}$ | $67.15_{\pm5.74}$ | $67.82_{\pm5.13}$ |
| | | 50 | $68.38_{\pm4.82}$ | $70.64_{\pm3.44}$ | $71.77_{\pm2.99}$ | $66.78_{\pm4.89}$ | $61.39_{\pm4.94}$ | $63.98_{\pm3.65}$ | $68.78_{\pm5.22}$ | $70.34_{\pm3.39}$ | $71.38_{\pm3.60}$ | $\mathbf{72.12}_{\pm3.29}$ |
| | | 100 | $73.19_{\pm2.46}$ | $75.36_{\pm2.56}$ | $74.98_{\pm2.58}$ | $72.68_{\pm2.98}$ | $69.62_{\pm3.53}$ | $73.11_{\pm2.39}$ | $72.16_{\pm2.58}$ | $74.22_{\pm2.32}$ | $\mathbf{75.85}_{\pm1.56}$ | $75.65_{\pm1.53}$ |
| | | 200 | $77.85_{\pm2.72}$ | $78.86_{\pm1.97}$ | $76.42_{\pm2.25}$ | $76.68_{\pm2.77}$ | $73.43_{\pm3.01}$ | $76.16_{\pm2.00}$ | $75.79_{\pm2.49}$ | $77.42_{\pm2.24}$ | $\mathbf{79.68}_{\pm1.74}$ | $79.22_{\pm1.70}$ |
| | | 500 | $81.42_{\pm0.73}$ | $82.03_{\pm1.02}$ | $81.88_{\pm0.87}$ | $81.71_{\pm1.54}$ | $80.50_{\pm1.21}$ | $81.43_{\pm1.26}$ | $81.34_{\pm1.58}$ | $81.94_{\pm0.85}$ | $\mathbf{82.38}_{\pm1.35}$ | $82.10_{\pm1.31}$ |
| | steel | 20 | $52.77_{\pm1.60}$ | $56.16_{\pm4.50}$ | $57.23_{\pm3.97}$ | $54.65_{\pm3.40}$ | $53.75_{\pm3.49}$ | $51.70_{\pm1.66}$ | $54.09_{\pm4.36}$ | $55.50_{\pm2.97}$ | $57.04_{\pm3.07}$ | $\mathbf{57.41}_{\pm2.67}$ |
| | | 50 | $59.75_{\pm3.11}$ | $62.12_{\pm2.46}$ | $60.65_{\pm1.96}$ | $58.09_{\pm1.75}$ | $54.69_{\pm2.44}$ | $58.14_{\pm4.21}$ | $57.67_{\pm2.52}$ | $60.34_{\pm2.90}$ | $65.07_{\pm3.11}$ | $\mathbf{67.74}_{\pm3.36}$ |
| | | 100 | $64.97_{\pm2.05}$ | $69.08_{\pm3.62}$ | $64.46_{\pm4.17}$ | $61.62_{\pm1.98}$ | $58.43_{\pm2.46}$ | $60.53_{\pm3.64}$ | $62.71_{\pm3.43}$ | $63.07_{\pm2.23}$ | $73.28_{\pm3.39}$ | $\mathbf{79.63}_{\pm3.41}$ |
| | | 200 | $75.45_{\pm3.26}$ | $74.71_{\pm3.79}$ | $71.45_{\pm2.18}$ | $68.52_{\pm3.80}$ | $62.15_{\pm2.80}$ | $68.10_{\pm3.59}$ | $67.61_{\pm1.81}$ | $67.36_{\pm1.63}$ | $85.12_{\pm4.44}$ | $\mathbf{88.85}_{\pm5.10}$ |
| | | 500 | $90.93_{\pm2.83}$ | $85.37_{\pm2.36}$ | $85.63_{\pm3.14}$ | $84.51_{\pm3.22}$ | $76.12_{\pm2.70}$ | $89.19_{\pm3.20}$ | $81.44_{\pm2.64}$ | $80.35_{\pm3.54}$ | $94.35_{\pm1.34}$ | $\mathbf{95.90}_{\pm1.06}$ |
| | stock | 20 | $79.47_{\pm5.83}$ | $81.99_{\pm4.49}$ | $77.94_{\pm5.21}$ | $72.53_{\pm7.16}$ | $73.20_{\pm9.98}$ | $80.99_{\pm7.01}$ | $72.57_{\pm8.76}$ | $78.10_{\pm5.91}$ | $83.96_{\pm5.57}$ | $\mathbf{84.73}_{\pm3.46}$ |
| | | 50 | $87.57_{\pm2.60}$ | $89.69_{\pm1.99}$ | $86.62_{\pm3.44}$ | $83.75_{\pm4.32}$ | $84.28_{\pm2.98}$ | $88.69_{\pm2.11}$ | $84.92_{\pm2.09}$ | $88.65_{\pm2.55}$ | $89.35_{\pm2.18}$ | $\mathbf{89.99}_{\pm2.63}$ |
| | | 100 | $91.44_{\pm1.59}$ | $91.47_{\pm2.16}$ | $91.07_{\pm2.08}$ | $89.82_{\pm2.69}$ | $89.33_{\pm1.92}$ | $91.33_{\pm2.07}$ | $90.48_{\pm2.38}$ | $92.00_{\pm2.36}$ | $92.07_{\pm1.22}$ | $\mathbf{92.17}_{\pm1.24}$ |
| | | 200 | $93.52_{\pm0.80}$ | $\mathbf{93.94}_{\pm1.09}$ | $93.35_{\pm1.05}$ | $92.62_{\pm1.02}$ | $92.77_{\pm1.25}$ | $93.65_{\pm1.08}$ | $93.08_{\pm0.53}$ | $93.87_{\pm1.25}$ | $93.65_{\pm1.02}$ | $93.67_{\pm1.07}$ |
| *More than 10 classes* | energy | 50 | $18.96_{\pm1.40}$ | N/A | $16.63_{\pm2.27}$ | $15.66_{\pm2.43}$ | $14.81_{\pm3.16}$ | $14.49_{\pm1.26}$ | $15.05_{\pm3.06}$ | $15.58_{\pm3.26}$ | N/A | $\mathbf{27.74}_{\pm3.71}$ |
| | | 100 | $30.85_{\pm2.19}$ | N/A | $24.59_{\pm2.27}$ | $28.59_{\pm2.63}$ | $27.59_{\pm2.86}$ | $27.23_{\pm2.39}$ | $27.99_{\pm2.18}$ | $25.43_{\pm2.46}$ | N/A | $\mathbf{41.03}_{\pm2.24}$ |
| | | 200 | $45.80_{\pm2.32}$ | N/A | $42.10_{\pm2.57}$ | $41.69_{\pm3.84}$ | $44.41_{\pm2.51}$ | $44.58_{\pm1.37}$ | $41.33_{\pm3.90}$ | $44.64_{\pm2.54}$ | N/A | $\mathbf{53.87}_{\pm2.81}$ |
| | collins | 100 | $10.41_{\pm1.61}$ | N/A | $6.75_{\pm0.69}$ | $8.23_{\pm1.76}$ | $7.34_{\pm1.46}$ | $12.84_{\pm1.61}$ | $6.73_{\pm1.36}$ | $8.43_{\pm0.94}$ | N/A | $\mathbf{13.35}_{\pm1.49}$ |
| | | 200 | $13.75_{\pm1.12}$ | $\mathbf{17.56}_{\pm1.79}$ | $10.51_{\pm1.41}$ | $11.00_{\pm1.37}$ | $9.85_{\pm1.38}$ | $15.15_{\pm1.22}$ | $9.90_{\pm0.72}$ | $13.40_{\pm1.09}$ | N/A | $16.51_{\pm1.53}$ |
| | texture | 50 | $71.27_{\pm1.99}$ | $71.17_{\pm3.89}$ | $57.41_{\pm3.33}$ | $62.78_{\pm4.21}$ | $65.24_{\pm4.52}$ | $69.45_{\pm2.15}$ | $62.93_{\pm4.84}$ | $64.33_{\pm3.57}$ | N/A | $\mathbf{75.79}_{\pm3.07}$ |
| | | 100 | $80.40_{\pm2.45}$ | $80.38_{\pm2.67}$ | $65.63_{\pm4.21}$ | $75.38_{\pm3.99}$ | $77.67_{\pm2.62}$ | $79.31_{\pm1.88}$ | $75.98_{\pm2.56}$ | $77.30_{\pm2.44}$ | N/A | $\mathbf{82.30}_{\pm2.21}$ |
| | | 200 | $84.00_{\pm1.56}$ | $85.12_{\pm3.07}$ | $76.98_{\pm2.25}$ | $84.44_{\pm2.41}$ | $85.30_{\pm1.96}$ | $84.00_{\pm1.20}$ | $83.70_{\pm2.05}$ | $80.02_{\pm1.60}$ | N/A | $\mathbf{85.92}_{\pm2.18}$ |
| | | 500 | $89.43_{\pm0.80}$ | $90.17_{\pm1.25}$ | $88.97_{\pm1.44}$ | $90.00_{\pm1.66}$ | $89.99_{\pm1.06}$ | $90.17_{\pm1.32}$ | $\mathbf{91.01}_{\pm1.32}$ | $88.98_{\pm1.26}$ | N/A | $90.77_{\pm1.10}$ |
| **Average rank** | | | $4.36_{\pm1.95}$ | $3.02_{\pm1.14}$ | $6.88_{\pm2.25}$ | $7.82_{\pm1.61}$ | $8.45_{\pm1.95}$ | $6.12_{\pm2.23}$ | $7.85_{\pm1.77}$ | $6.00_{\pm1.75}$ | $3.12_{\pm1.75}$ | $\mathbf{1.38}_{\pm0.57}$ |

Table 11: **Classification accuracy** (%) of XGBoost, comparing data augmentation on eight real-world tabular datasets with varied real data availability. We report the mean ± std balanced accuracy and average accuracy rank across datasets. A higher rank implies higher accuracy. Note that "N/A" denotes that a specific generator was not applicable or the downstream predictor failed to converge, and the rank is computed with the mean balanced accuracy of other methods. We **bold** the highest accuracy for each dataset of different sample sizes. TabEBM achieves the best overall performance against Baseline and benchmark generators.

| Datasets | | $N_{\text{real}}$ | Baseline (Real data) | SMOTE | TVAE | CTGAN | NFLOW | TabDDPM | ARF | GOGGLE | TabPFGen | **TabEBM** |
|---|---|---|---|---|---|---|---|---|---|---|---|---|
| *At most 10 classes* | protein | 20 | $19.70_{\pm6.33}$ | N/A | $19.44_{\pm4.11}$ | $17.32_{\pm2.75}$ | $18.11_{\pm3.07}$ | $16.15_{\pm3.80}$ | $20.71_{\pm5.24}$ | $17.40_{\pm4.89}$ | $24.00_{\pm3.64}$ | $\mathbf{24.18}_{\pm3.05}$ |
| | | 50 | $39.01_{\pm4.92}$ | $37.68_{\pm5.40}$ | $33.07_{\pm4.18}$ | $24.38_{\pm3.45}$ | $23.09_{\pm4.55}$ | $30.87_{\pm5.70}$ | $34.13_{\pm6.45}$ | $33.62_{\pm3.67}$ | $39.78_{\pm6.03}$ | $\mathbf{44.46}_{\pm4.97}$ |
| | | 100 | $57.59_{\pm3.69}$ | $60.16_{\pm5.75}$ | $49.23_{\pm5.51}$ | $43.33_{\pm7.92}$ | $37.69_{\pm5.96}$ | $48.36_{\pm4.08}$ | $43.97_{\pm5.45}$ | $47.00_{\pm3.29}$ | $53.74_{\pm7.94}$ | $\mathbf{62.77}_{\pm5.85}$ |
| | | 200 | $74.05_{\pm2.92}$ | $76.90_{\pm4.96}$ | $69.71_{\pm4.28}$ | $67.46_{\pm4.39}$ | $58.29_{\pm7.96}$ | $69.68_{\pm4.23}$ | $63.69_{\pm6.32}$ | $66.09_{\pm4.78}$ | $73.19_{\pm6.06}$ | $\mathbf{79.25}_{\pm3.83}$ |
| | | 500 | $88.89_{\pm1.71}$ | $90.02_{\pm1.51}$ | $90.10_{\pm1.80}$ | $89.37_{\pm1.81}$ | $86.03_{\pm2.31}$ | $87.29_{\pm2.08}$ | $90.05_{\pm2.70}$ | $85.04_{\pm2.07}$ | $89.66_{\pm1.17}$ | $\mathbf{91.81}_{\pm1.44}$ |
| | fourier | 20 | $10.00_{\pm0.00}$ | N/A | $14.64_{\pm3.13}$ | $13.58_{\pm2.57}$ | $13.82_{\pm4.14}$ | $11.72_{\pm4.19}$ | $16.38_{\pm3.86}$ | $12.34_{\pm3.59}$ | $23.50_{\pm1.56}$ | $\mathbf{26.78}_{\pm4.82}$ |
| | | 50 | $42.10_{\pm6.19}$ | $43.40_{\pm5.22}$ | $34.32_{\pm3.98}$ | $24.68_{\pm6.47}$ | $17.66_{\pm4.64}$ | $24.82_{\pm6.35}$ | $27.74_{\pm5.86}$ | $35.42_{\pm7.51}$ | $35.60_{\pm3.11}$ | $\mathbf{45.08}_{\pm6.47}$ |
| | | 100 | $54.84_{\pm2.78}$ | $52.92_{\pm5.69}$ | $48.22_{\pm3.28}$ | $36.90_{\pm5.15}$ | $30.36_{\pm3.94}$ | $42.46_{\pm4.13}$ | $40.28_{\pm3.41}$ | $48.78_{\pm4.36}$ | $49.80_{\pm1.98}$ | $\mathbf{54.94}_{\pm5.72}$ |
| | | 200 | $63.88_{\pm3.35}$ | $65.34_{\pm3.57}$ | $58.36_{\pm3.27}$ | $53.20_{\pm5.26}$ | $46.96_{\pm4.58}$ | $61.40_{\pm4.12}$ | $52.10_{\pm3.32}$ | $56.66_{\pm2.67}$ | $66.60_{\pm4.24}$ | $\mathbf{67.68}_{\pm3.19}$ |
| | | 500 | $74.56_{\pm1.97}$ | $74.18_{\pm2.10}$ | $68.28_{\pm2.82}$ | $67.98_{\pm2.07}$ | $61.24_{\pm2.35}$ | $72.78_{\pm2.54}$ | $67.50_{\pm2.57}$ | $68.28_{\pm3.43}$ | N/A | $\mathbf{76.25}_{\pm3.18}$ |
| | biodeg | 20 | $62.95_{\pm7.95}$ | $66.51_{\pm5.84}$ | $62.72_{\pm5.69}$ | $55.24_{\pm6.28}$ | $59.20_{\pm7.83}$ | $54.65_{\pm5.56}$ | $62.78_{\pm5.98}$ | $61.09_{\pm10.49}$ | $65.52_{\pm6.08}$ | $\mathbf{66.64}_{\pm6.71}$ |
| | | 50 | $67.96_{\pm3.45}$ | $67.69_{\pm4.42}$ | $66.22_{\pm5.70}$ | $61.64_{\pm6.73}$ | $60.72_{\pm5.73}$ | $57.48_{\pm8.28}$ | $\mathbf{69.48}_{\pm5.35}$ | $65.93_{\pm4.98}$ | $67.76_{\pm4.90}$ | $67.90_{\pm3.27}$ |
| | | 100 | $\mathbf{73.88}_{\pm2.55}$ | $72.05_{\pm4.75}$ | $72.11_{\pm3.17}$ | $70.41_{\pm3.60}$ | $66.02_{\pm6.25}$ | $69.35_{\pm4.66}$ | $71.11_{\pm3.88}$ | $69.03_{\pm4.33}$ | $72.58_{\pm2.91}$ | $71.05_{\pm5.70}$ |
| | | 200 | $76.38_{\pm4.85}$ | $74.98_{\pm3.15}$ | $73.93_{\pm3.29}$ | $75.68_{\pm4.15}$ | $67.82_{\pm3.91}$ | $72.58_{\pm5.07}$ | $74.74_{\pm2.24}$ | $73.84_{\pm3.82}$ | $75.85_{\pm1.80}$ | $\mathbf{76.74}_{\pm2.44}$ |
| | | 500 | $78.45_{\pm3.37}$ | $79.38_{\pm1.99}$ | $78.88_{\pm3.42}$ | $\mathbf{80.15}_{\pm1.87}$ | $76.72_{\pm3.44}$ | $77.10_{\pm2.96}$ | $78.14_{\pm2.65}$ | $78.83_{\pm2.21}$ | $79.40_{\pm1.49}$ | $78.80_{\pm3.76}$ |
| | steel | 20 | $53.12_{\pm5.62}$ | $55.64_{\pm4.76}$ | $53.32_{\pm7.25}$ | $55.36_{\pm6.24}$ | $52.38_{\pm3.55}$ | $52.44_{\pm4.08}$ | $51.34_{\pm4.15}$ | $50.74_{\pm2.53}$ | $55.43_{\pm5.57}$ | $\mathbf{55.78}_{\pm4.53}$ |
| | | 50 | $66.73_{\pm9.11}$ | $60.79_{\pm5.52}$ | $59.51_{\pm4.15}$ | $54.82_{\pm4.23}$ | $54.79_{\pm4.69}$ | $59.71_{\pm6.94}$ | $57.66_{\pm5.19}$ | $55.89_{\pm4.50}$ | $63.78_{\pm7.20}$ | $\mathbf{74.18}_{\pm13.67}$ |
| | | 100 | $83.17_{\pm9.36}$ | $66.95_{\pm6.51}$ | $61.72_{\pm6.80}$ | $65.12_{\pm3.02}$ | $60.56_{\pm4.37}$ | $72.02_{\pm12.47}$ | $59.67_{\pm4.77}$ | $59.04_{\pm4.76}$ | $90.52_{\pm7.47}$ | $\mathbf{96.55}_{\pm2.66}$ |
| | | 200 | $95.94_{\pm2.73}$ | $81.21_{\pm5.01}$ | $73.14_{\pm5.45}$ | $70.64_{\pm10.67}$ | $70.26_{\pm9.25}$ | $74.50_{\pm23.57}$ | $74.57_{\pm9.36}$ | $65.41_{\pm6.70}$ | $99.14_{\pm1.19}$ | $\mathbf{99.54}_{\pm0.62}$ |
| | | 500 | $99.95_{\pm0.10}$ | $97.04_{\pm2.14}$ | $95.27_{\pm2.88}$ | $89.46_{\pm6.88}$ | $83.25_{\pm8.10}$ | $91.72_{\pm15.34}$ | $87.59_{\pm6.72}$ | $79.54_{\pm15.29}$ | $100.00_{\pm0.00}$ | $\mathbf{100.00}_{\pm0.00}$ |
| | stock | 20 | $76.42_{\pm4.34}$ | $78.92_{\pm5.21}$ | $67.46_{\pm13.93}$ | $60.56_{\pm9.69}$ | $73.36_{\pm9.57}$ | $77.45_{\pm9.80}$ | $69.15_{\pm9.35}$ | $70.88_{\pm8.52}$ | $79.82_{\pm4.52}$ | $\mathbf{83.44}_{\pm3.74}$ |
| | | 50 | $83.71_{\pm3.40}$ | $86.23_{\pm2.54}$ | $84.65_{\pm4.44}$ | $79.31_{\pm6.58}$ | $76.27_{\pm3.89}$ | $85.70_{\pm3.96}$ | $81.61_{\pm1.97}$ | $84.98_{\pm4.44}$ | $87.28_{\pm3.65}$ | $\mathbf{88.21}_{\pm3.31}$ |
| | | 100 | $88.19_{\pm3.04}$ | $89.01_{\pm2.07}$ | $85.66_{\pm6.01}$ | $84.68_{\pm2.87}$ | $82.50_{\pm3.73}$ | $\mathbf{90.07}_{\pm3.41}$ | $86.09_{\pm4.08}$ | $84.67_{\pm7.29}$ | $90.01_{\pm3.46}$ | $89.66_{\pm3.28}$ |
| | | 200 | $\mathbf{92.32}_{\pm1.35}$ | $92.26_{\pm2.33}$ | $90.94_{\pm1.98}$ | $89.01_{\pm2.53}$ | $88.92_{\pm2.67}$ | $91.36_{\pm3.79}$ | $91.04_{\pm1.46}$ | $91.42_{\pm2.66}$ | $91.72_{\pm2.77}$ | $92.17_{\pm1.51}$ |
| *More than 10 classes* | energy | 50 | $12.05_{\pm2.42}$ | N/A | $11.60_{\pm3.83}$ | $14.47_{\pm5.32}$ | $10.95_{\pm4.68}$ | $10.21_{\pm5.11}$ | $12.81_{\pm2.51}$ | $12.34_{\pm3.55}$ | N/A | $\mathbf{21.07}_{\pm3.99}$ |
| | | 100 | $\mathbf{29.37}_{\pm1.72}$ | N/A | $20.61_{\pm5.39}$ | $19.81_{\pm4.52}$ | $22.71_{\pm6.15}$ | $22.27_{\pm2.12}$ | $22.02_{\pm3.54}$ | $10.01_{\pm3.40}$ | N/A | $27.93_{\pm4.16}$ |
| | | 200 | $\mathbf{44.96}_{\pm3.31}$ | N/A | $36.73_{\pm6.03}$ | $35.92_{\pm8.45}$ | $33.71_{\pm6.54}$ | $34.73_{\pm5.89}$ | $37.06_{\pm5.26}$ | $18.81_{\pm7.27}$ | N/A | $40.95_{\pm5.59}$ |
| | collins | 100 | $7.77_{\pm2.21}$ | N/A | $7.76_{\pm0.95}$ | $6.52_{\pm1.16}$ | $6.11_{\pm1.09}$ | $\mathbf{8.95}_{\pm1.90}$ | $6.21_{\pm1.14}$ | $5.96_{\pm1.07}$ | N/A | $8.73_{\pm1.64}$ |
| | | 200 | $10.58_{\pm2.57}$ | $11.46_{\pm2.11}$ | $9.43_{\pm2.20}$ | $9.84_{\pm1.56}$ | $8.26_{\pm1.75}$ | $9.80_{\pm1.96}$ | $8.90_{\pm0.83}$ | $9.79_{\pm0.80}$ | N/A | $\mathbf{11.72}_{\pm1.34}$ |
| | texture | 50 | $56.72_{\pm6.12}$ | $60.99_{\pm4.35}$ | $45.76_{\pm6.50}$ | $39.50_{\pm6.46}$ | $43.02_{\pm6.12}$ | $50.22_{\pm6.28}$ | $43.71_{\pm5.98}$ | $46.21_{\pm7.95}$ | N/A | $\mathbf{69.11}_{\pm3.27}$ |
| | | 100 | $68.96_{\pm2.59}$ | $69.77_{\pm4.63}$ | $54.95_{\pm5.99}$ | $55.52_{\pm7.80}$ | $63.23_{\pm4.80}$ | $65.59_{\pm3.62}$ | $57.04_{\pm6.59}$ | $62.06_{\pm6.11}$ | N/A | $\mathbf{76.35}_{\pm2.64}$ |
| | | 200 | $77.91_{\pm1.98}$ | $81.55_{\pm2.22}$ | $70.70_{\pm4.40}$ | $71.60_{\pm4.19}$ | $73.76_{\pm5.69}$ | $77.06_{\pm2.17}$ | $72.56_{\pm4.09}$ | $70.31_{\pm6.55}$ | N/A | $\mathbf{82.59}_{\pm2.15}$ |
| | | 500 | $89.37_{\pm1.11}$ | $\mathbf{89.87}_{\pm1.24}$ | $85.06_{\pm2.40}$ | $86.80_{\pm2.25}$ | $86.83_{\pm1.89}$ | $86.52_{\pm1.66}$ | $85.70_{\pm2.75}$ | $87.07_{\pm2.43}$ | N/A | $89.69_{\pm1.10}$ |
| **Average rank** | | | $3.64_{\pm2.09}$ | $3.45_{\pm1.48}$ | $6.32_{\pm1.86}$ | $7.33_{\pm2.19}$ | $8.64_{\pm1.82}$ | $6.30_{\pm2.44}$ | $6.64_{\pm2.18}$ | $7.62_{\pm1.84}$ | $3.44_{\pm1.54}$ | $\mathbf{1.62}_{\pm1.29}$ |

Table 12: **Classification accuracy** (%) of TabPFN, comparing data augmentation on eight real-world tabular datasets with varied real data availability. We report the mean ± std balanced accuracy and average accuracy rank across datasets. A higher rank implies higher accuracy. Note that "N/A" denotes that a specific generator was not applicable or the downstream predictor failed to converge, and the rank is computed with the mean balanced accuracy of other methods. We **bold** the highest accuracy for each dataset of different sample sizes. TabEBM achieves the best overall performance against Baseline and benchmark generators.

| Datasets | | $N_{real}$ | Baseline (Real data) | SMOTE | TVAE | CTGAN | NFLOW | TabDDPM | ARF | GOGGLE | TabPFGen | **TabEBM** |
|---|---|---|---|---|---|---|---|---|---|---|---|---|
| *At most 10 classes* | protein | 20 | $27.80_{\pm4.37}$ | N/A | $19.21_{\pm3.80}$ | $20.58_{\pm4.63}$ | $20.80_{\pm4.34}$ | $18.89_{\pm4.37}$ | $23.97_{\pm3.05}$ | $10.55_{\pm1.61}$ | $33.42_{\pm5.95}$ | $\mathbf{34.63}_{\pm5.78}$ |
| | | 50 | $55.24_{\pm3.46}$ | $\mathbf{59.85}_{\pm3.87}$ | $43.58_{\pm6.20}$ | $37.37_{\pm8.01}$ | $34.42_{\pm6.65}$ | $21.70_{\pm7.43}$ | $46.02_{\pm2.60}$ | $13.54_{\pm4.22}$ | $57.63_{\pm2.82}$ | $58.88_{\pm3.99}$ |
| | | 100 | $74.31_{\pm3.49}$ | $\mathbf{80.05}_{\pm3.16}$ | $68.15_{\pm5.54}$ | $71.10_{\pm2.55}$ | $57.89_{\pm6.13}$ | $59.28_{\pm8.23}$ | $65.84_{\pm3.03}$ | $23.69_{\pm10.40}$ | $77.60_{\pm4.03}$ | $78.26_{\pm3.75}$ |
| | | 200 | $88.67_{\pm1.53}$ | $\mathbf{91.79}_{\pm1.42}$ | $87.05_{\pm2.85}$ | $86.69_{\pm2.85}$ | $83.29_{\pm2.42}$ | $87.39_{\pm2.95}$ | $85.49_{\pm2.49}$ | $77.63_{\pm6.11}$ | $90.77_{\pm1.37}$ | $90.94_{\pm1.46}$ |
| | | 500 | $97.31_{\pm0.69}$ | $\mathbf{97.69}_{\pm0.77}$ | $97.51_{\pm0.85}$ | $97.58_{\pm0.85}$ | $96.89_{\pm0.62}$ | $97.44_{\pm0.85}$ | $97.40_{\pm0.60}$ | $97.35_{\pm0.61}$ | $97.24_{\pm0.80}$ | $97.28_{\pm0.62}$ |
| | fourier | 20 | $30.06_{\pm6.85}$ | N/A | $22.00_{\pm4.62}$ | $20.10_{\pm4.31}$ | $14.52_{\pm3.96}$ | $12.22_{\pm2.40}$ | $21.64_{\pm5.91}$ | $14.64_{\pm3.94}$ | N/A | $\mathbf{36.56}_{\pm4.96}$ |
| | | 50 | $53.62_{\pm4.71}$ | $53.08_{\pm3.34}$ | $45.82_{\pm4.29}$ | $37.46_{\pm5.82}$ | $28.78_{\pm2.78}$ | $22.74_{\pm5.11}$ | $42.14_{\pm3.09}$ | $11.30_{\pm1.50}$ | $53.15_{\pm3.50}$ | $\mathbf{53.82}_{\pm3.92}$ |
| | | 100 | $64.62_{\pm4.14}$ | $63.66_{\pm3.92}$ | $56.68_{\pm3.02}$ | $54.78_{\pm2.80}$ | $45.50_{\pm4.50}$ | $49.36_{\pm8.51}$ | $54.74_{\pm2.78}$ | $21.40_{\pm4.29}$ | $\mathbf{65.95}_{\pm3.49}$ | $65.40_{\pm3.61}$ |
| | | 200 | $71.62_{\pm2.59}$ | $70.56_{\pm3.61}$ | $66.48_{\pm3.82}$ | $66.14_{\pm4.02}$ | $62.64_{\pm2.60}$ | $72.12_{\pm2.64}$ | $65.04_{\pm3.20}$ | $52.18_{\pm7.35}$ | $69.93_{\pm3.91}$ | $\mathbf{72.48}_{\pm3.08}$ |
| | | 500 | $77.66_{\pm1.61}$ | $77.50_{\pm1.08}$ | $76.80_{\pm1.24}$ | $77.82_{\pm1.24}$ | $73.90_{\pm1.76}$ | $\mathbf{79.16}_{\pm2.05}$ | $75.70_{\pm2.11}$ | $74.36_{\pm2.53}$ | $77.30_{\pm0.42}$ | $77.40_{\pm1.28}$ |
| | biodeg | 20 | $65.26_{\pm8.01}$ | $68.72_{\pm4.50}$ | $69.02_{\pm5.37}$ | $59.39_{\pm6.25}$ | $58.28_{\pm8.30}$ | $50.00_{\pm0.00}$ | $58.45_{\pm8.16}$ | $51.80_{\pm4.07}$ | $70.68_{\pm4.94}$ | $\mathbf{71.18}_{\pm5.25}$ |
| | | 50 | $75.27_{\pm2.63}$ | $74.65_{\pm3.28}$ | $73.44_{\pm4.02}$ | $70.21_{\pm3.61}$ | $55.68_{\pm9.27}$ | $50.00_{\pm0.00}$ | $72.74_{\pm3.74}$ | $55.75_{\pm7.45}$ | $\mathbf{75.69}_{\pm2.44}$ | $75.56_{\pm3.22}$ |
| | | 100 | $78.92_{\pm1.98}$ | $77.78_{\pm2.65}$ | $77.27_{\pm3.15}$ | $77.71_{\pm1.81}$ | $63.50_{\pm10.77}$ | $57.50_{\pm6.27}$ | $77.25_{\pm1.66}$ | $65.87_{\pm6.72}$ | $78.15_{\pm1.45}$ | $\mathbf{79.00}_{\pm1.99}$ |
| | | 200 | $\mathbf{82.59}_{\pm1.84}$ | $81.42_{\pm1.27}$ | $80.48_{\pm1.82}$ | $80.19_{\pm2.48}$ | $79.16_{\pm2.49}$ | $80.45_{\pm1.48}$ | $80.88_{\pm1.68}$ | $80.66_{\pm1.49}$ | $82.56_{\pm1.68}$ | $82.58_{\pm1.90}$ |
| | | 500 | $\mathbf{85.00}_{\pm0.70}$ | $84.37_{\pm0.75}$ | $84.40_{\pm0.68}$ | $84.67_{\pm0.98}$ | $84.45_{\pm0.91}$ | $84.58_{\pm0.70}$ | $84.68_{\pm1.06}$ | $83.66_{\pm0.67}$ | $84.56_{\pm0.98}$ | $84.55_{\pm0.92}$ |
| | steel | 20 | $56.77_{\pm4.17}$ | $55.95_{\pm4.30}$ | $56.03_{\pm4.37}$ | $55.62_{\pm4.80}$ | $52.52_{\pm4.64}$ | $50.00_{\pm0.00}$ | $52.39_{\pm3.13}$ | $50.05_{\pm0.17}$ | $64.80_{\pm5.66}$ | $\mathbf{65.87}_{\pm6.14}$ |
| | | 50 | $82.34_{\pm8.38}$ | $63.42_{\pm3.93}$ | $62.08_{\pm2.69}$ | $63.98_{\pm4.08}$ | $52.92_{\pm4.72}$ | $50.64_{\pm2.01}$ | $61.32_{\pm4.55}$ | $50.36_{\pm1.09}$ | $84.70_{\pm7.84}$ | $\mathbf{86.30}_{\pm6.73}$ |
| | | 100 | $97.37_{\pm1.37}$ | $73.06_{\pm4.46}$ | $71.96_{\pm5.40}$ | $72.23_{\pm4.15}$ | $56.34_{\pm6.30}$ | $80.87_{\pm20.44}$ | $69.29_{\pm5.70}$ | $51.18_{\pm3.24}$ | $97.49_{\pm1.21}$ | $\mathbf{97.81}_{\pm1.49}$ |
| | | 200 | $98.84_{\pm0.70}$ | $82.32_{\pm2.88}$ | $81.78_{\pm3.36}$ | $83.24_{\pm2.68}$ | $82.92_{\pm6.21}$ | $\mathbf{99.35}_{\pm0.70}$ | $86.40_{\pm4.22}$ | $64.42_{\pm11.35}$ | $98.80_{\pm0.73}$ | $98.96_{\pm0.71}$ |
| | | 500 | $99.74_{\pm0.29}$ | $94.27_{\pm2.39}$ | $94.93_{\pm1.89}$ | $96.98_{\pm1.34}$ | $98.32_{\pm1.19}$ | $\mathbf{99.88}_{\pm0.15}$ | $95.70_{\pm1.50}$ | $98.56_{\pm0.52}$ | $99.77_{\pm0.30}$ | $99.74_{\pm0.29}$ |
| | stock | 20 | $83.18_{\pm4.37}$ | $83.69_{\pm3.10}$ | $74.01_{\pm5.09}$ | $56.92_{\pm16.52}$ | $74.99_{\pm6.60}$ | $78.73_{\pm12.25}$ | $69.64_{\pm6.88}$ | $73.40_{\pm4.88}$ | $82.95_{\pm4.44}$ | $\mathbf{83.81}_{\pm4.94}$ |
| | | 50 | $90.01_{\pm2.07}$ | $90.01_{\pm2.43}$ | $82.27_{\pm4.30}$ | $78.91_{\pm4.14}$ | $78.94_{\pm8.78}$ | $89.68_{\pm1.92}$ | $83.72_{\pm2.50}$ | $79.00_{\pm6.87}$ | $89.95_{\pm2.08}$ | $\mathbf{90.15}_{\pm1.76}$ |
| | | 100 | $92.39_{\pm1.06}$ | $92.09_{\pm1.45}$ | $90.75_{\pm2.20}$ | $89.43_{\pm3.29}$ | $86.16_{\pm3.83}$ | $92.12_{\pm1.16}$ | $90.17_{\pm1.92}$ | $89.30_{\pm1.33}$ | $92.12_{\pm1.12}$ | $\mathbf{92.57}_{\pm1.27}$ |
| | | 200 | $94.16_{\pm0.92}$ | $93.99_{\pm0.70}$ | $93.57_{\pm1.10}$ | $93.28_{\pm1.59}$ | $91.92_{\pm2.00}$ | $\mathbf{94.22}_{\pm1.10}$ | $93.05_{\pm1.35}$ | $92.07_{\pm1.76}$ | $94.17_{\pm0.89}$ | $94.16_{\pm1.07}$ |
| **Average rank** | | | $3.08_{\pm1.22}$ | $4.23_{\pm2.32}$ | $6.12_{\pm1.57}$ | $6.29_{\pm2.07}$ | $8.42_{\pm1.32}$ | $6.12_{\pm3.38}$ | $6.54_{\pm1.61}$ | $8.83_{\pm1.46}$ | $3.12_{\pm1.80}$ | $\mathbf{2.23}_{\pm1.83}$ |

### D.5.2 Results on six UCI Datasets

Table 13: Details of the six real-world tabular datasets from UCI.

| Dataset | UCI ID | Not evaluated in TabPFN [33] | # Samples ($N$) | # Features ($D$) | # Classes | N/D | # Samples per class (Min) | # Samples per class (Max) |
|---|---|---|---|---|---|---|---|---|
| clinical | 890 | ✔ | 2,139 | 23 | 2 | 93 | 521 | 1,618 |
| support2 | 880 | ✔ | 9,105 | 42 | 2 | 217 | 2,904 | 6,201 |
| mushroom | 73 | ✔ | 8,124 | 22 | 2 | 369 | 3,916 | 4,208 |
| auction | 713 | ✔ | 2,043 | 7 | 2 | 292 | 262 | 1,781 |
| abalone | 1 | ✔ | 4,153 | 8 | 19 | 519 | 14 | 689 |
| statlog | 144 | ✔ | 1,000 | 20 | 2 | 50 | 300 | 700 |

Table 14: Test classification accuracy (%) aggregated over six downstream predictors, comparing data augmentation on six leakage-free UCI datasets. Note that "N/A" denotes that a specific generator was not applicable. TabEBM still achieves the best overall performance against benchmark methods.

| Datasets ($N_{real} = 100$) | Baseline | SMOTE | TVAE | CTGAN | TabDDPM | TabPFGen | TabEBM (Ours) |
|---|---|---|---|---|---|---|---|
| clinical | $68.63_{\pm5.81}$ | $71.07_{\pm4.67}$ | $61.80_{\pm2.76}$ | $65.21_{\pm5.77}$ | $54.03_{\pm5.36}$ | $69.66_{\pm3.65}$ | $\mathbf{71.20}_{\pm3.54}$ |
| support2 | $64.23_{\pm1.89}$ | $\mathbf{65.60}_{\pm1.52}$ | $60.70_{\pm0.90}$ | $59.14_{\pm1.88}$ | $58.31_{\pm1.74}$ | $64.34_{\pm1.19}$ | $65.28_{\pm1.15}$ |
| mushroom | $95.51_{\pm2.48}$ | $95.84_{\pm1.99}$ | $93.75_{\pm1.18}$ | $93.26_{\pm2.46}$ | $79.87_{\pm2.29}$ | $\mathbf{97.05}_{\pm1.56}$ | $96.82_{\pm1.51}$ |
| auction | $51.90_{\pm1.91}$ | $57.35_{\pm1.53}$ | $53.09_{\pm0.91}$ | $52.35_{\pm1.90}$ | $51.14_{\pm1.76}$ | $56.82_{\pm1.20}$ | $\mathbf{57.97}_{\pm1.16}$ |
| abalone | $11.59_{\pm2.69}$ | N/A | $8.49_{\pm1.28}$ | $7.72_{\pm2.67}$ | $9.95_{\pm2.48}$ | N/A | $\mathbf{13.56}_{\pm1.64}$ |
| statlog | $56.22_{\pm3.20}$ | $57.30_{\pm2.57}$ | $53.12_{\pm1.52}$ | $55.55_{\pm3.18}$ | $53.07_{\pm2.95}$ | $57.65_{\pm2.01}$ | $\mathbf{57.85}_{\pm1.95}$ |

### D.5.3 Results on larger sample sizes

Table 15: Test classification accuracy (%) aggregated over six downstream predictors, comparing data augmentation with increased real data availability of the "texture" dataset. Note that "N/A" denotes that a specific generator was not applicable. On larger datasets, TabEBM still outperforms other generators, but training on real data alone appears sufficient. This highlights TabEBM's usefulness in fields with limited training samples.

| $N_{real}$ | Baseline | SMOTE | TVAE | CTGAN | TabDDPM | TabPFGen | TabEBM (Ours) | Accuracy improvements by TabEBM (%) |
|---|---|---|---|---|---|---|---|---|
| 50 | $72.40_{\pm13.07}$ | $76.40_{\pm10.50}$ | $55.33_{\pm6.20}$ | $54.80_{\pm12.97}$ | $62.94_{\pm12.06}$ | N/A | $\mathbf{78.90}_{\pm7.96}$ | +6.50 |
| 100 | $82.42_{\pm10.38}$ | $84.35_{\pm9.67}$ | $66.00_{\pm7.21}$ | $69.49_{\pm10.93}$ | $76.34_{\pm9.55}$ | N/A | $\mathbf{86.01}_{\pm7.36}$ | +3.59 |
| 200 | $87.54_{\pm7.62}$ | $89.29_{\pm6.20}$ | $78.37_{\pm6.03}$ | $82.44_{\pm7.15}$ | $82.53_{\pm7.99}$ | N/A | $\mathbf{89.77}_{\pm5.77}$ | +2.23 |
| 500 | $92.96_{\pm4.07}$ | $93.69_{\pm3.83}$ | $90.09_{\pm3.56}$ | $91.48_{\pm3.50}$ | $91.24_{\pm3.56}$ | N/A | $\mathbf{93.76}_{\pm3.64}$ | +0.80 |
| 1000 | $\mathbf{96.37}_{\pm2.17}$ | $96.21_{\pm2.37}$ | $93.61_{\pm2.10}$ | $95.36_{\pm1.71}$ | $94.56_{\pm1.59}$ | N/A | $96.30_{\pm2.30}$ | -0.07 |
| 2000 | $97.76_{\pm1.16}$ | $96.84_{\pm1.46}$ | $96.62_{\pm1.24}$ | $97.10_{\pm0.84}$ | $97.13_{\pm0.71}$ | N/A | $\mathbf{97.83}_{\pm1.45}$ | +0.15 |
| 3000 | $98.20_{\pm0.62}$ | $98.28_{\pm0.90}$ | $97.60_{\pm0.73}$ | $97.60_{\pm0.41}$ | $97.73_{\pm0.31}$ | N/A | $\mathbf{98.35}_{\pm0.91}$ | +0.15 |
| 4000 | $98.51_{\pm0.33}$ | $\mathbf{98.59}_{\pm0.56}$ | $98.11_{\pm0.43}$ | $98.00_{\pm0.20}$ | $98.46_{\pm0.14}$ | N/A | $98.55_{\pm0.58}$ | +0.04 |

### D.6 Results on Statistical Fidelity

We aim to provide a fair and coherent comparison between TabEBM and existing methods and thus we follow the widely adopted evaluation process in prior studies. Specifically, we compute the three statistical fidelity metrics with the open-source implementations from the well-established benchmark, Synthcity. We note that the previous studies [87, 67] often operate under the assumption that the issues associated with multiple comparisons are less pronounced in generating low-dimensional tabular data, hence correction methods for multiple hypothesis testing are seldom employed. Following such assumptions, correction methods are not employed in this work. In addition, we would like to point out the imperfection of widely adopted univariate metrics (i.e., Inverse KL, KS test and test) in existing work. However, evaluating generators' ability to capture the joining feature relationships remains an open research question [78]. We leave this for future work to explore.

### D.6.1 Similarity between Real Train Data and Synthetic Data

Table 16: **Inverse KL between real train data and synthetic data** on eight real-world tabular datasets with varied real data availability. We report the mean $\pm$ std balanced accuracy and average accuracy rank across datasets. A higher rank implies higher fidelity. Note that "N/A" denotes that a specific generator was not applicable, and the rank is computed with the mean result of other methods. We **bold** the highest result for each dataset of different sample sizes. TabEBM achieves the best overall performance against benchmark generators.

| Datasets | | $N_{\text{real}}$ | SMOTE | TVAE | CTGAN | NFLOW | TabDDPM | ARF | GOGGLE | TabPFGen | **TabEBM** |
|---|---|---|---|---|---|---|---|---|---|---|---|
| *At most 10 classes* | protein | 20 | N/A | $0.11_{\pm0.01}$ | $0.20_{\pm0.02}$ | $0.34_{\pm0.05}$ | $0.07_{\pm0.01}$ | $0.22_{\pm0.02}$ | $0.07_{\pm0.00}$ | $0.46_{\pm0.13}$ | $\mathbf{0.77}_{\pm0.04}$ |
| | | 50 | $0.88_{\pm0.01}$ | $0.80_{\pm0.02}$ | $0.66_{\pm0.05}$ | $0.87_{\pm0.01}$ | $0.07_{\pm0.00}$ | $0.87_{\pm0.01}$ | $0.50_{\pm0.04}$ | $0.82_{\pm0.06}$ | $\mathbf{0.94}_{\pm0.02}$ |
| | | 100 | $0.93_{\pm0.01}$ | $0.79_{\pm0.02}$ | $0.78_{\pm0.03}$ | $0.90_{\pm0.03}$ | $0.07_{\pm0.00}$ | $0.91_{\pm0.01}$ | $0.32_{\pm0.05}$ | $0.92_{\pm0.02}$ | $\mathbf{0.96}_{\pm0.01}$ |
| | | 200 | $0.95_{\pm0.01}$ | $0.75_{\pm0.03}$ | $0.83_{\pm0.03}$ | $0.93_{\pm0.01}$ | $0.08_{\pm0.01}$ | $0.93_{\pm0.01}$ | $0.11_{\pm0.01}$ | $0.94_{\pm0.01}$ | $\mathbf{0.96}_{\pm0.01}$ |
| | | 500 | $0.96_{\pm0.00}$ | $0.70_{\pm0.02}$ | $0.87_{\pm0.03}$ | $0.94_{\pm0.00}$ | $0.09_{\pm0.00}$ | $0.95_{\pm0.01}$ | $0.13_{\pm0.01}$ | $0.96_{\pm0.01}$ | $\mathbf{0.97}_{\pm0.01}$ |
| | fourier | 20 | N/A | $0.12_{\pm0.03}$ | $0.15_{\pm0.02}$ | $0.27_{\pm0.04}$ | $0.50_{\pm0.03}$ | $0.50_{\pm0.03}$ | $0.50_{\pm0.04}$ | $\mathbf{0.97}_{\pm0.00}$ | $0.87_{\pm0.01}$ |
| | | 50 | $0.93_{\pm0.01}$ | $0.79_{\pm0.02}$ | $0.66_{\pm0.05}$ | $0.90_{\pm0.01}$ | $0.07_{\pm0.00}$ | $0.90_{\pm0.00}$ | $0.47_{\pm0.05}$ | $0.87_{\pm0.02}$ | $\mathbf{0.95}_{\pm0.01}$ |
| | | 100 | $0.95_{\pm0.01}$ | $0.76_{\pm0.03}$ | $0.81_{\pm0.03}$ | $0.93_{\pm0.00}$ | $0.07_{\pm0.00}$ | $0.94_{\pm0.01}$ | $0.20_{\pm0.05}$ | $0.94_{\pm0.01}$ | $\mathbf{0.97}_{\pm0.01}$ |
| | | 200 | $0.97_{\pm0.01}$ | $0.61_{\pm0.01}$ | $0.82_{\pm0.02}$ | $0.95_{\pm0.00}$ | $0.09_{\pm0.01}$ | $0.96_{\pm0.00}$ | $0.08_{\pm0.01}$ | $0.97_{\pm0.01}$ | $\mathbf{0.98}_{\pm0.00}$ |
| | | 500 | $0.97_{\pm0.00}$ | $0.52_{\pm0.03}$ | $0.90_{\pm0.02}$ | $0.95_{\pm0.01}$ | $0.10_{\pm0.01}$ | $0.97_{\pm0.00}$ | $0.09_{\pm0.00}$ | $0.98_{\pm0.00}$ | $\mathbf{0.98}_{\pm0.00}$ |
| | biodeg | 20 | $0.47_{\pm0.04}$ | $0.43_{\pm0.04}$ | $0.43_{\pm0.03}$ | $0.50_{\pm0.05}$ | $0.34_{\pm0.03}$ | $0.51_{\pm0.04}$ | $0.37_{\pm0.04}$ | $0.60_{\pm0.07}$ | $\mathbf{0.87}_{\pm0.04}$ |
| | | 50 | $0.62_{\pm0.03}$ | $0.59_{\pm0.02}$ | $0.56_{\pm0.05}$ | $0.63_{\pm0.05}$ | $0.28_{\pm0.02}$ | $0.65_{\pm0.03}$ | $0.41_{\pm0.03}$ | $0.75_{\pm0.05}$ | $\mathbf{0.90}_{\pm0.02}$ |
| | | 100 | $0.69_{\pm0.05}$ | $0.66_{\pm0.03}$ | $0.65_{\pm0.05}$ | $0.67_{\pm0.04}$ | $0.30_{\pm0.04}$ | $0.69_{\pm0.05}$ | $0.38_{\pm0.04}$ | $0.76_{\pm0.04}$ | $\mathbf{0.90}_{\pm0.04}$ |
| | | 200 | $0.71_{\pm0.03}$ | $0.65_{\pm0.03}$ | $0.69_{\pm0.02}$ | $0.68_{\pm0.03}$ | $0.29_{\pm0.04}$ | $0.69_{\pm0.03}$ | $0.34_{\pm0.01}$ | $0.79_{\pm0.04}$ | $\mathbf{0.91}_{\pm0.02}$ |
| | | 500 | $0.80_{\pm0.02}$ | $0.68_{\pm0.02}$ | $0.73_{\pm0.02}$ | $0.75_{\pm0.02}$ | $0.26_{\pm0.02}$ | $0.73_{\pm0.02}$ | $0.37_{\pm0.02}$ | $0.81_{\pm0.04}$ | $\mathbf{0.92}_{\pm0.02}$ |
| | steel | 20 | $0.45_{\pm0.05}$ | $0.37_{\pm0.03}$ | $0.40_{\pm0.04}$ | $0.47_{\pm0.04}$ | $0.17_{\pm0.03}$ | $0.43_{\pm0.05}$ | $0.29_{\pm0.05}$ | $0.53_{\pm0.08}$ | $\mathbf{0.84}_{\pm0.03}$ |
| | | 50 | $0.70_{\pm0.02}$ | $0.57_{\pm0.03}$ | $0.59_{\pm0.04}$ | $0.64_{\pm0.04}$ | $0.13_{\pm0.01}$ | $0.63_{\pm0.01}$ | $0.23_{\pm0.03}$ | $0.71_{\pm0.08}$ | $\mathbf{0.91}_{\pm0.02}$ |
| | | 100 | $0.71_{\pm0.04}$ | $0.55_{\pm0.02}$ | $0.63_{\pm0.02}$ | $0.67_{\pm0.02}$ | $0.13_{\pm0.01}$ | $0.66_{\pm0.02}$ | $0.20_{\pm0.02}$ | $0.75_{\pm0.05}$ | $\mathbf{0.92}_{\pm0.03}$ |
| | | 200 | $0.75_{\pm0.02}$ | $0.50_{\pm0.04}$ | $0.65_{\pm0.04}$ | $0.70_{\pm0.04}$ | $0.13_{\pm0.02}$ | $0.67_{\pm0.02}$ | $0.17_{\pm0.01}$ | $0.77_{\pm0.04}$ | $\mathbf{0.93}_{\pm0.01}$ |
| | | 500 | $0.75_{\pm0.01}$ | $0.51_{\pm0.04}$ | $0.66_{\pm0.04}$ | $0.70_{\pm0.01}$ | $0.14_{\pm0.01}$ | $0.68_{\pm0.01}$ | $0.19_{\pm0.01}$ | $0.80_{\pm0.06}$ | $\mathbf{0.94}_{\pm0.02}$ |
| | stock | 20 | $0.55_{\pm0.07}$ | $0.45_{\pm0.07}$ | $0.43_{\pm0.05}$ | $0.60_{\pm0.05}$ | $0.24_{\pm0.04}$ | $0.52_{\pm0.06}$ | $0.35_{\pm0.11}$ | $0.68_{\pm0.12}$ | $\mathbf{0.89}_{\pm0.02}$ |
| | | 50 | $0.92_{\pm0.02}$ | $0.73_{\pm0.06}$ | $0.78_{\pm0.07}$ | $0.86_{\pm0.03}$ | $0.37_{\pm0.07}$ | $0.88_{\pm0.01}$ | $0.32_{\pm0.11}$ | $0.91_{\pm0.03}$ | $\mathbf{0.95}_{\pm0.02}$ |
| | | 100 | $0.96_{\pm0.02}$ | $0.67_{\pm0.07}$ | $0.83_{\pm0.05}$ | $0.91_{\pm0.03}$ | $0.49_{\pm0.10}$ | $0.93_{\pm0.01}$ | $0.17_{\pm0.04}$ | $0.95_{\pm0.02}$ | $\mathbf{0.97}_{\pm0.01}$ |
| | | 200 | $0.98_{\pm0.01}$ | $0.63_{\pm0.04}$ | $0.80_{\pm0.08}$ | $0.92_{\pm0.02}$ | $0.83_{\pm0.09}$ | $0.95_{\pm0.01}$ | $0.15_{\pm0.00}$ | $\mathbf{0.98}_{\pm0.01}$ | $0.98_{\pm0.00}$ |
| *More than 10 classes* | energy | 50 | N/A | $0.25_{\pm0.06}$ | $0.34_{\pm0.06}$ | $0.42_{\pm0.06}$ | $0.24_{\pm0.05}$ | $0.49_{\pm0.10}$ | $0.24_{\pm0.09}$ | N/A | $\mathbf{0.80}_{\pm0.04}$ |
| | | 100 | N/A | $0.28_{\pm0.07}$ | $0.42_{\pm0.09}$ | $0.44_{\pm0.04}$ | $0.22_{\pm0.07}$ | $0.41_{\pm0.08}$ | $0.16_{\pm0.04}$ | N/A | $\mathbf{0.89}_{\pm0.01}$ |
| | | 200 | N/A | $0.30_{\pm0.08}$ | $0.43_{\pm0.08}$ | $0.47_{\pm0.09}$ | $0.25_{\pm0.06}$ | $0.40_{\pm0.08}$ | $0.12_{\pm0.04}$ | N/A | $\mathbf{0.91}_{\pm0.01}$ |
| | collins | 100 | N/A | $0.72_{\pm0.02}$ | $0.84_{\pm0.04}$ | $0.90_{\pm0.02}$ | $0.44_{\pm0.11}$ | $0.91_{\pm0.02}$ | $0.28_{\pm0.08}$ | N/A | $\mathbf{0.94}_{\pm0.01}$ |
| | | 200 | $0.94_{\pm0.01}$ | $0.64_{\pm0.05}$ | $0.87_{\pm0.04}$ | $0.92_{\pm0.02}$ | $0.44_{\pm0.05}$ | $0.93_{\pm0.01}$ | $0.23_{\pm0.10}$ | N/A | $\mathbf{0.96}_{\pm0.01}$ |
| | texture | 50 | $0.89_{\pm0.04}$ | $0.74_{\pm0.04}$ | $0.71_{\pm0.06}$ | $0.88_{\pm0.03}$ | $0.10_{\pm0.01}$ | $0.88_{\pm0.02}$ | $0.45_{\pm0.10}$ | N/A | $\mathbf{0.93}_{\pm0.04}$ |
| | | 100 | $0.96_{\pm0.01}$ | $0.67_{\pm0.05}$ | $0.81_{\pm0.07}$ | $0.91_{\pm0.02}$ | $0.11_{\pm0.02}$ | $0.92_{\pm0.02}$ | $0.22_{\pm0.06}$ | N/A | $\mathbf{0.97}_{\pm0.01}$ |
| | | 200 | $0.96_{\pm0.02}$ | $0.56_{\pm0.04}$ | $0.80_{\pm0.07}$ | $0.93_{\pm0.01}$ | $0.12_{\pm0.01}$ | $0.95_{\pm0.01}$ | $0.08_{\pm0.01}$ | N/A | $\mathbf{0.98}_{\pm0.01}$ |
| | | 500 | $0.97_{\pm0.02}$ | $0.63_{\pm0.06}$ | $0.84_{\pm0.04}$ | $0.93_{\pm0.02}$ | $0.14_{\pm0.01}$ | $0.96_{\pm0.01}$ | $0.11_{\pm0.01}$ | N/A | $\mathbf{0.98}_{\pm0.01}$ |
| **Average rank** | | | $3.24_{\pm1.30}$ | $6.76_{\pm0.75}$ | $5.91_{\pm1.10}$ | $4.12_{\pm1.08}$ | $8.42_{\pm1.17}$ | $3.97_{\pm0.95}$ | $8.21_{\pm0.89}$ | $3.30_{\pm1.79}$ | $\mathbf{1.06}_{\pm0.24}$ |

Table 17: **KS test between real train data and synthetic data** on eight real-world tabular datasets with varied real data availability. We report the mean $\pm$ std result and average rank across datasets. A higher rank implies higher fidelity. Note that "N/A" denotes that a specific generator was not applicable, and the rank is computed with the mean result of other methods. We **bold** the highest result for each dataset of different sample sizes. TabEBM achieves the best overall performance against benchmark generators.

| Datasets | | $N_{\text{real}}$ | SMOTE | TVAE | CTGAN | NFLOW | TabDDPM | ARF | GOGGLE | TabPFGen | **TabEBM** |
|---|---|---|---|---|---|---|---|---|---|---|---|
| *At most 10 classes* | protein | 20 | N/A | $0.72_{\pm0.02}$ | $0.75_{\pm0.03}$ | $0.81_{\pm0.01}$ | $0.63_{\pm0.01}$ | $0.80_{\pm0.01}$ | $0.44_{\pm0.00}$ | $0.84_{\pm0.02}$ | $\mathbf{0.87}_{\pm0.01}$ |
| | | 50 | $0.90_{\pm0.01}$ | $0.87_{\pm0.00}$ | $0.87_{\pm0.01}$ | $0.89_{\pm0.01}$ | $0.62_{\pm0.01}$ | $0.89_{\pm0.00}$ | $0.73_{\pm0.02}$ | $0.91_{\pm0.00}$ | $\mathbf{0.93}_{\pm0.00}$ |
| | | 100 | $0.92_{\pm0.00}$ | $0.88_{\pm0.01}$ | $0.91_{\pm0.00}$ | $0.91_{\pm0.00}$ | $0.60_{\pm0.01}$ | $0.91_{\pm0.00}$ | $0.66_{\pm0.02}$ | $0.94_{\pm0.00}$ | $\mathbf{0.94}_{\pm0.00}$ |
| | | 200 | $0.94_{\pm0.00}$ | $0.87_{\pm0.01}$ | $0.92_{\pm0.00}$ | $0.93_{\pm0.00}$ | $0.59_{\pm0.01}$ | $0.93_{\pm0.00}$ | $0.58_{\pm0.01}$ | $0.95_{\pm0.00}$ | $\mathbf{0.95}_{\pm0.00}$ |
| | | 500 | $0.95_{\pm0.00}$ | $0.83_{\pm0.01}$ | $0.92_{\pm0.00}$ | $0.93_{\pm0.00}$ | $0.59_{\pm0.00}$ | $0.94_{\pm0.00}$ | $0.63_{\pm0.01}$ | $0.95_{\pm0.00}$ | $\mathbf{0.96}_{\pm0.00}$ |
| | fourier | 20 | N/A | $0.71_{\pm0.04}$ | $0.73_{\pm0.02}$ | $0.79_{\pm0.01}$ | $0.85_{\pm0.00}$ | $0.85_{\pm0.00}$ | $0.85_{\pm0.00}$ | $\mathbf{0.94}_{\pm0.00}$ | $0.90_{\pm0.00}$ |
| | | 50 | $0.92_{\pm0.00}$ | $0.87_{\pm0.00}$ | $0.88_{\pm0.01}$ | $0.91_{\pm0.00}$ | $0.64_{\pm0.01}$ | $0.90_{\pm0.00}$ | $0.73_{\pm0.02}$ | $0.93_{\pm0.00}$ | $\mathbf{0.94}_{\pm0.00}$ |
| | | 100 | $0.94_{\pm0.00}$ | $0.87_{\pm0.01}$ | $0.92_{\pm0.00}$ | $0.93_{\pm0.00}$ | $0.63_{\pm0.01}$ | $0.93_{\pm0.00}$ | $0.62_{\pm0.02}$ | $0.95_{\pm0.00}$ | $\mathbf{0.95}_{\pm0.00}$ |
| | | 200 | $0.95_{\pm0.00}$ | $0.83_{\pm0.01}$ | $0.92_{\pm0.00}$ | $0.94_{\pm0.00}$ | $0.62_{\pm0.01}$ | $0.94_{\pm0.00}$ | $0.61_{\pm0.03}$ | $\mathbf{0.97}_{\pm0.00}$ | $0.96_{\pm0.00}$ |
| | | 500 | $0.96_{\pm0.00}$ | $0.81_{\pm0.00}$ | $0.94_{\pm0.00}$ | $0.95_{\pm0.00}$ | $0.62_{\pm0.01}$ | $0.95_{\pm0.00}$ | $0.63_{\pm0.01}$ | $\mathbf{0.97}_{\pm0.00}$ | $0.97_{\pm0.00}$ |
| | biodeg | 20 | $0.63_{\pm0.04}$ | $0.62_{\pm0.04}$ | $0.64_{\pm0.03}$ | $0.64_{\pm0.04}$ | $0.56_{\pm0.04}$ | $0.63_{\pm0.04}$ | $0.65_{\pm0.03}$ | $0.59_{\pm0.02}$ | $\mathbf{0.70}_{\pm0.01}$ |
| | | 50 | $0.57_{\pm0.03}$ | $0.57_{\pm0.03}$ | $0.59_{\pm0.03}$ | $0.60_{\pm0.03}$ | $0.48_{\pm0.03}$ | $0.59_{\pm0.03}$ | $0.63_{\pm0.02}$ | $0.61_{\pm0.04}$ | $\mathbf{0.73}_{\pm0.00}$ |
| | | 100 | $0.56_{\pm0.03}$ | $0.55_{\pm0.03}$ | $0.57_{\pm0.03}$ | $0.59_{\pm0.03}$ | $0.46_{\pm0.03}$ | $0.58_{\pm0.03}$ | $0.59_{\pm0.03}$ | $0.59_{\pm0.02}$ | $\mathbf{0.73}_{\pm0.01}$ |
| | | 200 | $0.53_{\pm0.01}$ | $0.52_{\pm0.01}$ | $0.53_{\pm0.02}$ | $0.56_{\pm0.02}$ | $0.43_{\pm0.02}$ | $0.55_{\pm0.02}$ | $0.55_{\pm0.02}$ | $0.58_{\pm0.02}$ | $\mathbf{0.72}_{\pm0.01}$ |
| | | 500 | $0.53_{\pm0.00}$ | $0.50_{\pm0.01}$ | $0.52_{\pm0.01}$ | $0.55_{\pm0.01}$ | $0.43_{\pm0.01}$ | $0.56_{\pm0.01}$ | $0.56_{\pm0.01}$ | $0.57_{\pm0.01}$ | $\mathbf{0.72}_{\pm0.01}$ |
| | steel | 20 | $0.67_{\pm0.02}$ | $0.63_{\pm0.02}$ | $0.64_{\pm0.03}$ | $0.66_{\pm0.02}$ | $0.54_{\pm0.02}$ | $0.65_{\pm0.02}$ | $0.57_{\pm0.03}$ | $0.67_{\pm0.02}$ | $\mathbf{0.76}_{\pm0.01}$ |
| | | 50 | $0.69_{\pm0.02}$ | $0.64_{\pm0.03}$ | $0.65_{\pm0.03}$ | $0.67_{\pm0.02}$ | $0.51_{\pm0.02}$ | $0.67_{\pm0.03}$ | $0.56_{\pm0.02}$ | $0.72_{\pm0.02}$ | $\mathbf{0.79}_{\pm0.01}$ |
| | | 100 | $0.69_{\pm0.02}$ | $0.63_{\pm0.02}$ | $0.65_{\pm0.02}$ | $0.67_{\pm0.01}$ | $0.50_{\pm0.01}$ | $0.65_{\pm0.02}$ | $0.55_{\pm0.02}$ | $0.71_{\pm0.03}$ | $\mathbf{0.79}_{\pm0.01}$ |
| | | 200 | $0.70_{\pm0.01}$ | $0.61_{\pm0.02}$ | $0.66_{\pm0.02}$ | $0.69_{\pm0.01}$ | $0.50_{\pm0.01}$ | $0.65_{\pm0.02}$ | $0.55_{\pm0.01}$ | $0.71_{\pm0.02}$ | $\mathbf{0.79}_{\pm0.01}$ |
| | | 500 | $0.70_{\pm0.01}$ | $0.62_{\pm0.02}$ | $0.66_{\pm0.02}$ | $0.68_{\pm0.01}$ | $0.49_{\pm0.01}$ | $0.65_{\pm0.01}$ | $0.58_{\pm0.01}$ | $0.74_{\pm0.02}$ | $\mathbf{0.80}_{\pm0.01}$ |
| | stock | 20 | $0.86_{\pm0.02}$ | $0.82_{\pm0.01}$ | $0.82_{\pm0.02}$ | $0.86_{\pm0.01}$ | $0.74_{\pm0.03}$ | $0.86_{\pm0.02}$ | $0.63_{\pm0.07}$ | $0.89_{\pm0.01}$ | $\mathbf{0.91}_{\pm0.01}$ |
| | | 50 | $0.92_{\pm0.01}$ | $0.86_{\pm0.01}$ | $0.88_{\pm0.01}$ | $0.90_{\pm0.01}$ | $0.84_{\pm0.02}$ | $0.91_{\pm0.01}$ | $0.68_{\pm0.04}$ | $0.93_{\pm0.01}$ | $\mathbf{0.94}_{\pm0.00}$ |
| | | 100 | $0.94_{\pm0.01}$ | $0.86_{\pm0.01}$ | $0.90_{\pm0.01}$ | $0.92_{\pm0.01}$ | $0.88_{\pm0.02}$ | $0.93_{\pm0.01}$ | $0.63_{\pm0.02}$ | $\mathbf{0.95}_{\pm0.01}$ | $0.95_{\pm0.00}$ |
| | | 200 | $0.95_{\pm0.01}$ | $0.86_{\pm0.01}$ | $0.90_{\pm0.01}$ | $0.93_{\pm0.01}$ | $0.92_{\pm0.01}$ | $0.94_{\pm0.00}$ | $0.63_{\pm0.01}$ | $\mathbf{0.96}_{\pm0.00}$ | $0.95_{\pm0.00}$ |
| *More than 10 classes* | energy | 50 | N/A | $0.70_{\pm0.02}$ | $0.69_{\pm0.04}$ | $0.73_{\pm0.01}$ | $0.65_{\pm0.03}$ | $0.72_{\pm0.01}$ | $0.63_{\pm0.03}$ | N/A | $\mathbf{0.78}_{\pm0.01}$ |
| | | 100 | N/A | $0.69_{\pm0.02}$ | $0.74_{\pm0.01}$ | $0.74_{\pm0.01}$ | $0.69_{\pm0.01}$ | $0.74_{\pm0.01}$ | $0.63_{\pm0.03}$ | N/A | $\mathbf{0.81}_{\pm0.01}$ |
| | | 200 | N/A | $0.71_{\pm0.01}$ | $0.74_{\pm0.02}$ | $0.75_{\pm0.01}$ | $0.67_{\pm0.01}$ | $0.75_{\pm0.00}$ | $0.63_{\pm0.02}$ | N/A | $\mathbf{0.83}_{\pm0.01}$ |
| | collins | 100 | N/A | $0.85_{\pm0.01}$ | $0.89_{\pm0.01}$ | $0.90_{\pm0.01}$ | $0.82_{\pm0.04}$ | $0.90_{\pm0.01}$ | $0.65_{\pm0.04}$ | N/A | $\mathbf{0.93}_{\pm0.00}$ |
| | | 200 | $0.93_{\pm0.00}$ | $0.83_{\pm0.02}$ | $0.91_{\pm0.01}$ | $0.92_{\pm0.00}$ | $0.80_{\pm0.03}$ | $0.92_{\pm0.00}$ | $0.63_{\pm0.05}$ | N/A | $\mathbf{0.94}_{\pm0.00}$ |
| | texture | 50 | $0.92_{\pm0.01}$ | $0.86_{\pm0.01}$ | $0.88_{\pm0.01}$ | $0.90_{\pm0.01}$ | $0.57_{\pm0.03}$ | $0.90_{\pm0.01}$ | $0.71_{\pm0.05}$ | N/A | $\mathbf{0.93}_{\pm0.01}$ |
| | | 100 | $0.94_{\pm0.01}$ | $0.86_{\pm0.01}$ | $0.91_{\pm0.01}$ | $0.92_{\pm0.00}$ | $0.61_{\pm0.02}$ | $0.92_{\pm0.01}$ | $0.63_{\pm0.03}$ | N/A | $\mathbf{0.95}_{\pm0.00}$ |
| | | 200 | $0.96_{\pm0.01}$ | $0.83_{\pm0.01}$ | $0.91_{\pm0.01}$ | $0.93_{\pm0.00}$ | $0.62_{\pm0.01}$ | $0.94_{\pm0.01}$ | $0.60_{\pm0.01}$ | N/A | $\mathbf{0.96}_{\pm0.00}$ |
| | | 500 | $0.97_{\pm0.00}$ | $0.85_{\pm0.01}$ | $0.91_{\pm0.01}$ | $0.93_{\pm0.00}$ | $0.61_{\pm0.01}$ | $0.95_{\pm0.00}$ | $0.64_{\pm0.01}$ | N/A | $\mathbf{0.97}_{\pm0.00}$ |
| **Average rank** | | | $3.91_{\pm1.65}$ | $6.97_{\pm0.92}$ | $5.76_{\pm1.00}$ | $3.97_{\pm1.07}$ | $8.33_{\pm1.05}$ | $4.15_{\pm0.94}$ | $7.55_{\pm2.22}$ | $3.21_{\pm2.16}$ | $\mathbf{1.15}_{\pm0.36}$ |

Table 18: $\chi^2$ **test between real train data and synthetic data** on eight real-world tabular datasets with varied real data availability. We report the mean $\pm$ std result and average rank across datasets. A higher rank implies higher fidelity. Note that "N/A" denotes that a specific generator was not applicable, and the rank is computed with the mean result of other methods. We **bold** the highest result for each dataset of different sample sizes. TabEBM achieves the best overall performance against benchmark generators.

| | Datasets | $N_{\text{real}}$ | SMOTE | TVAE | CTGAN | NFLOW | TabDDPM | ARF | GOGGLE | TabPFGen | **TabEBM** |
|---|---|---|---|---|---|---|---|---|---|---|---|
| *At most 10 classes* | protein | 20 | N/A | $0.02_{\pm0.01}$ | $0.08_{\pm0.03}$ | $0.19_{\pm0.07}$ | $0.02_{\pm0.01}$ | $0.05_{\pm0.03}$ | $0.02_{\pm0.01}$ | $0.33_{\pm0.23}$ | $\mathbf{0.92}_{\pm0.07}$ |
| | | 50 | $0.95_{\pm0.04}$ | $0.84_{\pm0.05}$ | $0.50_{\pm0.10}$ | $0.92_{\pm0.04}$ | $0.01_{\pm0.00}$ | $\mathbf{0.98}_{\pm0.02}$ | $0.53_{\pm0.05}$ | $0.63_{\pm0.14}$ | $0.96_{\pm0.04}$ |
| | | 100 | $0.86_{\pm0.06}$ | $0.62_{\pm0.07}$ | $0.53_{\pm0.09}$ | $0.83_{\pm0.10}$ | $0.01_{\pm0.00}$ | $0.89_{\pm0.03}$ | $0.27_{\pm0.07}$ | $0.70_{\pm0.10}$ | $\mathbf{0.91}_{\pm0.06}$ |
| | | 200 | $0.80_{\pm0.05}$ | $0.46_{\pm0.07}$ | $0.46_{\pm0.09}$ | $0.77_{\pm0.05}$ | $0.01_{\pm0.00}$ | $0.76_{\pm0.06}$ | $0.02_{\pm0.02}$ | $0.66_{\pm0.08}$ | $\mathbf{0.81}_{\pm0.08}$ |
| | | 500 | $0.62_{\pm0.06}$ | $0.25_{\pm0.05}$ | $0.36_{\pm0.06}$ | $0.61_{\pm0.04}$ | $0.01_{\pm0.00}$ | $0.57_{\pm0.05}$ | $0.01_{\pm0.00}$ | $0.65_{\pm0.07}$ | $\mathbf{0.70}_{\pm0.05}$ |
| | fourier | 20 | N/A | $0.02_{\pm0.02}$ | $0.05_{\pm0.02}$ | $0.15_{\pm0.05}$ | $0.37_{\pm0.04}$ | $0.35_{\pm0.05}$ | $0.36_{\pm0.06}$ | $\mathbf{1.00}_{\pm0.00}$ | $1.00_{\pm0.00}$ |
| | | 50 | $0.99_{\pm0.01}$ | $0.87_{\pm0.04}$ | $0.52_{\pm0.10}$ | $0.97_{\pm0.02}$ | $0.01_{\pm0.00}$ | $\mathbf{0.99}_{\pm0.01}$ | $0.49_{\pm0.06}$ | $0.74_{\pm0.05}$ | $0.99_{\pm0.01}$ |
| | | 100 | $0.96_{\pm0.02}$ | $0.75_{\pm0.04}$ | $0.69_{\pm0.07}$ | $0.95_{\pm0.02}$ | $0.01_{\pm0.00}$ | $\mathbf{0.98}_{\pm0.02}$ | $0.16_{\pm0.05}$ | $0.82_{\pm0.06}$ | $0.97_{\pm0.03}$ |
| | | 200 | $0.92_{\pm0.03}$ | $0.41_{\pm0.07}$ | $0.59_{\pm0.04}$ | $0.92_{\pm0.03}$ | $0.01_{\pm0.00}$ | $\mathbf{0.95}_{\pm0.02}$ | $0.02_{\pm0.00}$ | $0.85_{\pm0.05}$ | $0.95_{\pm0.03}$ |
| | | 500 | $0.80_{\pm0.06}$ | $0.14_{\pm0.04}$ | $0.59_{\pm0.06}$ | $0.76_{\pm0.07}$ | $0.01_{\pm0.00}$ | $0.84_{\pm0.04}$ | $0.01_{\pm0.00}$ | $0.81_{\pm0.04}$ | $\mathbf{0.84}_{\pm0.02}$ |
| | biodeg | 20 | $0.23_{\pm0.06}$ | $0.21_{\pm0.06}$ | $0.16_{\pm0.04}$ | $0.28_{\pm0.06}$ | $0.08_{\pm0.04}$ | $0.28_{\pm0.07}$ | $0.12_{\pm0.06}$ | $0.29_{\pm0.10}$ | $\mathbf{0.71}_{\pm0.06}$ |
| | | 50 | $0.39_{\pm0.02}$ | $0.33_{\pm0.05}$ | $0.28_{\pm0.06}$ | $0.41_{\pm0.07}$ | $0.05_{\pm0.02}$ | $0.45_{\pm0.05}$ | $0.15_{\pm0.03}$ | $0.44_{\pm0.09}$ | $\mathbf{0.75}_{\pm0.06}$ |
| | | 100 | $0.35_{\pm0.06}$ | $0.26_{\pm0.06}$ | $0.30_{\pm0.08}$ | $0.38_{\pm0.06}$ | $0.04_{\pm0.01}$ | $0.42_{\pm0.05}$ | $0.08_{\pm0.04}$ | $0.37_{\pm0.13}$ | $\mathbf{0.67}_{\pm0.14}$ |
| | | 200 | $0.25_{\pm0.06}$ | $0.19_{\pm0.05}$ | $0.24_{\pm0.05}$ | $0.30_{\pm0.07}$ | $0.03_{\pm0.01}$ | $0.31_{\pm0.06}$ | $0.02_{\pm0.00}$ | $0.37_{\pm0.06}$ | $\mathbf{0.59}_{\pm0.08}$ |
| | | 500 | $0.22_{\pm0.08}$ | $0.10_{\pm0.04}$ | $0.17_{\pm0.05}$ | $0.25_{\pm0.07}$ | $0.02_{\pm0.00}$ | $0.23_{\pm0.06}$ | $0.02_{\pm0.00}$ | $0.26_{\pm0.08}$ | $\mathbf{0.44}_{\pm0.09}$ |
| | steel | 20 | $0.37_{\pm0.08}$ | $0.32_{\pm0.05}$ | $0.32_{\pm0.04}$ | $0.40_{\pm0.08}$ | $0.04_{\pm0.02}$ | $0.40_{\pm0.08}$ | $0.23_{\pm0.07}$ | $0.34_{\pm0.13}$ | $\mathbf{0.81}_{\pm0.05}$ |
| | | 50 | $0.68_{\pm0.02}$ | $0.50_{\pm0.06}$ | $0.51_{\pm0.08}$ | $0.64_{\pm0.05}$ | $0.03_{\pm0.00}$ | $0.67_{\pm0.03}$ | $0.10_{\pm0.05}$ | $0.49_{\pm0.15}$ | $\mathbf{0.85}_{\pm0.06}$ |
| | | 100 | $0.61_{\pm0.09}$ | $0.37_{\pm0.07}$ | $0.47_{\pm0.08}$ | $0.61_{\pm0.04}$ | $0.03_{\pm0.00}$ | $0.61_{\pm0.07}$ | $0.06_{\pm0.02}$ | $0.51_{\pm0.07}$ | $\mathbf{0.81}_{\pm0.08}$ |
| | | 200 | $0.60_{\pm0.04}$ | $0.23_{\pm0.04}$ | $0.42_{\pm0.06}$ | $0.60_{\pm0.05}$ | $0.03_{\pm0.00}$ | $0.53_{\pm0.07}$ | $0.02_{\pm0.00}$ | $0.52_{\pm0.07}$ | $\mathbf{0.77}_{\pm0.06}$ |
| | | 500 | $0.46_{\pm0.06}$ | $0.17_{\pm0.07}$ | $0.35_{\pm0.07}$ | $0.55_{\pm0.06}$ | $0.03_{\pm0.00}$ | $0.42_{\pm0.04}$ | $0.02_{\pm0.00}$ | $0.48_{\pm0.10}$ | $\mathbf{0.69}_{\pm0.09}$ |
| | stock | 20 | $0.56_{\pm0.13}$ | $0.50_{\pm0.12}$ | $0.40_{\pm0.12}$ | $0.63_{\pm0.08}$ | $0.11_{\pm0.03}$ | $0.55_{\pm0.12}$ | $0.43_{\pm0.16}$ | $0.56_{\pm0.20}$ | $\mathbf{1.00}_{\pm0.00}$ |
| | | 50 | $0.99_{\pm0.03}$ | $0.85_{\pm0.10}$ | $0.88_{\pm0.13}$ | $0.99_{\pm0.03}$ | $0.12_{\pm0.06}$ | $\mathbf{1.00}_{\pm0.00}$ | $0.31_{\pm0.14}$ | $0.87_{\pm0.08}$ | $0.99_{\pm0.03}$ |
| | | 100 | $1.00_{\pm0.00}$ | $0.64_{\pm0.14}$ | $0.89_{\pm0.12}$ | $0.99_{\pm0.03}$ | $0.16_{\pm0.13}$ | $\mathbf{1.00}_{\pm0.00}$ | $0.13_{\pm0.05}$ | $0.91_{\pm0.10}$ | $0.99_{\pm0.03}$ |
| | | 200 | $0.99_{\pm0.03}$ | $0.58_{\pm0.09}$ | $0.76_{\pm0.20}$ | $0.98_{\pm0.04}$ | $0.71_{\pm0.22}$ | $\mathbf{1.00}_{\pm0.00}$ | $0.10_{\pm0.00}$ | $0.99_{\pm0.03}$ | $0.99_{\pm0.03}$ |
| *More than 10 classes* | energy | 50 | N/A | $0.14_{\pm0.08}$ | $0.26_{\pm0.09}$ | $0.35_{\pm0.09}$ | $0.17_{\pm0.06}$ | $0.41_{\pm0.14}$ | $0.19_{\pm0.10}$ | N/A | $\mathbf{0.80}_{\pm0.05}$ |
| | | 100 | N/A | $0.09_{\pm0.10}$ | $0.32_{\pm0.12}$ | $0.35_{\pm0.08}$ | $0.13_{\pm0.06}$ | $0.30_{\pm0.09}$ | $0.06_{\pm0.07}$ | N/A | $\mathbf{0.92}_{\pm0.01}$ |
| | | 200 | N/A | $0.15_{\pm0.12}$ | $0.30_{\pm0.09}$ | $0.39_{\pm0.12}$ | $0.15_{\pm0.07}$ | $0.28_{\pm0.08}$ | $0.03_{\pm0.05}$ | N/A | $\mathbf{0.96}_{\pm0.01}$ |
| | collins | 100 | N/A | $0.61_{\pm0.08}$ | $0.73_{\pm0.12}$ | $0.87_{\pm0.08}$ | $0.10_{\pm0.07}$ | $\mathbf{0.90}_{\pm0.08}$ | $0.20_{\pm0.09}$ | N/A | $0.89_{\pm0.04}$ |
| | | 200 | $0.75_{\pm0.09}$ | $0.35_{\pm0.10}$ | $0.62_{\pm0.13}$ | $0.76_{\pm0.10}$ | $0.07_{\pm0.03}$ | $0.78_{\pm0.06}$ | $0.05_{\pm0.04}$ | N/A | $\mathbf{0.80}_{\pm0.09}$ |
| | texture | 50 | $0.90_{\pm0.12}$ | $0.79_{\pm0.11}$ | $0.63_{\pm0.14}$ | $0.94_{\pm0.09}$ | $0.02_{\pm0.00}$ | $\mathbf{0.99}_{\pm0.02}$ | $0.47_{\pm0.13}$ | N/A | $0.93_{\pm0.12}$ |
| | | 100 | $0.97_{\pm0.04}$ | $0.51_{\pm0.13}$ | $0.67_{\pm0.12}$ | $0.92_{\pm0.07}$ | $0.02_{\pm0.00}$ | $0.94_{\pm0.09}$ | $0.17_{\pm0.08}$ | N/A | $\mathbf{0.97}_{\pm0.06}$ |
| | | 200 | $0.86_{\pm0.11}$ | $0.28_{\pm0.11}$ | $0.55_{\pm0.16}$ | $0.89_{\pm0.10}$ | $0.02_{\pm0.00}$ | $\mathbf{0.93}_{\pm0.07}$ | $0.00_{\pm0.01}$ | N/A | $0.91_{\pm0.08}$ |
| | | 500 | $0.74_{\pm0.17}$ | $0.21_{\pm0.09}$ | $0.51_{\pm0.08}$ | $0.73_{\pm0.07}$ | $0.02_{\pm0.00}$ | $0.73_{\pm0.07}$ | $0.00_{\pm0.00}$ | N/A | $\mathbf{0.82}_{\pm0.12}$ |
| **Average rank** | | | $3.52_{\pm1.08}$ | $6.88_{\pm1.02}$ | $6.03_{\pm1.10}$ | $3.55_{\pm1.03}$ | $8.30_{\pm1.02}$ | $2.82_{\pm1.76}$ | $8.18_{\pm0.88}$ | $4.27_{\pm1.62}$ | $\mathbf{1.45}_{\pm0.71}$ |

### D.6.2 Similarity between Real Test Data and Synthetic Data

Table 19: **Inverse KL between real test data and synthetic data** on eight real-world tabular datasets with varied real data availability. We report the mean $\pm$ std result and average rank across datasets. A higher rank implies higher fidelity. Note that "N/A" denotes that a specific generator was not applicable, and the rank is computed with the mean result of other methods. We **bold** the highest result for each dataset of different sample sizes. TabEBM achieves the best overall performance against benchmark generators.

| | Datasets | $N_{\text{real}}$ | SMOTE | TVAE | CTGAN | NFLOW | TabDDPM | ARF | GOGGLE | TabPFGen | **TabEBM** |
|---|---|---|---|---|---|---|---|---|---|---|---|
| *At most 10 classes* | protein | 20 | N/A | $0.26_{\pm 0.04}$ | $0.26_{\pm 0.03}$ | $0.32_{\pm 0.03}$ | $0.09_{\pm 0.01}$ | $0.34_{\pm 0.03}$ | $0.08_{\pm 0.00}$ | $0.35_{\pm 0.07}$ | $\mathbf{0.52}_{\pm 0.06}$ |
| | | 50 | $0.78_{\pm 0.02}$ | $\mathbf{0.82}_{\pm 0.02}$ | $0.63_{\pm 0.06}$ | $0.71_{\pm 0.04}$ | $0.08_{\pm 0.00}$ | $0.80_{\pm 0.03}$ | $0.52_{\pm 0.03}$ | $0.62_{\pm 0.04}$ | $0.75_{\pm 0.03}$ |
| | | 100 | $\mathbf{0.88}_{\pm 0.02}$ | $0.80_{\pm 0.02}$ | $0.76_{\pm 0.04}$ | $0.83_{\pm 0.03}$ | $0.08_{\pm 0.00}$ | $0.87_{\pm 0.02}$ | $0.33_{\pm 0.05}$ | $0.78_{\pm 0.03}$ | $0.85_{\pm 0.02}$ |
| | | 200 | $\mathbf{0.92}_{\pm 0.01}$ | $0.75_{\pm 0.03}$ | $0.81_{\pm 0.03}$ | $0.91_{\pm 0.01}$ | $0.08_{\pm 0.00}$ | $0.91_{\pm 0.01}$ | $0.12_{\pm 0.01}$ | $0.89_{\pm 0.02}$ | $0.92_{\pm 0.01}$ |
| | | 500 | $0.94_{\pm 0.00}$ | $0.68_{\pm 0.02}$ | $0.86_{\pm 0.02}$ | $0.93_{\pm 0.02}$ | $0.08_{\pm 0.00}$ | $0.94_{\pm 0.00}$ | $0.13_{\pm 0.01}$ | $0.94_{\pm 0.00}$ | $\mathbf{0.95}_{\pm 0.01}$ |
| | fourier | 20 | N/A | $0.14_{\pm 0.02}$ | $0.18_{\pm 0.01}$ | $0.20_{\pm 0.01}$ | $0.21_{\pm 0.02}$ | $0.21_{\pm 0.01}$ | $0.22_{\pm 0.02}$ | $0.24_{\pm 0.01}$ | $\mathbf{0.48}_{\pm 0.03}$ |
| | | 50 | $0.84_{\pm 0.02}$ | $0.78_{\pm 0.03}$ | $0.62_{\pm 0.05}$ | $0.77_{\pm 0.03}$ | $0.07_{\pm 0.00}$ | $\mathbf{0.85}_{\pm 0.03}$ | $0.48_{\pm 0.06}$ | $0.66_{\pm 0.03}$ | $0.79_{\pm 0.03}$ |
| | | 100 | $\mathbf{0.91}_{\pm 0.02}$ | $0.72_{\pm 0.05}$ | $0.76_{\pm 0.03}$ | $0.88_{\pm 0.02}$ | $0.08_{\pm 0.00}$ | $0.90_{\pm 0.01}$ | $0.20_{\pm 0.05}$ | $0.81_{\pm 0.02}$ | $0.88_{\pm 0.02}$ |
| | | 200 | $\mathbf{0.94}_{\pm 0.01}$ | $0.58_{\pm 0.02}$ | $0.79_{\pm 0.03}$ | $0.93_{\pm 0.01}$ | $0.09_{\pm 0.00}$ | $0.93_{\pm 0.01}$ | $0.08_{\pm 0.01}$ | $0.90_{\pm 0.02}$ | $0.93_{\pm 0.01}$ |
| | | 500 | $0.96_{\pm 0.00}$ | $0.50_{\pm 0.02}$ | $0.88_{\pm 0.02}$ | $0.93_{\pm 0.01}$ | $0.10_{\pm 0.01}$ | $0.95_{\pm 0.01}$ | $0.09_{\pm 0.00}$ | $0.95_{\pm 0.01}$ | $\mathbf{0.96}_{\pm 0.00}$ |
| | biodeg | 20 | $0.43_{\pm 0.03}$ | $0.44_{\pm 0.04}$ | $0.41_{\pm 0.03}$ | $0.41_{\pm 0.04}$ | $0.33_{\pm 0.03}$ | $0.44_{\pm 0.03}$ | $0.36_{\pm 0.03}$ | $0.45_{\pm 0.02}$ | $\mathbf{0.57}_{\pm 0.03}$ |
| | | 50 | $0.60_{\pm 0.05}$ | $0.59_{\pm 0.04}$ | $0.53_{\pm 0.04}$ | $0.55_{\pm 0.04}$ | $0.31_{\pm 0.03}$ | $0.57_{\pm 0.03}$ | $0.41_{\pm 0.03}$ | $0.60_{\pm 0.05}$ | $\mathbf{0.71}_{\pm 0.04}$ |
| | | 100 | $0.65_{\pm 0.04}$ | $0.65_{\pm 0.02}$ | $0.62_{\pm 0.03}$ | $0.63_{\pm 0.03}$ | $0.31_{\pm 0.03}$ | $0.64_{\pm 0.03}$ | $0.37_{\pm 0.03}$ | $0.65_{\pm 0.03}$ | $\mathbf{0.77}_{\pm 0.02}$ |
| | | 200 | $0.71_{\pm 0.02}$ | $0.66_{\pm 0.03}$ | $0.66_{\pm 0.04}$ | $0.65_{\pm 0.03}$ | $0.31_{\pm 0.04}$ | $0.68_{\pm 0.02}$ | $0.33_{\pm 0.02}$ | $0.73_{\pm 0.04}$ | $\mathbf{0.83}_{\pm 0.02}$ |
| | | 500 | $0.77_{\pm 0.03}$ | $0.64_{\pm 0.02}$ | $0.69_{\pm 0.03}$ | $0.72_{\pm 0.04}$ | $0.25_{\pm 0.03}$ | $0.69_{\pm 0.03}$ | $0.35_{\pm 0.01}$ | $0.76_{\pm 0.05}$ | $\mathbf{0.88}_{\pm 0.02}$ |
| | steel | 20 | $0.47_{\pm 0.03}$ | $0.45_{\pm 0.03}$ | $0.42_{\pm 0.02}$ | $0.44_{\pm 0.02}$ | $0.23_{\pm 0.01}$ | $0.46_{\pm 0.04}$ | $0.37_{\pm 0.04}$ | $0.42_{\pm 0.04}$ | $\mathbf{0.70}_{\pm 0.03}$ |
| | | 50 | $0.65_{\pm 0.03}$ | $0.59_{\pm 0.03}$ | $0.58_{\pm 0.03}$ | $0.60_{\pm 0.04}$ | $0.21_{\pm 0.01}$ | $0.63_{\pm 0.03}$ | $0.30_{\pm 0.03}$ | $0.62_{\pm 0.05}$ | $\mathbf{0.83}_{\pm 0.03}$ |
| | | 100 | $0.70_{\pm 0.02}$ | $0.59_{\pm 0.02}$ | $0.62_{\pm 0.03}$ | $0.66_{\pm 0.03}$ | $0.21_{\pm 0.01}$ | $0.66_{\pm 0.02}$ | $0.29_{\pm 0.02}$ | $0.72_{\pm 0.04}$ | $\mathbf{0.89}_{\pm 0.02}$ |
| | | 200 | $0.73_{\pm 0.01}$ | $0.55_{\pm 0.02}$ | $0.66_{\pm 0.02}$ | $0.69_{\pm 0.02}$ | $0.22_{\pm 0.02}$ | $0.68_{\pm 0.02}$ | $0.25_{\pm 0.01}$ | $0.75_{\pm 0.03}$ | $\mathbf{0.91}_{\pm 0.02}$ |
| | | 500 | $0.74_{\pm 0.01}$ | $0.55_{\pm 0.04}$ | $0.66_{\pm 0.02}$ | $0.69_{\pm 0.01}$ | $0.22_{\pm 0.01}$ | $0.70_{\pm 0.01}$ | $0.27_{\pm 0.01}$ | $0.78_{\pm 0.05}$ | $\mathbf{0.93}_{\pm 0.02}$ |
| | stock | 20 | $0.50_{\pm 0.09}$ | $0.50_{\pm 0.09}$ | $0.41_{\pm 0.05}$ | $0.47_{\pm 0.08}$ | $0.25_{\pm 0.04}$ | $0.54_{\pm 0.09}$ | $0.40_{\pm 0.13}$ | $0.35_{\pm 0.05}$ | $\mathbf{0.65}_{\pm 0.09}$ |
| | | 50 | $0.80_{\pm 0.06}$ | $0.68_{\pm 0.06}$ | $0.68_{\pm 0.05}$ | $0.76_{\pm 0.03}$ | $0.34_{\pm 0.05}$ | $\mathbf{0.86}_{\pm 0.04}$ | $0.33_{\pm 0.11}$ | $0.69_{\pm 0.09}$ | $0.85_{\pm 0.05}$ |
| | | 100 | $0.86_{\pm 0.04}$ | $0.61_{\pm 0.05}$ | $0.73_{\pm 0.06}$ | $0.85_{\pm 0.06}$ | $0.44_{\pm 0.07}$ | $0.90_{\pm 0.02}$ | $0.18_{\pm 0.04}$ | $0.84_{\pm 0.05}$ | $\mathbf{0.91}_{\pm 0.04}$ |
| | | 200 | $0.92_{\pm 0.02}$ | $0.59_{\pm 0.05}$ | $0.75_{\pm 0.07}$ | $0.90_{\pm 0.03}$ | $0.67_{\pm 0.09}$ | $0.94_{\pm 0.01}$ | $0.15_{\pm 0.00}$ | $0.94_{\pm 0.03}$ | $\mathbf{0.96}_{\pm 0.02}$ |
| *More than 10 classes* | energy | 50 | N/A | $0.26_{\pm 0.06}$ | $0.33_{\pm 0.06}$ | $0.36_{\pm 0.08}$ | $0.23_{\pm 0.05}$ | $0.46_{\pm 0.10}$ | $0.22_{\pm 0.07}$ | N/A | $\mathbf{0.77}_{\pm 0.03}$ |
| | | 100 | N/A | $0.27_{\pm 0.06}$ | $0.40_{\pm 0.08}$ | $0.43_{\pm 0.06}$ | $0.22_{\pm 0.06}$ | $0.39_{\pm 0.08}$ | $0.16_{\pm 0.05}$ | N/A | $\mathbf{0.87}_{\pm 0.01}$ |
| | | 200 | N/A | $0.30_{\pm 0.07}$ | $0.41_{\pm 0.07}$ | $0.46_{\pm 0.09}$ | $0.24_{\pm 0.06}$ | $0.39_{\pm 0.08}$ | $0.12_{\pm 0.04}$ | N/A | $\mathbf{0.89}_{\pm 0.01}$ |
| | collins | 100 | N/A | $0.68_{\pm 0.03}$ | $0.75_{\pm 0.04}$ | $0.79_{\pm 0.03}$ | $0.43_{\pm 0.07}$ | $\mathbf{0.81}_{\pm 0.02}$ | $0.28_{\pm 0.07}$ | N/A | $0.78_{\pm 0.02}$ |
| | | 200 | $0.87_{\pm 0.02}$ | $0.62_{\pm 0.03}$ | $0.81_{\pm 0.03}$ | $0.88_{\pm 0.02}$ | $0.44_{\pm 0.04}$ | $\mathbf{0.88}_{\pm 0.02}$ | $0.22_{\pm 0.09}$ | N/A | $0.87_{\pm 0.01}$ |
| | texture | 50 | $0.82_{\pm 0.04}$ | $0.80_{\pm 0.04}$ | $0.70_{\pm 0.05}$ | $0.80_{\pm 0.07}$ | $0.11_{\pm 0.01}$ | $\mathbf{0.92}_{\pm 0.01}$ | $0.48_{\pm 0.11}$ | N/A | $0.87_{\pm 0.03}$ |
| | | 100 | $0.89_{\pm 0.02}$ | $0.69_{\pm 0.03}$ | $0.79_{\pm 0.06}$ | $0.89_{\pm 0.02}$ | $0.13_{\pm 0.01}$ | $\mathbf{0.93}_{\pm 0.00}$ | $0.23_{\pm 0.06}$ | N/A | $0.92_{\pm 0.02}$ |
| | | 200 | $0.93_{\pm 0.02}$ | $0.58_{\pm 0.05}$ | $0.79_{\pm 0.05}$ | $0.92_{\pm 0.01}$ | $0.14_{\pm 0.01}$ | $0.95_{\pm 0.01}$ | $0.10_{\pm 0.02}$ | N/A | $\mathbf{0.95}_{\pm 0.01}$ |
| | | 500 | $0.96_{\pm 0.01}$ | $0.64_{\pm 0.06}$ | $0.85_{\pm 0.04}$ | $0.93_{\pm 0.01}$ | $0.15_{\pm 0.01}$ | $0.96_{\pm 0.00}$ | $0.12_{\pm 0.01}$ | N/A | $\mathbf{0.97}_{\pm 0.01}$ |
| **Average rank** | | | $2.94_{\pm 1.31}$ | $6.03_{\pm 1.69}$ | $5.82_{\pm 1.07}$ | $4.39_{\pm 1.30}$ | $8.48_{\pm 0.71}$ | $3.00_{\pm 1.52}$ | $8.24_{\pm 0.94}$ | $4.45_{\pm 1.91}$ | $\mathbf{1.64}_{\pm 1.03}$ |

Table 20: **KS test between real test data and synthetic data** on eight real-world tabular datasets with varied real data availability. We report the mean $\pm$ std result and average rank across datasets. A higher rank implies higher fidelity. Note that "N/A" denotes that a specific generator was not applicable, and the rank is computed with the mean result of other methods. We **bold** the highest result for each dataset of different sample sizes. TabEBM achieves the best overall performance against benchmark generators.

| Datasets | | $N_{\text{real}}$ | SMOTE | TVAE | CTGAN | NFLOW | TabDDPM | ARF | GOGGLE | TabPFGen | **TabEBM** |
|---|---|---|---|---|---|---|---|---|---|---|---|
| *At most 10 classes* | protein | 20 | N/A | $0.69_{\pm0.01}$ | $0.72_{\pm0.02}$ | $0.75_{\pm0.01}$ | $0.61_{\pm0.01}$ | $0.74_{\pm0.01}$ | $0.41_{\pm0.02}$ | $0.77_{\pm0.02}$ | $\mathbf{0.81}_{\pm0.01}$ |
| | | 50 | $0.88_{\pm0.01}$ | $0.87_{\pm0.01}$ | $0.87_{\pm0.01}$ | $0.88_{\pm0.01}$ | $0.61_{\pm0.01}$ | $\mathbf{0.89}_{\pm0.01}$ | $0.73_{\pm0.02}$ | $0.86_{\pm0.01}$ | $0.88_{\pm0.01}$ |
| | | 100 | $0.90_{\pm0.01}$ | $0.87_{\pm0.01}$ | $0.89_{\pm0.01}$ | $0.90_{\pm0.01}$ | $0.60_{\pm0.01}$ | $0.91_{\pm0.01}$ | $0.65_{\pm0.02}$ | $0.89_{\pm0.01}$ | $\mathbf{0.91}_{\pm0.01}$ |
| | | 200 | $0.92_{\pm0.01}$ | $0.86_{\pm0.01}$ | $0.90_{\pm0.01}$ | $0.91_{\pm0.01}$ | $0.59_{\pm0.01}$ | $0.92_{\pm0.01}$ | $0.58_{\pm0.01}$ | $0.91_{\pm0.01}$ | $\mathbf{0.92}_{\pm0.01}$ |
| | | 500 | $0.92_{\pm0.00}$ | $0.82_{\pm0.01}$ | $0.90_{\pm0.00}$ | $0.91_{\pm0.00}$ | $0.58_{\pm0.01}$ | $0.92_{\pm0.00}$ | $0.63_{\pm0.00}$ | $0.92_{\pm0.00}$ | $\mathbf{0.93}_{\pm0.00}$ |
| | fourier | 20 | N/A | $0.67_{\pm0.03}$ | $0.69_{\pm0.02}$ | $0.73_{\pm0.01}$ | $0.75_{\pm0.01}$ | $0.75_{\pm0.01}$ | $0.75_{\pm0.01}$ | $0.76_{\pm0.01}$ | $\mathbf{0.81}_{\pm0.01}$ |
| | | 50 | $0.89_{\pm0.00}$ | $0.86_{\pm0.01}$ | $0.88_{\pm0.01}$ | $0.89_{\pm0.01}$ | $0.64_{\pm0.00}$ | $\mathbf{0.91}_{\pm0.00}$ | $0.73_{\pm0.02}$ | $0.87_{\pm0.01}$ | $0.89_{\pm0.01}$ |
| | | 100 | $0.92_{\pm0.00}$ | $0.85_{\pm0.01}$ | $0.90_{\pm0.00}$ | $0.91_{\pm0.00}$ | $0.62_{\pm0.01}$ | $\mathbf{0.92}_{\pm0.00}$ | $0.61_{\pm0.02}$ | $0.90_{\pm0.00}$ | $0.91_{\pm0.00}$ |
| | | 200 | $0.93_{\pm0.01}$ | $0.82_{\pm0.01}$ | $0.91_{\pm0.01}$ | $0.93_{\pm0.00}$ | $0.62_{\pm0.01}$ | $\mathbf{0.93}_{\pm0.01}$ | $0.60_{\pm0.01}$ | $0.93_{\pm0.00}$ | $0.93_{\pm0.00}$ |
| | | 500 | $0.94_{\pm0.00}$ | $0.80_{\pm0.00}$ | $0.92_{\pm0.01}$ | $0.93_{\pm0.00}$ | $0.62_{\pm0.01}$ | $0.94_{\pm0.00}$ | $0.63_{\pm0.01}$ | $0.94_{\pm0.00}$ | $\mathbf{0.94}_{\pm0.00}$ |
| | biodeg | 20 | $0.61_{\pm0.03}$ | $0.61_{\pm0.03}$ | $0.63_{\pm0.03}$ | $0.63_{\pm0.03}$ | $0.55_{\pm0.04}$ | $0.62_{\pm0.03}$ | $0.64_{\pm0.03}$ | $0.56_{\pm0.03}$ | $\mathbf{0.67}_{\pm0.02}$ |
| | | 50 | $0.56_{\pm0.03}$ | $0.57_{\pm0.03}$ | $0.58_{\pm0.03}$ | $0.59_{\pm0.03}$ | $0.48_{\pm0.03}$ | $0.58_{\pm0.03}$ | $0.63_{\pm0.02}$ | $0.59_{\pm0.04}$ | $\mathbf{0.71}_{\pm0.01}$ |
| | | 100 | $0.55_{\pm0.02}$ | $0.55_{\pm0.02}$ | $0.57_{\pm0.03}$ | $0.58_{\pm0.02}$ | $0.46_{\pm0.03}$ | $0.58_{\pm0.02}$ | $0.59_{\pm0.03}$ | $0.58_{\pm0.02}$ | $\mathbf{0.71}_{\pm0.01}$ |
| | | 200 | $0.53_{\pm0.01}$ | $0.51_{\pm0.02}$ | $0.53_{\pm0.02}$ | $0.55_{\pm0.02}$ | $0.43_{\pm0.02}$ | $0.55_{\pm0.02}$ | $0.55_{\pm0.01}$ | $0.57_{\pm0.02}$ | $\mathbf{0.72}_{\pm0.01}$ |
| | | 500 | $0.53_{\pm0.00}$ | $0.49_{\pm0.01}$ | $0.51_{\pm0.01}$ | $0.54_{\pm0.01}$ | $0.43_{\pm0.01}$ | $0.55_{\pm0.01}$ | $0.56_{\pm0.01}$ | $0.57_{\pm0.01}$ | $\mathbf{0.72}_{\pm0.00}$ |
| | steel | 20 | $0.65_{\pm0.03}$ | $0.62_{\pm0.03}$ | $0.63_{\pm0.03}$ | $0.65_{\pm0.02}$ | $0.54_{\pm0.02}$ | $0.64_{\pm0.02}$ | $0.56_{\pm0.03}$ | $0.64_{\pm0.02}$ | $\mathbf{0.73}_{\pm0.02}$ |
| | | 50 | $0.68_{\pm0.02}$ | $0.64_{\pm0.03}$ | $0.65_{\pm0.03}$ | $0.67_{\pm0.02}$ | $0.51_{\pm0.01}$ | $0.68_{\pm0.03}$ | $0.56_{\pm0.03}$ | $0.70_{\pm0.02}$ | $\mathbf{0.77}_{\pm0.01}$ |
| | | 100 | $0.68_{\pm0.01}$ | $0.63_{\pm0.02}$ | $0.65_{\pm0.01}$ | $0.66_{\pm0.01}$ | $0.50_{\pm0.01}$ | $0.66_{\pm0.02}$ | $0.55_{\pm0.02}$ | $0.70_{\pm0.03}$ | $\mathbf{0.78}_{\pm0.01}$ |
| | | 200 | $0.69_{\pm0.01}$ | $0.61_{\pm0.02}$ | $0.66_{\pm0.02}$ | $0.68_{\pm0.02}$ | $0.50_{\pm0.01}$ | $0.65_{\pm0.02}$ | $0.54_{\pm0.01}$ | $0.71_{\pm0.02}$ | $\mathbf{0.78}_{\pm0.01}$ |
| | | 500 | $0.69_{\pm0.01}$ | $0.62_{\pm0.02}$ | $0.66_{\pm0.02}$ | $0.68_{\pm0.01}$ | $0.50_{\pm0.01}$ | $0.64_{\pm0.01}$ | $0.58_{\pm0.01}$ | $0.73_{\pm0.02}$ | $\mathbf{0.79}_{\pm0.01}$ |
| | stock | 20 | $0.83_{\pm0.04}$ | $0.81_{\pm0.04}$ | $0.82_{\pm0.04}$ | $0.84_{\pm0.04}$ | $0.74_{\pm0.04}$ | $0.84_{\pm0.04}$ | $0.64_{\pm0.06}$ | $0.83_{\pm0.04}$ | $\mathbf{0.86}_{\pm0.04}$ |
| | | 50 | $0.89_{\pm0.02}$ | $0.84_{\pm0.02}$ | $0.87_{\pm0.03}$ | $0.88_{\pm0.02}$ | $0.83_{\pm0.02}$ | $0.89_{\pm0.02}$ | $0.67_{\pm0.05}$ | $0.88_{\pm0.02}$ | $\mathbf{0.89}_{\pm0.02}$ |
| | | 100 | $0.91_{\pm0.02}$ | $0.84_{\pm0.02}$ | $0.88_{\pm0.02}$ | $0.90_{\pm0.02}$ | $0.87_{\pm0.03}$ | $0.90_{\pm0.02}$ | $0.62_{\pm0.01}$ | $0.91_{\pm0.02}$ | $\mathbf{0.91}_{\pm0.02}$ |
| | | 200 | $0.92_{\pm0.02}$ | $0.84_{\pm0.02}$ | $0.88_{\pm0.02}$ | $0.90_{\pm0.02}$ | $0.90_{\pm0.02}$ | $0.91_{\pm0.02}$ | $0.63_{\pm0.01}$ | $0.92_{\pm0.02}$ | $\mathbf{0.92}_{\pm0.02}$ |
| | energy | 50 | N/A | $0.69_{\pm0.02}$ | $0.70_{\pm0.03}$ | $0.72_{\pm0.02}$ | $0.64_{\pm0.02}$ | $0.72_{\pm0.01}$ | $0.62_{\pm0.03}$ | N/A | $\mathbf{0.76}_{\pm0.01}$ |
| | | 100 | N/A | $0.69_{\pm0.03}$ | $0.74_{\pm0.01}$ | $0.75_{\pm0.01}$ | $0.69_{\pm0.01}$ | $0.75_{\pm0.01}$ | $0.64_{\pm0.03}$ | N/A | $\mathbf{0.81}_{\pm0.01}$ |
| *More than 10 classes* | | 200 | N/A | $0.71_{\pm0.01}$ | $0.74_{\pm0.01}$ | $0.75_{\pm0.01}$ | $0.67_{\pm0.02}$ | $0.76_{\pm0.01}$ | $0.63_{\pm0.02}$ | N/A | $\mathbf{0.83}_{\pm0.01}$ |
| | collins | 100 | N/A | $0.83_{\pm0.01}$ | $0.88_{\pm0.01}$ | $0.90_{\pm0.01}$ | $0.81_{\pm0.04}$ | $\mathbf{0.90}_{\pm0.01}$ | $0.65_{\pm0.03}$ | N/A | $0.90_{\pm0.01}$ |
| | | 200 | $0.91_{\pm0.01}$ | $0.82_{\pm0.02}$ | $0.90_{\pm0.01}$ | $0.91_{\pm0.01}$ | $0.80_{\pm0.02}$ | $0.91_{\pm0.01}$ | $0.63_{\pm0.04}$ | N/A | $\mathbf{0.93}_{\pm0.01}$ |
| | texture | 50 | $0.90_{\pm0.01}$ | $0.87_{\pm0.02}$ | $0.88_{\pm0.01}$ | $0.91_{\pm0.01}$ | $0.55_{\pm0.04}$ | $0.92_{\pm0.01}$ | $0.73_{\pm0.05}$ | N/A | $\mathbf{0.92}_{\pm0.01}$ |
| | | 100 | $0.92_{\pm0.01}$ | $0.86_{\pm0.01}$ | $0.91_{\pm0.01}$ | $0.92_{\pm0.01}$ | $0.60_{\pm0.02}$ | $0.93_{\pm0.01}$ | $0.64_{\pm0.03}$ | N/A | $\mathbf{0.94}_{\pm0.01}$ |
| | | 200 | $0.94_{\pm0.01}$ | $0.82_{\pm0.01}$ | $0.90_{\pm0.01}$ | $0.93_{\pm0.00}$ | $0.62_{\pm0.01}$ | $0.94_{\pm0.00}$ | $0.60_{\pm0.01}$ | N/A | $\mathbf{0.94}_{\pm0.00}$ |
| | | 500 | $0.95_{\pm0.00}$ | $0.84_{\pm0.02}$ | $0.91_{\pm0.01}$ | $0.93_{\pm0.01}$ | $0.61_{\pm0.02}$ | $0.95_{\pm0.00}$ | $0.64_{\pm0.01}$ | N/A | $\mathbf{0.95}_{\pm0.00}$ |
| **Average rank** | | | $3.97_{\pm1.73}$ | $7.09_{\pm0.72}$ | $5.52_{\pm0.97}$ | $3.76_{\pm1.12}$ | $8.27_{\pm1.21}$ | $3.27_{\pm1.53}$ | $7.45_{\pm2.36}$ | $4.39_{\pm1.93}$ | $\mathbf{1.27}_{\pm0.72}$ |

Table 21: $\chi^2$ **test between real test data and synthetic data** on eight real-world tabular datasets with varied real data availability. We report the mean $\pm$ std result and average rank across datasets. A higher rank implies higher fidelity. Note that "N/A" denotes that a specific generator was not applicable, and the rank is computed with the mean result of other methods. We **bold** the highest result for each dataset of different sample sizes. TabEBM achieves the best overall performance against benchmark generators.

| Datasets | | $N_{\text{real}}$ | SMOTE | TVAE | CTGAN | NFLOW | TabDDPM | ARF | GOGGLE | TabPFGen | **TabEBM** |
|---|---|---|---|---|---|---|---|---|---|---|---|
| *At most 10 classes* | protein | 20 | N/A | $0.01_{\pm0.00}$ | $0.01_{\pm0.00}$ | $0.01_{\pm0.00}$ | $0.01_{\pm0.00}$ | $0.01_{\pm0.00}$ | $0.01_{\pm0.00}$ | $0.02_{\pm0.01}$ | $\mathbf{0.06}_{\pm0.03}$ |
| | | 50 | $0.26_{\pm0.05}$ | $\mathbf{0.32}_{\pm0.06}$ | $0.09_{\pm0.04}$ | $0.17_{\pm0.04}$ | $0.01_{\pm0.00}$ | $0.25_{\pm0.04}$ | $0.22_{\pm0.05}$ | $0.05_{\pm0.03}$ | $0.15_{\pm0.04}$ |
| | | 100 | $\mathbf{0.39}_{\pm0.04}$ | $0.33_{\pm0.05}$ | $0.18_{\pm0.05}$ | $0.29_{\pm0.09}$ | $0.01_{\pm0.00}$ | $0.34_{\pm0.04}$ | $0.13_{\pm0.04}$ | $0.12_{\pm0.04}$ | $0.26_{\pm0.05}$ |
| | | 200 | $\mathbf{0.48}_{\pm0.06}$ | $0.30_{\pm0.07}$ | $0.28_{\pm0.06}$ | $0.48_{\pm0.04}$ | $0.01_{\pm0.00}$ | $0.44_{\pm0.08}$ | $0.02_{\pm0.01}$ | $0.34_{\pm0.08}$ | $0.43_{\pm0.06}$ |
| | | 500 | $0.54_{\pm0.05}$ | $0.24_{\pm0.07}$ | $0.33_{\pm0.06}$ | $\mathbf{0.60}_{\pm0.04}$ | $0.01_{\pm0.00}$ | $0.55_{\pm0.04}$ | $0.01_{\pm0.00}$ | $0.53_{\pm0.05}$ | $0.58_{\pm0.06}$ |
| | fourier | 20 | N/A | $0.00_{\pm0.00}$ | $0.01_{\pm0.00}$ | $0.01_{\pm0.00}$ | $0.01_{\pm0.00}$ | $0.01_{\pm0.00}$ | $0.01_{\pm0.00}$ | $0.01_{\pm0.00}$ | $\mathbf{0.10}_{\pm0.04}$ |
| | | 50 | $0.42_{\pm0.08}$ | $\mathbf{0.45}_{\pm0.07}$ | $0.15_{\pm0.06}$ | $0.31_{\pm0.07}$ | $0.01_{\pm0.00}$ | $0.39_{\pm0.08}$ | $0.29_{\pm0.08}$ | $0.10_{\pm0.04}$ | $0.33_{\pm0.06}$ |
| | | 100 | $\mathbf{0.58}_{\pm0.08}$ | $0.44_{\pm0.07}$ | $0.32_{\pm0.07}$ | $0.52_{\pm0.09}$ | $0.01_{\pm0.00}$ | $0.48_{\pm0.07}$ | $0.08_{\pm0.04}$ | $0.26_{\pm0.07}$ | $0.49_{\pm0.07}$ |
| | | 200 | $0.67_{\pm0.05}$ | $0.28_{\pm0.03}$ | $0.38_{\pm0.06}$ | $\mathbf{0.68}_{\pm0.04}$ | $0.01_{\pm0.00}$ | $0.60_{\pm0.04}$ | $0.02_{\pm0.01}$ | $0.46_{\pm0.06}$ | $0.60_{\pm0.05}$ |
| | | 500 | $\mathbf{0.76}_{\pm0.04}$ | $0.15_{\pm0.06}$ | $0.55_{\pm0.08}$ | $0.72_{\pm0.07}$ | $0.01_{\pm0.00}$ | $0.71_{\pm0.05}$ | $0.01_{\pm0.00}$ | $0.69_{\pm0.04}$ | $0.74_{\pm0.03}$ |
| | biodeg | 20 | $0.05_{\pm0.03}$ | $0.05_{\pm0.04}$ | $0.04_{\pm0.03}$ | $0.04_{\pm0.02}$ | $0.03_{\pm0.01}$ | $0.05_{\pm0.03}$ | $0.03_{\pm0.01}$ | $0.10_{\pm0.01}$ | $\mathbf{0.13}_{\pm0.02}$ |
| | | 50 | $0.13_{\pm0.06}$ | $0.12_{\pm0.04}$ | $0.07_{\pm0.02}$ | $0.08_{\pm0.04}$ | $0.02_{\pm0.00}$ | $0.08_{\pm0.05}$ | $0.06_{\pm0.03}$ | $0.08_{\pm0.04}$ | $\mathbf{0.14}_{\pm0.05}$ |
| | | 100 | $0.15_{\pm0.04}$ | $0.14_{\pm0.05}$ | $0.10_{\pm0.03}$ | $0.12_{\pm0.05}$ | $0.02_{\pm0.00}$ | $0.12_{\pm0.04}$ | $0.04_{\pm0.02}$ | $0.07_{\pm0.03}$ | $\mathbf{0.18}_{\pm0.04}$ |
| | | 200 | $0.18_{\pm0.03}$ | $0.15_{\pm0.04}$ | $0.14_{\pm0.03}$ | $0.19_{\pm0.05}$ | $0.03_{\pm0.01}$ | $0.18_{\pm0.03}$ | $0.02_{\pm0.00}$ | $0.14_{\pm0.05}$ | $\mathbf{0.28}_{\pm0.07}$ |
| | | 500 | $0.25_{\pm0.07}$ | $0.13_{\pm0.05}$ | $0.21_{\pm0.07}$ | $0.28_{\pm0.05}$ | $0.03_{\pm0.01}$ | $0.28_{\pm0.07}$ | $0.02_{\pm0.00}$ | $0.23_{\pm0.07}$ | $\mathbf{0.38}_{\pm0.05}$ |
| | steel | 20 | $0.13_{\pm0.04}$ | $0.16_{\pm0.04}$ | $0.09_{\pm0.03}$ | $0.10_{\pm0.04}$ | $0.03_{\pm0.00}$ | $0.14_{\pm0.03}$ | $0.12_{\pm0.05}$ | $0.07_{\pm0.03}$ | $\mathbf{0.29}_{\pm0.06}$ |
| | | 50 | $0.26_{\pm0.07}$ | $0.24_{\pm0.07}$ | $0.17_{\pm0.04}$ | $0.21_{\pm0.06}$ | $0.03_{\pm0.00}$ | $0.31_{\pm0.04}$ | $0.05_{\pm0.02}$ | $0.17_{\pm0.05}$ | $\mathbf{0.43}_{\pm0.07}$ |
| | | 100 | $0.32_{\pm0.04}$ | $0.23_{\pm0.06}$ | $0.24_{\pm0.07}$ | $0.31_{\pm0.06}$ | $0.03_{\pm0.00}$ | $0.33_{\pm0.06}$ | $0.04_{\pm0.02}$ | $0.29_{\pm0.06}$ | $\mathbf{0.49}_{\pm0.07}$ |
| | | 200 | $0.36_{\pm0.04}$ | $0.17_{\pm0.03}$ | $0.29_{\pm0.06}$ | $0.34_{\pm0.05}$ | $0.03_{\pm0.00}$ | $0.36_{\pm0.06}$ | $0.02_{\pm0.00}$ | $0.35_{\pm0.06}$ | $\mathbf{0.53}_{\pm0.07}$ |
| | | 500 | $0.37_{\pm0.04}$ | $0.15_{\pm0.07}$ | $0.28_{\pm0.05}$ | $0.36_{\pm0.04}$ | $0.03_{\pm0.00}$ | $0.36_{\pm0.05}$ | $0.02_{\pm0.00}$ | $0.38_{\pm0.06}$ | $\mathbf{0.57}_{\pm0.06}$ |
| | stock | 20 | $0.31_{\pm0.11}$ | $0.38_{\pm0.13}$ | $0.22_{\pm0.08}$ | $0.25_{\pm0.08}$ | $0.10_{\pm0.00}$ | $0.33_{\pm0.11}$ | $\mathbf{0.42}_{\pm0.18}$ | $0.11_{\pm0.03}$ | $0.41_{\pm0.19}$ |
| | | 50 | $0.60_{\pm0.09}$ | $0.56_{\pm0.12}$ | $0.44_{\pm0.15}$ | $0.54_{\pm0.08}$ | $0.10_{\pm0.00}$ | $0.73_{\pm0.14}$ | $0.31_{\pm0.14}$ | $0.41_{\pm0.16}$ | $\mathbf{0.76}_{\pm0.10}$ |
| | | 100 | $0.71_{\pm0.11}$ | $0.40_{\pm0.11}$ | $0.53_{\pm0.15}$ | $0.68_{\pm0.12}$ | $0.12_{\pm0.06}$ | $\mathbf{0.85}_{\pm0.05}$ | $0.13_{\pm0.05}$ | $0.65_{\pm0.12}$ | $0.84_{\pm0.07}$ |
| | | 200 | $0.86_{\pm0.07}$ | $0.45_{\pm0.12}$ | $0.60_{\pm0.18}$ | $0.91_{\pm0.11}$ | $0.35_{\pm0.20}$ | $0.97_{\pm0.05}$ | $0.10_{\pm0.00}$ | $0.92_{\pm0.08}$ | $\mathbf{0.97}_{\pm0.05}$ |
| *More than 10 classes* | energy | 50 | N/A | $0.15_{\pm0.08}$ | $0.25_{\pm0.09}$ | $0.28_{\pm0.10}$ | $0.17_{\pm0.06}$ | $0.40_{\pm0.14}$ | $0.15_{\pm0.10}$ | N/A | $\mathbf{0.78}_{\pm0.05}$ |
| | | 100 | N/A | $0.10_{\pm0.09}$ | $0.31_{\pm0.12}$ | $0.34_{\pm0.08}$ | $0.14_{\pm0.06}$ | $0.30_{\pm0.09}$ | $0.06_{\pm0.07}$ | N/A | $\mathbf{0.92}_{\pm0.02}$ |
| | | 200 | N/A | $0.16_{\pm0.11}$ | $0.30_{\pm0.09}$ | $0.38_{\pm0.12}$ | $0.15_{\pm0.07}$ | $0.28_{\pm0.08}$ | $0.03_{\pm0.05}$ | N/A | $\mathbf{0.96}_{\pm0.01}$ |
| | collins | 100 | N/A | $0.23_{\pm0.08}$ | $0.21_{\pm0.08}$ | $\mathbf{0.25}_{\pm0.09}$ | $0.05_{\pm0.02}$ | $0.24_{\pm0.06}$ | $0.07_{\pm0.03}$ | N/A | $0.18_{\pm0.07}$ |
| | | 200 | $0.32_{\pm0.06}$ | $0.16_{\pm0.06}$ | $0.24_{\pm0.06}$ | $\mathbf{0.38}_{\pm0.05}$ | $0.06_{\pm0.02}$ | $0.33_{\pm0.07}$ | $0.02_{\pm0.04}$ | N/A | $0.31_{\pm0.07}$ |
| | texture | 50 | $0.33_{\pm0.11}$ | $0.44_{\pm0.10}$ | $0.21_{\pm0.08}$ | $0.29_{\pm0.10}$ | $0.02_{\pm0.00}$ | $\mathbf{0.55}_{\pm0.15}$ | $0.27_{\pm0.08}$ | N/A | $0.41_{\pm0.04}$ |
| | | 100 | $0.44_{\pm0.13}$ | $0.27_{\pm0.06}$ | $0.27_{\pm0.09}$ | $0.48_{\pm0.15}$ | $0.02_{\pm0.00}$ | $\mathbf{0.54}_{\pm0.06}$ | $0.07_{\pm0.05}$ | N/A | $0.49_{\pm0.08}$ |
| | | 200 | $0.51_{\pm0.11}$ | $0.19_{\pm0.10}$ | $0.34_{\pm0.10}$ | $0.58_{\pm0.11}$ | $0.02_{\pm0.00}$ | $\mathbf{0.64}_{\pm0.09}$ | $0.00_{\pm0.00}$ | N/A | $0.59_{\pm0.09}$ |
| | | 500 | $0.63_{\pm0.14}$ | $0.21_{\pm0.09}$ | $0.49_{\pm0.09}$ | $\mathbf{0.67}_{\pm0.08}$ | $0.02_{\pm0.00}$ | $0.64_{\pm0.08}$ | $0.00_{\pm0.00}$ | N/A | $0.66_{\pm0.12}$ |
| **Average rank** | | | $3.06_{\pm1.32}$ | $5.45_{\pm2.35}$ | $6.15_{\pm0.87}$ | $3.64_{\pm1.75}$ | $8.36_{\pm0.90}$ | $3.18_{\pm1.47}$ | $7.76_{\pm1.77}$ | $5.33_{\pm1.71}$ | $\mathbf{2.06}_{\pm1.43}$ |

### D.7  Results on Privacy Preservation

### D.7.1  Downstream Accuracy in Data Sharing

Table 22: **Classification accuracy** (%) aggregated over six downstream predictors, comparing data sharing on eight real-world tabular datasets with varied real data availability. We report the mean $\pm$ std balanced accuracy and average accuracy rank across datasets. A higher rank implies higher accuracy. Note that "N/A" denotes the inapplicability of a specific generator. Different from Table 1, "–" denotes a generator cannot satisfy the requirement of generating 500 stratified samples even after generating 10,000 synthetic samples. The results of these inapplicable or failed generators are computed with the mean results of other methods. We **bold** the highest result for each dataset of different sample sizes. TVAE learns the joint distribution $p(\mathbf{x}, y)$ and fails to maintain the original training label distribution. TabEBM achieves the best overall performance against Baseline and benchmark generators.

| Datasets | | $N_{real}$ | SMOTE | TVAE | CTGAN | NFLOW | TabDDPM | ARF | GOGGLE | TabPFGen | **TabEBM** |
|---|---|---|---|---|---|---|---|---|---|---|---|
| *At most 10 classes* | protein | 20 | N/A | $16.38_{\pm3.34}$ | $13.06_{\pm3.57}$ | $12.99_{\pm4.16}$ | $12.64_{\pm3.06}$ | $19.42_{\pm3.96}$ | $12.88_{\pm2.32}$ | $33.11_{\pm3.64}$ | $\mathbf{33.97}_{\pm3.48}$ |
| | | 50 | $54.55_{\pm3.94}$ | $27.29_{\pm3.43}$ | $17.18_{\pm4.87}$ | $13.20_{\pm3.08}$ | $14.04_{\pm3.29}$ | $29.34_{\pm4.46}$ | $13.05_{\pm3.34}$ | $54.82_{\pm3.74}$ | $\mathbf{55.83}_{\pm4.35}$ |
| | | 100 | $72.46_{\pm3.54}$ | $40.04_{\pm4.54}$ | $26.65_{\pm3.89}$ | $12.80_{\pm3.48}$ | $17.65_{\pm3.25}$ | $35.90_{\pm5.29}$ | $13.55_{\pm3.03}$ | $71.63_{\pm3.95}$ | $\mathbf{72.99}_{\pm3.69}$ |
| | | 200 | $83.12_{\pm2.33}$ | $45.33_{\pm6.01}$ | $32.52_{\pm5.60}$ | $14.07_{\pm3.94}$ | $20.79_{\pm4.42}$ | $41.63_{\pm4.24}$ | $11.68_{\pm3.60}$ | $84.18_{\pm1.90}$ | $\mathbf{84.29}_{\pm2.03}$ |
| | | 500 | $89.63_{\pm1.57}$ | $55.24_{\pm5.30}$ | $44.35_{\pm5.67}$ | $12.76_{\pm2.73}$ | $21.63_{\pm4.90}$ | $54.26_{\pm3.47}$ | $11.09_{\pm3.19}$ | $\mathbf{91.19}_{\pm1.48}$ | $90.99_{\pm1.53}$ |
| | fourier | 20 | N/A | – | $13.30_{\pm3.14}$ | $10.72_{\pm3.03}$ | $10.03_{\pm2.42}$ | $17.14_{\pm4.25}$ | $10.29_{\pm2.81}$ | $33.87_{\pm4.93}$ | $\mathbf{37.06}_{\pm4.80}$ |
| | | 50 | $55.11_{\pm3.56}$ | – | N/A | $9.88_{\pm2.51}$ | $11.96_{\pm2.30}$ | $28.63_{\pm3.71}$ | $11.52_{\pm2.46}$ | $56.11_{\pm3.28}$ | $\mathbf{56.88}_{\pm2.90}$ |
| | | 100 | $64.06_{\pm3.34}$ | $34.80_{\pm4.60}$ | $23.35_{\pm4.20}$ | $10.85_{\pm3.02}$ | $17.68_{\pm3.42}$ | $32.59_{\pm4.02}$ | $9.82_{\pm2.84}$ | $64.42_{\pm2.96}$ | $\mathbf{64.93}_{\pm3.06}$ |
| | | 200 | $70.78_{\pm2.99}$ | $40.49_{\pm3.43}$ | $32.98_{\pm5.20}$ | $10.28_{\pm3.74}$ | $28.63_{\pm4.22}$ | $39.34_{\pm3.41}$ | $9.55_{\pm2.06}$ | $71.37_{\pm2.56}$ | $\mathbf{72.01}_{\pm2.67}$ |
| | | 500 | $74.84_{\pm1.47}$ | $46.83_{\pm3.74}$ | $46.13_{\pm4.33}$ | $11.13_{\pm2.47}$ | $27.25_{\pm6.04}$ | $47.19_{\pm2.89}$ | $10.31_{\pm2.19}$ | $75.86_{\pm1.91}$ | $\mathbf{76.58}_{\pm1.54}$ |
| | biodeg | 20 | $68.75_{\pm4.96}$ | $64.07_{\pm7.39}$ | $53.60_{\pm7.49}$ | $54.58_{\pm8.02}$ | $48.39_{\pm2.73}$ | $58.69_{\pm6.75}$ | $48.65_{\pm9.50}$ | $69.14_{\pm5.08}$ | $\mathbf{69.66}_{\pm5.18}$ |
| | | 50 | $72.08_{\pm3.01}$ | $67.11_{\pm5.03}$ | $62.21_{\pm7.04}$ | $55.12_{\pm8.81}$ | $47.86_{\pm3.58}$ | $70.14_{\pm3.92}$ | $52.14_{\pm7.45}$ | $73.53_{\pm3.39}$ | $\mathbf{73.96}_{\pm3.13}$ |
| | | 100 | $75.61_{\pm2.48}$ | $70.90_{\pm4.96}$ | $70.33_{\pm3.28}$ | $58.08_{\pm6.71}$ | $48.78_{\pm2.92}$ | $70.79_{\pm2.88}$ | $49.15_{\pm7.71}$ | $\mathbf{76.65}_{\pm2.03}$ | $76.56_{\pm2.29}$ |
| | | 200 | $78.97_{\pm1.46}$ | $71.54_{\pm4.84}$ | $71.42_{\pm4.30}$ | $55.78_{\pm7.76}$ | $47.63_{\pm3.98}$ | $72.74_{\pm3.46}$ | $50.75_{\pm6.70}$ | $79.66_{\pm2.49}$ | $\mathbf{79.80}_{\pm2.15}$ |
| | | 500 | $81.26_{\pm1.42}$ | $74.57_{\pm3.49}$ | $76.32_{\pm2.93}$ | $52.78_{\pm4.17}$ | $47.31_{\pm2.99}$ | $75.67_{\pm2.27}$ | $48.05_{\pm7.50}$ | $\mathbf{81.36}_{\pm1.69}$ | $81.10_{\pm1.52}$ |
| | steel | 20 | $57.55_{\pm4.83}$ | $54.39_{\pm7.65}$ | $52.13_{\pm5.78}$ | $51.21_{\pm6.68}$ | $51.28_{\pm4.58}$ | $52.03_{\pm4.88}$ | $49.25_{\pm4.36}$ | $63.27_{\pm5.66}$ | $\mathbf{63.30}_{\pm5.47}$ |
| | | 50 | $64.84_{\pm4.07}$ | $59.06_{\pm3.85}$ | $56.70_{\pm4.96}$ | $53.24_{\pm4.77}$ | $49.18_{\pm4.67}$ | $57.16_{\pm4.35}$ | $49.90_{\pm3.96}$ | $78.55_{\pm5.17}$ | $\mathbf{79.99}_{\pm6.37}$ |
| | | 100 | $72.48_{\pm4.51}$ | $61.74_{\pm4.16}$ | $59.37_{\pm5.08}$ | $52.79_{\pm4.06}$ | $45.01_{\pm5.84}$ | $57.26_{\pm4.46}$ | $49.66_{\pm3.56}$ | $90.12_{\pm4.20}$ | $\mathbf{92.33}_{\pm2.57}$ |
| | | 200 | $77.85_{\pm3.50}$ | $65.45_{\pm4.07}$ | $63.31_{\pm4.63}$ | $51.02_{\pm2.94}$ | $43.71_{\pm7.51}$ | $61.41_{\pm4.21}$ | $50.20_{\pm3.70}$ | $94.61_{\pm1.75}$ | $\mathbf{95.54}_{\pm1.45}$ |
| | | 500 | $84.21_{\pm3.52}$ | $70.26_{\pm4.93}$ | $70.62_{\pm5.09}$ | $49.97_{\pm4.05}$ | $46.98_{\pm4.30}$ | $66.33_{\pm5.72}$ | $51.60_{\pm3.43}$ | $96.31_{\pm1.25}$ | $\mathbf{97.04}_{\pm1.17}$ |
| | stock | 20 | $81.62_{\pm4.62}$ | $69.35_{\pm8.30}$ | $51.71_{\pm12.09}$ | $67.05_{\pm11.69}$ | $75.94_{\pm14.43}$ | $64.16_{\pm9.18}$ | $48.55_{\pm12.54}$ | $82.55_{\pm4.42}$ | $\mathbf{83.60}_{\pm4.09}$ |
| | | 50 | $87.43_{\pm2.47}$ | $76.07_{\pm5.02}$ | $69.12_{\pm5.00}$ | $70.61_{\pm8.50}$ | $85.99_{\pm2.82}$ | $78.38_{\pm3.56}$ | $49.74_{\pm9.74}$ | $88.09_{\pm2.37}$ | $\mathbf{88.49}_{\pm2.34}$ |
| | | 100 | $89.63_{\pm1.30}$ | $81.18_{\pm4.24}$ | $78.44_{\pm3.91}$ | $72.05_{\pm5.43}$ | $88.27_{\pm2.34}$ | $84.02_{\pm2.86}$ | $51.07_{\pm10.98}$ | $90.13_{\pm1.57}$ | $\mathbf{90.58}_{\pm1.34}$ |
| | | 200 | $\mathbf{91.11}_{\pm1.29}$ | $84.05_{\pm2.63}$ | $82.12_{\pm2.93}$ | $75.44_{\pm3.54}$ | $89.92_{\pm1.56}$ | $85.66_{\pm2.26}$ | $49.74_{\pm12.06}$ | $91.09_{\pm1.52}$ | $91.07_{\pm1.07}$ |
| *More than 10 classes* | energy | 50 | N/A | $7.17_{\pm1.81}$ | $5.20_{\pm2.04}$ | $5.33_{\pm1.61}$ | $4.21_{\pm1.69}$ | $7.26_{\pm1.89}$ | $4.52_{\pm0.57}$ | N/A | $\mathbf{23.80}_{\pm2.60}$ |
| | | 100 | N/A | – | $7.66_{\pm1.96}$ | $5.52_{\pm1.49}$ | $4.08_{\pm1.40}$ | $9.87_{\pm2.01}$ | $4.01_{\pm1.12}$ | N/A | $\mathbf{30.15}_{\pm3.21}$ |
| | | 200 | N/A | – | $7.57_{\pm2.01}$ | $6.92_{\pm1.96}$ | $3.42_{\pm1.25}$ | $11.85_{\pm2.39}$ | $4.18_{\pm0.91}$ | N/A | $\mathbf{35.74}_{\pm3.59}$ |
| | collins | 100 | N/A | – | $5.51_{\pm0.87}$ | $4.58_{\pm0.95}$ | $11.74_{\pm1.73}$ | $5.34_{\pm1.48}$ | $3.77_{\pm0.63}$ | N/A | $\mathbf{13.12}_{\pm1.75}$ |
| | | 200 | $\mathbf{17.60}_{\pm1.83}$ | – | $5.55_{\pm1.34}$ | $4.70_{\pm0.83}$ | $13.64_{\pm1.54}$ | $5.46_{\pm0.90}$ | $3.92_{\pm0.92}$ | N/A | $16.80_{\pm1.55}$ |
| | texture | 50 | $75.50_{\pm2.93}$ | $23.86_{\pm8.05}$ | $12.00_{\pm4.19}$ | $12.16_{\pm3.20}$ | $17.18_{\pm5.34}$ | $16.82_{\pm6.16}$ | $9.63_{\pm2.75}$ | N/A | $\mathbf{78.84}_{\pm3.35}$ |
| | | 100 | $83.77_{\pm2.52}$ | $26.87_{\pm4.19}$ | $14.05_{\pm6.20}$ | $13.95_{\pm5.32}$ | $20.08_{\pm5.52}$ | $23.82_{\pm5.96}$ | $10.21_{\pm2.69}$ | N/A | $\mathbf{85.88}_{\pm1.87}$ |
| | | 200 | $87.96_{\pm1.79}$ | $21.59_{\pm8.99}$ | $15.36_{\pm4.43}$ | $12.17_{\pm4.57}$ | $20.59_{\pm5.96}$ | $42.28_{\pm5.81}$ | $9.52_{\pm2.91}$ | N/A | $\mathbf{88.84}_{\pm1.68}$ |
| | | 500 | $\mathbf{91.65}_{\pm1.14}$ | $34.48_{\pm9.28}$ | $26.45_{\pm8.77}$ | $12.37_{\pm4.63}$ | $20.07_{\pm5.56}$ | $57.94_{\pm4.77}$ | $10.22_{\pm3.69}$ | N/A | $91.36_{\pm1.12}$ |
| **Average rank** | | | $2.76_{\pm0.76}$ | $4.52_{\pm0.77}$ | $6.18_{\pm0.95}$ | $7.48_{\pm0.76}$ | $7.03_{\pm2.07}$ | $4.91_{\pm1.16}$ | $8.52_{\pm0.57}$ | $2.36_{\pm0.89}$ | $\mathbf{1.24}_{\pm0.56}$ |

Table 23: **Classification accuracy** (%) of LR, comparing data sharing on eight real-world tabular datasets with varied real data availability. We report the mean ± std balanced accuracy and average accuracy rank across datasets. A higher rank implies higher accuracy. Note that "N/A" denotes the inapplicability of a specific generator. Different from Table 1, "−" denotes a generator cannot satisfy the requirement of generating 500 stratified samples even after generating 10,000 synthetic samples. The results of these inapplicable or failed generators are computed with the mean results of other methods. We **bold** the highest result for each dataset of different sample sizes. TVAE learns the joint distribution $p(\mathbf{x}, y)$ and fails to maintain the original training label distribution. TabEBM achieves the best overall performance against Baseline and benchmark generators.

| Datasets | | $N_{\text{real}}$ | SMOTE | TVAE | CTGAN | NFLOW | TabDDPM | ARF | GOGGLE | TabPFGen | **TabEBM** |
|---|---|---|---|---|---|---|---|---|---|---|---|
| *At most 10 classes* | protein | 20 | N/A | $14.51_{\pm3.57}$ | $13.00_{\pm5.74}$ | $12.08_{\pm4.67}$ | $13.83_{\pm3.50}$ | $22.46_{\pm5.01}$ | $12.52_{\pm2.66}$ | $38.00_{\pm2.16}$ | $\mathbf{38.04}_{\pm2.34}$ |
| | | 50 | $61.15_{\pm4.22}$ | $24.43_{\pm3.31}$ | $16.81_{\pm5.60}$ | $12.55_{\pm2.78}$ | $14.74_{\pm4.36}$ | $33.77_{\pm5.53}$ | $12.26_{\pm4.83}$ | $\mathbf{63.01}_{\pm3.66}$ | $62.94_{\pm4.10}$ |
| | | 100 | $78.16_{\pm3.53}$ | $40.22_{\pm4.23}$ | $27.14_{\pm5.20}$ | $12.40_{\pm5.31}$ | $19.34_{\pm3.12}$ | $40.99_{\pm5.14}$ | $12.33_{\pm6.06}$ | $\mathbf{80.54}_{\pm3.22}$ | $80.18_{\pm2.97}$ |
| | | 200 | $88.75_{\pm1.37}$ | $45.70_{\pm7.29}$ | $34.17_{\pm6.11}$ | $13.87_{\pm4.39}$ | $22.01_{\pm5.64}$ | $44.76_{\pm4.63}$ | $12.78_{\pm5.20}$ | $\mathbf{90.89}_{\pm1.53}$ | $90.09_{\pm1.86}$ |
| | | 500 | $94.29_{\pm1.17}$ | $60.39_{\pm2.81}$ | $47.87_{\pm7.19}$ | $12.58_{\pm3.76}$ | $24.69_{\pm5.87}$ | $58.91_{\pm2.80}$ | $9.79_{\pm3.65}$ | $\mathbf{95.86}_{\pm1.59}$ | $95.38_{\pm1.40}$ |
| | fourier | 20 | N/A | — | $11.64_{\pm4.35}$ | $10.16_{\pm2.99}$ | $8.48_{\pm2.86}$ | $18.75_{\pm4.29}$ | $11.38_{\pm4.85}$ | $42.98_{\pm5.14}$ | $\mathbf{43.04}_{\pm5.10}$ |
| | | 50 | $58.24_{\pm2.05}$ | — | N/A | $9.82_{\pm4.25}$ | $13.48_{\pm2.85}$ | $33.97_{\pm3.71}$ | $12.00_{\pm2.92}$ | $\mathbf{60.30}_{\pm1.58}$ | $60.28_{\pm1.63}$ |
| | | 100 | $65.32_{\pm2.38}$ | $28.60_{\pm2.38}$ | $21.26_{\pm2.88}$ | $11.24_{\pm3.65}$ | $19.72_{\pm3.78}$ | $37.64_{\pm3.66}$ | $11.34_{\pm2.54}$ | $67.36_{\pm2.21}$ | $\mathbf{67.38}_{\pm2.34}$ |
| | | 200 | $70.74_{\pm2.13}$ | $37.50_{\pm4.11}$ | $28.86_{\pm4.32}$ | $10.12_{\pm4.56}$ | $33.58_{\pm3.65}$ | $42.75_{\pm2.57}$ | $7.52_{\pm2.05}$ | $\mathbf{72.20}_{\pm3.02}$ | $72.18_{\pm2.94}$ |
| | | 500 | $73.96_{\pm0.94}$ | $46.84_{\pm2.99}$ | $41.18_{\pm4.18}$ | $10.60_{\pm3.33}$ | $31.88_{\pm9.69}$ | $48.67_{\pm1.58}$ | $10.56_{\pm3.12}$ | $75.58_{\pm1.99}$ | $\mathbf{76.00}_{\pm2.93}$ |
| | biodeg | 20 | $69.93_{\pm5.59}$ | $66.92_{\pm7.16}$ | $50.10_{\pm10.26}$ | $56.08_{\pm10.54}$ | $46.15_{\pm4.22}$ | $61.15_{\pm7.27}$ | $45.12_{\pm9.23}$ | $70.78_{\pm3.95}$ | $\mathbf{71.24}_{\pm4.85}$ |
| | | 50 | $74.71_{\pm3.39}$ | $70.19_{\pm4.89}$ | $62.90_{\pm6.28}$ | $56.85_{\pm10.24}$ | $42.74_{\pm2.82}$ | $73.38_{\pm2.61}$ | $54.52_{\pm12.44}$ | $75.67_{\pm2.36}$ | $\mathbf{76.47}_{\pm2.99}$ |
| | | 100 | $77.48_{\pm1.83}$ | $71.77_{\pm5.56}$ | $71.42_{\pm3.27}$ | $59.17_{\pm7.35}$ | $46.24_{\pm3.89}$ | $74.48_{\pm2.84}$ | $48.03_{\pm12.56}$ | $77.94_{\pm2.45}$ | $\mathbf{78.34}_{\pm2.11}$ |
| | | 200 | $80.56_{\pm1.77}$ | $75.37_{\pm4.47}$ | $72.61_{\pm5.69}$ | $55.71_{\pm7.21}$ | $44.16_{\pm4.71}$ | $75.09_{\pm2.93}$ | $52.94_{\pm11.50}$ | $81.21_{\pm2.06}$ | $\mathbf{81.50}_{\pm1.84}$ |
| | | 500 | $\mathbf{82.68}_{\pm1.01}$ | $77.92_{\pm2.73}$ | $77.55_{\pm2.55}$ | $54.90_{\pm5.52}$ | $46.53_{\pm2.91}$ | $77.87_{\pm2.10}$ | $46.40_{\pm12.51}$ | $82.38_{\pm1.61}$ | $82.14_{\pm1.30}$ |
| | steel | 20 | $56.80_{\pm5.46}$ | $54.06_{\pm11.24}$ | $54.41_{\pm5.57}$ | $52.14_{\pm7.31}$ | $51.40_{\pm6.83}$ | $54.38_{\pm4.84}$ | $48.32_{\pm6.98}$ | $66.75_{\pm9.74}$ | $\mathbf{67.05}_{\pm9.39}$ |
| | | 50 | $66.80_{\pm6.52}$ | $61.11_{\pm3.73}$ | $58.25_{\pm6.67}$ | $52.95_{\pm6.68}$ | $49.07_{\pm7.58}$ | $61.31_{\pm4.45}$ | $48.86_{\pm7.10}$ | $\mathbf{93.51}_{\pm4.94}$ | $92.13_{\pm4.90}$ |
| | | 100 | $79.38_{\pm4.02}$ | $64.64_{\pm3.99}$ | $60.57_{\pm5.90}$ | $54.40_{\pm5.67}$ | $42.51_{\pm5.11}$ | $58.86_{\pm3.66}$ | $50.95_{\pm3.77}$ | $\mathbf{99.21}_{\pm0.86}$ | $99.17_{\pm0.93}$ |
| | | 200 | $82.66_{\pm3.91}$ | $71.78_{\pm3.68}$ | $65.44_{\pm5.02}$ | $51.72_{\pm4.16}$ | $38.19_{\pm8.24}$ | $63.31_{\pm5.98}$ | $53.46_{\pm2.96}$ | $99.45_{\pm0.69}$ | $\mathbf{99.51}_{\pm0.69}$ |
| | | 500 | $90.21_{\pm3.80}$ | $76.49_{\pm5.04}$ | $75.41_{\pm6.20}$ | $48.71_{\pm6.15}$ | $46.00_{\pm4.77}$ | $72.11_{\pm7.56}$ | $52.55_{\pm3.50}$ | $99.70_{\pm0.22}$ | $\mathbf{99.72}_{\pm0.20}$ |
| | stock | 20 | $80.31_{\pm4.03}$ | $71.68_{\pm8.45}$ | $54.58_{\pm13.04}$ | $70.64_{\pm9.77}$ | $71.05_{\pm11.98}$ | $68.03_{\pm7.38}$ | $49.84_{\pm19.78}$ | $79.58_{\pm4.43}$ | $\mathbf{80.33}_{\pm3.52}$ |
| | | 50 | $81.28_{\pm2.87}$ | $75.75_{\pm3.84}$ | $69.67_{\pm3.14}$ | $72.99_{\pm6.89}$ | $77.72_{\pm4.99}$ | $76.64_{\pm2.54}$ | $49.01_{\pm17.32}$ | $\mathbf{82.37}_{\pm3.13}$ | $82.06_{\pm2.58}$ |
| | | 100 | $\mathbf{83.90}_{\pm1.88}$ | $78.79_{\pm3.25}$ | $77.37_{\pm3.85}$ | $75.64_{\pm3.63}$ | $80.43_{\pm3.70}$ | $78.22_{\pm2.48}$ | $49.60_{\pm17.93}$ | $83.65_{\pm1.65}$ | $83.56_{\pm1.81}$ |
| | | 200 | $83.61_{\pm1.32}$ | $79.31_{\pm2.69}$ | $80.20_{\pm2.16}$ | $76.37_{\pm1.92}$ | $79.32_{\pm2.34}$ | $78.89_{\pm2.42}$ | $48.18_{\pm17.04}$ | $83.69_{\pm1.41}$ | $\mathbf{83.82}_{\pm1.34}$ |
| *More than 10 classes* | energy | 50 | N/A | $7.51_{\pm1.98}$ | $5.79_{\pm2.56}$ | $5.89_{\pm1.99}$ | $3.42_{\pm2.02}$ | $8.72_{\pm1.70}$ | $4.52_{\pm1.16}$ | N/A | $\mathbf{21.56}_{\pm1.66}$ |
| | | 100 | N/A | — | $7.16_{\pm1.69}$ | $6.15_{\pm1.09}$ | $4.88_{\pm1.99}$ | $10.67_{\pm2.27}$ | $4.03_{\pm1.33}$ | N/A | $\mathbf{27.59}_{\pm2.14}$ |
| | | 200 | N/A | — | $7.82_{\pm2.19}$ | $6.63_{\pm2.11}$ | $3.48_{\pm1.64}$ | $12.96_{\pm2.22}$ | $4.34_{\pm1.04}$ | N/A | $\mathbf{33.01}_{\pm2.44}$ |
| | collins | 100 | N/A | — | $5.33_{\pm0.98}$ | $5.49_{\pm1.10}$ | $12.30_{\pm2.14}$ | $5.98_{\pm1.61}$ | $3.61_{\pm1.35}$ | N/A | $\mathbf{14.02}_{\pm2.45}$ |
| | | 200 | $18.91_{\pm1.64}$ | — | $5.36_{\pm1.46}$ | $5.13_{\pm0.92}$ | $14.70_{\pm1.70}$ | $6.04_{\pm0.86}$ | $3.28_{\pm1.59}$ | N/A | $\mathbf{19.11}_{\pm1.47}$ |
| | texture | 50 | $86.20_{\pm2.75}$ | $25.72_{\pm6.89}$ | $13.32_{\pm4.63}$ | $13.86_{\pm4.99}$ | $23.90_{\pm6.56}$ | $18.17_{\pm8.82}$ | $10.11_{\pm3.94}$ | N/A | $\mathbf{88.46}_{\pm2.81}$ |
| | | 100 | $93.17_{\pm2.01}$ | $26.04_{\pm2.24}$ | $12.34_{\pm5.07}$ | $12.44_{\pm5.67}$ | $27.67_{\pm6.90}$ | $22.55_{\pm5.85}$ | $12.00_{\pm5.29}$ | N/A | $\mathbf{94.23}_{\pm1.30}$ |
| | | 200 | $95.70_{\pm1.24}$ | $16.65_{\pm9.20}$ | $17.15_{\pm5.04}$ | $13.14_{\pm6.40}$ | $29.67_{\pm6.59}$ | $43.54_{\pm7.07}$ | $11.12_{\pm5.24}$ | N/A | $\mathbf{95.99}_{\pm1.14}$ |
| | | 500 | $\mathbf{97.17}_{\pm0.38}$ | $39.69_{\pm10.43}$ | $27.24_{\pm8.08}$ | $11.68_{\pm4.59}$ | $27.75_{\pm7.52}$ | $60.15_{\pm5.45}$ | $10.95_{\pm5.25}$ | N/A | $96.53_{\pm0.54}$ |
| **Average rank** | | | $2.82_{\pm0.80}$ | $4.70_{\pm0.78}$ | $6.39_{\pm1.14}$ | $7.48_{\pm0.76}$ | $6.79_{\pm2.00}$ | $4.64_{\pm1.37}$ | $8.52_{\pm0.71}$ | $2.24_{\pm1.10}$ | $\mathbf{1.42}_{\pm0.61}$ |

Table 24: **Classification accuracy** (%) of KNN, comparing data sharing on eight real-world tabular datasets with varied real data availability. We report the mean ± std balanced accuracy and average accuracy rank across datasets. A higher rank implies higher accuracy. Note that "N/A" denotes the inapplicability of a specific generator. Different from Table 1, "−" denotes a generator cannot satisfy the requirement of generating 500 stratified samples even after generating 10,000 synthetic samples. The results of these inapplicable or failed generators are computed with the mean results of other methods. We **bold** the highest result for each dataset of different sample sizes. TVAE learns the joint distribution $p(\mathbf{x}, y)$ and fails to maintain the original training label distribution. TabEBM achieves the best overall performance against Baseline and benchmark generators.

| Datasets | | $N_\text{real}$ | SMOTE | TVAE | CTGAN | NFLOW | TabDDPM | ARF | GOGGLE | TabPFGen | **TabEBM** |
|---|---|---|---|---|---|---|---|---|---|---|---|
| *At most 10 classes* | protein | 20 | N/A | $16.84_{\pm3.19}$ | $12.84_{\pm3.39}$ | $11.26_{\pm3.06}$ | $11.63_{\pm1.92}$ | $16.53_{\pm3.52}$ | $11.76_{\pm2.51}$ | $35.75_{\pm4.48}$ | $\mathbf{35.76}_{\pm4.39}$ |
| | | 50 | $\mathbf{55.10}_{\pm3.65}$ | $27.34_{\pm2.43}$ | $15.55_{\pm4.09}$ | $13.25_{\pm4.00}$ | $12.03_{\pm3.72}$ | $24.09_{\pm4.43}$ | $12.53_{\pm2.50}$ | $53.42_{\pm3.59}$ | $53.44_{\pm3.32}$ |
| | | 100 | $\mathbf{69.44}_{\pm2.83}$ | $39.23_{\pm3.86}$ | $22.23_{\pm2.89}$ | $12.64_{\pm3.23}$ | $16.71_{\pm3.38}$ | $30.64_{\pm4.80}$ | $13.66_{\pm2.29}$ | $67.23_{\pm2.65}$ | $67.35_{\pm2.69}$ |
| | | 200 | $\mathbf{77.12}_{\pm2.73}$ | $41.50_{\pm4.16}$ | $27.68_{\pm5.17}$ | $14.16_{\pm4.10}$ | $20.29_{\pm3.05}$ | $36.01_{\pm2.74}$ | $11.91_{\pm4.20}$ | $75.56_{\pm2.00}$ | $76.17_{\pm2.04}$ |
| | | 500 | $83.82_{\pm1.86}$ | $50.56_{\pm5.32}$ | $35.07_{\pm4.52}$ | $13.44_{\pm2.78}$ | $19.90_{\pm3.06}$ | $46.40_{\pm3.68}$ | $11.99_{\pm2.62}$ | $83.63_{\pm1.63}$ | $\mathbf{84.35}_{\pm1.33}$ |
| | fourier | 20 | N/A | — | $19.60_{\pm4.25}$ | $11.06_{\pm3.51}$ | $9.16_{\pm1.73}$ | $15.58_{\pm3.19}$ | $9.78_{\pm2.29}$ | $42.66_{\pm5.78}$ | $\mathbf{42.78}_{\pm5.83}$ |
| | | 50 | $60.16_{\pm1.62}$ | — | N/A | $9.74_{\pm2.10}$ | $11.66_{\pm2.39}$ | $24.32_{\pm4.59}$ | $12.86_{\pm2.44}$ | $58.52_{\pm1.71}$ | $58.56_{\pm1.99}$ |
| | | 100 | $\mathbf{66.54}_{\pm2.75}$ | $38.18_{\pm5.70}$ | $19.86_{\pm3.82}$ | $10.68_{\pm3.49}$ | $18.42_{\pm3.40}$ | $27.64_{\pm4.26}$ | $9.48_{\pm3.68}$ | $64.64_{\pm2.26}$ | $64.98_{\pm2.58}$ |
| | | 200 | $\mathbf{71.08}_{\pm2.07}$ | $42.72_{\pm2.73}$ | $31.68_{\pm4.44}$ | $9.70_{\pm2.10}$ | $26.96_{\pm2.98}$ | $34.32_{\pm4.26}$ | $9.76_{\pm2.16}$ | $70.00_{\pm1.77}$ | $70.50_{\pm1.86}$ |
| | | 500 | $\mathbf{75.06}_{\pm1.95}$ | $47.58_{\pm3.46}$ | $42.68_{\pm3.18}$ | $10.00_{\pm1.07}$ | $26.18_{\pm3.18}$ | $41.87_{\pm3.25}$ | $10.88_{\pm1.70}$ | $73.46_{\pm1.80}$ | $73.95_{\pm1.45}$ |
| | biodeg | 20 | $69.11_{\pm3.28}$ | $65.06_{\pm6.01}$ | $54.27_{\pm6.17}$ | $52.02_{\pm5.98}$ | $47.87_{\pm4.28}$ | $55.35_{\pm6.64}$ | $47.41_{\pm12.04}$ | $67.89_{\pm4.68}$ | $\mathbf{69.62}_{\pm4.52}$ |
| | | 50 | $72.84_{\pm2.33}$ | $66.10_{\pm4.38}$ | $61.89_{\pm7.87}$ | $52.26_{\pm8.81}$ | $50.78_{\pm5.21}$ | $67.72_{\pm3.53}$ | $53.48_{\pm10.48}$ | $72.01_{\pm4.14}$ | $\mathbf{73.77}_{\pm3.46}$ |
| | | 100 | $\mathbf{75.82}_{\pm2.03}$ | $69.86_{\pm3.93}$ | $70.02_{\pm4.13}$ | $58.95_{\pm5.48}$ | $48.67_{\pm4.05}$ | $69.39_{\pm2.95}$ | $51.11_{\pm13.28}$ | $74.77_{\pm2.03}$ | $75.15_{\pm1.93}$ |
| | | 200 | $\mathbf{79.67}_{\pm0.95}$ | $70.25_{\pm5.58}$ | $71.89_{\pm3.71}$ | $53.54_{\pm7.72}$ | $51.40_{\pm7.02}$ | $70.21_{\pm3.87}$ | $49.42_{\pm6.95}$ | $77.78_{\pm2.42}$ | $78.23_{\pm2.08}$ |
| | | 500 | $\mathbf{82.51}_{\pm1.48}$ | $72.72_{\pm3.89}$ | $75.53_{\pm3.97}$ | $52.92_{\pm5.87}$ | $46.14_{\pm4.19}$ | $73.57_{\pm3.14}$ | $50.03_{\pm3.77}$ | $80.55_{\pm1.56}$ | $81.15_{\pm1.54}$ |
| | steel | 20 | $62.72_{\pm5.46}$ | $56.75_{\pm7.62}$ | $49.84_{\pm5.79}$ | $49.97_{\pm6.59}$ | $52.29_{\pm4.71}$ | $52.67_{\pm5.56}$ | $49.23_{\pm4.34}$ | $\mathbf{70.71}_{\pm3.87}$ | $69.36_{\pm4.02}$ |
| | | 50 | $69.72_{\pm3.48}$ | $60.43_{\pm5.12}$ | $57.65_{\pm4.62}$ | $54.41_{\pm4.04}$ | $48.61_{\pm6.80}$ | $57.24_{\pm5.82}$ | $49.66_{\pm1.98}$ | $\mathbf{82.03}_{\pm3.08}$ | $80.57_{\pm3.35}$ |
| | | 100 | $76.33_{\pm4.19}$ | $63.34_{\pm2.92}$ | $60.89_{\pm5.83}$ | $52.24_{\pm4.96}$ | $42.38_{\pm6.73}$ | $58.01_{\pm6.58}$ | $51.17_{\pm1.99}$ | $\mathbf{87.79}_{\pm3.06}$ | $87.76_{\pm3.25}$ |
| | | 200 | $79.90_{\pm1.97}$ | $68.00_{\pm2.85}$ | $66.31_{\pm4.27}$ | $50.19_{\pm4.02}$ | $43.82_{\pm7.44}$ | $63.29_{\pm4.68}$ | $50.99_{\pm2.53}$ | $90.90_{\pm1.80}$ | $\mathbf{91.07}_{\pm1.75}$ |
| | | 500 | $85.06_{\pm1.90}$ | $73.24_{\pm3.16}$ | $72.49_{\pm2.82}$ | $49.19_{\pm4.79}$ | $45.77_{\pm8.19}$ | $68.81_{\pm4.42}$ | $52.45_{\pm3.59}$ | $92.81_{\pm1.15}$ | $\mathbf{92.83}_{\pm1.35}$ |
| | stock | 20 | $84.27_{\pm5.28}$ | $66.05_{\pm9.50}$ | $52.15_{\pm10.98}$ | $58.69_{\pm15.22}$ | $77.28_{\pm18.40}$ | $58.03_{\pm12.41}$ | $45.13_{\pm15.32}$ | $84.45_{\pm4.02}$ | $\mathbf{84.69}_{\pm4.16}$ |
| | | 50 | $89.64_{\pm2.35}$ | $75.40_{\pm3.27}$ | $67.64_{\pm6.93}$ | $64.81_{\pm9.27}$ | $88.94_{\pm1.62}$ | $77.67_{\pm4.49}$ | $48.55_{\pm11.95}$ | $89.69_{\pm1.89}$ | $\mathbf{89.70}_{\pm1.90}$ |
| | | 100 | $91.93_{\pm0.77}$ | $80.87_{\pm2.69}$ | $78.37_{\pm3.93}$ | $67.33_{\pm6.34}$ | $91.03_{\pm1.15}$ | $84.53_{\pm3.34}$ | $53.19_{\pm11.60}$ | $91.98_{\pm0.63}$ | $\mathbf{92.35}_{\pm0.75}$ |
| | | 200 | $\mathbf{93.46}_{\pm0.93}$ | $84.65_{\pm1.61}$ | $81.64_{\pm3.69}$ | $70.56_{\pm3.41}$ | $91.81_{\pm0.69}$ | $85.58_{\pm2.64}$ | $49.98_{\pm11.39}$ | $92.53_{\pm1.09}$ | $92.94_{\pm0.99}$ |
| *More than 10 classes* | energy | 50 | N/A | $6.75_{\pm1.25}$ | $4.88_{\pm1.00}$ | $5.21_{\pm1.39}$ | $4.71_{\pm1.44}$ | $6.06_{\pm1.55}$ | $4.63_{\pm0.47}$ | N/A | $\mathbf{25.36}_{\pm2.29}$ |
| | | 100 | N/A | — | $7.53_{\pm1.81}$ | $5.33_{\pm1.43}$ | $3.68_{\pm1.03}$ | $9.05_{\pm2.14}$ | $4.37_{\pm1.22}$ | N/A | $\mathbf{29.63}_{\pm2.43}$ |
| | | 200 | N/A | — | $7.22_{\pm0.99}$ | $6.01_{\pm1.31}$ | $3.30_{\pm0.91}$ | $10.56_{\pm1.57}$ | $3.99_{\pm0.73}$ | N/A | $\mathbf{33.85}_{\pm3.11}$ |
| | collins | 100 | N/A | — | $5.61_{\pm0.95}$ | $4.32_{\pm0.72}$ | $13.46_{\pm1.68}$ | $4.71_{\pm1.10}$ | $3.80_{\pm0.09}$ | N/A | $\mathbf{15.20}_{\pm2.00}$ |
| | | 200 | $19.94_{\pm2.15}$ | — | $5.24_{\pm0.54}$ | $4.49_{\pm0.68}$ | $15.29_{\pm1.82}$ | $4.58_{\pm0.70}$ | $3.89_{\pm0.28}$ | N/A | $17.66_{\pm1.80}$ |
| | texture | 50 | $78.67_{\pm2.72}$ | $22.15_{\pm10.01}$ | $12.36_{\pm3.58}$ | $11.19_{\pm3.12}$ | $15.85_{\pm5.14}$ | $13.67_{\pm6.44}$ | $9.05_{\pm0.13}$ | N/A | $75.57_{\pm2.61}$ |
| | | 100 | $85.31_{\pm2.44}$ | $25.80_{\pm5.61}$ | $13.82_{\pm7.03}$ | $11.67_{\pm4.14}$ | $19.32_{\pm4.92}$ | $25.54_{\pm5.86}$ | $9.09_{\pm0.10}$ | N/A | $84.69_{\pm1.70}$ |
| | | 200 | $88.17_{\pm1.71}$ | $18.33_{\pm7.83}$ | $13.85_{\pm3.95}$ | $10.55_{\pm4.55}$ | $23.17_{\pm7.57}$ | $42.92_{\pm6.67}$ | $9.07_{\pm0.56}$ | N/A | $\mathbf{89.16}_{\pm1.75}$ |
| | | 500 | $90.78_{\pm1.16}$ | $29.29_{\pm7.03}$ | $24.91_{\pm9.99}$ | $10.23_{\pm3.48}$ | $23.85_{\pm9.99}$ | $57.58_{\pm4.98}$ | $10.12_{\pm2.55}$ | N/A | $\mathbf{91.23}_{\pm0.95}$ |
| **Average rank** | | | $2.03_{\pm0.99}$ | $4.39_{\pm0.84}$ | $6.03_{\pm1.07}$ | $7.70_{\pm0.81}$ | $6.97_{\pm2.05}$ | $5.30_{\pm0.92}$ | $8.33_{\pm0.82}$ | $2.73_{\pm0.83}$ | $\mathbf{1.52}_{\pm0.51}$ |

Table 25: **Classification accuracy** (%) of MLP, comparing data sharing on eight real-world tabular datasets with varied real data availability. We report the mean ± std balanced accuracy and average accuracy rank across datasets. A higher rank implies higher accuracy. Note that "N/A" denotes the inapplicability of a specific generator. Different from Table 1, "−" denotes a generator cannot satisfy the requirement of generating 500 stratified samples even after generating 10,000 synthetic samples. The results of these inapplicable or failed generators are computed with the mean results of other methods. We **bold** the highest result for each dataset of different sample sizes. TVAE learns the joint distribution $p(\mathbf{x}, y)$ and fails to maintain the original training label distribution. TabEBM achieves the best overall performance against Baseline and benchmark generators.

| Datasets | | $N_{\text{real}}$ | SMOTE | TVAE | CTGAN | NFLOW | TabDDPM | ARF | GOGGLE | TabPFGen | **TabEBM** |
|---|---|---|---|---|---|---|---|---|---|---|---|
| *At most 10 classes* | protein | 20 | N/A | $15.81_{\pm3.26}$ | $12.08_{\pm4.27}$ | $13.06_{\pm5.22}$ | $12.93_{\pm3.80}$ | $20.17_{\pm4.44}$ | $12.98_{\pm2.50}$ | $36.01_{\pm2.68}$ | $\mathbf{36.23_{\pm2.49}}$ |
| | | 50 | $56.85_{\pm4.56}$ | $26.83_{\pm4.86}$ | $14.19_{\pm4.81}$ | $13.68_{\pm1.80}$ | $14.92_{\pm3.38}$ | $31.36_{\pm4.84}$ | $13.78_{\pm3.91}$ | $58.69_{\pm4.22}$ | $\mathbf{58.83_{\pm4.59}}$ |
| | | 100 | $75.86_{\pm3.03}$ | $41.60_{\pm4.43}$ | $25.78_{\pm4.77}$ | $11.34_{\pm3.38}$ | $20.82_{\pm4.55}$ | $40.31_{\pm5.63}$ | $14.54_{\pm3.04}$ | $77.47_{\pm3.39}$ | $\mathbf{77.56_{\pm3.65}}$ |
| | | 200 | $87.85_{\pm1.99}$ | $50.87_{\pm6.26}$ | $32.49_{\pm6.76}$ | $12.90_{\pm3.77}$ | $23.85_{\pm5.06}$ | $45.40_{\pm5.31}$ | $10.24_{\pm3.54}$ | $\mathbf{90.01_{\pm2.00}}$ | $89.48_{\pm2.00}$ |
| | | 500 | $94.37_{\pm1.66}$ | $62.81_{\pm4.29}$ | $47.22_{\pm6.43}$ | $12.06_{\pm2.49}$ | $24.51_{\pm7.67}$ | $60.31_{\pm4.02}$ | $10.41_{\pm2.60}$ | $\mathbf{96.26_{\pm1.35}}$ | $96.03_{\pm1.56}$ |
| | fourier | 20 | N/A | — | $12.16_{\pm2.77}$ | $10.56_{\pm3.94}$ | $10.38_{\pm2.81}$ | $19.66_{\pm4.97}$ | $11.04_{\pm3.61}$ | $34.40_{\pm3.85}$ | $\mathbf{35.04_{\pm3.63}}$ |
| | | 50 | $52.72_{\pm2.11}$ | — | N/A | $9.42_{\pm3.23}$ | $13.78_{\pm3.07}$ | $33.94_{\pm2.46}$ | $10.88_{\pm3.14}$ | $55.12_{\pm1.81}$ | $\mathbf{55.30_{\pm1.65}}$ |
| | | 100 | $60.82_{\pm2.85}$ | $33.16_{\pm3.91}$ | $21.44_{\pm3.15}$ | $10.88_{\pm4.21}$ | $21.96_{\pm2.34}$ | $39.68_{\pm4.04}$ | $8.12_{\pm3.25}$ | $63.54_{\pm1.81}$ | $\mathbf{64.08_{\pm1.87}}$ |
| | | 200 | $68.64_{\pm2.78}$ | $42.18_{\pm3.95}$ | $30.56_{\pm5.91}$ | $10.94_{\pm4.87}$ | $33.80_{\pm5.10}$ | $43.92_{\pm2.17}$ | $8.46_{\pm2.68}$ | $\mathbf{71.76_{\pm1.62}}$ | $71.36_{\pm1.94}$ |
| | | 500 | $73.78_{\pm1.12}$ | $50.48_{\pm4.03}$ | $45.94_{\pm3.55}$ | $13.10_{\pm4.09}$ | $35.18_{\pm5.84}$ | $51.10_{\pm3.53}$ | $9.66_{\pm2.62}$ | $76.50_{\pm1.87}$ | $\mathbf{77.65_{\pm1.81}}$ |
| | biodeg | 20 | $68.59_{\pm4.24}$ | $62.24_{\pm8.67}$ | $53.56_{\pm5.65}$ | $55.15_{\pm9.09}$ | $45.87_{\pm3.29}$ | $57.76_{\pm4.42}$ | $45.98_{\pm11.23}$ | $72.04_{\pm4.95}$ | $\mathbf{72.10_{\pm4.69}}$ |
| | | 50 | $73.48_{\pm2.67}$ | $65.96_{\pm6.24}$ | $62.15_{\pm7.11}$ | $55.84_{\pm11.19}$ | $45.55_{\pm8.63}$ | $72.40_{\pm3.40}$ | $49.95_{\pm6.48}$ | $77.20_{\pm2.84}$ | $\mathbf{77.24_{\pm3.18}}$ |
| | | 100 | $77.20_{\pm1.72}$ | $70.20_{\pm5.67}$ | $71.90_{\pm3.27}$ | $57.99_{\pm6.21}$ | $49.10_{\pm5.76}$ | $71.42_{\pm2.59}$ | $45.52_{\pm6.61}$ | $78.58_{\pm2.25}$ | $\mathbf{78.93_{\pm2.11}}$ |
| | | 200 | $81.30_{\pm0.91}$ | $72.26_{\pm5.86}$ | $71.59_{\pm4.85}$ | $55.75_{\pm7.93}$ | $44.37_{\pm7.27}$ | $73.82_{\pm4.38}$ | $48.45_{\pm8.83}$ | $82.06_{\pm1.67}$ | $\mathbf{82.25_{\pm1.69}}$ |
| | | 500 | $83.55_{\pm0.62}$ | $74.36_{\pm4.32}$ | $77.55_{\pm2.54}$ | $53.37_{\pm4.84}$ | $43.97_{\pm4.60}$ | $76.51_{\pm2.60}$ | $47.04_{\pm9.41}$ | $\mathbf{83.63_{\pm1.19}}$ | $83.33_{\pm0.86}$ |
| | steel | 20 | $57.49_{\pm5.60}$ | $53.28_{\pm11.47}$ | $50.57_{\pm7.01}$ | $52.01_{\pm8.53}$ | $52.76_{\pm9.11}$ | $51.25_{\pm4.91}$ | $47.98_{\pm5.99}$ | $\mathbf{64.34_{\pm6.21}}$ | $64.33_{\pm5.70}$ |
| | | 50 | $66.70_{\pm3.60}$ | $62.02_{\pm5.08}$ | $57.41_{\pm5.18}$ | $54.51_{\pm6.50}$ | $50.58_{\pm7.20}$ | $59.85_{\pm5.16}$ | $50.78_{\pm6.19}$ | $\mathbf{83.06_{\pm6.05}}$ | $82.38_{\pm5.79}$ |
| | | 100 | $76.17_{\pm4.30}$ | $63.05_{\pm4.11}$ | $60.62_{\pm4.71}$ | $54.81_{\pm5.06}$ | $44.64_{\pm9.29}$ | $59.60_{\pm4.83}$ | $48.93_{\pm2.99}$ | $95.36_{\pm3.13}$ | $\mathbf{95.45_{\pm3.29}}$ |
| | | 200 | $80.23_{\pm2.17}$ | $68.45_{\pm3.46}$ | $64.73_{\pm5.71}$ | $50.83_{\pm4.76}$ | $39.73_{\pm12.23}$ | $62.36_{\pm3.96}$ | $47.89_{\pm7.71}$ | $98.34_{\pm1.12}$ | $\mathbf{98.37_{\pm0.81}}$ |
| | | 500 | $85.76_{\pm3.61}$ | $73.89_{\pm4.48}$ | $71.48_{\pm4.21}$ | $49.32_{\pm6.51}$ | $43.65_{\pm5.66}$ | $68.12_{\pm4.63}$ | $53.56_{\pm4.70}$ | $\mathbf{99.56_{\pm0.34}}$ | $99.45_{\pm0.44}$ |
| | stock | 20 | $82.93_{\pm4.66}$ | $71.78_{\pm7.33}$ | $55.09_{\pm13.94}$ | $68.46_{\pm8.87}$ | $76.71_{\pm15.45}$ | $62.13_{\pm10.43}$ | $52.68_{\pm12.66}$ | $83.83_{\pm3.86}$ | $\mathbf{83.91_{\pm3.99}}$ |
| | | 50 | $89.48_{\pm2.12}$ | $75.93_{\pm3.72}$ | $67.04_{\pm4.78}$ | $73.18_{\pm9.70}$ | $87.47_{\pm2.35}$ | $78.81_{\pm3.68}$ | $48.79_{\pm5.17}$ | $90.25_{\pm1.96}$ | $\mathbf{90.36_{\pm2.20}}$ |
| | | 100 | $90.85_{\pm0.72}$ | $81.61_{\pm3.37}$ | $79.76_{\pm3.51}$ | $71.03_{\pm5.88}$ | $90.17_{\pm1.66}$ | $85.83_{\pm2.70}$ | $50.41_{\pm14.72}$ | $91.60_{\pm1.00}$ | $\mathbf{91.72_{\pm0.97}}$ |
| | | 200 | $92.03_{\pm0.79}$ | $84.67_{\pm2.45}$ | $82.92_{\pm2.01}$ | $74.97_{\pm4.41}$ | $91.21_{\pm1.00}$ | $87.06_{\pm2.18}$ | $44.90_{\pm17.08}$ | $\mathbf{92.24_{\pm0.91}}$ | $92.00_{\pm0.93}$ |
| *More than 10 classes* | energy | 50 | N/A | $7.77_{\pm1.39}$ | $5.64_{\pm2.40}$ | $5.75_{\pm1.72}$ | $4.30_{\pm2.07}$ | $8.28_{\pm1.91}$ | $4.46_{\pm0.25}$ | N/A | $\mathbf{23.91_{\pm1.49}}$ |
| | | 100 | N/A | — | $7.44_{\pm2.38}$ | $5.34_{\pm1.60}$ | $3.68_{\pm1.68}$ | $10.38_{\pm1.67}$ | $4.29_{\pm0.71}$ | N/A | $\mathbf{29.24_{\pm2.45}}$ |
| | | 200 | N/A | — | $8.74_{\pm2.17}$ | $7.38_{\pm2.33}$ | $3.84_{\pm1.29}$ | $12.88_{\pm2.09}$ | $4.08_{\pm1.12}$ | N/A | $\mathbf{38.27_{\pm3.50}}$ |
| | collins | 100 | N/A | — | $6.06_{\pm0.96}$ | $5.04_{\pm1.41}$ | $12.80_{\pm1.48}$ | $5.69_{\pm1.80}$ | $3.67_{\pm0.43}$ | N/A | $\mathbf{13.66_{\pm1.86}}$ |
| | | 200 | $18.91_{\pm1.62}$ | — | $5.78_{\pm1.75}$ | $4.90_{\pm0.94}$ | $15.94_{\pm1.74}$ | $5.84_{\pm1.19}$ | $4.42_{\pm1.18}$ | N/A | $\mathbf{19.49_{\pm1.56}}$ |
| | texture | 50 | $83.45_{\pm3.05}$ | $28.11_{\pm7.00}$ | $14.57_{\pm6.49}$ | $12.56_{\pm2.92}$ | $22.50_{\pm6.71}$ | $18.07_{\pm4.88}$ | $9.88_{\pm4.02}$ | N/A | $\mathbf{85.41_{\pm2.73}}$ |
| | | 100 | $90.73_{\pm1.80}$ | $33.20_{\pm4.80}$ | $15.18_{\pm7.46}$ | $13.72_{\pm6.03}$ | $29.36_{\pm8.98}$ | $25.80_{\pm5.24}$ | $10.77_{\pm5.02}$ | N/A | $\mathbf{91.87_{\pm1.42}}$ |
| | | 200 | $93.31_{\pm1.19}$ | $26.95_{\pm15.32}$ | $18.10_{\pm5.37}$ | $10.39_{\pm4.68}$ | $28.79_{\pm10.29}$ | $44.44_{\pm5.46}$ | $7.70_{\pm2.74}$ | N/A | $\mathbf{93.78_{\pm1.41}}$ |
| | | 500 | $\mathbf{95.61_{\pm1.10}}$ | $39.39_{\pm9.19}$ | $29.60_{\pm8.79}$ | $13.15_{\pm4.59}$ | $27.70_{\pm6.21}$ | $62.88_{\pm5.56}$ | $10.30_{\pm3.10}$ | N/A | $95.06_{\pm0.96}$ |
| **Average rank** | | | $2.91_{\pm0.71}$ | $4.73_{\pm0.75}$ | $6.45_{\pm1.12}$ | $7.48_{\pm0.87}$ | $6.79_{\pm2.13}$ | $4.70_{\pm1.33}$ | $8.39_{\pm0.70}$ | $2.21_{\pm1.05}$ | $\mathbf{1.33_{\pm0.60}}$ |

Table 26: **Classification accuracy** (%) of RF, comparing data sharing on eight real-world tabular datasets with varied real data availability. We report the mean $\pm$ std balanced accuracy and average accuracy rank across datasets. A higher rank implies higher accuracy. Note that "N/A" denotes the inapplicability of a specific generator. Different from Table 1, "−" denotes a generator cannot satisfy the requirement of generating 500 stratified samples even after generating 10,000 synthetic samples. The results of these inapplicable or failed generators are computed with the mean results of other methods. We **bold** the highest result for each dataset of different sample sizes. TVAE learns the joint distribution $p(\mathbf{x}, y)$ and fails to maintain the original training label distribution. TabEBM achieves the best overall performance against Baseline and benchmark generators.

| Datasets | | $N_{\text{real}}$ | SMOTE | TVAE | CTGAN | NFLOW | TabDDPM | ARF | GOGGLE | TabPFGen | **TabEBM** |
|---|---|---|---|---|---|---|---|---|---|---|---|
| *At most 10 classes* | protein | 20 | N/A | $19.29_{\pm3.85}$ | $13.63_{\pm2.95}$ | $12.71_{\pm2.84}$ | $12.75_{\pm2.09}$ | $20.47_{\pm4.29}$ | $13.25_{\pm2.90}$ | $31.86_{\pm2.24}$ | $\mathbf{34.05_{\pm2.24}}$ |
| | | 50 | $56.09_{\pm2.77}$ | $32.29_{\pm3.05}$ | $19.37_{\pm5.89}$ | $12.62_{\pm2.05}$ | $14.87_{\pm2.75}$ | $32.60_{\pm4.15}$ | $13.29_{\pm2.50}$ | $54.25_{\pm1.99}$ | $\mathbf{56.96_{\pm3.32}}$ |
| | | 100 | $71.14_{\pm2.65}$ | $44.50_{\pm3.08}$ | $29.72_{\pm3.69}$ | $12.37_{\pm3.12}$ | $17.71_{\pm3.51}$ | $38.22_{\pm4.85}$ | $13.67_{\pm1.00}$ | $70.68_{\pm3.24}$ | $\mathbf{72.74_{\pm2.97}}$ |
| | | 200 | $81.22_{\pm2.51}$ | $50.10_{\pm3.89}$ | $37.08_{\pm5.17}$ | $14.34_{\pm3.15}$ | $22.23_{\pm3.43}$ | $43.48_{\pm3.74}$ | $11.76_{\pm3.27}$ | $81.93_{\pm2.47}$ | $\mathbf{82.35_{\pm2.09}}$ |
| | | 500 | $87.93_{\pm1.52}$ | $58.17_{\pm5.02}$ | $49.44_{\pm5.57}$ | $12.03_{\pm2.40}$ | $22.88_{\pm2.67}$ | $53.67_{\pm2.94}$ | $10.59_{\pm3.14}$ | $\mathbf{90.81_{\pm1.37}}$ | $89.78_{\pm1.73}$ |
| | fourier | 20 | N/A | − | $11.28_{\pm1.63}$ | $11.52_{\pm2.65}$ | $9.24_{\pm2.16}$ | $17.76_{\pm3.96}$ | $9.78_{\pm1.33}$ | $31.94_{\pm3.25}$ | $\mathbf{38.16_{\pm4.48}}$ |
| | | 50 | $65.32_{\pm3.86}$ | − | N/A | $9.98_{\pm1.98}$ | $12.12_{\pm2.56}$ | $31.76_{\pm3.66}$ | $11.24_{\pm2.70}$ | $65.64_{\pm3.73}$ | $\mathbf{66.04_{\pm2.91}}$ |
| | | 100 | $73.82_{\pm2.70}$ | $44.84_{\pm5.88}$ | $29.68_{\pm5.97}$ | $11.04_{\pm2.51}$ | $21.90_{\pm4.68}$ | $35.60_{\pm4.77}$ | $9.30_{\pm2.87}$ | $74.28_{\pm2.81}$ | $\mathbf{74.92_{\pm3.12}}$ |
| | | 200 | $78.66_{\pm1.64}$ | $48.56_{\pm3.52}$ | $41.94_{\pm5.54}$ | $10.24_{\pm2.93}$ | $31.36_{\pm4.10}$ | $44.80_{\pm3.88}$ | $9.88_{\pm2.32}$ | $79.48_{\pm2.18}$ | $\mathbf{79.52_{\pm2.03}}$ |
| | | 500 | $79.64_{\pm1.39}$ | $54.80_{\pm4.51}$ | $57.06_{\pm4.59}$ | $11.94_{\pm2.26}$ | $25.08_{\pm4.65}$ | $54.04_{\pm3.10}$ | $10.66_{\pm2.71}$ | $81.64_{\pm1.61}$ | $\mathbf{81.85_{\pm0.53}}$ |
| | biodeg | 20 | $\mathbf{69.34_{\pm6.13}}$ | $64.52_{\pm7.42}$ | $52.83_{\pm7.85}$ | $53.88_{\pm7.31}$ | $49.80_{\pm0.45}$ | $60.95_{\pm6.69}$ | $50.47_{\pm9.67}$ | $67.37_{\pm5.45}$ | $67.06_{\pm4.88}$ |
| | | 50 | $70.17_{\pm3.69}$ | $68.14_{\pm3.98}$ | $62.50_{\pm6.74}$ | $54.84_{\pm7.67}$ | $49.94_{\pm1.03}$ | $67.21_{\pm4.25}$ | $51.99_{\pm6.12}$ | $\mathbf{71.94_{\pm4.16}}$ | $71.91_{\pm2.77}$ |
| | | 100 | $74.80_{\pm2.96}$ | $71.82_{\pm3.70}$ | $68.29_{\pm2.67}$ | $56.44_{\pm5.66}$ | $49.62_{\pm0.68}$ | $67.68_{\pm2.88}$ | $50.71_{\pm5.41}$ | $\mathbf{75.68_{\pm2.01}}$ | $74.93_{\pm2.08}$ |
| | | 200 | $77.49_{\pm2.13}$ | $69.22_{\pm3.80}$ | $69.63_{\pm2.86}$ | $55.47_{\pm7.86}$ | $48.70_{\pm1.53}$ | $70.40_{\pm3.01}$ | $48.04_{\pm5.01}$ | $\mathbf{78.64_{\pm2.29}}$ | $76.96_{\pm3.02}$ |
| | | 500 | $\mathbf{80.31_{\pm1.32}}$ | $74.83_{\pm1.76}$ | $74.44_{\pm2.90}$ | $51.30_{\pm3.00}$ | $49.68_{\pm0.61}$ | $73.88_{\pm1.59}$ | $45.73_{\pm8.13}$ | $79.56_{\pm1.95}$ | $78.66_{\pm1.76}$ |
| | steel | 20 | $55.71_{\pm3.47}$ | $55.23_{\pm4.31}$ | $52.14_{\pm3.64}$ | $50.93_{\pm6.29}$ | $49.93_{\pm0.56}$ | $52.02_{\pm5.80}$ | $48.88_{\pm4.70}$ | $57.20_{\pm2.92}$ | $\mathbf{58.27_{\pm2.40}}$ |
| | | 50 | $62.30_{\pm2.94}$ | $56.05_{\pm3.07}$ | $55.40_{\pm4.01}$ | $52.16_{\pm3.88}$ | $49.98_{\pm0.29}$ | $54.55_{\pm2.89}$ | $50.65_{\pm1.92}$ | $64.26_{\pm2.55}$ | $\mathbf{66.98_{\pm3.30}}$ |
| | | 100 | $67.63_{\pm3.75}$ | $58.93_{\pm4.82}$ | $56.65_{\pm4.04}$ | $51.86_{\pm2.52}$ | $47.84_{\pm2.76}$ | $56.13_{\pm2.99}$ | $49.33_{\pm2.96}$ | $72.47_{\pm4.01}$ | $\mathbf{77.42_{\pm3.85}}$ |
| | | 200 | $71.41_{\pm3.46}$ | $62.15_{\pm4.45}$ | $59.89_{\pm4.05}$ | $51.44_{\pm1.39}$ | $48.94_{\pm1.59}$ | $58.23_{\pm2.57}$ | $49.25_{\pm3.05}$ | $81.56_{\pm4.46}$ | $\mathbf{86.14_{\pm4.04}}$ |
| | | 500 | $77.17_{\pm2.01}$ | $67.42_{\pm3.15}$ | $65.34_{\pm4.17}$ | $50.86_{\pm3.85}$ | $49.83_{\pm0.48}$ | $60.96_{\pm4.42}$ | $51.04_{\pm3.62}$ | $86.49_{\pm4.38}$ | $\mathbf{90.60_{\pm4.65}}$ |
| | stock | 20 | $80.81_{\pm4.71}$ | $71.58_{\pm6.48}$ | $48.00_{\pm11.14}$ | $66.07_{\pm12.69}$ | $77.42_{\pm15.77}$ | $61.80_{\pm9.34}$ | $46.62_{\pm11.62}$ | $84.30_{\pm4.96}$ | $\mathbf{84.65_{\pm3.38}}$ |
| | | 50 | $89.45_{\pm2.09}$ | $74.30_{\pm8.64}$ | $66.75_{\pm7.07}$ | $69.95_{\pm10.51}$ | $87.73_{\pm2.21}$ | $79.90_{\pm4.82}$ | $47.28_{\pm12.21}$ | $89.27_{\pm1.89}$ | $\mathbf{90.04_{\pm2.09}}$ |
| | | 100 | $91.38_{\pm1.80}$ | $84.09_{\pm3.75}$ | $79.16_{\pm4.14}$ | $72.00_{\pm4.77}$ | $90.96_{\pm2.04}$ | $86.38_{\pm2.97}$ | $47.78_{\pm10.15}$ | $92.07_{\pm1.56}$ | $\mathbf{92.22_{\pm1.36}}$ |
| | | 200 | $\mathbf{93.46_{\pm0.79}}$ | $86.85_{\pm2.22}$ | $83.47_{\pm3.47}$ | $78.62_{\pm3.19}$ | $92.72_{\pm0.81}$ | $87.66_{\pm0.89}$ | $49.91_{\pm13.88}$ | $93.35_{\pm1.32}$ | $93.07_{\pm0.99}$ |
| *More than 10 classes* | energy | 50 | N/A | $6.42_{\pm1.49}$ | $4.52_{\pm2.22}$ | $4.88_{\pm1.49}$ | $4.67_{\pm1.31}$ | $6.62_{\pm2.47}$ | $4.47_{\pm0.38}$ | N/A | $\mathbf{28.79_{\pm3.70}}$ |
| | | 100 | N/A | − | $7.92_{\pm1.14}$ | $5.40_{\pm1.11}$ | $4.21_{\pm0.94}$ | $9.03_{\pm1.53}$ | $3.37_{\pm1.21}$ | N/A | $\mathbf{39.20_{\pm3.05}}$ |
| | | 200 | N/A | − | $7.12_{\pm1.77}$ | $6.01_{\pm1.38}$ | $3.26_{\pm0.91}$ | $11.78_{\pm3.10}$ | $4.32_{\pm0.75}$ | N/A | $\mathbf{42.55_{\pm3.93}}$ |
| | collins | 100 | N/A | − | $6.10_{\pm0.63}$ | $4.09_{\pm0.82}$ | $11.95_{\pm1.92}$ | $4.95_{\pm1.23}$ | $3.99_{\pm0.64}$ | N/A | $\mathbf{13.67_{\pm1.31}}$ |
| | | 200 | $17.74_{\pm1.75}$ | − | $5.86_{\pm1.39}$ | $4.56_{\pm0.98}$ | $13.22_{\pm1.22}$ | $5.23_{\pm0.63}$ | $4.08_{\pm0.62}$ | N/A | $\mathbf{16.24_{\pm1.45}}$ |
| | texture | 50 | $71.18_{\pm2.80}$ | $22.41_{\pm7.88}$ | $10.33_{\pm1.47}$ | $10.55_{\pm1.87}$ | $12.35_{\pm4.09}$ | $19.83_{\pm5.72}$ | $9.50_{\pm2.91}$ | N/A | $\mathbf{75.77_{\pm3.60}}$ |
| | | 100 | $79.85_{\pm2.02}$ | $26.34_{\pm1.17}$ | $14.84_{\pm6.52}$ | $15.77_{\pm5.11}$ | $14.16_{\pm4.42}$ | $26.07_{\pm6.44}$ | $8.97_{\pm0.33}$ | N/A | $\mathbf{81.99_{\pm2.39}}$ |
| | | 200 | $84.17_{\pm2.07}$ | $22.70_{\pm5.57}$ | $12.59_{\pm4.81}$ | $11.61_{\pm3.40}$ | $11.68_{\pm3.73}$ | $46.83_{\pm4.78}$ | $10.17_{\pm3.08}$ | N/A | $\mathbf{84.90_{\pm1.98}}$ |
| | | 500 | $\mathbf{88.31_{\pm1.36}}$ | $32.92_{\pm10.58}$ | $27.62_{\pm9.73}$ | $13.33_{\pm4.79}$ | $10.84_{\pm2.37}$ | $62.04_{\pm4.01}$ | $9.50_{\pm3.84}$ | N/A | $88.14_{\pm1.50}$ |
| **Average rank** | | | $2.55_{\pm0.84}$ | $4.39_{\pm0.80}$ | $6.09_{\pm1.01}$ | $7.42_{\pm0.90}$ | $7.09_{\pm1.94}$ | $5.15_{\pm1.03}$ | $8.55_{\pm0.62}$ | $2.36_{\pm0.89}$ | $\mathbf{1.39_{\pm0.70}}$ |

Table 27: **Classification accuracy** (%) of XGBoost, comparing data sharing on eight real-world tabular datasets with varied real data availability. We report the mean ± std balanced accuracy and average accuracy rank across datasets. A higher rank implies higher accuracy. Note that "N/A" denotes the inapplicability of a specific generator. Different from Table 1, "−" denotes a generator cannot satisfy the requirement of generating 500 stratified samples even after generating 10,000 synthetic samples. The results of these inapplicable or failed generators are computed with the mean results of other methods. We **bold** the highest result for each dataset of different sample sizes. TVAE learns the joint distribution $p(\mathbf{x}, y)$ and fails to maintain the original training label distribution. TabEBM achieves the best overall performance against Baseline and benchmark generators.

| Datasets | | $N_{\text{real}}$ | SMOTE | TVAE | CTGAN | NFLOW | TabDDPM | ARF | GOGGLE | TabPFGen | **TabEBM** |
|---|---|---|---|---|---|---|---|---|---|---|---|
| *At most 10 classes* | protein | 20 | N/A | $15.47_{\pm2.77}$ | $13.60_{\pm3.62}$ | $15.03_{\pm4.60}$ | $11.77_{\pm3.36}$ | $18.32_{\pm3.84}$ | $14.61_{\pm2.21}$ | $23.54_{\pm4.33}$ | $\mathbf{25.22}_{\pm3.56}$ |
| | | 50 | $38.51_{\pm4.83}$ | $25.36_{\pm3.19}$ | $18.88_{\pm4.31}$ | $13.84_{\pm4.50}$ | $12.72_{\pm1.99}$ | $24.39_{\pm4.09}$ | $13.13_{\pm4.44}$ | $41.88_{\pm6.20}$ | $\mathbf{43.79}_{\pm6.81}$ |
| | | 100 | $60.71_{\pm5.70}$ | $33.23_{\pm4.65}$ | $23.54_{\pm3.74}$ | $14.36_{\pm3.09}$ | $14.08_{\pm1.84}$ | $28.49_{\pm6.26}$ | $14.48_{\pm4.96}$ | $54.91_{\pm6.96}$ | $\mathbf{61.11}_{\pm6.11}$ |
| | | 200 | $73.31_{\pm3.15}$ | $35.50_{\pm6.72}$ | $26.74_{\pm5.01}$ | $15.45_{\pm4.45}$ | $14.71_{\pm3.38}$ | $36.61_{\pm3.61}$ | $10.41_{\pm2.52}$ | $75.20_{\pm2.13}$ | $\mathbf{76.96}_{\pm2.90}$ |
| | | 500 | $83.11_{\pm1.82}$ | $41.18_{\pm6.31}$ | $37.95_{\pm5.62}$ | $13.56_{\pm2.40}$ | $14.47_{\pm2.40}$ | $48.92_{\pm4.09}$ | $11.25_{\pm3.92}$ | $85.30_{\pm1.56}$ | $\mathbf{85.39}_{\pm1.90}$ |
| | fourier | 20 | N/A | − | − | $11.16_{\pm2.78}$ | $10.36_{\pm2.58}$ | $12.38_{\pm3.08}$ | $14.64_{\pm4.90}$ | $9.18_{\pm1.60}$ | $21.80_{\pm4.86}$ | $\mathbf{26.80}_{\pm4.82}$ |
| | | 50 | $41.70_{\pm8.00}$ | − | N/A | $10.06_{\pm1.96}$ | $10.00_{\pm1.41}$ | $19.96_{\pm4.44}$ | $12.24_{\pm3.23}$ | $43.10_{\pm6.37}$ | $\mathbf{47.16}_{\pm5.42}$ |
| | | 100 | $54.52_{\pm5.80}$ | $27.78_{\pm5.49}$ | $19.40_{\pm4.93}$ | $10.98_{\pm2.29}$ | $12.22_{\pm3.24}$ | $24.24_{\pm4.70}$ | $10.34_{\pm2.64}$ | $51.40_{\pm5.42}$ | $\mathbf{52.62}_{\pm5.05}$ |
| | | 200 | $65.96_{\pm4.60}$ | $31.60_{\pm2.49}$ | $24.76_{\pm5.83}$ | $11.84_{\pm3.99}$ | $13.12_{\pm3.24}$ | $31.74_{\pm3.75}$ | $11.36_{\pm2.39}$ | $62.68_{\pm4.32}$ | $\mathbf{66.26}_{\pm4.52}$ |
| | | 500 | $71.44_{\pm1.57}$ | $34.72_{\pm1.86}$ | $36.56_{\pm5.98}$ | $11.64_{\pm2.58}$ | $12.72_{\pm3.83}$ | $37.86_{\pm3.31}$ | $9.42_{\pm1.66}$ | $72.00_{\pm2.94}$ | $\mathbf{74.45}_{\pm1.66}$ |
| | biodeg | 20 | $66.52_{\pm5.96}$ | $60.35_{\pm7.63}$ | $58.55_{\pm6.57}$ | $54.65_{\pm8.42}$ | $50.67_{\pm4.16}$ | $59.80_{\pm7.09}$ | $52.58_{\pm13.73}$ | $65.99_{\pm6.54}$ | $\mathbf{66.60}_{\pm6.87}$ |
| | | 50 | $68.01_{\pm2.67}$ | $63.62_{\pm4.85}$ | $60.53_{\pm7.91}$ | $56.10_{\pm6.94}$ | $48.13_{\pm3.77}$ | $\mathbf{68.90}_{\pm4.66}$ | $52.89_{\pm9.18}$ | $68.60_{\pm4.27}$ | $68.82_{\pm3.18}$ |
| | | 100 | $71.98_{\pm4.21}$ | $68.71_{\pm4.56}$ | $68.58_{\pm3.57}$ | $60.67_{\pm9.00}$ | $49.08_{\pm3.11}$ | $68.74_{\pm3.58}$ | $49.51_{\pm8.42}$ | $\mathbf{74.73}_{\pm1.88}$ | $72.69_{\pm3.36}$ |
| | | 200 | $74.29_{\pm2.14}$ | $68.94_{\pm4.03}$ | $69.87_{\pm3.15}$ | $59.26_{\pm7.34}$ | $47.26_{\pm2.99}$ | $70.89_{\pm3.66}$ | $55.64_{\pm7.90}$ | $76.03_{\pm4.77}$ | $\mathbf{77.15}_{\pm2.53}$ |
| | | 500 | $76.04_{\pm3.09}$ | $70.53_{\pm5.19}$ | $73.65_{\pm3.54}$ | $53.73_{\pm4.78}$ | $47.86_{\pm4.57}$ | $74.39_{\pm2.84}$ | $49.08_{\pm11.19}$ | $\mathbf{78.99}_{\pm2.89}$ | $77.61_{\pm2.83}$ |
| | steel | 20 | $\mathbf{55.76}_{\pm4.40}$ | $53.17_{\pm6.28}$ | $53.71_{\pm9.38}$ | $51.36_{\pm5.69}$ | $51.32_{\pm6.29}$ | $51.36_{\pm6.40}$ | $51.11_{\pm4.18}$ | $55.71_{\pm5.46}$ | $54.95_{\pm5.04}$ |
| | | 50 | $61.07_{\pm3.58}$ | $56.97_{\pm3.93}$ | $54.38_{\pm4.00}$ | $53.54_{\pm3.82}$ | $46.83_{\pm6.18}$ | $53.60_{\pm4.17}$ | $49.46_{\pm6.57}$ | $63.64_{\pm6.52}$ | $\mathbf{71.58}_{\pm14.28}$ |
| | | 100 | $64.12_{\pm5.76}$ | $59.03_{\pm5.74}$ | $57.01_{\pm5.40}$ | $52.86_{\pm3.06}$ | $44.22_{\pm7.97}$ | $55.21_{\pm4.00}$ | $46.92_{\pm7.56}$ | $88.61_{\pm12.72}$ | $\mathbf{96.34}_{\pm2.73}$ |
| | | 200 | $74.96_{\pm6.42}$ | $58.42_{\pm6.66}$ | $59.25_{\pm3.48}$ | $51.94_{\pm3.30}$ | $43.96_{\pm10.64}$ | $58.92_{\pm3.56}$ | $49.58_{\pm5.96}$ | $98.85_{\pm1.71}$ | $\mathbf{99.28}_{\pm0.72}$ |
| | | 500 | $81.79_{\pm5.79}$ | $61.11_{\pm8.40}$ | $65.70_{\pm7.71}$ | $51.68_{\pm2.71}$ | $46.63_{\pm6.69}$ | $59.19_{\pm7.43}$ | $49.98_{\pm5.15}$ | $99.62_{\pm1.11}$ | $\mathbf{99.92}_{\pm0.13}$ |
| | stock | 20 | $77.64_{\pm6.07}$ | $64.61_{\pm10.04}$ | $50.93_{\pm6.99}$ | $67.70_{\pm12.94}$ | $74.63_{\pm13.41}$ | $68.29_{\pm7.28}$ | $45.59_{\pm10.96}$ | $80.14_{\pm4.76}$ | $\mathbf{84.22}_{\pm4.50}$ |
| | | 50 | $84.83_{\pm2.98}$ | $77.89_{\pm7.53}$ | $73.57_{\pm5.76}$ | $70.29_{\pm6.08}$ | $84.75_{\pm4.01}$ | $79.19_{\pm2.51}$ | $54.82_{\pm11.81}$ | $86.97_{\pm3.25}$ | $\mathbf{88.59}_{\pm3.54}$ |
| | | 100 | $87.69_{\pm1.26}$ | $80.12_{\pm8.75}$ | $77.23_{\pm4.37}$ | $71.03_{\pm8.08}$ | $85.72_{\pm4.00}$ | $83.91_{\pm3.27}$ | $55.60_{\pm10.94}$ | $89.21_{\pm3.39}$ | $\mathbf{91.14}_{\pm1.97}$ |
| | | 200 | $90.58_{\pm3.02}$ | $83.34_{\pm3.80}$ | $81.04_{\pm3.24}$ | $76.60_{\pm5.86}$ | $\mathbf{91.52}_{\pm3.49}$ | $87.49_{\pm3.41}$ | $55.43_{\pm12.95}$ | $90.78_{\pm3.56}$ | $90.94_{\pm1.18}$ |
| *More than 10 classes* | energy | 50 | N/A | $7.37_{\pm2.92}$ | $5.16_{\pm2.03}$ | $4.90_{\pm1.44}$ | $3.95_{\pm1.60}$ | $6.62_{\pm1.84}$ | N/A | N/A | $\mathbf{19.38}_{\pm3.84}$ |
| | | 100 | N/A | − | $8.25_{\pm2.78}$ | $5.40_{\pm2.24}$ | $3.95_{\pm1.37}$ | $10.21_{\pm2.45}$ | N/A | N/A | $\mathbf{25.08}_{\pm5.99}$ |
| | | 200 | N/A | − | $6.97_{\pm2.92}$ | $8.56_{\pm2.68}$ | $3.21_{\pm1.53}$ | $11.08_{\pm2.96}$ | N/A | N/A | $\mathbf{31.03}_{\pm4.94}$ |
| | collins | 100 | N/A | − | $4.45_{\pm0.82}$ | $3.98_{\pm0.70}$ | $8.18_{\pm1.43}$ | $5.39_{\pm1.68}$ | N/A | N/A | $\mathbf{9.03}_{\pm1.14}$ |
| | | 200 | $\mathbf{12.52}_{\pm1.98}$ | − | $5.52_{\pm1.57}$ | $4.43_{\pm0.63}$ | $9.06_{\pm1.21}$ | $5.61_{\pm1.10}$ | N/A | N/A | $11.49_{\pm1.45}$ |
| | texture | 50 | $57.99_{\pm3.35}$ | $20.94_{\pm8.45}$ | $9.44_{\pm4.78}$ | $12.63_{\pm3.07}$ | $11.32_{\pm4.19}$ | $14.35_{\pm4.95}$ | N/A | N/A | $\mathbf{68.97}_{\pm4.98}$ |
| | | 100 | $69.80_{\pm4.34}$ | $23.00_{\pm7.10}$ | $14.06_{\pm4.92}$ | $16.18_{\pm5.64}$ | $9.90_{\pm2.39}$ | $19.13_{\pm6.41}$ | N/A | N/A | $\mathbf{76.62}_{\pm2.55}$ |
| | | 200 | $78.46_{\pm2.72}$ | $23.32_{\pm7.04}$ | $15.09_{\pm2.96}$ | $15.15_{\pm3.79}$ | $9.66_{\pm1.64}$ | $33.67_{\pm5.07}$ | N/A | N/A | $\mathbf{80.37}_{\pm2.13}$ |
| | | 500 | $\mathbf{86.39}_{\pm1.70}$ | $31.13_{\pm9.16}$ | $22.88_{\pm7.24}$ | $13.46_{\pm5.72}$ | $10.22_{\pm1.70}$ | $47.06_{\pm3.83}$ | N/A | N/A | $85.83_{\pm1.68}$ |
| **Average rank** | | | $2.76_{\pm0.90}$ | $4.97_{\pm0.92}$ | $6.39_{\pm1.32}$ | $7.42_{\pm0.75}$ | $7.48_{\pm2.35}$ | $4.97_{\pm1.26}$ | $7.11_{\pm2.15}$ | $2.62_{\pm0.95}$ | $\mathbf{1.27}_{\pm0.52}$ |

Table 28: **Classification accuracy** (%) of TabPFN, comparing data sharing on eight real-world tabular datasets with varied real data availability. We report the mean $\pm$ std balanced accuracy and average accuracy rank across datasets. A higher rank implies higher accuracy. Note that "N/A" denotes the inapplicability of a specific generator. Different from Table 1, "$-$" denotes a generator cannot satisfy the requirement of generating 500 stratified samples even after generating 10,000 synthetic samples. The results of these inapplicable or failed generators are computed with the mean results of other methods. We **bold** the highest result for each dataset of different sample sizes. TVAE learns the joint distribution $p(\mathbf{x}, y)$ and fails to maintain the original training label distribution. TabEBM achieves the best overall performance against Baseline and benchmark generators.

| Datasets | | $N_{\text{real}}$ | SMOTE | TVAE | CTGAN | NFLOW | TabDDPM | ARF | GOGGLE | TabPFGen | **TabEBM** |
|---|---|---|---|---|---|---|---|---|---|---|---|
| *At most 10 classes* | protein | 20 | N/A | $16.36_{\pm3.39}$ | $13.22_{\pm1.42}$ | $13.80_{\pm4.56}$ | $12.93_{\pm3.67}$ | $18.55_{\pm2.64}$ | $12.14_{\pm1.16}$ | $33.46_{\pm5.96}$ | $\mathbf{34.50}_{\pm5.86}$ |
| | | 50 | $\mathbf{59.60}_{\pm3.63}$ | $27.51_{\pm3.72}$ | $18.26_{\pm4.54}$ | $13.28_{\pm3.34}$ | $14.95_{\pm3.52}$ | $29.82_{\pm3.75}$ | $13.30_{\pm1.88}$ | $57.68_{\pm2.79}$ | $59.00_{\pm3.97}$ |
| | | 100 | $\mathbf{79.47}_{\pm3.48}$ | $41.49_{\pm6.98}$ | $31.50_{\pm3.05}$ | $13.69_{\pm2.75}$ | $17.21_{\pm3.12}$ | $36.73_{\pm5.05}$ | $12.61_{\pm0.81}$ | $78.93_{\pm4.23}$ | $79.02_{\pm3.74}$ |
| | | 200 | $90.49_{\pm2.23}$ | $48.30_{\pm7.73}$ | $36.99_{\pm5.39}$ | $13.71_{\pm3.79}$ | $21.67_{\pm5.92}$ | $43.53_{\pm5.42}$ | $12.96_{\pm2.86}$ | $\mathbf{91.47}_{\pm1.28}$ | $90.71_{\pm1.30}$ |
| | | 500 | $94.28_{\pm1.36}$ | $58.31_{\pm8.05}$ | $48.52_{\pm4.70}$ | $12.91_{\pm2.56}$ | $23.35_{\pm7.76}$ | $57.37_{\pm3.28}$ | $12.51_{\pm3.23}$ | $\mathbf{95.26}_{\pm1.37}$ | $95.00_{\pm1.28}$ |
| | fourier | 20 | N/A | $-$ | $13.98_{\pm3.05}$ | $10.64_{\pm2.52}$ | $10.56_{\pm1.86}$ | $16.44_{\pm4.20}$ | $10.56_{\pm3.20}$ | $29.42_{\pm6.72}$ | $\mathbf{36.56}_{\pm4.95}$ |
| | | 50 | $52.52_{\pm3.71}$ | $-$ | N/A | $10.24_{\pm1.53}$ | $10.72_{\pm1.53}$ | $27.82_{\pm3.41}$ | $9.90_{\pm0.33}$ | $\mathbf{53.98}_{\pm4.48}$ | $53.92_{\pm3.82}$ |
| | | 100 | $63.36_{\pm3.56}$ | $36.24_{\pm4.25}$ | $28.44_{\pm4.45}$ | $10.30_{\pm1.94}$ | $11.84_{\pm3.10}$ | $30.76_{\pm2.70}$ | $10.32_{\pm2.04}$ | $65.32_{\pm3.26}$ | $\mathbf{65.58}_{\pm3.38}$ |
| | | 200 | $69.62_{\pm4.71}$ | $40.36_{\pm3.77}$ | $40.08_{\pm5.17}$ | $8.82_{\pm3.98}$ | $32.98_{\pm6.25}$ | $38.50_{\pm3.82}$ | $10.30_{\pm0.76}$ | $72.08_{\pm2.45}$ | $\mathbf{72.26}_{\pm2.70}$ |
| | | 500 | $75.14_{\pm1.83}$ | $46.56_{\pm5.61}$ | $53.38_{\pm4.52}$ | $9.52_{\pm1.46}$ | $32.46_{\pm9.02}$ | $49.62_{\pm2.60}$ | $10.68_{\pm1.36}$ | $\mathbf{75.98}_{\pm1.25}$ | $75.60_{\pm0.85}$ |
| | biodeg | 20 | $69.01_{\pm4.56}$ | $65.35_{\pm7.46}$ | $52.29_{\pm8.45}$ | $55.71_{\pm6.77}$ | $50.00_{\pm0.00}$ | $57.11_{\pm8.37}$ | $50.35_{\pm1.11}$ | $70.77_{\pm4.90}$ | $\mathbf{71.36}_{\pm5.30}$ |
| | | 50 | $73.27_{\pm3.31}$ | $68.69_{\pm5.88}$ | $63.31_{\pm6.35}$ | $54.85_{\pm8.00}$ | $50.00_{\pm0.00}$ | $71.23_{\pm5.04}$ | $50.00_{\pm0.00}$ | $\mathbf{75.78}_{\pm2.59}$ | $75.56_{\pm3.19}$ |
| | | 100 | $76.39_{\pm2.12}$ | $73.06_{\pm6.33}$ | $71.75_{\pm2.75}$ | $55.26_{\pm6.54}$ | $50.00_{\pm0.00}$ | $73.02_{\pm2.42}$ | $50.00_{\pm0.00}$ | $78.22_{\pm1.55}$ | $\mathbf{79.28}_{\pm2.16}$ |
| | | 200 | $80.52_{\pm0.85}$ | $73.19_{\pm5.28}$ | $72.93_{\pm5.24}$ | $54.94_{\pm8.51}$ | $49.88_{\pm0.38}$ | $76.03_{\pm2.88}$ | $50.00_{\pm0.00}$ | $82.22_{\pm1.72}$ | $\mathbf{82.70}_{\pm1.73}$ |
| | | 500 | $82.47_{\pm1.02}$ | $77.06_{\pm3.03}$ | $79.18_{\pm2.08}$ | $50.47_{\pm1.03}$ | $49.67_{\pm1.05}$ | $77.79_{\pm1.36}$ | $50.00_{\pm0.00}$ | $83.03_{\pm0.96}$ | $\mathbf{83.67}_{\pm0.83}$ |
| | steel | 20 | $56.80_{\pm4.60}$ | $53.88_{\pm5.00}$ | $52.12_{\pm3.28}$ | $50.86_{\pm5.69}$ | $50.00_{\pm0.00}$ | $50.47_{\pm1.75}$ | $50.00_{\pm0.00}$ | $64.92_{\pm5.75}$ | $\mathbf{65.82}_{\pm6.29}$ |
| | | 50 | $62.45_{\pm4.31}$ | $57.80_{\pm2.19}$ | $57.11_{\pm5.27}$ | $51.84_{\pm3.72}$ | $50.00_{\pm0.00}$ | $56.40_{\pm3.61}$ | $50.00_{\pm0.00}$ | $84.81_{\pm7.86}$ | $\mathbf{86.32}_{\pm6.58}$ |
| | | 100 | $71.25_{\pm5.06}$ | $61.44_{\pm3.38}$ | $60.45_{\pm4.62}$ | $50.59_{\pm3.10}$ | $48.48_{\pm3.17}$ | $55.74_{\pm4.69}$ | $50.66_{\pm2.08}$ | $97.29_{\pm1.40}$ | $\mathbf{97.85}_{\pm1.38}$ |
| | | 200 | $77.96_{\pm3.06}$ | $63.92_{\pm3.33}$ | $64.22_{\pm5.27}$ | $50.00_{\pm0.00}$ | $47.65_{\pm4.95}$ | $62.33_{\pm4.50}$ | $50.00_{\pm0.01}$ | $98.58_{\pm0.72}$ | $\mathbf{98.89}_{\pm0.69}$ |
| | | 500 | $85.29_{\pm4.02}$ | $69.41_{\pm5.33}$ | $73.34_{\pm5.43}$ | $50.09_{\pm0.27}$ | $50.00_{\pm0.00}$ | $68.78_{\pm5.84}$ | $50.00_{\pm0.00}$ | $99.71_{\pm0.30}$ | $99.71_{\pm0.27}$ |
| | stock | 20 | $\mathbf{83.79}_{\pm3.01}$ | $70.41_{\pm7.99}$ | $49.52_{\pm16.46}$ | $70.75_{\pm10.65}$ | $78.58_{\pm11.59}$ | $66.66_{\pm8.26}$ | $51.45_{\pm4.91}$ | $82.98_{\pm4.47}$ | $83.79_{\pm4.96}$ |
| | | 50 | $89.91_{\pm2.44}$ | $77.18_{\pm3.13}$ | $70.04_{\pm2.32}$ | $72.42_{\pm8.52}$ | $89.35_{\pm1.75}$ | $78.06_{\pm3.32}$ | $50.00_{\pm0.00}$ | $89.97_{\pm2.12}$ | $\mathbf{90.17}_{\pm1.73}$ |
| | | 100 | $92.05_{\pm1.34}$ | $81.60_{\pm3.66}$ | $78.73_{\pm3.67}$ | $75.24_{\pm3.91}$ | $91.32_{\pm1.47}$ | $85.24_{\pm2.39}$ | $49.83_{\pm0.54}$ | $92.25_{\pm1.21}$ | $\mathbf{92.46}_{\pm1.18}$ |
| | | 200 | $93.50_{\pm0.91}$ | $85.46_{\pm3.04}$ | $83.47_{\pm3.02}$ | $75.51_{\pm2.44}$ | $92.95_{\pm1.04}$ | $87.25_{\pm2.03}$ | $50.00_{\pm0.00}$ | $93.93_{\pm0.85}$ | $93.65_{\pm1.01}$ |
| **Average rank** | | | $2.77_{\pm0.69}$ | $4.73_{\pm0.87}$ | $5.88_{\pm1.26}$ | $7.42_{\pm1.06}$ | $7.33_{\pm1.72}$ | $5.21_{\pm0.83}$ | $8.42_{\pm0.52}$ | $1.85_{\pm0.62}$ | $\mathbf{1.40}_{\pm0.49}$ |

### D.7.2 DCR Evaluation

Table 29: **DCR between real train data and synthetic data** on eight real-world tabular datasets with varied real data availability. We report the mean ± std result and average rank across datasets. A higher rank implies better privacy preservation. Note that "N/A" denotes that a specific generator was not applicable, and the rank is computed with the mean result of other methods. We **bold** the highest result for each dataset of different sample sizes. Even though ARF and NFLOW show high DCR, our experiments demonstrate that they do not learn the data distribution well, leading to poor downstream accuracy. TabEBM achieves competitive overall DCR against benchmark generators.

| Datasets | | $N_{\text{real}}$ | SMOTE | TVAE | CTGAN | NFLOW | TabDDPM | ARF | TabPFGen | **TabEBM** |
|---|---|---|---|---|---|---|---|---|---|---|
| *At most 10 classes* | protein | 20 | N/A | $0.24_{\pm0.06}$ | $0.36_{\pm0.10}$ | $0.29_{\pm0.07}$ | $\mathbf{0.60}_{\pm0.06}$ | $0.49_{\pm0.03}$ | $0.24_{\pm0.11}$ | $0.39_{\pm0.05}$ |
| | | 50 | $0.20_{\pm0.03}$ | $0.34_{\pm0.09}$ | $0.42_{\pm0.06}$ | $0.21_{\pm0.06}$ | $\mathbf{0.62}_{\pm0.01}$ | $0.47_{\pm0.08}$ | $0.21_{\pm0.11}$ | $0.37_{\pm0.11}$ |
| | | 100 | $0.20_{\pm0.03}$ | $0.33_{\pm0.07}$ | $0.37_{\pm0.05}$ | $0.26_{\pm0.10}$ | $\mathbf{0.54}_{\pm0.03}$ | $0.46_{\pm0.06}$ | $0.12_{\pm0.07}$ | $0.27_{\pm0.15}$ |
| | | 200 | $0.19_{\pm0.03}$ | $0.31_{\pm0.05}$ | $0.35_{\pm0.04}$ | $0.31_{\pm0.06}$ | $\mathbf{0.51}_{\pm0.04}$ | $0.44_{\pm0.05}$ | $0.10_{\pm0.03}$ | $0.26_{\pm0.06}$ |
| | | 500 | $0.19_{\pm0.02}$ | $0.32_{\pm0.06}$ | $0.30_{\pm0.05}$ | $0.33_{\pm0.02}$ | $\mathbf{0.48}_{\pm0.03}$ | $0.43_{\pm0.05}$ | $0.09_{\pm0.06}$ | $0.23_{\pm0.13}$ |
| | fourier | 20 | N/A | $0.19_{\pm0.17}$ | $\mathbf{0.61}_{\pm0.04}$ | $0.52_{\pm0.08}$ | $0.56_{\pm0.05}$ | $0.57_{\pm0.04}$ | $0.48_{\pm0.00}$ | $0.40_{\pm0.00}$ |
| | | 50 | $0.20_{\pm0.02}$ | $0.29_{\pm0.26}$ | $0.48_{\pm0.17}$ | $0.33_{\pm0.10}$ | $\mathbf{0.67}_{\pm0.05}$ | $0.54_{\pm0.07}$ | $0.31_{\pm0.07}$ | $0.43_{\pm0.03}$ |
| | | 100 | $0.23_{\pm0.02}$ | $0.53_{\pm0.06}$ | $0.50_{\pm0.04}$ | $0.37_{\pm0.09}$ | $\mathbf{0.60}_{\pm0.08}$ | $0.59_{\pm0.06}$ | $0.31_{\pm0.07}$ | $0.44_{\pm0.04}$ |
| | | 200 | $0.22_{\pm0.02}$ | $0.56_{\pm0.06}$ | $0.53_{\pm0.05}$ | $0.37_{\pm0.08}$ | $\mathbf{0.58}_{\pm0.02}$ | $0.56_{\pm0.04}$ | $0.30_{\pm0.03}$ | $0.46_{\pm0.04}$ |
| | | 500 | $0.25_{\pm0.03}$ | $\mathbf{0.67}_{\pm0.04}$ | $0.54_{\pm0.06}$ | $0.40_{\pm0.06}$ | $0.62_{\pm0.04}$ | $0.61_{\pm0.05}$ | $0.28_{\pm0.03}$ | $0.45_{\pm0.05}$ |
| | biodeg | 20 | $0.29_{\pm0.05}$ | $0.19_{\pm0.08}$ | $0.26_{\pm0.07}$ | $0.33_{\pm0.08}$ | $0.26_{\pm0.15}$ | $\mathbf{0.46}_{\pm0.05}$ | $0.38_{\pm0.03}$ | $0.39_{\pm0.04}$ |
| | | 50 | $0.18_{\pm0.05}$ | $0.17_{\pm0.06}$ | $0.16_{\pm0.04}$ | $0.24_{\pm0.04}$ | $0.14_{\pm0.05}$ | $\mathbf{0.31}_{\pm0.05}$ | $0.31_{\pm0.07}$ | $0.30_{\pm0.07}$ |
| | | 100 | $0.11_{\pm0.04}$ | $0.17_{\pm0.04}$ | $0.17_{\pm0.04}$ | $0.22_{\pm0.06}$ | $0.10_{\pm0.02}$ | $0.21_{\pm0.02}$ | $\mathbf{0.24}_{\pm0.07}$ | $0.21_{\pm0.08}$ |
| | | 200 | $0.08_{\pm0.02}$ | $0.14_{\pm0.02}$ | $0.14_{\pm0.03}$ | $0.20_{\pm0.04}$ | $0.11_{\pm0.04}$ | $\mathbf{0.20}_{\pm0.04}$ | $0.13_{\pm0.08}$ | $0.15_{\pm0.07}$ |
| | | 500 | $0.08_{\pm0.03}$ | $0.16_{\pm0.03}$ | $0.13_{\pm0.03}$ | $0.18_{\pm0.05}$ | $0.10_{\pm0.04}$ | $\mathbf{0.18}_{\pm0.03}$ | $0.05_{\pm0.03}$ | $0.09_{\pm0.04}$ |
| | steel | 20 | $0.38_{\pm0.09}$ | $0.21_{\pm0.10}$ | $0.25_{\pm0.12}$ | $0.33_{\pm0.07}$ | $\mathbf{0.48}_{\pm0.15}$ | $0.43_{\pm0.08}$ | $0.23_{\pm0.05}$ | $0.21_{\pm0.03}$ |
| | | 50 | $0.27_{\pm0.11}$ | $0.27_{\pm0.09}$ | $0.22_{\pm0.09}$ | $0.34_{\pm0.05}$ | $\mathbf{0.45}_{\pm0.15}$ | $0.40_{\pm0.06}$ | $0.15_{\pm0.06}$ | $0.24_{\pm0.08}$ |
| | | 100 | $0.20_{\pm0.11}$ | $0.28_{\pm0.07}$ | $0.22_{\pm0.08}$ | $0.30_{\pm0.08}$ | $0.37_{\pm0.19}$ | $\mathbf{0.41}_{\pm0.07}$ | $0.15_{\pm0.09}$ | $0.25_{\pm0.10}$ |
| | | 200 | $0.19_{\pm0.08}$ | $0.28_{\pm0.04}$ | $0.22_{\pm0.04}$ | $0.32_{\pm0.06}$ | $0.31_{\pm0.11}$ | $\mathbf{0.40}_{\pm0.04}$ | $0.14_{\pm0.06}$ | $0.30_{\pm0.09}$ |
| | | 500 | $0.17_{\pm0.04}$ | $0.29_{\pm0.07}$ | $0.24_{\pm0.07}$ | $0.32_{\pm0.05}$ | $0.21_{\pm0.07}$ | $\mathbf{0.37}_{\pm0.05}$ | $0.10_{\pm0.05}$ | $0.25_{\pm0.09}$ |
| | stock | 20 | $0.24_{\pm0.05}$ | $0.37_{\pm0.08}$ | $0.42_{\pm0.07}$ | $0.46_{\pm0.05}$ | $0.45_{\pm0.12}$ | $\mathbf{0.50}_{\pm0.05}$ | $0.41_{\pm0.06}$ | $0.46_{\pm0.03}$ |
| | | 50 | $0.16_{\pm0.03}$ | $0.41_{\pm0.08}$ | $0.34_{\pm0.04}$ | $0.43_{\pm0.06}$ | $0.28_{\pm0.07}$ | $0.39_{\pm0.03}$ | $0.37_{\pm0.14}$ | $\mathbf{0.46}_{\pm0.02}$ |
| | | 100 | $0.15_{\pm0.04}$ | $0.39_{\pm0.04}$ | $0.33_{\pm0.05}$ | $\mathbf{0.46}_{\pm0.04}$ | $0.17_{\pm0.02}$ | $0.33_{\pm0.03}$ | $0.34_{\pm0.09}$ | $0.44_{\pm0.03}$ |
| | | 200 | $0.14_{\pm0.02}$ | $0.38_{\pm0.04}$ | $0.28_{\pm0.03}$ | $0.45_{\pm0.05}$ | $0.11_{\pm0.01}$ | $0.32_{\pm0.03}$ | $0.39_{\pm0.11}$ | $\mathbf{0.46}_{\pm0.04}$ |
| | energy | 50 | N/A | $0.18_{\pm0.20}$ | $0.36_{\pm0.08}$ | $\mathbf{0.48}_{\pm0.07}$ | $0.00_{\pm0.00}$ | $0.46_{\pm0.06}$ | N/A | $0.44_{\pm0.02}$ |
| | | 100 | N/A | $0.08_{\pm0.17}$ | $0.40_{\pm0.03}$ | $\mathbf{0.46}_{\pm0.05}$ | $0.00_{\pm0.00}$ | $0.40_{\pm0.04}$ | N/A | $0.40_{\pm0.04}$ |
| | | 200 | N/A | $0.04_{\pm0.11}$ | $0.30_{\pm0.16}$ | $\mathbf{0.45}_{\pm0.05}$ | $0.00_{\pm0.00}$ | $0.38_{\pm0.03}$ | N/A | $0.42_{\pm0.04}$ |
| *More than 10 classes* | collins | 100 | N/A | $0.00_{\pm0.00}$ | $0.30_{\pm0.05}$ | $0.33_{\pm0.05}$ | $0.26_{\pm0.05}$ | $0.33_{\pm0.06}$ | N/A | $\mathbf{0.38}_{\pm0.11}$ |
| | | 200 | $0.18_{\pm0.03}$ | $0.00_{\pm0.00}$ | $0.23_{\pm0.06}$ | $0.29_{\pm0.08}$ | $0.19_{\pm0.03}$ | $0.33_{\pm0.09}$ | N/A | $\mathbf{0.36}_{\pm0.12}$ |
| | texture | 50 | $0.21_{\pm0.05}$ | $0.00_{\pm0.00}$ | $0.17_{\pm0.19}$ | $0.26_{\pm0.08}$ | $0.26_{\pm0.13}$ | $\mathbf{0.42}_{\pm0.07}$ | N/A | $0.40_{\pm0.13}$ |
| | | 100 | $0.16_{\pm0.04}$ | $0.08_{\pm0.14}$ | $0.35_{\pm0.06}$ | $0.26_{\pm0.03}$ | $0.37_{\pm0.09}$ | $\mathbf{0.46}_{\pm0.08}$ | N/A | $0.42_{\pm0.10}$ |
| | | 200 | $0.13_{\pm0.03}$ | $0.24_{\pm0.18}$ | $0.29_{\pm0.12}$ | $0.32_{\pm0.05}$ | $0.40_{\pm0.08}$ | $0.37_{\pm0.05}$ | N/A | $\mathbf{0.42}_{\pm0.07}$ |
| | | 500 | $0.13_{\pm0.02}$ | $0.04_{\pm0.11}$ | $0.15_{\pm0.11}$ | $0.34_{\pm0.05}$ | $0.37_{\pm0.07}$ | $0.31_{\pm0.03}$ | N/A | $\mathbf{0.44}_{\pm0.05}$ |
| **Average rank** | | | $6.64_{\pm1.35}$ | $5.48_{\pm2.08}$ | $4.82_{\pm1.42}$ | $3.45_{\pm1.75}$ | $4.03_{\pm2.82}$ | $\mathbf{2.21}_{\pm1.17}$ | $5.85_{\pm1.85}$ | $3.52_{\pm1.95}$ |

### D.7.3 Delta-presence Evaluation

Table 30: $\delta$-**presence between real train data and synthetic data** on eight real-world tabular datasets with varied real data availability. We report the mean $\pm$ std result and average rank across datasets. A higher rank implies better privacy preservation. Note that "N/A" denotes that a specific generator was not applicable, and the rank is computed with the mean result of other methods. We **bold** the best result for each dataset of different sample sizes. TabEBM achieves the best overall performance against benchmark generators.

| Datasets | | $N_{\text{real}}$ | SMOTE | TVAE | CTGAN | NFLOW | TabDDPM | ARF | TabPFGen | **TabEBM** |
|---|---|---|---|---|---|---|---|---|---|---|
| *At most 10 classes* | protein | 20 | N/A | **0.03**$_{\pm0.00}$ | 0.33$_{\pm0.94}$ | 0.13$_{\pm0.23}$ | 0.07$_{\pm0.11}$ | 0.05$_{\pm0.04}$ | 0.03$_{\pm0.00}$ | 0.03$_{\pm0.00}$ |
| | | 50 | **0.07**$_{\pm0.00}$ | 0.07$_{\pm0.00}$ | 0.07$_{\pm0.00}$ | 0.39$_{\pm0.92}$ | 0.11$_{\pm0.10}$ | 0.09$_{\pm0.03}$ | 0.07$_{\pm0.00}$ | 0.07$_{\pm0.00}$ |
| | | 100 | 0.19$_{\pm0.02}$ | 0.36$_{\pm0.19}$ | 0.50$_{\pm0.88}$ | 2.45$_{\pm3.23}$ | 0.23$_{\pm0.03}$ | 0.36$_{\pm0.14}$ | **0.17**$_{\pm0.01}$ | 0.17$_{\pm0.01}$ |
| | | 200 | 0.62$_{\pm0.23}$ | 2.84$_{\pm2.07}$ | 2.03$_{\pm1.05}$ | 11.55$_{\pm7.74}$ | 1.71$_{\pm0.72}$ | 2.73$_{\pm1.46}$ | 0.57$_{\pm0.17}$ | 0.57$_{\pm0.17}$ |
| | | 500 | **1.15**$_{\pm0.12}$ | 4.62$_{\pm2.61}$ | 2.41$_{\pm1.32}$ | 26.85$_{\pm9.98}$ | 6.51$_{\pm1.41}$ | 2.26$_{\pm0.53}$ | 1.20$_{\pm0.32}$ | 1.20$_{\pm0.32}$ |
| | fourier | 20 | N/A | 0.02$_{\pm0.00}$ | 0.02$_{\pm0.00}$ | 0.06$_{\pm0.07}$ | 0.04$_{\pm0.03}$ | 0.04$_{\pm0.04}$ | 0.02$_{\pm0.00}$ | 0.02$_{\pm0.00}$ |
| | | 50 | 0.08$_{\pm0.00}$ | 0.10$_{\pm0.01}$ | 0.09$_{\pm0.01}$ | 0.09$_{\pm0.01}$ | 0.08$_{\pm0.00}$ | 0.08$_{\pm0.00}$ | 0.08$_{\pm0.00}$ | 0.08$_{\pm0.00}$ |
| | | 100 | 0.18$_{\pm0.01}$ | 0.31$_{\pm0.10}$ | 0.26$_{\pm0.12}$ | 2.74$_{\pm2.76}$ | 0.30$_{\pm0.07}$ | 0.53$_{\pm0.16}$ | 0.17$_{\pm0.01}$ | **0.17**$_{\pm0.01}$ |
| | | 200 | 0.73$_{\pm0.48}$ | 1.56$_{\pm0.63}$ | 3.52$_{\pm1.67}$ | 10.60$_{\pm4.58}$ | 1.52$_{\pm0.67}$ | 2.52$_{\pm1.14}$ | 0.45$_{\pm0.05}$ | **0.44**$_{\pm0.05}$ |
| | | 500 | 1.42$_{\pm0.31}$ | 6.06$_{\pm3.50}$ | 5.65$_{\pm4.16}$ | 25.63$_{\pm12.53}$ | 3.39$_{\pm1.53}$ | 2.99$_{\pm0.80}$ | 1.18$_{\pm0.20}$ | **1.16**$_{\pm0.18}$ |
| | biodeg | 20 | 0.03$_{\pm0.00}$ | 0.09$_{\pm0.10}$ | 0.12$_{\pm0.15}$ | 0.22$_{\pm0.32}$ | **0.03**$_{\pm0.00}$ | 0.14$_{\pm0.30}$ | 0.03$_{\pm0.00}$ | 0.03$_{\pm0.00}$ |
| | | 50 | 0.08$_{\pm0.01}$ | 0.10$_{\pm0.04}$ | 0.08$_{\pm0.01}$ | 0.12$_{\pm0.04}$ | 0.10$_{\pm0.04}$ | 0.10$_{\pm0.04}$ | 0.08$_{\pm0.00}$ | **0.08**$_{\pm0.00}$ |
| | | 100 | 0.35$_{\pm0.25}$ | 0.52$_{\pm0.29}$ | 0.98$_{\pm0.92}$ | 1.14$_{\pm0.89}$ | 0.59$_{\pm0.41}$ | 0.51$_{\pm0.30}$ | 0.24$_{\pm0.06}$ | 0.24$_{\pm0.06}$ |
| | | 200 | 2.35$_{\pm1.76}$ | 2.27$_{\pm1.01}$ | 2.80$_{\pm1.74}$ | 4.18$_{\pm1.74}$ | 5.10$_{\pm4.44}$ | 1.70$_{\pm0.82}$ | 0.74$_{\pm0.23}$ | 0.74$_{\pm0.23}$ |
| | | 500 | 2.91$_{\pm1.43}$ | 3.95$_{\pm1.08}$ | 3.58$_{\pm1.71}$ | 8.85$_{\pm3.07}$ | 11.92$_{\pm8.19}$ | 2.37$_{\pm1.04}$ | **1.57**$_{\pm0.40}$ | 1.61$_{\pm0.42}$ |
| | steel | 20 | 0.04$_{\pm0.01}$ | 0.08$_{\pm0.08}$ | 0.10$_{\pm0.12}$ | 0.08$_{\pm0.06}$ | 0.06$_{\pm0.02}$ | 0.15$_{\pm0.30}$ | 0.03$_{\pm0.00}$ | 0.03$_{\pm0.00}$ |
| | | 50 | 0.08$_{\pm0.01}$ | 0.12$_{\pm0.03}$ | 0.09$_{\pm0.02}$ | 0.17$_{\pm0.08}$ | 1.04$_{\pm0.98}$ | 0.09$_{\pm0.01}$ | 0.08$_{\pm0.01}$ | 0.08$_{\pm0.01}$ |
| | | 100 | 0.22$_{\pm0.03}$ | 0.39$_{\pm0.12}$ | 0.44$_{\pm0.40}$ | 0.73$_{\pm0.30}$ | 2.17$_{\pm2.47}$ | 0.27$_{\pm0.06}$ | **0.20**$_{\pm0.03}$ | 0.20$_{\pm0.03}$ |
| | | 200 | 1.39$_{\pm0.78}$ | 2.05$_{\pm0.99}$ | 3.39$_{\pm2.50}$ | 5.55$_{\pm3.58}$ | 6.68$_{\pm4.55}$ | 1.29$_{\pm0.69}$ | 0.49$_{\pm0.07}$ | **0.48**$_{\pm0.06}$ |
| | | 500 | 1.93$_{\pm0.60}$ | 4.39$_{\pm3.30}$ | 5.23$_{\pm3.90}$ | 10.52$_{\pm7.22}$ | 33.92$_{\pm25.31}$ | 2.39$_{\pm0.88}$ | **1.68**$_{\pm0.88}$ | 1.70$_{\pm0.89}$ |
| | stock | 20 | 0.03$_{\pm0.00}$ | 0.03$_{\pm0.00}$ | 0.03$_{\pm0.00}$ | 0.03$_{\pm0.00}$ | 0.04$_{\pm0.00}$ | **0.03**$_{\pm0.00}$ | 0.03$_{\pm0.00}$ | 0.03$_{\pm0.00}$ |
| | | 50 | 0.08$_{\pm0.00}$ | 0.09$_{\pm0.00}$ | 0.09$_{\pm0.00}$ | 0.08$_{\pm0.00}$ | 0.09$_{\pm0.01}$ | 0.08$_{\pm0.00}$ | 0.08$_{\pm0.00}$ | **0.08**$_{\pm0.00}$ |
| | | 100 | 0.19$_{\pm0.02}$ | 0.23$_{\pm0.04}$ | 0.26$_{\pm0.12}$ | 0.23$_{\pm0.04}$ | 0.22$_{\pm0.04}$ | 0.20$_{\pm0.02}$ | 0.18$_{\pm0.01}$ | **0.18**$_{\pm0.01}$ |
| | | 200 | 0.51$_{\pm0.08}$ | 0.97$_{\pm0.30}$ | 1.83$_{\pm1.14}$ | 0.90$_{\pm0.32}$ | 0.57$_{\pm0.07}$ | 0.60$_{\pm0.16}$ | 0.49$_{\pm0.07}$ | **0.48**$_{\pm0.08}$ |
| *More than 10 classes* | energy | 50 | N/A | 0.09$_{\pm0.02}$ | 0.08$_{\pm0.00}$ | 0.08$_{\pm0.00}$ | **0.06**$_{\pm0.00}$ | 0.08$_{\pm0.00}$ | N/A | 0.08$_{\pm0.00}$ |
| | | 100 | N/A | 1.56$_{\pm1.51}$ | 0.19$_{\pm0.03}$ | 0.19$_{\pm0.03}$ | 0.99$_{\pm2.46}$ | 0.16$_{\pm0.01}$ | N/A | **0.16**$_{\pm0.00}$ |
| | | 200 | N/A | 4.15$_{\pm3.04}$ | 1.67$_{\pm0.84}$ | 0.72$_{\pm0.16}$ | 13.28$_{\pm9.21}$ | 0.44$_{\pm0.04}$ | N/A | **0.38**$_{\pm0.03}$ |
| | collins | 100 | N/A | 1.79$_{\pm1.23}$ | 0.18$_{\pm0.02}$ | 0.18$_{\pm0.03}$ | 0.17$_{\pm0.02}$ | 0.17$_{\pm0.01}$ | N/A | **0.16**$_{\pm0.01}$ |
| | | 200 | 0.43$_{\pm0.08}$ | 4.20$_{\pm2.55}$ | 0.76$_{\pm0.26}$ | 1.73$_{\pm1.09}$ | 0.94$_{\pm0.59}$ | 0.95$_{\pm0.47}$ | N/A | **0.39**$_{\pm0.05}$ |
| | texture | 50 | 0.08$_{\pm0.00}$ | 0.08$_{\pm0.00}$ | 0.08$_{\pm0.00}$ | **0.08**$_{\pm0.00}$ | 0.10$_{\pm0.02}$ | 0.08$_{\pm0.00}$ | N/A | 0.08$_{\pm0.00}$ |
| | | 100 | 0.17$_{\pm0.01}$ | 0.42$_{\pm0.22}$ | 0.19$_{\pm0.02}$ | 0.41$_{\pm0.16}$ | 0.44$_{\pm0.32}$ | 0.25$_{\pm0.05}$ | N/A | **0.17**$_{\pm0.01}$ |
| | | 200 | 0.57$_{\pm0.18}$ | 2.34$_{\pm1.14}$ | 1.76$_{\pm1.05}$ | 5.78$_{\pm2.82}$ | 7.02$_{\pm4.14}$ | 1.54$_{\pm0.94}$ | N/A | **0.45**$_{\pm0.08}$ |
| | | 500 | 1.33$_{\pm0.61}$ | 2.88$_{\pm1.09}$ | 2.42$_{\pm1.35}$ | 23.40$_{\pm10.74}$ | 16.64$_{\pm12.35}$ | 2.36$_{\pm1.06}$ | N/A | **1.03**$_{\pm0.14}$ |
| **Average rank** | | | 3.30$_{\pm1.37}$ | 5.91$_{\pm1.68}$ | 5.30$_{\pm1.74}$ | 6.45$_{\pm1.86}$ | 5.82$_{\pm2.11}$ | 4.45$_{\pm1.66}$ | 3.00$_{\pm1.90}$ | **1.76**$_{\pm1.05}$ |

