# OpenReview forum: "TabEBM: A Tabular Data Augmentation Method with Distinct Class-Specific Energy-Based Models"
_NeurIPS.cc/2024/Conference — NeurIPS 2024 poster_

### Official Review · Reviewer_XdB4 · 2024-07-06

**Soundness:** 2
**Presentation:** 3
**Contribution:** 3
**Rating:** 6
**Confidence:** 4

**Summary:**

The authors address the problem of data augmentation for classification tasks by proposing a novel approach that utilizes separate energy-based models (EBMs) to generate synthetic data for each class. These EBMs are derived directly from the logits of a binary classifier whose goal is to distinguish real data from negative data. These logits (their logsumexp over positive and negative classes) are reinterpreted them as an energy function for each class.
In order to avoid training these classifiers on the data itself, the authors take advantage of a pre-trained Prior-Data Fitted Network (PFN).

The authors perform a comprehensive evaluation of their method comparing it to competing methods for data augmentation on a number of small real world datasets. They show that their method is the only one that offers a consistent improvement in classification from the addition of synthetic data.

**Strengths:**

- The paper is well-structured and clearly written, effectively communicating the key ideas and concepts.
- The authors introduce a novel approach by leveraging a Prior-Data Fitted Network (PFN) to obtain energy-based models (EBMs), which, to the best of my knowledge, has not been previously explored.
- The experimental results section is thorough and comprehensive. They compare their approach to a wide range of relevant and competitive methods for synthetic data generation.
- The appendices are well-organized and provide valuable supplementary information. The paper strikes a good balance with the most relevant information presented in main text and additional details provided in the appendix.

**Weaknesses:**

- Interpreting classifier logits as energy functions may not necessarily yield good density models. Given a classifier with logits $f(x)[y]$, one can add an arbitrary function $g(x)$ that is constant over $y$, $f(x)[y] \rightarrow f(x)[y] + g(x)$, without affecting the conditional probabilities $p(y\vert x)$ due to the invariance of the softmax function to the addition of a constant. However, this changes the energy function for the density model to $E(x) \rightarrow E(x) - g(x)$. Consequently, there exists a functional degree of freedom that can arbitrarily alter the energy function without impacting the discriminative loss optimized by a classifier. As a result, a classifier trained solely to distinguish between positive and negative samples is not inherently encouraged to learn a useful energy function for the density over $x$.
My interpretation of the results of [1], from which the authors seem to draw inspiration is that reinterpreting the logits of a classification model as an energy based model by itself accomplishes nothing if the training process of the original classifier is not changed to encourage a good density model to be learned and encoded into the free degree of freedom. [1] achieves this by optimizing the joint likelihood, $E_{x,y} p_\theta(y,x)$, instead of the conditional likelihood, $E_{x,y} p_\theta(y\vert x)$, used in training classifiers. This significantly complicates training by requiring sampling from the EBM during training. Therefore, it is unclear why the authors assume that the pre-trained PFN has learned useful energy functions for density modeling when it is derived from a simple classifier and this is not discussed in the paper.

- Another weakness of the paper is in the experimental setup where only default hyperparameters are used for all competing methods. For many of these methods, tuning hyperparameters can substantially improve their results and the default hyperparameters may not be optimal.
Personally, I would find the presented results more convincing if the paper compared with fewer competing methods but made a better effort of getting good results out of those as it is impossible to know how much the authors tuned the degrees of freedom available during experimentation with their own proposed method.

[1] Will Grathwohl, Kuan-Chieh Wang, Jörn-Henrik Jacobsen, David Duvenaud, Mohammad Norouzi, Kevin Swersky. Your Classifier is Secretly an Energy Based Model and You Should Treat it Like One, 2022 (https://arxiv.org/abs/1912.03263)

**Questions:**

- Can you provide a convincing explanation for why the logits from discriminative classifiers derived from TabPFN should be useful when reinterpreted as a class-conditional energy function in light of the first point above? Without a clear justification, it is difficult to understand the basis for expecting this method to work effectively.

- Since the authors marginalize over $y$ by taking the LogSumExp over the $y=0$ and $y=1$ logits corresponding to negative and empirical class data, respectively, one would expect the energy function to represent a mixture distribution between these two densities. Why aren't the logits for $y=1$ used directly as an energy function instead? In other words, what is the motivation for marginalizing over $y$ when it is simply an indicator of whether the data is real or negative?

- The details of how the negative examples are generated are not clear. The text simply states:
"These surrogate negative samples $\mathcal{X}^{neg}_c$ are constructed to be far from the distribution of real data, ensuring that the classifier can easily distinguish them. This placement prevents class ambiguity and facilitates a robust energy landscape". The appendix does provide some additional details, but it is still ambiguous how this data is generated. Given the potential importance of this process in understanding why the method works, could you provide more clarity on how the negative data is generated and discuss how changing the distribution of negative data affects the results?

- The authors helpfully point out all datasets that were also used as test sets for TabPFN's evaluation since these could potentially be tainted if good practices were not followed on the original TabPFN paper [1].
I am more concerned however with datasets that were part of the meta-validation set that was used to tune the hyperparameter priors of TabPFN. It seems that one of the datasets used for evaluation of TabEBM (stock) is part of this meta-validation set (Tables 7 and 8 in the appendix of [1]).
Even though this is only one of 150 datasets that comprise this validation set and it is not clear in the original paper the extent of this finetuning, I would still consider it good practice to avoid using this dataset for evaluating downstream methods for fear of data leakage. I also would have liked the paper to prioritize evaluating on more datasets that weren't used at all in [1]. I understand if [1] might have somewhat exhausted the pool of small datasets in OpenML but evaluating on larger datasets by first subsampling them would be an option. This would also allow for the use of real data as an ideal baseline to measure TabEBM's improvement against.

- For TabEBM and the other methods that learn class conditional densities (CTGAN, TabDDPM, ARF, GOGGLE, TabPFGen) what distribution of classes (i.e., $p(y)$) was used for the synthetic data? This is not specified in the paper so I am assuming it was an empirical estimate of the p(y) from the training data otherwise this would be somewhat unfair when comparing to methods that learn a joint distribution. Furthermore it would be potentially problematic when evaluating a balanced accuracy in a test set if the p(y) of generated data is not the same across methods (for conditional methods).

- How are statistical fidelity results computed? The metrics reported seem to be univariate ones so how are they being used to compare multivariate distributions? Are only marginal distributions being compared? Is it an average over univariate metrics computed for each marginal separately? How is the KS test used on categorical variables? This section of the experiments is sorely lacking in details and the appendices seem to provide no further information. If it is indeed the case that only marginals are being compared I find this evaluation lackluster since it is easy to achieve similar marginal distributions without learning the relationships between different input variables which is the main difficulty in learning multivariate density models.


[1] Noah Hollmann, Samuel Müller, Katharina Eggensperger, Frank Hutter. TabPFN: A Transformer That Solves Small Tabular Classification Problems in a Second, 2022 (https://arxiv.org/abs/2207.01848)

**Limitations:**

The authors aknowledge that due to the reliance of the method on a PFN it is currently constrained for use in small datasets.

---

> ### Author Rebuttal · Authors · 2024-08-06
>
> Thank you for the detailed review and constructive feedback! We address **all** your questions and comments below, and provide a **rebuttal PDF** in the general rebuttal. Due to space limits, we summarise the new results. We will update our manuscript with additional clarifications and results.
>
> ## >Q3a: Clarify the generation of the negative samples
> The surrogate classification task is to determine if a sample belongs to class $c$ by comparing $\mathcal{X}_c$ against a set of "negative samples" $\mathcal{X}_c^{\text{neg}}$. We label the true samples $\mathcal{X}_c$ as 1 and the negative samples $\mathcal{X}_c^{\text{neg}}$ as 0, and train TabPFN on this datasets, resulting in a class-specific binary classifier used to define the $E_c(x)$ (using Equation 3)
>
> We generate the negative samples at the corners of a hypercube in $R^D$. For each dimension $d$, the coordinates of a negative sample are either $\alpha\sigma_d$ or $-\alpha\sigma_d$, where $\alpha$ is a fixed constant and $\sigma_d$ is the standard deviation of dimension $d$. For example, in $R^3$, a negative sample might have coordinates $[\alpha\sigma_1, \alpha\sigma_2, -\alpha\sigma_3]$. In the paper we use four negative samples with $\alpha = 5$, placing them far from the real data.
>
> ## >Q3b: How does the distribution of negative data affect the results?
> We found it essential to use negative samples far from the real data, to allow the binary classifier to easily differentiate between $\mathcal{X}_c$ and $\mathcal{X}_c^{\text{neg}}$.
>
> Figure `F1` from the PDF shows a **new experiment** varying the distribution of the negative samples. TabEBM infers an accurate energy surface with distant negative samples, and the energy surface becomes inaccurate when negative samples resemble real samples. This occurs because TabPFN is uncertain, affecting its logits magnitude and making them unsuitable for density estimation (we further investigate the logits in our next answer to Q1/W1).
>
> TabEBM is robust to the distribution of negative samples if they are easily distinguishable from real data. We ran **two new experiments**, keeping the negative samples at the corners of a hypercube and varying their distance and number. When varying the per-dimension distance of those samples $\alpha \in [0.1, 5]$, TabEBM provides consistent improvements between 2.85-3.18%. Additionally, our results in Table `R2` (in our answer to Q1 from reviewer Cc3y) demonstrate that TabEBM performs similarly regardless of the number of negative samples in the surrogate binary tasks.
>
> ## >Q1/W1: Why can TabPFN's logits be useful for energy estimation?
> Indeed, we agree with your observation! As the TabPFN classifier is trained on different class-specific surrogate tasks, it learns different logits for each task. We found it essential to place the negative samples far from the real samples, because TabPFN's confidence depends on the distance to the training data [1], as it was pre-trained to approximate Bayesian inference.
>
> We present **new experiments** in Figure `F2` from the PDF. The top row shows that as the distance to the real data increases, the logit $f(x)[1]$ for the real data smoothly decreases until the two logits become similar. Thus, the classifier predicts is uncertain in these far-away regions. As maximum logits decrease, TabEBM's inferred density drops significantly (as shown on the bottom row) because $p_c(x) \propto (\exp(f(x)[0]) + \exp(f(x)[1]))$. One could possibly fine-tune TabPFN on the surrogate tasks to further improve the logits for density estimation.
>
> ## >Q2: Why fit p(x) rather than p(x|y=1) in the surrogate binary tasks?
> Both approaches lead to virtually identical results due to the design of our surrogate task. The **new results** in Figure `F2` (bottom row) from the PDF shows that the TabEBM's inferred energy using $p(x)$ (in red) is essentially identical to the energy of $p(x|y=1)$ (in grey), especially near the real data. As the SGLD sampling starts near the original points (line 676), it will visit the neighbourhood of the real samples, where the energy surfaces are virtually identical, leading to similar results.
>
> ## >Q4: Evaluating more datasets
> We run **new experiments** on six leakage-free datasets from UCI, with 1,000-9105 samples of 7-42 features. Table `R2` (in our response to Q2 for `tdoQ`) shows that TabEBM consistently outperforms the baseline and all other benchmark methods.
>
> ## >Q5: What's the class distribution of the synthetic data?
> The synthetic data has the same class distribution as the real training data.
>
> ## >Q6: Is statistical fidelity computed over the univariate marginals?
> Yes, these metrics are univariate, and we computed them using the open-source library Synthcity. We acknowledge the reviewer's point about the imperfection of univariate metrics, although evaluating generators' ability to capture the joint feature relationships remains underexplored ``[2]``.
>
> ## >W2: How does tuning the generators/predictors affect the ranking?
> We run **new experiments** tuning three generators on three datasets using the average validation accuracy of six downstream classifiers. The ranges are: SMOTE ($k \in \{3, 5, 10\}$), TVAE ($\text{lr} \in \{5e-4, 1e-3, 5e-3\}$) and CTGAN ($\text{lr} \in \{1e-3, 5e-3, 1e-2\}$). TabEBM remains the most competitive data augmentation method, improving accuracy by +2.25, followed by SMOTE (+1.81) and CTGAN (-5.71). We can provide the full results in the discussion.
>
> We also tuned the downstream predictors, and the **new results** in Table `R7` from the PDF show that TabEBM remains the best-performing method, providing the largest improvements for data augmentation, even after tuning the predictors.
>
> Thank you again for your constructive review! We would appreciate it if you would consider raising your score in light of our response.
>
> References:
> - `[1]` McCarter, *What exactly has TabPFN learned to do?*, ICLR Blogposts, 2024
> - `[2]` Tu, Ruibo, et al., 2024, (https://arxiv.org/abs/2406.08311)

---

> > ### Comment · Reviewer_XdB4 · 2024-08-07
> > **Response**
> >
> > I thank the authors for the clarifications and for taking the time to run these additional experiments.
> >
> > I must confess that the reason this approach works is still somewhat mysterious to me and I would like to have a better theoretical justification for it.
> > However, the linked blogpost and the author's additional experiments do shed some light on the matter and seem to suggest that this may be due to some property of TabPFN.
> > The authors have, at least, successfully convinced me that their surrogate task procedure works and is somewhat reliable and I believe these are valuable empirical results.
> >
> > I believe the clarification on the generation of negative data should be in the paper.
> >
> > Since I consider my concerns mostly addressed, I am happy to raise my score to a 6.

---

### Official Review · Reviewer_iojh · 2024-07-09

**Soundness:** 3
**Presentation:** 3
**Contribution:** 3
**Rating:** 5
**Confidence:** 4

**Summary:**

The paper proposes TabEBM, a novel data augmentation method designed for low-sample-size tabular classification tasks. TabEBM generates synthetic tabular data using class-specific Energy-Based Models (EBMs) to learn the marginal distribution for each class. Experimental results on various real-world datasets demonstrate that TabEBM improves downstream performance via data augmentation, generates high-fidelity synthetic data, and strikes a competitive balance between accuracy and privacy in data sharing.

**Strengths:**

- Novel approach using class-specific EBMs for tabular data generation.
- Comprehensive evaluation across multiple datasets, metrics, and downstream tasks.
- Strong performance, especially in low data regimes.
- Thorough analysis of statistical fidelity and privacy preservation.
- Open-source implementation provided.

**Weaknesses:**

- Scalability Issues: The reliance on TabPFN, which struggles with large sample sizes, limits TabEBM's scalability.
- Implementation Complexity: The need for class-specific surrogate tasks and the iterative nature of the sampling process may complicate implementation and increase computational overhead.

**Questions:**

- How does TabEBM perform on datasets with  hundreds of features?
- Could the approach be extended to use other pre-trained models besides TabPFN?
- How does the method handle highly imbalanced datasets where some classes have significantly fewer samples than others?
- What is the computational complexity of generating samples compared to other methods?
- How sensitive is the performance to the choice of hyperparameters in the SGLD sampling process?
- Clarify the meaning of the word "fit" which is used in line 122 of the paper.

**Limitations:**

- Performance may degrade for datasets much larger than those tested.
- Inherits limitations of the underlying TabPFN model.
- May not be suitable for datasets with extremely high dimensionality.

---

> ### Author Rebuttal · Authors · 2024-08-06
>
> Thank you for your thoughtful review! We address **all** your questions and comments below. Due to limited space, we summarised the new results. We will update the manuscript to include the complete new experiments and clarifications.
>
> ## >Q3: How does TabEBM perform on highly imbalanced datasets?
> We conducted **new experiments**, adjusting the class imbalance on two binary datasets (with $N_\text{real}=100$). We keep the setup from Section 3 and report the balanced accuracy averaged over six downstream predictors. Table `R4` shows that under high class imbalanced, TabEBM outperforms the Baseline and SMOTE, which is a method specifically designed to handle imbalanced datasets.
>
> *Table `R4`.* Test balanced classification accuracy (%) varying the class imbalance.
> |**Datasets**|`(#Minority:#Majority)`|Baseline|SMOTE|TabEBM(Ours)|
> |---|---|---|---|---|
> |**steel**|`(50:50)`|88.79|77.74|**93.77**|
> | |`(10:90)`|60.74|60.76|**73.02**|
> | |`(5:95)`|52.29|56.11|**65.95**|
> |**stock**|`(50:50)`|88.64|**90.21**|90.16|
> | |`(10:90)`|76.05|83.86|**84.92**|
> | |`(5:95)`|65.98|75.24|**77.68**|
>
> ## >Q5: How sensitive is TabEBM to the hyperparameters of the SGLD sampling?
> We run **new experiments** varying two key hyperparameters of SGLD on the “biodeg” dataset with $N_{\text{real}}=100$. Specifically, we vary the step size $\alpha_{\text{step}}$ and the noise scale $\alpha_{\text{noise}}$, reporting the accuracy averaged over six downstream predictors.
>
> Table `R5` shows that TabEBM is stable to these hyperparameters, as the difference between the highest and lowest accuracy is less than 1.5%. Note that increasing $\alpha_{\text{noise}}$ (which is added at each SGLD step) is expected to degrade performance because we standardized the data to have unit standard deviation.
>
> *Table `R5`.* Classification accuracy (%) of TabEBM with different SGLD settings
> |$\alpha_{\text{step}}\rightarrow$|0.1|0.3|0.5|1.0|
> |---|---|---|---|---|
> |$\alpha_{\text{noise}}\downarrow$| | | | |
> |0.01|76.45|77.09|77.04|76.58|
> |0.02|76.86|76.96|76.77|76.26|
> |0.05|75.93|75.89|75.94|75.70|
>
> ## >Q6: Clarify the meaning of the word "fit" on line 122
> We use the term “fit” to mean “train” the in-context model TabPFN. For TabPFN, “training” is similar to the K Nearest Neighbour algorithm, where "training" means defining the training dataset used for inference. Thus, we simply update TabPFN's training datasets and the model in inference mode, thus never updating its parameters.
>
> ## >Q4: How does TabEBM's computational complexity compare to other methods?
> Figure 4 from the paper illustrates the trade-off between accuracy and the time required for training and generating 500 synthetic samples. The results show that TabEBM is the fastest method for generating data besides SMOTE. Other methods are 3-30 times slower than TabEBM. Also, TabEBM surpasses all other methods in improving downstream accuracy through data augmentation.
>
> ## >W2: Class-specific surrogate tasks and iterative sampling can complicate implementation and increase computational overhead
> **On implementation complexity**: TabEBM simplifies implementation by eliminating the need for dedicated training, pipelines, protocols, and resources—making it ready to use for generating data. We provide an open-source implementation of TabEBM, allowing users to generate data with just two lines of code for new datasets. We believe this ease of use will encourage the widespread application of tabular data augmentation on small datasets.
>
> **On the computational overhead**: As mentioned in our answer to Q4 and Figure 4, TabEBM is computationally efficient, being the fastest method for generating data besides SMOTE —while outperforming all competing methods.
>
> ## >Q2: Can TabEBM use other pre-trained models besides TabPFN?
> Yes, TabEBM can be used with other pre-trained models besides TabPFN. In the paper, we use TabPFN because it is the only tabular model of this type with open-sourced weights. However, TabEBM is a general method for transforming classifiers into class-specific generators and can use any gradient-based in-context classifier that computes logits (through Equation 3). As new tabular foundational models are developed `[1]`, they can be readily integrated into TabEBM.
>
> ## >Q1/W1/L1/L2/L3: On the scalability and limitations of TabEBM
> TabEBM is a method that uses an underlying binary classifier, thus its capabilities and limitations are directly influenced by the in-context model it employs (TabPFN in our case), as discussed in the paper (Lines 317-323). Since TabPFN handles only up to 100 features, this limitation is also inherited by TabEBM.
>
> Our **new experiment** (Table `R6`) shows that TabEBM can be applied to larger datasets beyond TabPFN's limitation of 1000 samples. On larger datasets TabEBM still outperforms other generators (not shown due to space limits), but training on real data alone appears sufficient. This highlights TabEBM's usefulness in fields with limited training samples. Note that despite TabPFN's 10-class limit, TabEBM can handle unlimited classes by using TabPFN only for surrogate binary tasks.
>
> As foundational models for tabular data evolve `[1]`, new models that can accommodate more features are anticipated. Integrating these models into the TabEBM framework will enable it to handle high-dimensional datasets, thus increasing its versatility and utility.
>
> *Table `R6`.* Classification accuracy (%) comparing data augmentation with increased real data availability on the “texture” dataset
> |$N_{\text{real}}$|Baseline|Improvement by TabEBM (%)|
> |---|---|---|
> |50|72.40|+6.50|
> |100|82.42|+3.59|
> |1000|96.37|-0.07|
> |2000|97.76|+0.07|
> |3000|98.20|+0.15|
> |4000|98.51|+0.04|
>
> Thank you again for your thoughtful review! We would appreciate it if you would consider raising your score in light of our response.
>
> **References:**
> - `[1]` Boris van Breugel, Mihaela van der Schaar, *Why Tabular Foundation Models Should Be a Research Priority*, ICML 2024 (https://arxiv.org/abs/2405.01147)

---

> > ### Comment · Reviewer_iojh · 2024-08-11
> >
> > Thank you to the authors for clarifying and conducting additional experiments. These efforts have significantly enhanced the quality of the research. However, I believe my initial evaluation remains valid, so my score remains unchanged.

---

> ### Author Response · Authors · 2024-08-12
> **Additional clarifications and results on imbalanced datasets and large datasets**
>
> Dear Reviewer iojh,
>
> Thank you for acknowledging the additional experiments and clarifications we provided. We are glad that you found that our efforts have significantly enhanced the quality of our research. We are also pleased that, in your original review, you highlighted several strengths of our work, including its novelty, comprehensive evaluation, strong performance, and open-source implementation.
>
> We conducted **additional experiments** to further support our argument, which we believe will interest the reviewer. Specifically, we evaluated four additional generative models—TVAE, CTGAN, TabDDPM, and TabPFGen—addressing your questions on handling imbalanced datasets and the effectiveness of data augmentation on large datasets.
> - Table `R5-extended` demonstrates that TabEBM outperforms these methods, especially on highly imbalanced datasets. The results indicate that TabEBM is particularly effective for class balancing, and we will highlight this application in the revised manuscript.
> - Table `R6-extended` suggests that data augmentation may not be necessary when ample real data is available for model training. As the availability of real data increases, training downstream predictors solely on this real data can guarantee optimal performance. Existing methods, such as TVAE, CTGAN and TabDDPM, tend to decrease performance on smaller datasets, where performance enhancement is most needed, while TabEBM consistently delivers the largest improvements. This suggests that data augmentation is beneficial primarily in data-scarce scenarios, where TabEBM excels. We will include these results in the updated manuscript.
>
> Given the new results and your positive feedback, we kindly ask if you might reconsider your initial score in light of these results. If there are any remaining concerns, please let us know, as we would be more than happy to discuss them further while the discussion period is open. We would greatly appreciate your feedback in order to further improve our paper.
>
> Thank you again for your time and consideration,
>
> Authors
>
> Table `R5-extended`.  Test balanced classification accuracy (%) varying the class imbalance. We aggregated the performance over six downstream predictors. TabEBM is effective for datasets with high class imbalance.
>
> | Datasets `(#Minority:#Majority)` | Baseline | SMOTE | TVAE | CTGAN | TabDDPM | TabPFGen | TabEBM (Ours) |
> | --- | --- | --- | --- | --- | --- | --- | --- |
> | **steel** |  |  |  |  |  |  |  |
> | `(50:50)` | 88.79 | 77.74 | 74.21 | 77.61 | 83.78 | 89.67 | **93.77** |
> | `(20:80)` | 74.33 | 66.56 | 68.10 | 61.52 | 69.06 | 83.38 | **84.65** |
> | `(10:90)` | 60.74 | 60.76 | 54.85 | 53.50 | 58.46 | 70.36 | **73.02** |
> | `(5:95)` | 52.29 | 56.11 | 51.12 | 51.30 | 51.17 | 64.84 | **65.95** |
> | **stock** |  |  |  |  |  |  |  |
> | `(50:50)` | 88.64 | **90.21** | 88.67 | 85.52 | 89.23 | 88.16 | 90.16 |
> | `(20:80)` | 85.34 | 88.95 | 85.87 | 85.37 | 87.16 | 88.72 | **89.38** |
> | `(10:90)` | 76.05 | 83.86 | 74.60 | 76.93 | 78.81 | 83.52 | **84.92** |
> | `(5:95)` | 65.98 | 75.24 | 62.66 | 61.11 | 68.69 | 75.56 | **77.68** |
>
>
> Table `R6-extended`. Test balanced classification accuracy (%) aggregated over six downstream predictors, comparing data augmentation with increased real data availability of the “texture” dataset. TabPFGen is not applicable because it supports only datasets with up to 10 classes.
>
> | N_real | Baseline | SMOTE | TVAE | CTGAN | TabDDPM | TabPFGen | TabEBM (Ours) | Improvement vs Baseline by TabEBM (%) |
> | --- | --- | --- | --- | --- | --- | --- | --- | --- |
> | 50 | 72.40 | 76.40 | 55.332 | 54.80 | 62.94 | N/A | **78.90** | +6.50 |
> | 100 | 82.42 | 84.35 | 66.00 | 69.49 | 76.34 | N/A | **86.01** | +3.59 |
> | 200 | 87.54 | 89.29 | 78.37 | 82.44 | 82.53 | N/A | **89.77** | +2.23 |
> | 500 | 92.96 | 93.69 | 90.09 | 91.48 | 91.24 | N/A | **93.76** | +0.80 |
> | 1000 | **96.37** | 96.21 | 93.61 | 95.36 | 94.56 | N/A | 96.30 | -0.07 |
> | 2000 | 97.76 | 96.84 | 96.62 | 97.10 | 97.13 | N/A | **97.83** | +0.07 |
> | 3000 | 98.20 | 98.28 | 97.60 | 97.60 | 97.73 | N/A | **98.35** | +0.15 |
> | 4000 | 98.51 | **98.59** | 98.11 | 98.00 | 98.46 | N/A | 98.55 | +0.04 |

---

### Official Review · Reviewer_Cc3y · 2024-07-09

**Soundness:** 4
**Presentation:** 4
**Contribution:** 4
**Rating:** 7
**Confidence:** 4

**Summary:**

The authors present a new method of tabular data augmentation, called TabEBM. The unique feature of TabEBM is that it creates distinct generative models for each class in a classification problem setting. With extensive and thorough evaluations, the authors prove that TabEBM sets the new state of the art.

**Strengths:**

1. The manuscript is well organized. The text is clear. The quality of the figures is high.
2. The authors address a very common problem for the broad scientific community. Application of machine learning algorithms on small tabular datasets is limited by design. Introducing TabEBM as an effective data augmentation method, this work can have an extremely broad impact in the future.
3. The experiments are rich and rigorous. The method is compared against many existing methods on a variety of datasets. The evidence proving its effectiveness is convincing.
4. The authors claim TabEBM will be available as an open-source library.

**Weaknesses:**

Some important details on study design, model training, and practical considerations of TabEBM application are missing. See questions.

**Questions:**

1. The definition of a surrogate binary classification task (lines 111-113) might result in class imbalance. How is this potential problem addressed?
2. Can the authors provide more details on training the models for the surrogate binary classification task?  This could eliminate some other questions.
3. Have the authors specifically investigated the propensity of TabEBM to generate outliers? How does TabEBM compare to other methods in this regard?
4. In section 3.2, the authors describe many statistical evaluations performed. However, the multiple hypothesis testing aspect is not described in sufficient detail. How were the p-values adjusted? How strong is the impact of a particular correction method on the final results (tables of C3)?
5. Can the authors share more insights about the open-source library? This is positioned as one of the key contributions, but the code is not provided and the practical considerations of using the library are not clear from the submission.

**Limitations:**

Briefly discussed in section 4.

---

> ### Author Rebuttal · Authors · 2024-08-06
>
> Thank you for your thoughtful review! We address **all** of your questions and comments below. Due to space constraints, we summarised the new results. We will update the manuscript to include the complete new experiments and clarifications.
>
> ## >Q2: Provide details on training models for the proposed surrogate binary classification tasks
> In TabEBM, for each class $c$, we train a class-specific EBM, $E_c(x)$, using exclusively the data from that class, denoted as $\mathcal{X}_c$. The energy function $E_c(x)$ is derived by adapting the logits from a binary classifier, as outlined in Equation 3. To train this classifier with only class-specific data, we create surrogate binary classification tasks.
>
> Our surrogate classification tasks determine if a sample belongs to class $c$ by comparing $\mathcal{X}_c$ against a set of "negative samples," $\mathcal{X}_c^{\text{neg}}$. We intentionally create these negative samples at a distance of five standard deviations from the original data to ensure the binary classifier can easily differentiate between the real class data and these artificially distant samples.
>
> We label the true class samples $\mathcal{X}_c$ as 1 and the negative samples $\mathcal{X}_c^{\text{neg}}$ as 0, forming a combined dataset $\mathcal{D}_c = (\mathcal{X}_c \cup \mathcal{X}_c^{\text{neg}}, \{1\}^{|\mathcal{X}_c|} \cup \{0\}^{|\mathcal{X}_c^{\text{neg}}|})$. Following this, we train the TabPFN classifier on $\mathcal{D}_c$, resulting in a binary classifier specific to class $c$. This classifier is then used to define the energy function $E_c(x)$ using Equation 3, from which we can generate data using SGLD.
>
> ## >Q1: Does the surrogate binary classification task result in class imbalance issues?
> No, the imbalance between the "negative samples" and the real samples used in the surrogate binary classification task has a negligible impact on TabEBM's performance.
>
> We conducted a new experiment to evaluate the effects of varying the ratio $\|\mathcal{X}^{\text{neg}}_c\|:\|\mathcal{X}_c\|$. To ensure reliable outcomes, we maintained a consistent ratio across all classes, keeping the same proportion of negative samples for each class. We varied the number of negative samples $\|\mathcal{X}^{\text{neg}}_c\|$ to represent various ratios of $\|\mathcal{X}_c\|$, thus simulating both balanced and highly imbalanced scenarios.
>
> The results, presented in Table `R3`, computed across six datasets with $N_{\text{real}} = 100$ real samples, demonstrate that TabEBM is robust to potential imbalances in the surrogate binary tasks.
>
> *Table `R3`.* Classification accuracy (%) on data augmentation, showing the impact of the number of negative samples $\|\mathcal{X}^{\text{neg}}_c\|$ in TabEBM across six datasets. TabEBM's performance is robust regarding the number of negative samples.
> |Ratio$\|\mathcal{X}^{\text{neg}}_c\|:\|\mathcal{X}_c\|$|`1`|`0.5`|`0.2`|`0.1`|Fixed$\|\mathcal{X}^{\text{neg}}_c\|=4$|
> |---|---|---|---|---|---|
> |Average accuracy improvments|+2.91|+2.87|+2.88|+2.92|+2.90|
>
> ## >Q3: TabEBM's propensity to generate outliers
> We provide fair and thorough comparisons by (i) including a wide range of evaluation metrics and (ii) utilising a large synthetic set (Lines 183-184). Given our extensive test scope (i.e., eight datasets $\times$ five sample sizes), it is computationally infeasible to employ complex and specialised methods to detect outliers in synthetic data. However, the downstream accuracy and statistical fidelity metrics are highly indicative of the distribution shifts between real and synthetic data. Notably, Figure 3 illustrates that TabEBM consistently outperforms the baseline across various sample sizes and class numbers. We believe these two metrics adequately demonstrate that TabEBM is more stable to generate in-distribution samples than benchmark methods.
>
> ## >Q4: Details on the multiple hypothesis testing in Section 3.2
> We aim to provide a fair and coherent comparison between TabEBM and existing methods, and thus, we follow the widely-adopted evaluation process in prior studies. Specifically, we compute the statistical fidelity metrics with the open-source implementations from the well-established open-source library Synthcity. However, the previous studies often operate under the assumption that the issues associated with multiple comparisons are less pronounced in generating low-dimensional tabular data, hence correction methods for multiple hypothesis testing are seldom employed. We will further clarify this in the revised manuscript.
>
> ## >Q5: The code for the open-source library is not provided
> We included the TabEBM library as a zip file in the Supplementary material. The library has two core functionalities:
>
> 1. *Generate synthetic data*: The library can generate data that can be used as additional training material for data augmentation. We have included a demo notebook, `TabEBM_generate_data.ipynb`, in the attached codebase. This notebook demonstrates how to use the generated data for data augmentation and shows a toy example comparing the quality and position of the synthetic data with the real data.
>
> ```python
> from TabEBM import TabEBM
>
> tabebm = TabEBM()
> augmented_data = tabebm.generate(X_train, y_train, num_samples=100)
> # augmented_data[class_id] = generated data for a specific `class_id`
> ```
>
> 2. *Compute & visualise the energy function*: We allow users to compute TabEBM's energy function and the unnormalised data density. The demo notebook, `TabEBM_approximated_density.ipynb`, shows TabEBM's energy under conditions of data noise and class imbalance.
>
> We will make the TabEBM library freely available after publication. It is easy to use, domain-agnostic, and requires no training, making it suitable for data augmentation, especially for small-sized datasets.
>
>
> Thank you again for your thoughtful review! We would appreciate it if you would consider raising your score in light of our response.

---

> > ### Comment · Reviewer_Cc3y · 2024-08-11
> >
> > Thanks to the authors for the clarifications and additional experiments!
> > I believe my initial evaluation was fair, so my score remains the same.

---

### Official Review · Reviewer_tdoQ · 2024-07-11

**Soundness:** 2
**Presentation:** 3
**Contribution:** 2
**Rating:** 5
**Confidence:** 4

**Summary:**

The paper introduces TabEBM, a class-conditional generative method for tabular data augmentation using distinct EBMs for each class. By modeling each class's data distribution individually, TabEBM generates high-quality synthetic data. Experiments demonstrate that using TabEBM for data augmentation improves classification performance across various datasets, particularly small ones.

**Strengths:**

The paper introduces distinct class-specific EBMs for tabular data augmentation. Moreover, it presents thorough experiments demonstrating effectiveness across various datasets, especially small ones. Additionally, the writing is well-structured and clear.

**Weaknesses:**

1. On the technical level, TabPFN, EBMs, and SGLD have all been introduced in TabPFGen [48]. The main difference in the proposed TabEBM method is the use of class-specific EBMs. However, this idea is straightforward and lacks novelty.
2. The most significant difference between tabular tasks and common tasks like images is that they include both continuous and categorical features. However, TabEBM treats all features as continuous, lacking targeted consideration for categorical features.
3. In TabEBM, although the choice of surrogate binary classifier is arbitrary, not using training-free models like TabPFN will result in training costs being positively correlated with the number of classes, significantly increasing the training overhead.
4. The datasets used in the experiments have a maximum dimensionality of 77, lacking experimental results on high-dimensional data (e.g., hundreds of dimensions).

**Questions:**

1. Do the datasets used in the experiments only contain continuous features and no categorical features?
2. The TabEBM method is similar to TabPFGen [48]. Why didn't you use the 18 datasets mentioned in the TabPFGen paper for experiments, but instead chose 6 different datasets (with a smaller test scope) for evaluation?

---

> ### Author Rebuttal · Authors · 2024-08-06
>
> Thank you for your thoughtful review! We address **all** of your questions and comments below. Due to space constraints, we summarised the new results. We will update the manuscript to include the complete new experiments and clarifications.
>
> ## >Q1/W2: Do the datasets contain categorical features? How does TabEBM handle them?
> Yes, we evaluated TabEBM on datasets containing both continuous and categorical features. Specifically, Table `R1` shows the two considered datasets with mixed feature types. In lines 639-640, we also clarify that TabEBM encodes the categorical features with leave-one-out target statistics. As the reviewer mentioned, the evaluation results (Table 1) show that TabEBM is effective across various datasets, including those containing categorical features.
>
> *Table `R1`.* Number of categorical features in the datasets considered.
> |Dataset|#Categorical|
> |---|---|
> |**Main text**||
> |protein|4|
> |energy|1|
> |**New datasets in our response to Q2**||
> |support2|7|
> |mushroom|22|
> |abalone|1|
> |statlog|13|
> ## >Q2: Why not evaluate the 18 datasets in the TabPFGen paper?
> **Firstly,** we wanted to avoid data leakage and ensure a robust comparison by also evaluating datasets not in TabPFN's meta-test (Appendix F.3 in its original paper). As the 18 datasets in the TabPFGen paper were all used for constructing and evaluating TabPFN, we also use different, leakage-free datasets. **Secondly,** our test scope is not smaller than that of the TabPFGen paper. We evaluate a wide range of test scopes, considering different datasets and varying the availability of real data. We also include datasets with categorical data, unlike TabPFGen, which investigates only numerical features. Our setup (i.e., eight datasets × five sample sizes) leads up to 33 different test cases. For reference, ARF has the most test cases among prior studies, with 20 test cases, while TabPFGen follows with 18 test cases. **Thirdly,** we further provide **new results** on six more leakage-free datasets for UCI with 7-42 features, some including categorical features. We set $N_\text{real}=100$ and provide results for the Top-5 benchmark methods. Table `R2` shows that TabEBM continues to outperform all other benchmark methods.
>
> *Table `R2`.* Classification accuracy (%) comparing data augmentation on six *new* leakage-free datasets.
> |Datasets|Baseline|SMOTE|TVAE|CTGAN|TabDDPM|TabPFGen|TabEBM (Ours)|
> |---|---|---|---|---|---|---|---|
> |clinical|68.63|71.07|61.80|65.21|54.03|69.66|**71.20**|
> |support2|64.23|65.60|60.70|59.14|58.31|64.34|**65.28**|
> |mushroom|95.51|95.84|93.75|93.26|79.87|**97.05**|96.82|
> |auction|51.90|57.35|53.09|52.35|51.14|56.82|**57.97**|
> |abalone|11.59|N/A|8.49|7.72|9.95|N/A|**13.56**|
> |statlog|56.22|57.30|53.12|55.55|53.07|57.65|**57.85**|
>
> ## >W1: The core ideas “have all been introduced in TabPFGen” … "this idea is straightforward and lacks novelty”
> TabEBM makes a novel contribution to the field of tabular data augmentation by being the first to introduce class-specific generators, as we discussed in our related work, as well as acknowledged by all other reviewers.
>
> Specifically, TabEBM is the first method to create distinct class-specific EBMs. It consists of individual models—one for each class—designed to learn distinct marginal distributions for the inputs associated with each class. TabEBM's *class-specific generation* is not available to TabPFGen, and it is only possible due to our proposed *surrogate binary tasks*, which enable the creation of a binary classification task for each class and the obtaining of the class-specific EBM via Equation 3. The results demonstrate that training class-specific EBMs significantly improves the inferred energy over TabPFGen under high levels of noise (Figure 7) and high data imbalance (Figure 8). Unlike TabPFGen, which supports only 10 classes, TabEBM can handle an unlimited number of classes. As you rightly point out, our extensive experiments demonstrate that our method, with its class-specific generation, outperforms all other methods in accuracy, statistical fidelity, and privacy.
>
> In addition to the method, our paper makes two additional key contributions. First, we conduct the first extensive analysis of tabular data augmentation across various dataset sizes. Our benchmark reveals that existing tabular augmentation methods may improve performance only on small datasets, a previously uninvestigated issue. Second, we will release our method as an open-source library for tabular data augmentation, allowing users to generate data immediately without additional training.
>
> ## >W3: “In TabEBM … not using training-free models like TabPFN will result in … significantly increasing the training overhead.”
> Because we tackle classification problems on *small-size* datasets, training models from scratch is not practical, often leading to suboptimal performance due to overfitting. Instead, using pre-trained models, such as TabPFN, is integral to our approach. From a practical standpoint, using pre-trained models makes TabEBM immediately usable. A significant limitation of existing tabular generators is their need for training, which can be time-consuming, error-prone, and typically requires GPUs. In contrast, our TabEBM library (included with the submission) allows users to start generating data immediately without additional training or tuning.
> ## >W4: Lacking experimental results on high-dimensional data
> As noted in Lines 317-323, scaling up in-context tabular models for high-dimensional data remains an unresolved challenge and is not the primary focus of our study. Instead, TabEBM is designed to demonstrate the effectiveness of using pretrained in-context tabular classifiers for generating tabular data. As acknowledged by the reviewer, and further shown in our response to Q2, our "thorough experiments" demonstrate TabEBM's effectiveness.
>
> Thank you again for your thoughtful review! We would appreciate it if you would consider raising your score in light of our response.

---

> ### Author Response · Authors · 2024-08-12
>
> Dear Reviewer tdoQ,
>
> Thank you once more for taking the time to provide your feedback on our work. In light of our response and the new experiments, and with the discussion window closing soon, please let us know if you have any further questions or if there is anything else we can clarify. If not, we would appreciate it if you would consider updating your review based on our rebuttal.
>
> Thank you,
>
> Authors

---

> > ### Comment · Reviewer_tdoQ · 2024-08-13
> >
> > I appreciate the additional experiments conducted by the author. These evaluations addressed my concerns regarding the experiment scale and performance for datasets with many categorical features. I have increased my overall rating for this paper from Reject to Borderline Accept.

---

### Author Rebuttal · Authors · 2024-08-06

We thank the reviewers for the feedback!

## **(1) Summary of positive things**

- **Novel method**
    - `Cc3y`: *“a new method of tabular data augmentation”; “the unique feature is that it creates distinct generative models for each class”*
    - `iojh`: *“Novel approach using class-specific EBMs for tabular data generation”*
    - `XdB4`: *“a novel approach that utilizes separate energy-based models (EBMs) to generate synthetic data for each class”*
- **Comprehensive evaluation showing TabEBM's effectiveness**
    - `tdoQ`: *“thorough experiments demonstrating effectiveness”*
    - `Cc3y`: *“extensive and thorough evaluations”; ”TabEBM sets the new state of the art”*
    - `iojh`: *”Comprehensive evaluation across multiple datasets, metrics, and downstream tasks”; “TabEBM improves downstream performance … generates high-fidelity synthetic data”*
    - `XdB4`: *“The experimental results section is thorough and comprehensive”*
- **Clear presentation**
    - `tdoQ`: *“well-structured and clear”*
    - `Cc3y`: *“The manuscript is well organized”; “The text is clear”; “The quality of the figures is high”*
    - `XdB4`: *“The paper is well-structured and clearly written, effectively communicating the key ideas and concepts.”*
- **Impactful method**
    - `Cc3y`: *“The authors address a very common problem for the broad scientific community”; “this work can have an extremely broad impact in the future”*
- **Reproducibility**
    - `iojh`: *“Open-source implementation provided”*

## **(2) Summary of our responses and new experiments**

We replied to **all** questions and concerns raised by the reviewers:

*Table R0*. Number of our responses to the reviewers’ comments (`#Raised`/`#Replied`).

| Reviewer | # Questions | # Weaknesses | # Limitations | # New experiments |
| --- | --- | --- | --- | --- |
| `tdoQ` | 2/2 | 4/4 | N/A | 1 |
| `Cc3y` | 5/5 | N/A | N/A | 1 |
| `iojh` | 6/6 | 2/2 | 3/3 | 3 |
| `XdB4` | 6/6 | 2/2 | N/A | 5 |

We provide **10 new experiments** and attach a **rebuttal PDF**. The new results are *consistent* with the main text and further support TabEBM’s effectiveness. We will add them to the revised manuscript. Below, we detail the new experiments for each reviewer.

1. `[tdoQ, XdB4]` Evaluation on six new, leakage-free datasets with mixed feature types.
2. `[Cc3y]` Ablation studies on the surrogate binary classification task w.r.t. the imbalance between the negative samples and the real samples.
3. `[iojh]` Evaluation on highly imbalanced datasets to show TabEBM’s robustness.
4. `[iojh]` Ablation studies on TabEBM’s sensitivity to the hyperparameters of the SGLD sampling process.
5. `[iojh]` Evaluation on larger datasets to show TabEBM’s scalability.
6. `[XdB4]` Ablation studies on the distance of the negative samples and the real samples.
7. `[XdB4]` Evaluating the impact of the distribution of the negative samples.
8. `[XdB4]` Investigation into TabPFN's logit values on the surrogate binary tasks.
9. `[XdB4]` Ablation studies on tuning the hyper-parameters of generative benchmarks.
10. `[XdB4]` Ablation studies on tuning the downstream predictors.

---

### Decision · Program_Chairs · 2024-09-25

**Decision:**

Accept (poster)

**Comment:**

The authors propose a generative method for tabular data augmentation, utilizing a distinct generative process for each class in a classification setting.

- **Reviews**: The paper received mostly positive scores, though some were near and one below the threshold. The weaknesses noted were in the areas of novelty, clarity, and missing technical details.
- **Rebuttal**: The authors made significant efforts to address the reviewers' concerns, particularly by providing additional experiments. As a result, two reviewers raised their scores. It appears that the authors successfully addressed most of the points that were raised.
- **Post-decision Revision**: As noted by one reviewer,
    + the new dataset `collins` seems to have been used to tune TabPFN and hence is not meaningful to derive conclusions from;
    + the statistical fidelity metrics should be better explained, as it is 1) not clear how they are aggregated, 2) the KS metric only considers numerical features, and 3) the KL is potentially based only on categorical features.